# Mouse liver assembloids model periportal architecture and biliary fibrosis

Anna M. Dowbaj[1,7,11], Aleksandra Sljukic[1,11], Armin Niksic[1], Cedric Landerer[1,2], Julien Delpierre[1,8], Haochen Yang[1,2,3,4], Aparajita Lahree[1], Ariane C. Kühn[1], David Beers[3,9], Helen M. Byrne[3,4], Sarah Seifert[1], Heather A. Harrington[1,2,3,5,6], Marino Zerial[1,2,10] & Meritxell Huch[1,2,6✉]

Modelling liver disease in vitro requires systems that replicate disease progression[1,2]. Current tissue-derived organoids do not reproduce the complex cellular composition and tissue architecture observed in vivo[3]. Here, we describe a multicellular organoid system composed of adult hepatocytes, cholangiocytes and mesenchymal cells that recapitulates the architecture of the liver periportal region and, when manipulated, models aspects of cholestatic injury and biliary fibrosis. We first generate reproducible hepatocyte organoids with a functional bile canaliculi network that retain morphological features of in vivo tissue. By combining these with cholangiocytes and portal fibroblasts, we generate assembloids that mimic the cellular interactions of the periportal region. Assembloids are functional, consistently draining bile from bile canaliculi into the bile duct. Of note, manipulating the relative number of portal mesenchymal cells is sufficient to induce a fibrotic-like state, independently of an immune compartment. By generating chimeric assembloids of mutant and wild-type cells, or after gene knockdown, we show proof of concept that our system is amenable to investigating gene function and cell-autonomous mechanisms. Together, we demonstrate that liver assembloids represent a suitable in vitro system to study bile canaliculi formation, bile drainage and how different cell types contribute to cholestatic disease and biliary fibrosis in an all-in-one model.

The liver is organized in lobules, hexagonal functional units with a portal region in each corner and a central vein as the mid-point[4]. At the cellular level, the liver is composed of two epithelial cell types: hepatocytes and cholangiocytes, also known as ductal cells, and several stromal, endothelial and immune resident subpopulations[5,6]. Hepatocytes are uniquely polarized cells, with one or more apical poles that contribute to bile canaliculi, tubular lumina formed by adjacent hepatocytes. Bile canaliculi form a continuous network of tubes 1.5 to 2.5 μm in diameter, which drain bile into cholangiocyte-lined bile ductules and bile ducts[7]. Dysregulation of bile flow or the connection between bile canaliculi and bile duct results in cholestatic liver diseases, which can progress to fibrosis, cirrhosis and ultimately liver failure[2,6].

In cholestatic disease, portal mesenchyme and, specifically, portal fibroblasts and cholangiocytes become activated and expand, in both animal models (bile duct ligation and $Mdr2^{-/-}$ mice ($Mdr2$ is also known as $Abcb4$))[4,8] and human disease[1,9]. During cholestatic disease, the interactions between the different cell populations are disrupted, which suggests that restoring them could help the repair response[2,10]. Spatial transcriptomic analysis revealed many of the molecules involved in the cell–cell communication in the periportal region in healthy[11,12] and cholestatic[13] liver tissue. However, how these contribute to homeostasis, regeneration or fibrosis remains largely unknown. This is because

these interactions are highly dynamic, occur over short timescales, and require imaging with cellular resolution, which makes them extremely challenging to study in vivo, in living animals.

None of the in vitro systems that have been developed to date have modelled the complex homeostatic or fibrotic periportal cellular interactions and the periportal specialized functions and mesoscale architecture of the liver[3]. Tissue-derived adult mouse ductal (cholangiocyte)[14] or hepatocyte[15,16] organoids are made up solely of epithelial cells. Epithelial co-cultures of cholangiocyte and hepatocyte progenitors in 2D have been reported[17], but similar to hepatocytes cultured in 2D sandwich culture[18], they do not recapitulate the 3D network properties of the bile canaliculi network or contain stromal cells to recapitulate the periportal epithelial–stromal interactions[17]. Similarly, 3D liver organoids derived from pluripotent stem cells remain immature and do not replicate adult architecture[19]. Previous studies showed that cholangiocyte–mesenchyme organoid co-cultures retain aspects of 3D tissue morphology and the binary cell interactions[20]. However, this model lacks the other cell types in the periportal region.

Here, we developed hepatocyte organoids (HepOrgs) with a physiological 3D bile canaliculi network and combined them with cholangiocytes and portal mesenchyme to generate periportal liver organoids (periportal assembloids) that recapitulate the in vivo bile

[1]Max Planck Institute of Molecular Cell Biology and Genetics, Dresden, Germany. [2]Center for Systems Biology (CSBD), Dresden, Germany. [3]Mathematical Institute, University of Oxford, Oxford, UK. [4]Ludwig Institute for Cancer Research, University of Oxford, Oxford, UK. [5]Faculty of Mathematics, Technische Universität Dresden, Dresden, Germany. [6]Cluster of Excellence Physics of Life, TU Dresden, Dresden, Germany. [7]Present address: Center for Organoid Systems, Technische Universität München, Munich, Germany. [8]Present address: Institut Curie, Département de Génétique et Biologie du Développement, Paris, France. [9]Present address: Department of Mathematics, University of California, Los Angeles, Los Angeles, CA, USA. [10]Present address: Human Technopole, Milan, Italy. [11]These authors contributed equally: Anna M. Dowbaj, Aleksandra Sljukic. ✉e-mail: huch@mpi-cbg.de

canaliculi–bile duct cell–cell interactions and architectural organization. Using this system, we model many aspects of biliary liver fibrosis, including hepatocyte death, matrix deposition and ductal cell expansion, but not the inflammatory reaction. Additionally, we provide proof of concept that liver assembloids can be used to investigate molecular and cell-autonomous mechanisms in liver disease.

## HepOrgs form 3D bile canaliculi network

To ensure proper bile flow, hepatocytes must form a functional and physiological bile canaliculi network to connect to the bile duct lined by cholangiocytes surrounded by portal fibroblasts (Fig. 1a). Hepatocytes cultured in 2D sandwich form bile canaliculi[18], but do not generate the in vivo 3D network (Extended Data Fig. 1a,b). HepOrgs express bile canaliculi markers—however, the presence of a network has not been studied[15,16]. By staining for canaliculi and quantifying bile canaliculi length and branching, we found that the reported models[15,16] lack sufficient bile canaliculi to generate a network, which would ensure a reliable bile canaliculi–bile duct connection (Fig. 1d,e, compare Hu et al.[15] and Peng et al.[16] with tissue and Extended Data Fig. 1b–f). Thus, building upon the existing models, we first optimized HepOrgs to generate a physiologically relevant bile canaliculi network, while maintaining sufficient organoid expansion potential.

Wnt signalling is essential to increase hepatocyte[21–23] and cholangiocyte[14] proliferation during mouse liver regeneration and in liver cancer[24]. Consequently, we supplemented our previously published medium[25] with WNT3a ligand[26] (HM-Wnt) or Wnt surrogate[27] (HM-WntS). Organoids increased in size and cell numbers, presented significantly higher organoid formation efficiency (twofold to threefold increase) compared with the previous method, and could be maintained long term (Fig. 1b–f and Extended Data Fig. 1g–j). This size increase was not due to organoid merging, as cells seeded in sparse culture conditions also expanded over time (Extended Data Fig. 1i).

We observed heterogeneity in the shape of the generated structures, with some being spherical with smooth surfaces (referred to here as ball) and others exhibiting a folded shape with coarse surfaces (bubbly/grape-like) (Extended Data Fig. 2a). Using topological data analysis and the shape descriptor algorithm DETECT[28] (detecting temporal shape changes with the Euler characteristic transform; Methods and Supplementary Methods, section 2), we confirmed that the ball and bubbly/grape-like structures represent quantitatively different shapes that cluster independently of each other (Extended Data Fig. 2b–d). Live imaging revealed that the majority of the ball-shaped structures started from single hepatocytes, whereas hepatocyte doublets or clusters of more than two cells generated bubbly/grape-like structures (23% versus 87% efficiency) (Extended Data Fig. 2e,f). Both types of structures presented similar gene expression and function (Extended Data Fig. 2g–j), however they differed in their bile canaliculi architecture. Bubbly/grape-like structures presented thinner, longer and more interconnected bile canaliculi and consistently secreted the bile acid analogues cholyl-L-Lys-fluorescein (CLF) and metabolized 5-carboxyfluorescein diacetate, acetoxymethyl ester (5-CFDA). Ball-like structures did not secrete CLF or metabolize 5-CFDA, but accumulated the bile acid analogues inside of hepatocytes, confirming aberrant or absent bile canaliculi (Extended Data Fig. 2k,l and Supplementary Videos 1 and 2). This observation correlated with an increase in cleaved-caspase 3 staining (Extended Data Fig. 2k), as expected from intracellular bile acid accumulation.

These results indicate that marker gene expression is not sufficient to guarantee a functional and physiological bile canaliculi network or metabolite transport. Hereafter we used HepOrg cultures that were enriched for bubbly/grape-like organoid shape.

The optimized HepOrgs retained hepatocyte identity, expressed hepatocyte markers (such as HNF4α and albumin) and the bile canaliculi transporters MDR2 (encoded by *Abcb4*) and MRP2 (encoded

by *Abcc2*), but not ductal cell (for example, SOX9 and KRT19) or non-epithelial markers, thereby resembling freshly isolated hepatocytes (Fig. 1g, Extended Data Fig. 3a–f and Supplementary Table 2). HepOrgs secreted similar amounts of albumin as previously published 3D organoid models and more albumin than 2D hepatocyte sandwich cultures (Extended Data Fig. 3g,h). The optimized HepOrgs also recapitulated the complex hepatocyte cell polarity of the in vivo tissue, with the polarity markers ZO-1 and CD13 localized to the apical surface of adjacent hepatocytes and E-cadherin (ECAD) localized to the basolateral side (Fig. 1h and Extended Data Fig. 3c,e). Image analysis revealed that the optimized HepOrgs formed a significantly longer and more branched bile canaliculi network within each organoid and were closer to the tissue values compared with the previous methods (Fig.1e and Extended Data Fig. 1e,f).

Occasionally, we found cholangiocyte organoids growing in close proximity to HepOrgs, forming independent structures that were neither embedded nor formed physiological bile canaliculi–bile duct connections with hepatocytes (Extended Data Fig. 4a–e). We therefore further tested additional culture conditions with the aim of generating structures that would retain the complex hepatocyte polarity and support the culture of the three cell types while preserving their physiological ratios. In all conditions tested, HepOrgs were viable and retained the expression of hepatocyte markers (Extended Data Fig. 4f–i). Our previously published minimal medium (MM) for co-culture of ductal and portal mesenchyme cells[20] was superior to the other media that we tested, and supported the maintenance of cholangiocytes and mesenchyme without their overgrowth (Extended Data Fig. 4j). Hepatocyte function was similar but hepatocytes from HepOrgs cultured in MM were more multinucleated, which is considered a hallmark of mature hepatocytes (Extended Data Fig. 4k–m). RNA-sequencing (RNA-seq) analysis revealed that in both culture conditions, HepOrgs maintain the expression of genes encoding hepatocyte markers such as *Alb*, *Hpx* and *Cyp3a1*, and bile canaliculi transporters (*Abcb11* (also known as *Bsep*), *Abcc2* (also known as *Mrp2*) and *Abcb4* (also known as *Mdr2*)), in similar amounts as freshly isolated hepatocytes (Extended Data Fig. 5a and Supplementary Table 2). However, HepOrgs cultured in MM expressed higher levels of periportally zonated genes (*Fbp1*, *Arg1* and *Aldob*), whereas pericentral genes such as *Glul* (encoding glutamine synthetase) and *Axin2* were expressed in both conditions (Extended Data Fig. 5a). These observations suggested that the cultures exhibited some degree of zonation. To assess this, we performed single-cell RNA-seq (scRNA-seq) analysis of HepOrgs cultured in MM and our optimized expansion medium HM-Wnt (Methods). To underscore potential zonated gene expression, we generated a consensus pericentral or periportal score using publicly available datasets[13,29,30] (Methods). HepOrgs cultured in HM-Wnt scored more pericentrally, whereas MM HepOrgs scored more periportal, in agreement with the expression of selected zonated markers (Extended Data Fig. 5b–d). RNAscope and immunofluorescence analysis for periportal (*Gls2* RNA and albumin and ECAD protein) and pericentral (*Cyp1a1* RNA and CYP2E1 and glutamine synthetase protein) markers confirmed the heterogeneous distribution of these markers, with pericentral genes being more highly expressed on the periphery of an organoid, where cells are more exposed to the WNT ligand from the medium (Fig. 2a and Extended Data Fig. 5e,f). Together, the results suggested that HepOrg cultured in MM acquired a more periportal gene expression signature.

Of note, image analysis and reconstruction demonstrated that HepOrgs cultured in MM presented longer bile canaliculi, as well as narrower distributions of bile canaliculi diameter, which were within the physiological range[31] of 1.5 to 2.5 μm, narrower than the 2 to 7 μm diameter of bile canaliculi formed by HepOrgs cultured in HM-Wnt (Fig. 2b–d and Extended Data Fig. 6a–c). Notably, the bile canaliculi network connectivity was further improved in MM compared with HM-Wnt, resembling that of the liver tissue (Fig. 2d, right). We tested the functionality of MRP2, BSEP and MDR2 bile transporters located in the canalicular membrane using fluorescently labelled bile acid analogues

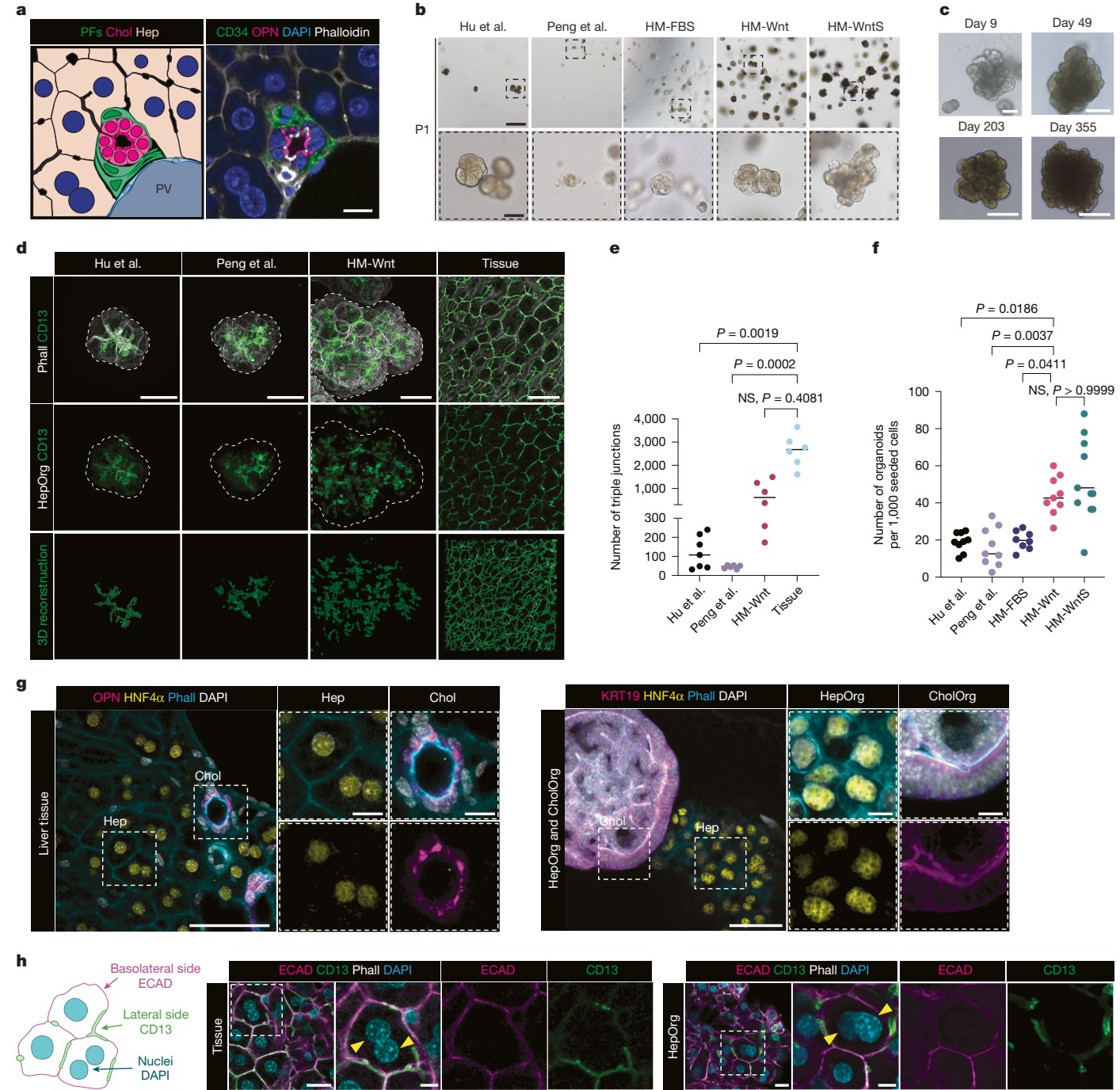

**Fig. 1 | HepOrgs retain marker expression and cell polarity of tissue.**
**a**, Schematic (left) and representative immunofluorescence image (right; *n* = 5) of the liver periportal region. CD34 marks portal fibroblasts (PFs), osteopontin (OPN) marks ductal cells, phalloidin marks membranes and DAPI labels nuclei. Chol, cholangiocyte; Hep, hepatocyte; PV, portal vein. Scale bar, 10 μm.
**b**, Representative bright-field images of HepOrgs at passage 1 (P1) cultured under the conditions described in Hu et al.[15], Peng et al.[16] or in HM-FBS, HM-Wnt or HM-WntS (*n* = 5 experiments). Scale bars: 500 μm (top row), 10 μm (bottom row). FBS, foetal bovine serum. **c**, Representative BF images of HepOrgs cultured in HM-Wnt at the indicated time points. Scale bars: 50 μm (day 9), 100 μm (day 49), 200 μm (day 203 and day 355). **d**, Immunofluorescence staining and 3D reconstruction of bile canaliculi (marked by CD13) in HepOrgs cultured in indicated conditions and mouse tissue. HepOrgs cultured in HM-Wnt have longer bile canaliculi compared with previous studies[15,16]. Cell borders are indicated by filamentous actin (F-actin) staining with phalloidin (Phall). Top, maximum-intensity projections of confocal images. HepOrgs are outlined with dashed lines. Bottom, bile canaliculi segmentation and 3D reconstruction.

Scale bars, 50 μm. **e**, Number of triple junctions in the largest bile canaliculi network in tissue and organoids cultured in indicated conditions. Dots show the total number of triple junctions per structure and the line represents the mean. Kruskal–Wallis test with Dunn's multiple comparisons post hoc test. NS, not significant. **f**, Organoid formation efficiency. Dots represent biologically independent samples (*n* = 5 biological replicates with at least 2 technical replicates) and the line represents the mean. Kruskal–Wallis test with Dunn's multiple comparisons post hoc test. **g**, Immunofluorescence staining of tissue (left) and hepatocyte and cholangiocyte organoids (HepOrgs and CholOrg, respectively) grown in HM-Wnt (right), for the hepatocyte marker HNF4α and cholangiocyte markers KRT19 and OPN (*n* = 2 experiments). Scale bars: 50 μm (main images), 10 μm (smaller images). Phalloidin marks membranes and DAPI labels nuclei. **h**, Left, schematic showing hepatocyte polarity. CD13 (apical) and ECAD (basolateral) staining in tissue (middle) and organoids (right) shows similar distributions of markers. Yellow arrowheads indicate binucleated cells. Scale bars: 20 μm (left most images), 10 μm (expanded views). All data were obtained from *n* = 3–5 independent experiments.

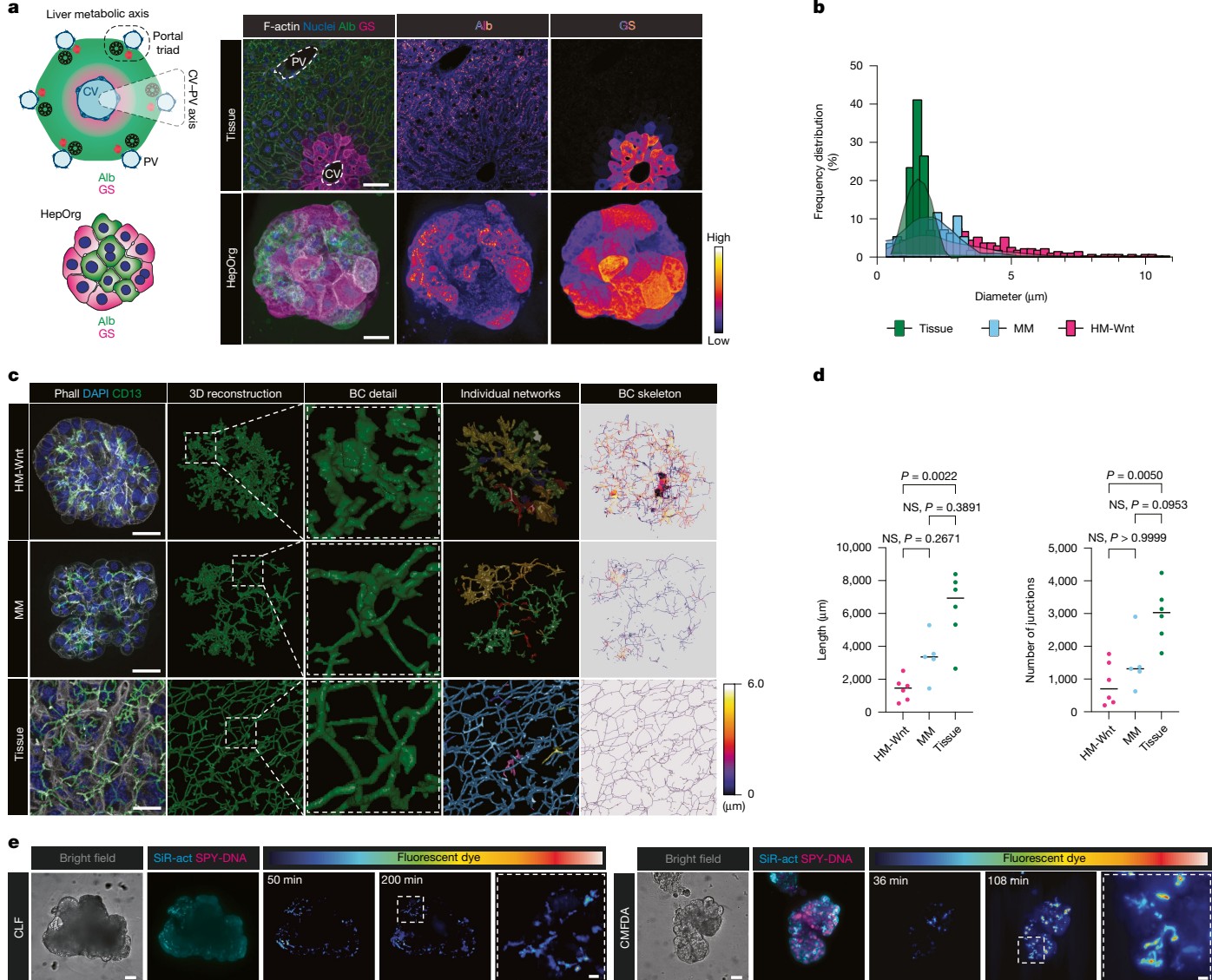

**Fig. 2 | HepOrgs exhibit physiological and functional bile canaliculi size and network resembling tissue. a**, Left, schematic of liver zonation. Right, maximum-intensity projections of staining from periportal (albumin (Alb)) and pericentral (glutamine synthetase (GS)) markers in tissue (top row) and HepOrgs cultured in MM (bottom row). Albumin and glutamine synthetase also shown as a Fire look-up table in the middle and right images (*n* = 3 independent experiments). Cells expressing glutamine synthetase are detected in the periphery of the organoids. Scale bars: 50 μm (top row), 20 μm (bottom row). CV, central vein. **b**, Histogram showing distribution of bile canaliculi diameters in tissue and HepOrgs. The curve represents the kernel density estimate, used here to estimate the probability density function of a continuous random variable, showing a smooth curve that represents the distribution of data points provided in Source Data. *n* = 3 independent experiments. **c**, Immunofluorescence staining (left), image reconstruction and analysis of bile canaliculi (BC) from HepOrgs cultured in HM-Wnt or MM, and tissue stained for CD13 (marking bile canaliculi), DAPI (nuclei) and phalloidin (F-actin). Individual interconnected networks are pseudo-coloured. The skeleton of the bile canaliculi network shows that the networks are longer and more interconnected in HepOrgs cultured in MM. Scale bar, 20 μm. *n* = 3 independent experiments. **d**, Left, length of the largest bile canaliculi network in HepOrgs cultured in HM-Wnt or MM, and tissue. Right, total number of junctions in the largest bile canaliculi network. Dots show the largest network in an individual organoid. Horizontal line indicates the median. HM-Wnt, *n* = 6; MM, *n* = 5; tissue, *n* = 6. Kruskal–Wallis test with Dunn's multiple comparisons post hoc test. **e**, Transport of CLF and CMFDA in HepOrgs cultured in MM confirm the functionality of bile acid transporters. Compounds are shown in Royal look-up tables. Nuclei (SPY555-DNA (SPY-DNA)) and actin (SiR-actin (SiR-act)) are labelled (*n* = 3 independent experiments). Scale bars for each set of images are 50 μm and 10 μm (magnification). Images are stills from time-lapse imaging shown in Supplementary Videos 3 and 4.

CLF and chloromethylfluorescein diacetate (CMFDA) (which indicate active BSEP and MRP2 transport) and the fluorescent phosphatidylcholine marker (for MDR2-mediated transport). We consistently observed uptake and transport of the fluorescent labels to the canalicular lumen (Fig. 2e, Extended Data Fig. 6d and Supplementary Videos 3–5), confirming the functionality of the bile canaliculi network, in agreement with the gene expression of apical transporters in our HepOrg cultures (Extended Data Fig. 6e).

Thus our optimized HepOrg system enables the growth of adult hepatocytes that preserve hepatocyte polarity, show partial hepatocyte zonation and generate a physiological and functional bile canaliculi 3D network resembling the in vivo adult liver tissue, without the emergence of cholestatic features that have been found in previous 2D hepatocyte cultures. Critically, our results highlight the importance of mimicking physiological properties at the cellular and tissue scale level, as these directly impact the overall fitness of organoid models.

## Assembloids mimic periportal architecture

We next aimed to reconstruct the periportal region of the liver lobule, specifically the heterotypic cellular interactions between hepatocytes, cholangiocytes and portal mesenchyme. We adapted the concept of assembloids[32] by mixing defined numbers of single cholangiocytes from ductal organoids (grown as described[14]) and portal mesenchymal cells (grown as described[20]) with a defined number of HepOrgs and cultured them in MM (Methods). We opted for a two-pronged strategy, using either a rocking platform or AggreWell plates (Fig. 3a and Methods). Both methods attained multicellular structures with high efficiency (around 70% of the total HepOrgs), which was further increased (to around 90%) when HepOrgs were pre-conditioned in the co-culture medium (Fig. 3b,c and Extended Data Fig. 7a–c). We selected the rocking platform method for subsequent experiments. The majority of assembloids were formed within the first 48 h, although assembloids containing mesenchyme could also aggregate after seeding (Extended Data Fig. 7d,e and Supplementary Videos 6 and 7). Notably, the composite structures retained the cellular proportions of the tissue (Fig. 3d) and recapitulated the tissue organization, with cholangiocytes forming bile duct structures with opened lumens (marked by KRT19, SOX9 and tdTomato, asterisk) surrounded by portal mesenchymal cells (marked by vimentin, elastin and PDGFRα–H2B–GFP) and embedded in the hepatocyte parenchyma (marked by HNF4α) (Fig. 3b,e and Extended Data Fig. 7a,f,g, comparing organoid versus tissue), thus recapitulating the mesoscale architecture of the native tissue.

scRNA-seq analysis indicated that the assembloid cells retain their identities and expression signatures, with hepatocytes, cholangiocytes and mesenchymal cells expressing known lineage markers such as *Alb* (hepatocytes), *Krt19* (cholangiocytes) and *Col1a1* (mesenchyme), respectively (Fig. 3f). Mesenchymal cells were enriched for basement membrane, metallopeptidase activity and extracellular matrix signatures, cholangiocytes were enriched for cell–cell junctions, and hepatocytes were enriched for metabolic processes and cholesterol transport, among others (Fig. 3f, Extended Data Fig. 7h–m and Supplementary Table 3). The assembloid populations mostly overlapped with in vivo hepatocyte, cholangiocyte and portal mesenchymal cells from several mouse liver cell atlases[11–13,33–39] (*n* = 10; Fig. 3g–i, Extended Data Fig. 8a–d and Methods). Marker gene analysis and quantitative PCR with reverse transcription (RT–qPCR) on sorted cells from assembloids compared with freshly isolated cells further confirmed that the cells mostly retain the expression profile of adult mouse liver tissue (Extended Data Fig. 8f–i). As expected, the mesenchyme most closely resembled portal fibroblasts and clustered further away from other mesenchymal populations (hepatic stellate cells (HSCs) and vascular smooth muscle cells (VSMCs)) (Extended Data Fig. 8c,d), whereas the hepatocytes of the assembloids retained a certain degree of zonation, with the expression of periportally zonated markers (Extended Data Fig. 8e).

Next, we investigated the fine detail of the tissue architecture at the cellular scale—specifically, whether the bile duct was functionally connected to the hepatocyte bile canaliculi network, akin to the tissue's canal of Hering structure[4]. Immunofluorescence staining revealed that the connection between bile canaliculi and bile duct in periportal assembloids recapitulated that of the native liver tissue in 100% of the cases where cholangiocytes were incorporated inside the structure (Fig. 3j, Extended Data Fig. 9a–g and Supplementary Videos 8–11; compare assembloid to tissue). Frequently, several bile canaliculi joined one bile duct lumen, usually surrounded by portal mesenchymal cells in close vicinity, as in the tissue (Fig. 3j, Extended Data Fig. 9a–g, bottom and Supplementary Videos 10 and 11). The bile canaliculi–bile duct connection was functional—since we readily detected CLF transported from the hepatocyte bile canaliculi network into the bile duct lumen (Fig. 3k,l, Extended Data Fig. 9h and Supplementary Videos 12 and 13)—at timescales that are close to those in the native tissue[40]. This fine cellular-scale detail was not observed in assembloids with aberrant ratio

and non-physiological arrangements of cholangiocytes to hepatocytes or in cholangiocyte-only organoids (Extended Data Fig. 9i and Supplementary Video 14). Furthermore, we found that the bile canaliculi–bile duct connection further improved the differentiation of the cells in the structures. We observed that hepatocytes increased expression of several bile acid transporter genes (*Abcb4*, *Abcc2* and *Abcc3*), whereas cholangiocytes expressed *Ezr* and *Ano1*, apical polarity and transporter markers, respectively (Extended Data Fig. 9j,k).

Collectively, these results confirm that periportal assembloids recapitulate the expression pattern, cellular function and periportal tissue architecture of the native tissue, at both the meso and cellular scale. In addition, our results further highlight that modelling a physiological bile canaliculi architecture, function and connection to the bile duct has benefits for the overall maturation of cells and the overall robustness of the assembloid model.

## Assembloids model biliary fibrosis

Since portal mesenchyme contributes to mouse and human biliary fibrosis[1,9], we investigated whether our periportal assembloid model could recapitulate aspects of this disease. We performed experiments in which we kept the hepatocyte and cholangiocyte numbers constant, but increased the initial number of portal mesenchymal cells (Fig. 4a). Non-physiological numbers of portal mesenchyme (10× excess) consistently resulted in structures with altered morphology compared with structures with physiological numbers (Fig. 4b). Gene expression analysis (RNA-seq) indicated that assembloids with high mesenchymal cell ratio exhibited increased expression of ductal cell markers (*Krt7*, *Sox9* and *Krt19*) and several collagens (*Col1a2* and *Col3a1*), which were persistent or even further increased in assembloids cultured longer-term (2.5 weeks; Extended Data Fig. 10a and Supplementary Table 2). Next, we compared scRNA-seq analysis of fibrotic versus homeostatic assembloids to nine publicly available single-cell liver datasets from in vivo mouse models of liver fibrosis[11–13,33–35,38,39] (Fig. 4c and Extended Data Fig. 10b–h). Cholangiocytes from fibrotic-like assembloids clustered closely to cholangiocytes from various damage models, whereas hepatocytes from fibrotic-like assembloids clustered closely to cholangiocytes, potentially suggestive of a degree of transdifferentiation. Similarly, mesenchyme from fibrotic-like assembloids closely correlated to fibroblasts from several damage models (Extended Data Fig. 10h). We observed a remarkable similarity, both in strength and magnitude, between the inferred cell–cell interactions from in vivo biliary fibrosis models[33] and our fibrotic assembloids. The predicted mesenchyme–mesenchyme interactions included *Col1a1*–*Ddr2*, *Gas6*–*Axl* and the metalloproteases inhibitor gene interactions *Timp2*–*Itgb1*, *Timp1*–*Cd63*, which are implicated in fibrosis progression[33,41–43]. Among the hepatocyte–mesenchyme interactions, the most notable were *Fgb* with *Itgb1* (Fig. 4d and Supplementary Table 4). We found a significant enrichment in signalling pathways involved in liver fibrosis and known to be activated in cholestatic injury such as IL-6 signalling, collagen remodelling, deposition and degradation, matrix metalloproteases, extracellular matrix turnover and cytokine signalling (Fig. 4e and Supplementary Table 5). Hepatocytes showed upregulation of stress-related pathways such as p53 signalling and downregulation of bile acid and fatty acid metabolism (Extended Data Fig. 10e). Conversely, cholangiocytes showed a reduction in apoptosis and TGFβ signalling (Extended Data Fig. 10f), whereas mesenchymal cells presented changes in signatures and pathways known to be involved in liver fibrosis such as PI3K–AKT–mTOR[44] (Extended Data Fig. 10g).

Together, these results suggested that the structures with high mesenchyme cell ratios exhibited a fibrotic-like phenotype; therefore, we called them 'fibrotic-like', in contrast to 'homeostatic' for the assembloids with physiological mesenchymal cell numbers.

Notably, in fibrotic-like assembloids, but not in homeostatic assembloids, we observed that the increase of the initial numbers of portal

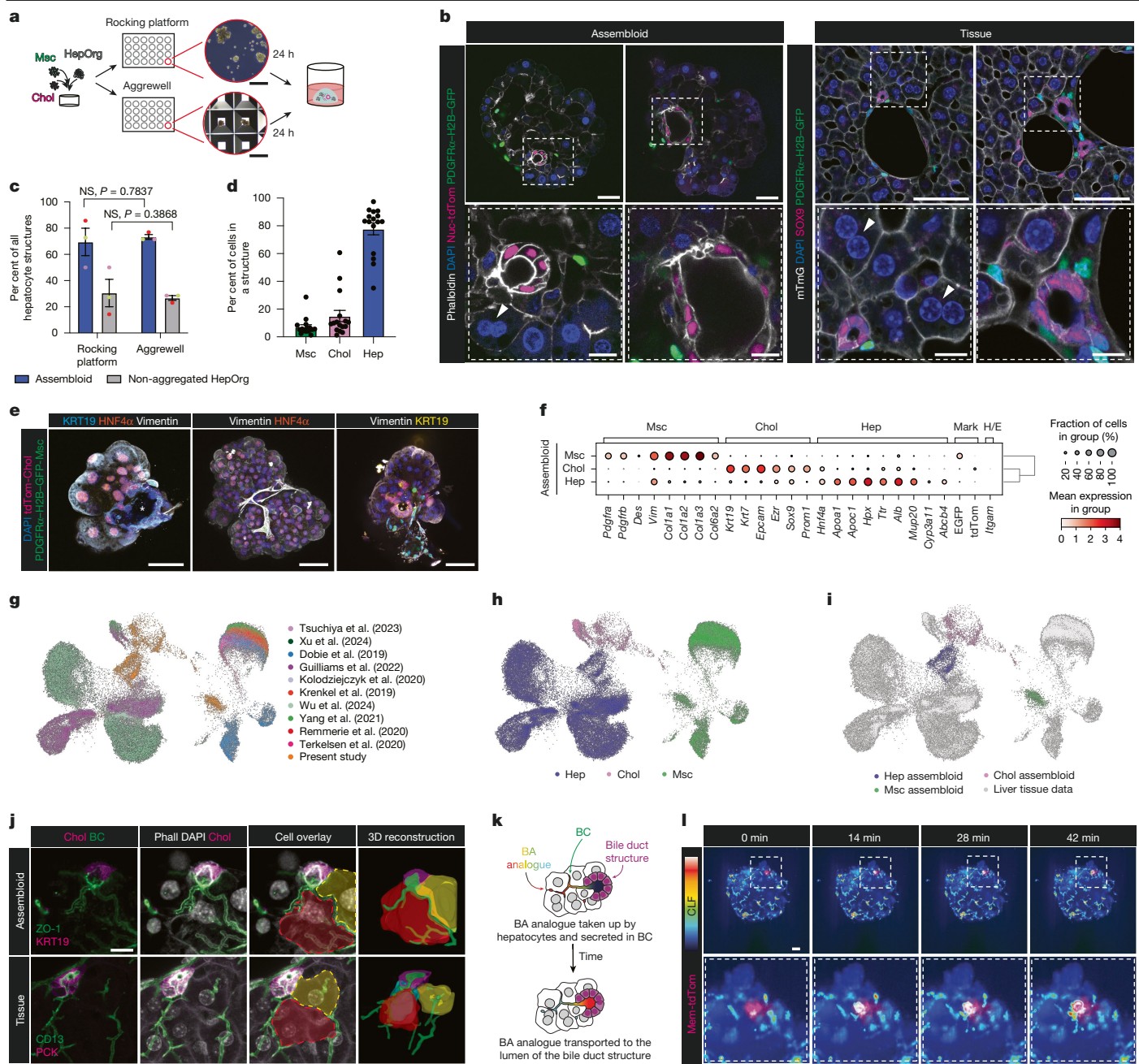

**Fig. 3 | Assembloids recapitulate mesoscale tissue architecture and gene expression. a**, Schematic of the experimental approach. Scale bars: 200 μm (top), 500 μm (bottom). Msc, mesenchyme. **b**, Representative images (*n* > 3 experiments) of assembloids cultured on a rocking platform (left) compared with tissue (right). PDGFRα–H2B–GFP marks mesenchyme, nuclear-tdTom and SOX9 mark cholangiocytes. Arrowheads indicate binucleated hepatocytes. Scale bars: 50 μm (main images), 20 μm (assembloids, zoom view), 10 μm (tissue, zoom view). **c**, Aggregation efficiency. Data are mean ± s.e.m. of *n* = 3 biological replicates from *n* = 3 independent experiments; Mann–Whitney test, two-tailed. **d**, Cellular composition of assembloids. Data are mean ± s.e.m. of assembloids from at least 3 independent experiments (*n* = 13 organoids total). Dots represent the percentage of hepatocyte, cholangiocyte or portal mesenchyme cells per structure. **e**, Representative confocal images of assembloids stained for the indicated markers. *n* > 3 experiments. Asterisks indicate bile duct lumen. Scale bars, 50 μm. **f**, Hepatocyte, cholangiocyte and mesenchyme marker expression in assembloids. Haemopoietic/endothelial markers (H/E)

are not expressed. EGFP marks mesenchyme, tdTomato (tdTom) marks cholangiocytes. **g**, Uniform manifold approximation and projection (UMAP) from liver atlas datasets and assembloids (this study). **h**, Mesenchyme (green), hepatocytes (blue) and cholangiocytes (magenta) superimposed on UMAP data from **g**. **i**, Assembloid data superimposed on data from **g**. **j**, Immunofluorescence staining and image reconstruction of the connection between bile canaliculi from hepatocytes (ZO-1, CD13) and the lumen from bile duct (KRT19, PCK) in assembloids (top) and tissue (bottom). Right, 3D reconstruction visualizes hepatocytes (red, yellow) whose bile canaliculi (green) enter the bile duct lumen (magenta). *n* = 6 independent experiments, see Supplementary Videos 8 and 9. Scale bar, 10 μm. **k,l**, Schematic (**k**) and still images (**l**) of CLF transport (shown as Fire look-up table) from the live imaging shown in Supplementary Video 12 in assembloids indicates functional connection between bile canaliculi and bile duct lumen (mem-tdTomato). *n* = 3 independent experiments with *n* = 3 biological replicates. BA, bile acid analogue. Scale bar, 50 μm.

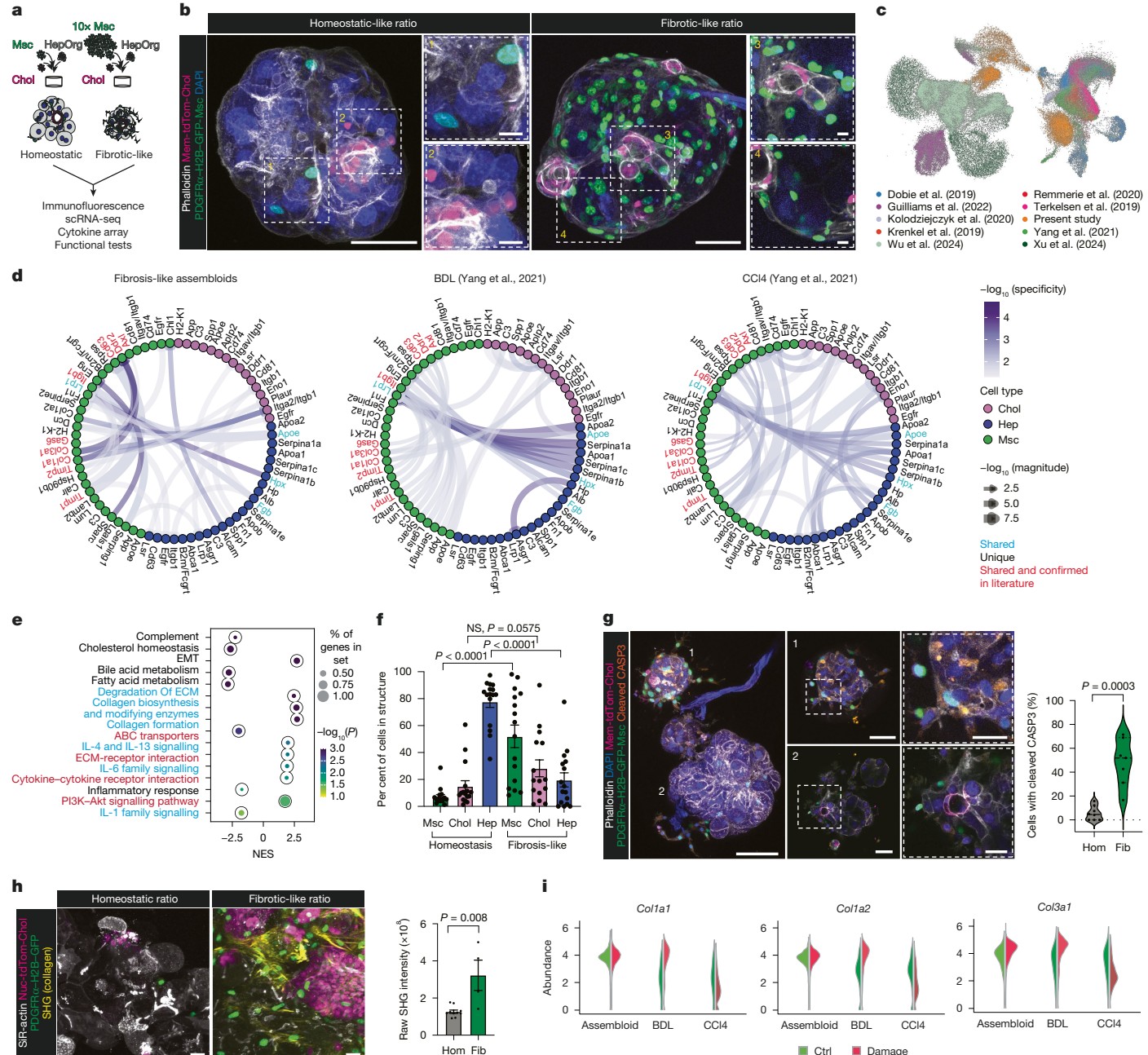

**Fig. 4 | Periportal assembloids mimic aspects of biliary fibrosis in vitro.**
**a**, Schematic of the experimental design. **b**, Immunofluorescence images of assembloids with homeostatic (left) and 10× excess (right) of mesenchyme. $n > 3$ experiments. Scale bars: 50 μm (main images), 10 μm (zoom view). **c**, UMAP of fibrotic-like assembloid data (from this study) integrated with datasets from liver damage models. **d**, Circular plots represent the inferred cell–cell interactions from fibrotic-like assembloids, bile duct ligation (BDL) and CCl4 models from Yang et al.[33]. Interactions reported in the literature and shared with BDL and CCl4 models are shown in red, interactions that are shared but not mentioned in the literature are in cyan and unique interactions are shown in black. Selected from the top 100 significant interactions for BDL. **e**, Gene set enrichment analysis (GSEA) of fibrosis-like versus homeostatic-like assembloids using MSigDB_Hallmark_2020 (black), KEGG_2019 (red) and Reactome_2022 (cyan) gene datasets. EMT, epithelial to mesenchymal transition; NES, normalized enrichment score. **f**, Cell composition of homeostatic-like and fibrotic-like assembloids. Dots show the percentage of cells per structure. Data are mean ± s.e.m. (3 independent experiments, $n = 13$ organoids); homeostatic-like data are reproduced from Fig. 3d. Mann–Whitney test, two-tailed. **g**, Left, immunofluorescence staining for cleaved caspase-3 (CASP3). Right, violin plot showing the median and quartiles of the percentage distribution of hepatocytes containing cleaved caspase-3 from three independent biological replicates. Each dot represents one organoid. Mann–Whitney test, two-tailed. Hom, homeostatic ($n = 8$); Fib, fibrotic-like ($n = 7$). Scale bars: 100 μm (left), 50 μm (middle), 25 μm (right). **h**, Left, SHG imaging reveals fibrous collagen deposition; cholangiocytes (nuc-tdTomato), mesenchyme (PDGFRα–H2B–GFP) and SiR-actin staining. Right, mean ± s.e.m. of the intensity of SHG signal from three biological replicates from $n = 3$ independent experiments. Mann–Whitney test, two-tailed. Homeostatic assembloids, $n = 10$; fibrotic-like assembloids, $n = 4$. Scale bars, 20 μm. **i**, Expression of selected genes in mesenchyme, presented as abundance, in homeostatic (Ctrl) and fibrosis-like (Damage) assembloids, or in damage or control tissue from BDL and CCl4 models from ref. 33.

mesenchymal cells (SCA1⁺PDGFRα⁺) affected the other two populations: hepatocytes and ductal cells (Fig. 4f). We observed a significant reduction in hepatocyte number, which also exhibited aberrant morphologies, distorted cell membranes and weak chromatin staining (Fig. 4g and Extended Data Fig. 11a). Live imaging consistently revealed big bursts of DNA signal coming from hepatocytes (Supplementary

Video 15). Cleaved caspase-3 staining indicated that hepatocyte death occurred, at least partially, through apoptosis, although gene expression analysis also indicates that other forms of cell death might be involved (Fig. 4g and Extended Data Fig. 11a–c). Hepatocyte death was also induced by freshly isolated portal mesenchymal cells in excess (fibrotic-like) numbers (Extended Data Fig. 11c). Consistently, fibrotic-like assembloids lost functional bile acid (CLF) uptake and reduced bile acid drainage from the bile canaliculi into the bile duct (Extended Data Fig. 11d–f and Supplementary Video 16), although we did not detect differences in albumin secretion, total bile acid or cytochrome activity between homeostasis and fibrotic-like assembloids (Extended Data Fig. 11g). Concomitant with the hepatocyte death, we observed increased numbers of ductal cells in fibrotic-like assembloids (Fig. 4f and Extended Data Fig. 11h), which were further increased in long-term cultured assembloids (more than 40 days; Extended Data Fig. 11i,j), reminiscent of the ductular reaction observed in biliary fibrosis patients[2]. Notably, second-harmonic generation (SHG) microscopy confirmed a significant increase in fibrillar collagen deposition in fibrotic-like assembloids (Fig. 4h), in agreement with the gene expression data (Fig. 4i) and a mouse model of biliary fibrosis[2,8]. Immunostaining confirmed the identity of portal fibroblasts (Extended Data Fig. 11k). In addition, several genes encoding pro-inflammatory molecules, including *Ccl11*, *Cxcl1* and *Cxcl12* and the metalloproteases *Mmp2* and *Mmp3*—all of which are implicated in fibrosis—were also highly upregulated in the mesenchyme of fibrotic-like assembloids (Extended Data Fig. 12a,b and Supplementary Table 4). Cytokine array analysis confirmed the secretion of CCL11, CXCL1, MMP2 and MMP3 by fibrotic-like assembloids, but not from matching control homeostatic assembloids (Extended Data Fig. 12c).

Together, the described features—namely: (1) fibrotic gene expression similar to in vivo models of biliary fibrosis; (2) hepatocyte death; (3) bile flow obstruction; (4) ductal cell expansion; and (5) collagen deposition—indicate that fibrotic assembloids recapitulate in vitro many aspects of the in vivo biliary fibrosis, except for the inflammatory reaction, as expected given that our system lacks the immune compartment.

## Assembloids as a tool to study fibrosis

We next investigated whether assembloids could be used as a tool to study liver fibrosis. We first examined mesenchyme-derived inflammatory cytokines, as recent human studies on biliary fibrosis[45] suggest that epithelium and mesenchyme express many cytokines, although their function remains unknown. Addition of CXCL12, CCL11 or CXCL1 had no effect on mesenchymal cells (Extended Data Fig. 12d). Similarly, blocking antibodies against selected cytokines did not rescue the fibrotic-like phenotype (Extended Data Fig. 12e). These results suggested that the paracrine signalling from mesenchymal cells was secondary to the fibrotic-like phenotype, at least for the cytokines tested, and notably, in the absence of the immune compartment.

Changes in cell adhesion and cell–extracellular matrix (ECM) interactions are common features of the fibrotic response. Therefore, we next tested whether assembloids could be exploited to investigate cell–cell and cell–ECM interactions in fibrosis. We first analysed our scRNA-seq results for potential cell adhesion, cell–ECM or ligand–receptor interactions that would be increased in the fibrotic assembloids compared with the homeostatic ones, focusing on the mesenchymal cell interactions that were also present in in vivo models of fibrosis (Fig. 4d and Extended Data Fig. 12a,b). We found that expression of the cell adhesion gene *Cdh11* was increased in fibrotic-like mesenchymal cells, and the ligand–receptor interactions involving molecules including the ECM integrin subunit ITGβ1 and the ECM modulators TIMP1 and TIMP2 (Extended Data Fig. 12a,b and Supplementary Table 4) significantly changed in intensity and magnitude. We selected some of these genes to perform a small knockdown screen in assembloids cultured

in fibrotic-like conditions (tenfold excess mesenchymal cells). As a first screening readout, we discriminated shape changes over time, as fibrotic-like assembloids are more compact compared to structures with physiological mesenchymal cell numbers. We again applied the shape descriptor algorithm DETECT[28] and developed the DETECT metric to quantitatively and statistically compare spatiotemporal changes in morphology of organoids under different conditions. We found that inhibiting mesenchymal cell adhesion by CDH11 knockdown as well disrupting mesenchyme–ECM interactions by knocking out ITGβ1 specifically in mesenchyme cells consistently prevented the fibrotic-like phenotype in assembloids. Specifically, assembloids did not change shape over time compared with controls, which became significantly more compact over time (Fig. 5a–d and Extended Data Figs. 12f–j and 13a–i). Additionally, ITGβ1 knockout or CDH11 knockdown in mesenchymal cells also resulted in a significant reduction in collagen deposition (Fig. 5d and Extended Data Figs. 12j and 13h,i), which could be related to the disruption of the mesenchyme–mesenchyme interactions as monocultures grown at similarly high density deposited large amounts of fibrillar collagen (Extended Data Fig. 13j). None of the other molecules tested resulted in similar rescue (Extended Data Fig. 12e–h).

Finally, we investigated whether our model would be relevant for studies of other aspects of the pathophysiology of cholestatic biliary fibrosis, such as the cell-autonomous mechanisms of liver fibrogenesis. We utilized a *Mdr2*−/− mouse model, a knockout mouse that lacks the phospholipid transporter MDR2, resulting in altered bile composition, and models primary sclerosing cholangitis[8], a cholangiopathy that develops biliary fibrosis. HepOrgs from MDR2-knockout mice exhibited signs of cholestatic liver disease, with the presence of dilated bile canaliculi, apical bulkheads, inward blebs and accumulation of liver rosettes (Fig. 5e), recognized hallmarks of in vivo cholestasis in mouse and humans[46–49]. These features were also observed in *Mdr2*−/− liver tissue, but were absent in control tissue and HepOrg derived from control wild-type littermates (Fig. 5e and Extended Data Fig. 14a–d). To investigate the effect of *Mdr2*−/− hepatocytes in the mechanisms of fibrosis, we then mixed the *Mdr2*−/− cholestatic HepOrgs with wild-type cholangiocytes and wild-type portal mesenchyme to generate chimeric assembloids (Fig. 5f,g). As in the in vivo tissue, both *Mdr2*−/− and wild-type control assembloids readily connected bile canaliculi with the bile duct (Extended Data Fig. 14e). Notably, assembloids formed by *Mdr2*−/− hepatocytes, but not wild-type controls, exhibited ductal cell expansion (Fig. 5g,h and Extended Data Fig. 14f), resembling the ductular proliferation observed in vivo in *Mdr2*−/− mice[8]. These results suggest that alterations in hepatocyte function contribute to ductular expansion, akin to early stages of biliary fibrosis in vivo.

In summary, we obtained a periportal assembloid model that recapitulates aspects of biliary fibrosis and cholestatic liver disease and is amenable for investigation of cell-autonomous and non-autonomous mechanisms in liver disease.

## Discussion

The study of the cellular mechanisms that regulate cholestatic liver injury and biliary fibrosis has been a major challenge, owing to the difficulty of modelling physiological bile canaliculi in vitro and the shortfall of cellular systems that recapitulate the architecture and cellular interactions between different cells in the periportal region.

Here, we generated a periportal assembloid model that combines portal mesenchyme, cholangiocytes and hepatocytes, and readily recapitulates the cellular and mesoscale architecture of the periportal region of the mouse liver, albeit lacking the portal endothelium and resident immune cells. By generating HepOrgs that retain apical polarity, we obtained a model that forms physiological and functional bile canaliculi 3D network. Notably, when HepOrgs are generated from the cholestatic *Mdr2*−/− mouse model, they faithfully recapitulate the specific histological features of cholestatic liver disease, including liver rosettes and bile

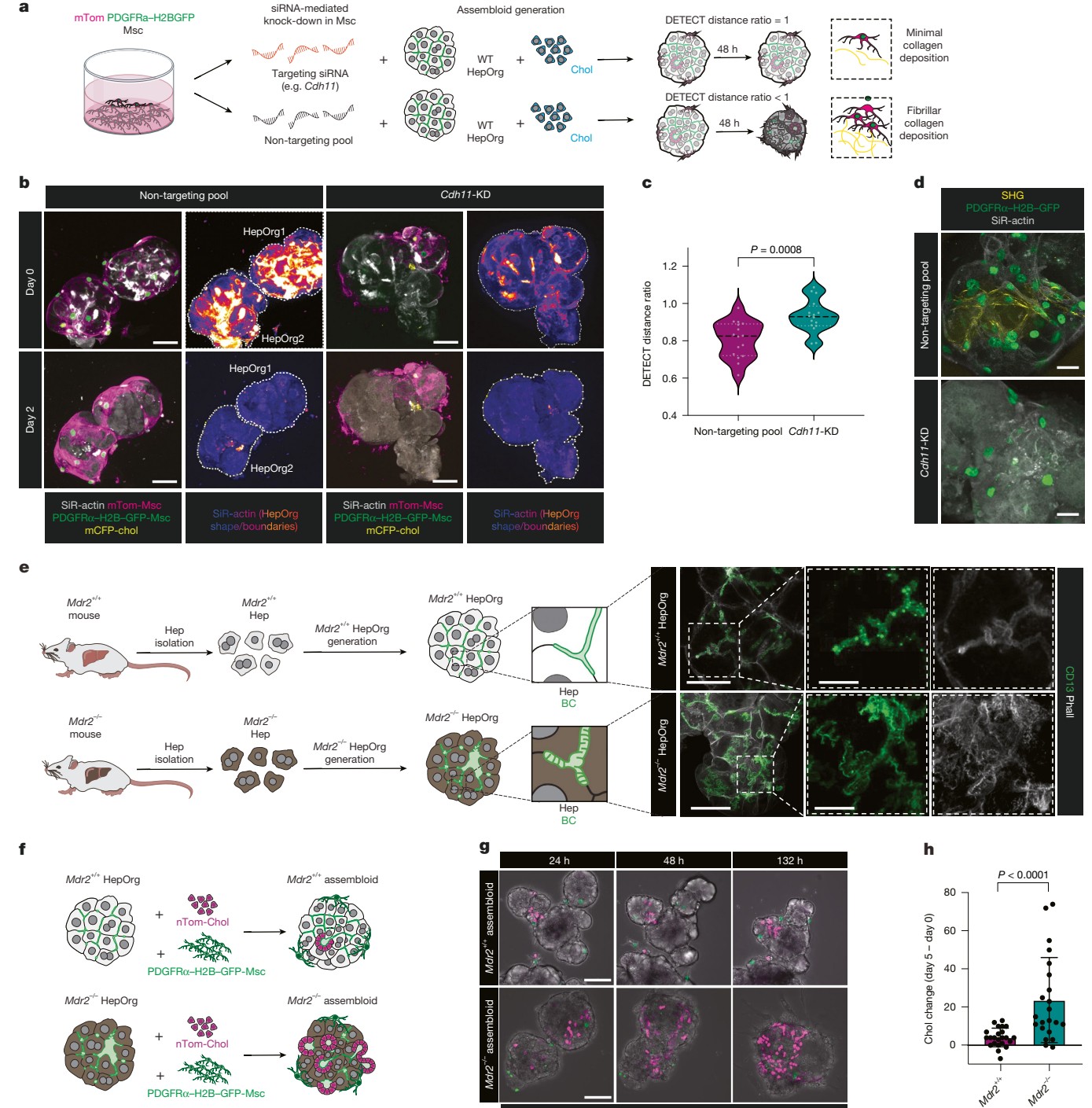

**Fig. 5 | Periportal assembloids as tools to investigate disease mechanisms.**
**a**–**d**, Short interfering RNA (siRNA)-mediated knockdown experiments in
assembloids. Mesenchyme cells (green nuclei and magenta membrane)
were transfected with targeting or non-targeting siRNAs before assembly.
**a**, Experimental design, including schematic of the DETECT distance ratio metric
($d_1/d_0$). **b**, Live imaging analysis of assembloids formed with mesenchyme cells
transfected with non-targeting (left) or *Cdh11* (right) siRNA. Cell boundaries
are indicated by SiR-actin. The white dotted line indicates segmentation of the
organoid border. Scale bars, 50 μm. **c**, Segmented assembloids from **b** were used
as input for the DETECT algorithm and to calculate the DETECT distance metric.
Violin plots show median and quartiles of the DETECT distance ratio for the
non-targeting and *Cdh11*-knockdown (KD) group. Mann–Whitney test, two-tailed.
Dots represent individual assembloids (*n* = 16 from 2 independent experiments).
**d**, SHG images of assembloids showing fibrillar collagen deposition in non-
targeting control and *Cdh11*-knockdown groups. PDGFRα–H2B–GFP marks

portal fibroblasts. *n* = 2 independent experiments. Scale bars, 25 μm. **e**, Left,
schematic of *Mdr2*[+/+] and *Mdr2*[−/−] HepOrgs. Right, immunofluorescence images
of HepOrgs derived from wild-type (top) or *Mdr2*[−/−] (bottom) livers show
dilated bile canaliculi on *Mdr2*[−/−] organoids. CD13 (green) marks bile canaliculi.
Phalloidin marks cell borders. *n* = 3 independent experiments with *n* = 3 biological
replicates. Scale bars: 20 μm (main images), 10 μm (zoom views). **f**–**h**, Chimeric
assembloids were formed by *Mdr2*[−/−] HepOrgs (bright field, grey), wild-type
cholangiocytes (nTom-Chol) and wild-type portal mesenchyme (PDGFRα–
H2B–GFP). **f**, Schematic of experimental design. **g**, Still images from live imaging
experiments (*n* = 2 independent experiments) of assembloids formed with
*Mdr2*[+/+] or *Mdr2*[−/−] HepOrgs. Scale bars, 50 μm. **h**, Change in cholangiocyte
numbers between day 0 and day 5 in assembloids formed with *Mdr2*[+/+] or *Mdr2*[−/−]
HepOrgs. Data are mean ± s.e.m. of 2 biological replicates from *n* = 2 independent
experiments with *n* = 23 (*Mdr2*[+/+]) and *n* = 22 (*Mdr2*[−/−]) assembloids. Dots represent
individual assembloids. Mann–Whitney test, two-tailed.

canaliculi bulkheads[8,46]. This is in stark contrast to previous models, which were cholestatic at baseline (presented bile canaliculi bulkheads)[46] and lacked the 3D network organization typical of transport networks[50].

The physiological bile canaliculi architecture generated by HepOrgs enabled us to reconstruct a physiological and functional connection between the bile canaliculi and bile duct. A limitation of this study was that it did not achieve full zonation as in the liver lobule—as expected, given that the organoids are too small in size to provide a full liver axis.

Paradoxically, although recapitulating native tissue architecture is essential, too much complexity could obstruct investigations of the precise role of specific niche cell types in the physiology or pathophysiology of a tissue without introducing confounding factors. Our system represents a modular and tractable in vitro tool for investigating the dynamics of cholestatic biliary fibrosis and the contribution of different cell types. Notably, this is not possible in current epithelial-only organoid models or in organoids derived from induced pluripotent stem cells, in which the different populations co-develop from a single clone. By targeting CDH11 or ITGβ1 specifically in mesenchyme, we provide proof of the concept that assembloid cells can be used as platform for mechanistic discovery and to investigate cell-autonomous mechanisms in liver disease. How this affects the exact intracellular molecular effectors in mesenchyme–ECM or mesenchyme–mesenchyme interactions that regulate hepatocyte cell death and duct cell expansion remains unknown.

In conclusion, our periportal assembloid model recapitulates architecture and cell–cell interactions of the liver at the meso- and cellular scale and provides an in vitro liver organoid system for the study of bile canaliculi formation, bile drainage and cell-autonomous or cell-specific contributions of hepatocytes, cholangiocytes and portal mesenchyme to cholestatic liver disease.

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

# Methods

## Mouse models

Mouse experiments were performed in accordance with the German animal welfare legislation and in strict pathogen-free conditions in the animal facility of the MPI-CBG. Protocols were approved by the Institutional Animal Welfare Officer (Tierschutzbeauftragter), and all necessary licenses were obtained from the regional Ethical Commission for Animal Experimentation of Dresden, Germany (Tierversuchskommission, Landesdirektion Dresden, number DD24-5131/346/3; TVT08/2023; TVV42/2021; TVV49/2021). The laboratory animal housing of the MPI-CBG is exclusively barrier housing. All mice are kept in individually ventilated cages under a 12 h:12 h light:dark cycle. The animal room temperature is maintained between 20 and 24 °C and the relative humidity is 55 ± 10%. Both are subject to constant monitoring. Sterile food and water were given ad libitum. Healthy adult mice (8–25 weeks of age) of both sexes were used for experiments. For MDR2 experiments, mice were used at 8 weeks of age. Wild-type C57/Bl6 mice, *Rosa26*-mTmG, *Rosa26*-nTnG, *Pdgfra*[H2B-GFP], *Pdgfra*[H2B-GFP] × *Rosa26*-mTmG, *Prom1*[creERT2] × *R26-LSL-ZsGreen*, *Itgb1*[fl/fl] × *R26-LSL-ZsGreen* or *Mdr2*-knockout mice were used for experiments. *Rosa26*-mTmG [Gt(ROSA)26Sortm4(ACTB-tdTomato, -EGFP)Luo/J] and *Rosa26*-nTnG [B6;129S6-Gt(ROSA)26Sortm1(CAG-tdTomato*,-EGFP*)Ees/J]] were obtained from the Jackson Laboratory (JAX). The *Pdgfra*[H2B-GFP] ([B6.129S4-Pdgfratm11(EGFP)Sor/J]) mice were described previously[51] and obtained from M. Zernicka-Goetz. The *Pdgfra*[H2B-GFP] × *Rosa26*-mTmG was generated by crossing the *Pdgfra*[H2B-GFP] mice with *Rosa26*-mTmG were obtained and described above. The *Mdr2*-knockout line [FVB.129P2-Abcb4tm1Bor/J] was described before[8]. The *R26-LSL-ZsGreen* B6.Cg-Gt(ROSA)26Sortm6(CAG-ZsGreen1)Hze/J) mice were obtained from JAX. *Itgb1*[fl/fl] (B6;129-Itgb1tm1Efu/J) was described previously[52], and obtained from M. Zernicka-Goetz. *Itgb1*[fl/fl] × *R26-ZsGreen* was generated by crossing *Itgb1*[fl/fl] mice with *R26-LSL-ZsGreen* mice. The *Prom1*[creERT2] × *R26-LSL-ZsGreen* mice were generated by crossing *Prom1*[creERT2] (B6N;129S-Prom1tm1(cre/ERT2)Gilb/J) mice[53], obtained from R. Gilbertson, with *R26-LSL-ZsGreen* mice. Confetti mouse (Gt(ROSA)26Sortm1(CAG-Brainbow2.1)Cle/J)) has been described[54] and was recombined in vitro to obtain mCFP cholangiocyte organoids. Mice were bred onto a C57/B6 background.

## HepOrg culture

Primary hepatocytes were isolated from mice following either euthanasia by cervical dislocation, or anaesthesia by intraperitoneal injection of 90 mg kg⁻¹ bodyweight ketamine and 10 mg kg⁻¹ Rompun (xylazine) as previously described[55]. When isolating hepatocytes from *Mdr2*[−/−] fibrotic livers, collagenase perfusion was longer compared to standard protocol, between 25–30 min. After isolation, hepatocytes were counted, and 5,000 viable hepatocyte cells were resuspended in 5 µl of suspension buffer and mixed with Matrigel to a total of 25 µl and seeded forming a Matrigel dome as for 25 µl Matrigel per well of a 48-well plate. Following Matrigel solidification (~15 min), 250 µl of medium per 48-well plate was overlaid. Hepatocytes were cultured in an adapted medium as described[25]: AdDMEM/F12 (ThermoFisher, 12634010) medium containing 1% HEPES (ThermoFisher, 15630-056), 1% penicillin/streptomycin (ThermoFisher, 15140-122), Glutamax (ThermoFisher, 35050-068), 1× B27 (Invitrogen, 12587010) and 1.25 mM *N*-acetylcysteine (Merck/Sigma, A9165) –referred to as basal medium, which was further supplemented with 10 nM gastrin (Merck/Sigma, G9145), 50 ng ml⁻¹ mEGF (ThermoFisher, PMG8043), 15% RSPO1 conditioned medium (homemade), 100 ng ml⁻¹ FGF10 (Peprotech, 100-2), 50 ng ml⁻¹ FGF7 (Peprotech, 100-19-50), 10 mM nicotinamide (Merck/Sigma, N0636), 25 ng ml⁻¹ HGF (Peprotech, 100-39), 3 µM CHIR9902 (Tocris, 4423), 1 µM A83-01 (Tocris, 2939) and 10 µM ROCK inhibitor (Y-27632, Merck/Sigma, Y0503), supplemented with 30% WNT3a conditioned medium (Wnt-CM) (homemade) –referred to as hepatocyte medium + 30% Wnt-CM (HM-Wnt), or Wnt surrogate (HM-WntS) (WNT Surrogate: Live Science Incubator, N001) or 3% foetal bovine serum (HM-FBS) (FBS, Merck/Sigma, F7524), or in medium described in Hu et al.[15] with modified FGF10 concentration (100 ng ml⁻¹) –or in medium described in Peng et al.[16], containing TNF (Peprotech, 300-01 A) with the modification that the basal medium was Advanced DMEM/F12 and EGF and HGF concentrations were 25 ng ml⁻¹. HepOrgs were cultured at 37 °C and 5% CO₂. The medium was changed 2–3 times per week. To enrich for bubbly/grape-like HepOrg structures, the initial cell preparation was enriched with cell clusters, or alternatively, ball HepOrgs were removed by hand-picking or by centrifugation at passage 1 and 2. To enrich for ball-like structures, cells were seeded as single cells at lower density (1,000 cells) to avoid fusion and promote single cells growing out in the aberrant morphology. Sometimes unwanted cholangiocyte organoids or mesenchyme was also observed in the cultures until passage 2, and also removed by hand-picking from the hepatocyte cultures. HepOrg cultures were regularly tested for the absence of mycoplasma using MycoAlert Mycoplasma Detection Kit (Lonza LT07-118).

For co-culture media test experiments, HepOrg grown in HM-Wnt up to passage 2 (P2) were cultured in one of the following media for 6–14 days, with a medium change every 2–3 days. Organoids were transferred to indicated medium from HM-Wnt medium, and then collected for analysis (7 days, 14 days) or immunofluorescence staining (6–7 days). Media regimes used were: (1) basal medium supplemented with Wnt-CM (30%) and 10 µM ROCK inhibitor (Y-27632, Merck/Sigma, Y0503) (MM)[20]; (2) and our previously published medium that supports hepatocyte differentiation from cholangiocytes with addition of small-molecule Wnt inhibitors[21], differentiation medium (DM) composed of basal medium supplemented with 10 nM gastrin (Merck/Sigma, G9145), 50 ng ml⁻¹ mEGF (ThermoFisher, PMG8043), 100 ng ml⁻¹ FGF19 (Peprotech, 100-32-25), 25 ng ml⁻¹ HGF (Peprotech, 100-39), 3 µM CHIR9902 (Tocris, 4423), 0.5 µM A83-01 (Tocris, 2939), 25 ng ml⁻¹ hBMP7 (Peprotech, 120-03), 10 µM DAPT (Merck/Sigma, D5942-5MG), 3 µM IWP2 (Merck/Sigma, I0536-5mg), 25 µM iCRT3 (Merck/Sigma, SML0211-5MG) and 3 µM dexamethasone (Tocris, 4489).

For organoid formation efficiency, primary hepatocytes were isolated as described above. To prevent organoids from fusing, 1,000 viable hepatocytes (viability >80%) were plated in 25 µl Matrigel (BD Bioscience; 356231) droplet and cultured as described above. After 9 days, organoid numbers were counted and results expressed as a percentage relative to the initial seeding cell numbers.

To determine cholangiocyte contamination, *Prom1*[creERT2] × *R26-LSL-ZsGreen* transgenic mouse were used to induce specific cholangiocyte labelling with ZsGreen protein in the adult mouse prior to isolation of hepatocytes. Following tamoxifen injection, cholangiocytes were labelled by fluorescent protein ZsGreen. This process has been shown to occur with close to 80% efficiency[53,56]. We then allowed for a tamoxifen wash-out period of 14 days, to exclude any effects relating to the injection and tamoxifen in murine system, and followed by standard hepatocyte isolation, as described in methods. For the analysis of cholangiocytes from hepatocyte isolation and HepOrg culture, primary hepatocytes and other liver cells were isolated from mice as described above. The cells were then strained with 100 µM strainer, washed one time with AdDMEM/F12 (ThermoFisher, 12634010) medium containing 1% HEPES (ThermoFisher, 15630-056), 1% penicillin/streptomycin (ThermoFisher, 15140-122), Glutamax (ThermoFisher, 35050-068), spun 5 min at 100*g* and stained 30 min with EpCAM antibody conjugated to APC (CD326 (EpCAM) Monoclonal Antibody (G8.8), APC, eBioscience, ThermoFisher, 17-5791-80). Before sorting, cells were washed one time with above medium, spun 5 min at 100*g* and resuspended in the same medium for sorting. In case of cholangiocytes from HepOrg culture, all cells from the culture were collected in above media, spun 5 min at 200*g* and dissociated by incubation with TrypLE Express (ThermoFisher, 12605010) for 5 min at 37 °C, before being strained and stained as outlined above.

## Liver ductal and mesenchymal cell isolation and sorting

Mouse livers were collected and enzymatically digested as described[14,57,58]. In brief, minced livers were incubated in a solution containing 0.0125% collagenase (Merck/Sigma, C9407), 0.0125% dispase II (ThermoFisher, 17105-041) and 1% foetal bovine serum (FBS) (Merck/Sigma, F7524) in DMEM/Glutamax (ThermoFisher, 31966-021) supplemented with 1% HEPES (ThermoFisher, 15630-056) and 1% penicillin/streptomycin (ThermoFisher, 15140-122) and 0.1 mg ml$^{-1}$ of DNAase (Merck/Sigma, DN25) in a shaker at 37 °C and 150 rpm for 1.5–3 h. The biliary tree fragments and associated stroma were then dissociated into single cells with TrypLE diluted to 5× (Gibco, A12177-01). Following dissociation, cells were isolated by fluorescence-activated cell sorting (FACS) using the described approach[20,58]. Single cells were incubated with fluorophore-conjugated antibodies for 30 min and FACS-sorted using BD FACS Aria (BD Biosciences) or SH800S (SONY) cell sorters. Cells were sequentially gated based on size and granularity (forward scatter (FSC) versus side scatter (SSC)) and singlets (FSC area versus FSC height); after which ductal cells were selected based on EPCAM positivity and negative exclusion of the haematopoietic/endothelial markers CD31, CD45 and CD11b. The mesenchyme was sorted as double positive PDGFRα–GFP$^+$SCA1$^+$ cells gated from the EpCAM$^-$CD31$^-$CD45$^-$CD11b$^-$ fraction. Following FACS isolation, ductal cells and portal mesenchymal cells were used for ductal organoid and mesenchymal cell culture, respectively. Cholangiocyte organoid and mesenchymal cell cultures were regularly tested for the absence of mycoplasma using MycoAlert Mycoplasma Detection Kit (Lonza LT07-118).

## Mesenchymal cell culture

Mesenchymal cells were cultured in the basal medium described above supplemented with Wnt-CM (30%) and 10 μM ROCK inhibitor (Y-27632, Merck/Sigma, Y0503), as described[20]. Cells were passaged at 1:3 and 1:2 ratios, through enzymatic digestion using TrypLE Express (ThermoFisher, 12605010) for 5 min at 37 °C. Mesenchymal cells were cultured at 37 °C and 5% CO$_2$. For 3D culture in Matrigel, mesenchyme cells were seeded in Matrigel at density 2,500 cells per 25 μl Matrigel in a 48-well plate as the sparse condition, overlaid with medium. Alternatively, for the confluent/aggregated 3D condition, 2,500 mesenchyme cells were placed in a well of a low-attachment 24-well plate in 0.5 ml of medium supplemented with 1× methylcellulose (Sigma, M6385) and the plate was transferred to a rocker-shaker (Biosan, MR-1), incubated at 10 rpm for 18–24 h at 37 °C and 5% CO$_2$. Mesenchyme aggregates were collected into 1.5 ml Eppendorf tube, spun down for 5 min at 200$g$ and seeded in Matrigel as described above and overlaid with medium.

## Hepatocyte 2D sandwich culture

Primary hepatocytes were isolated from mouse livers via collagenase perfusion as described above. Cells were plated onto collagen (0.9 mg ml$^{-1}$) coated 24-well plates at 200,000 cells per well in Williams E medium (PAN Biotech), substituted with 10% FBS, 100 nM dexamethasone and penicillin/streptomycin and maintained at 37 °C in an atmosphere with 5% CO$_2$. After 3–4 h of attachment, cultures were washed with phosphate buffer saline (PBS) and coated with a second layer of collagen (0.6 mg ml$^{-1}$) to obtain a sandwich culture as previously described[59]. Medium was changed every day. For immunofluorescence analysis, cells were seeded in collagen-coated glass cover slips and cultured as above. When required, cells were fixed in 4% paraformaldehyde at room temperature for 30 min, washed twice with PBS, permeabilized for one hour with 0.1% Triton X-100, washed and blocked in 10% horse serum for 2 h. Holes were applied to the top layer collagen using fine aspiration to ensure better antibody penetration. Cells were incubated with primary antibodies at room temperature overnight, washed for 2 h in wash buffer (300 mM NaCl, 0.1% Tween, 10 nM Tris/HCl) with frequent (8–12×) exchanges of buffer. Secondary antibodies and dyes were incubated for 5 h at 37 °C in a humidified chamber. Thereafter

cells were washed extensively and mounted onto glass slides using 0.1 g ml$^{-1}$ Mowiol (Calbiochem).

## *Itgb1*-knockout induction in mesenchyme cell culture

To generate *Itgb1*-knockout mesenchyme, the mesenchymal cells were isolated as specified above from *Itgb1-fl/fl (Itgb1$^{tm1Ref}$) R26-ZsGreen* transgenic mice, and sorted as positive SCA1$^+$ cells gated from the EpCAM$^-$CD31$^-$CD45$^-$CD11b$^-$ fraction. The cells were then expanded as specified above in 2D culture until passage 1, when the *Itgb1$^{fl/fl}$ × R26-ZsGreen* allele was recombined by transducing the cells using a adenovirus (Ad5-CMV-Cre; University of Iowa) at a multiplicity of infection of 10. Medium was changed up to 24 h after infection. Following culture and expansion, the cells were then FACS-sorted gated on GFP$^+$ to enrich for the recombined cells, and plated again to passage 2. Cells were used for experiments at passage 2.

## Cholangiocyte organoid culture

Cholangiocyte organoids were generated and cultured as described[14]. In brief, sorted EpCAM$^+$ cholangiocytes were embedded in 50 μl Matrigel per well of a 24-well plate and cultured in EM medium (AdDMEM/F12 medium (ThermoFisher, 12634010) containing 1% HEPES, 1% penicillin/streptomycin, Glutamax, 1× B27 and 1.25 mM $N$-acetylcysteine supplemented with 10 nM gastrin (Merck/Sigma, G9145), 50 ng ml$^{-1}$ mEGF (ThermoFisher, PMG8043), 5% RSPO1 conditioned medium (home made), 100 ng ml$^{-1}$ FGF10 (Peprotech, 100-26), 10 mM nicotinamide (Merck/Sigma, N0636) and 50 ng ml$^{-1}$ HGF (Peprotech, 100-39)) supplemented with 30% WNT3a conditioned medium (Wnt-CM) (homemade), 25 ng ml$^{-1}$ Noggin (Peprotech, 120-10C) and 10 μM ROCK inhibitor (Y-27632, Merck/Sigma, Y0503). Then, after the first 3 days in culture, cells were switched to EM medium only (without Wnt-CM, ROCK inhibitor and Noggin). The grown cholangiocyte organoids were passaged at a 1:3 ratio once a week by mechanical dissociation, re-embedded in fresh Matrigel and cultured in EM as described in[57]. Cholangiocyte organoids were cultured at 37 °C and 5% CO$_2$.

## Periportal assembloid generation

For periportal assembloids generation, portal mesenchyme, cholangiocyte cells from cholangiocyte organoids and HepOrgs were aggregated in 24-well plates in rocking platform or in 24-well AggreWell plates, and mixed at different ratios according to the purpose. To define the numbers of the different cell types needed, we took advantage of our previous studies on the homeostatic proportion of hepatocytes and cholangiocytes (97% versus 3%)[25], as well as of portal mesenchymal:ductal cells (3:10 ratio[20]). The use of cells labelled with different endogenous fluorescent proteins ensured that the structures containing three cell types had originated following the aggregation of the different cells. For healthy homeostasis ratios: 250 mesenchyme cells, 1,000 cholangiocyte cells and 10 HepOrgs were used per well. For non-physiological (fibrotic-like) ratio 2,500 mesenchyme cells, 1,000 cholangiocyte cells and 10 HepOrgs were used.

The cells were prepared as follows: cholangiocyte organoids grown in EM (passage 2–9) were dissociated to single cells by collecting them from Matrigel using cold AdDMEM/F12 (ThermoFisher, 12634010) containing 1% HEPES (ThermoFisher, 15630-056), 1% penicillin/streptomycin (ThermoFisher, 15140-122) and 1% Glutamax (ThermoFisher, 35050-068) and dissociating them into single cells using TrypLE 1× (ThermoFisher, 12605010) for 7 min at 37 °C and filtered through 40-μm cell strainers. In parallel, 80–90% confluent mesenchyme cultures (fresh from sorting or passage 0–2, as specified in legend) grown as specified above, were washed with PBS and dissociated to single cells by incubating with TrypLE 1× for 5 min at 37 °C. Both single cells suspensions were spun at 300$g$ for 5 min and the cell concentration was determined by manual counting in haemocytometer. In parallel, HepOrgs were grown in HM-Wnt for at least 1 or 2 passages and cultured (or not) with MM for 48 h before assembling (as specified in

the figure legend). Then, HepOrgs were removed from Matrigel using 2 washes with cold AdDMEM/F12 (ThermoFisher, 12634010) supplemented with 1% HEPES, 1% penicillin/streptomycin and 1% Glutamax, and incubated with cold Cell Recovery Solution (Corning, 354253) or cold PBS for 10 min on ice. Then, bubbly/grape-like shape organoids were hand-picked on a stereoscope and placed in the aggregation plate as described below. The efficiency of aggregation was increased when using HepOrgs that had been pre-conditioned for 48 h with MM (Extended Data Fig. 7c).

For rocking platform aggregation, cells were mixed together in a well of a low-attachment 24-well plate, in 0.5 ml MM, which in some cases, was supplemented with 1× methylcellulose (Sigma, M6385) to facilitate aggregation. Then, the plate was transferred to a rocker-shaker (Biosan, MR-1) and incubated at 10 rpm for 14–24 h at 37 °C and 5% $CO_2$. Following aggregation, assembloids were collected into 1.5 ml Eppendorf tube, spun down for 5 min at 200$g$ and seeded in Matrigel as described below. For AggreWell aggregation, cells and organoids were mixed in specified ratios in 1.5 ml MM in AggreWell plates (AggreWell800, Stem Cell Technologies, 34811, pre-treated as recommended by the manufacturer), spun down 5 min at 100$g$ and incubated for 14–24 h at 37 °C and 5% $CO_2$. To collect the structures from AggreWell wells, the solution was disrupted by pipetting with 1 ml pipette, and all the solution was collected on a 40-µm strainer. The strainer was then placed upside down over a 6-well plate, washed with AdDMEM/F12 (ThermoFisher, 12634010) supplemented with HEPES, penicillin/streptomycin and Glutamax, and all suspension was collected and spun down for 5 min at 200$g$. The supernatant was removed and the structures seeded in Matrigel. Structures from both aggregation types were seeded either in 25 µl Matrigel dome in a pre-warmed 48-well plate, or in a Matrigel layer (20 µl Matrigel in a 96-well plate centrifuged in a cold centrifuge 200$g$ for 5 min, which is then overlaid with structures in another 20 µl Matrigel on top), solidified 10 min in 37 °C incubator and overlaid with further 150–200 µl MM. The structures were grown for 7–14 days with media changes to fresh MM every 2–3 days. Unless specified in the figure legend, we opted for shaker aggregation as, in AggreWell, some assembloids tended to acquire a ball-shape structure compared to rocking platform/shaker and in addition, it did not require any specialized equipment.

Generally, for periportal assembloid generation, HepOrgs derived from wild-type mouse livers, cholangiocyte cells from cholangiocyte organoids from nuc-Tdtom mouse livers and portal mesenchymal cells from PDGFRα–H2B–GFP⁺SCA1⁺ cells were assembled either at a normal physiological ratio (10 HepOrgs: 1,000 cholagiocyte cells: 250 mesenchyme cells) or at a ratio with 10× excess mesenchyme cells, unless stated otherwise in the figure for other fluorescent colour combinations.

A step-by-step protocol for mouse periportal assembloid generation can be found on protocols.io[60].

**Blocking antibody treatment of assembloids**
To investigate whether assembloids could be used to study cellular interactions in fibrotic-like phenotype, blocking antibodies were added to the assembloid culture at the time of assembly (rocking platform). The list of compounds and antibodies is presented in Supplementary Table 1 with relevant concentrations tested.

**Assembloid experiments with siRNA-treated mesenchyme cells**
Passage 2 portal fibroblasts derived from PDGFRα-H2B-GFP × *Rosa26*-mTmG mouse were detached using 1:1 mixture of 10X TrypLE and Accutase (StemPRO Accutase, ThermoFisher A11105015). Single cell suspensions were spun at 300$g$ for 5 min. Cells were seeded at either 5,000 cells per well or 10,000 cells per well to be subsequently used for assembloid formation or RT-qPCR analysis to determine knockdown efficiency. Then, portal fibroblasts were transfected with 20 pmol of a pool of 4 ON-Targetplus siRNA targeting specific genes (Supplementary

Table 1) for 3 h according to manufacturer instructions. After 16–24 h transfected cells were collected using pre-warmed digestion mix of Accutase and TrypLE select Enzyme (10X) at 1:1 ratio, and incubated 5 min at 37 °C. Single cells suspensions were spun at 300$g$ and resuspended in 100 µl MM. mCFP cholangiocyte organoids were expanded in our standard cholangiocyte medium and processed to single cells as described above. To make fibrotic-like assembloids, 2,500 mesenchyme cells, 1,000 single-cell cholangiocytes and 10 HepOrgs were combined and incubated 14–24 h on a rocking platform at 37 °C as described above. Formed assembloids were hand-picked and seeded on 10 µl Matrigel pre-coated in a 96-well plate (Greiner, 655090) and incubated 15 minutes at 37 °C, following by overlay with 10 µl of Matrigel. After 30 minutes incubation at 37 °C, 200 µl MM supplemented with 0.5 µM of SiR-actin (Spirochrome, SC001) was added. Imaging started 2 h later and was performed every 24 h for 2 days on CellVoyager CV7000 spinning disc microscope with 20× objective, with $z$-step size 1 µm. Lasers with 405, 488, 561 and 647 nm were used to detect the mCFP, nGFP, TdTomato and SiR-actin signal, respectively, on a sCMOS camera. To determine knockdown efficiency, total RNA was extracted from cells after 16 h of siRNA incubation as described above using the Arcturus PicoPure RNA Isolation Kit (Applied Biosystems, 12204-01) according to the manufacturer's protocol; including a 15-min digestion step with DNAse.

To quantify the number of mesenchyme cells per assembloid in the siRNA experiments, the PDGFRα–H2B–GFP signal was segmented as follows: First a max projection was used to reduce the 3D stack to a 2D image, then a log transformation was applied to compress the high dynamic range of the signal. The image was then down-sampled by a factor of four. Subsequently, the StarDist[61] algorithm, using the pre-trained 2D_versatile_fluo model, was used for segmentation. Although this method effectively segmented mesenchyme nuclei, it also included other structures, such as hepatocyte nuclei, which exhibited lower intensity. To ensure accurate counting, only segments with an average intensity above a manually selected threshold were included.

**HepOrg formation and assembloid morphology imaging**
Freshly isolated hepatocytes were seeded in a 96-well plate (Greiner, 655090) in 7 µl Matrigel (BD Bioscience; 356231) droplet in the concentrations of 40–800 cells per µl, of Matrigel, supplemented with HM-Wnt. For imaging, 2 × 2 tile $z$-stack images (10 µm $z$-step size) of each well were acquired every 24 h for 13 days on a CellVoyager CV7000 spinning disc microscope with 10× objective (Yokogawa). Membranes of HepOrgs were detected by endogenous membrane-tdTomato signal and used to determine ball or bubbly/grape-like shape of the structures. Ball structures are defined with round edges resembling a sphere, while bubbly/grape-like shape is determined by the appearance of irregular surfaces and surface invaginations. Endogenous tdTomato signal was excited with 561 nm laser. Maximum-intensity images were made using 'Macro CV7000 $z$-projection'. For each well, time-lapse maximum-intensity projection videos were made using either stack-list files that contained information about every time point, $z$-stacks and channels (Supplementary Methods 1, appendix A, Method_section_FIJI_stack_files_KNIME). Stack lists were converted into time-lapse videos using the Fiji macro Macro_make_HyperStack_Zmax_005.ijm. For quantification, 30 randomly assigned ball or bubbly/grape-like HepOrgs were traced back to the seeding time point (day 0), and the number of cells of origin (one cell or cell cluster) was quantified.

For DETECT analysis in HepOrgs or in assembloids (details of the analysis below), maximum-intensity projections from the SiR-actin channel or bright-field channel for each time point were used to make XY coordinates of the assembloids outlines. In short, outlines of the assembloids were created in Fiji using the polygon selection tool in the clockwise orientation. $xy$ coordinates were saved using the 'save xy coordinates' command as a .txt file (as described in Supplementary Methods 1, appendix A).

### Organoid whole-mount staining and imaging

For in vitro staining, organoids and assembloids were first extracted from Matrigel with ice-cold cell recovery solution (Corning, 354253) or PBS, and then fixed with 4% paraformaldehyde (PFA) in PBS for 30 min on ice. Blocking and permeabilization was performed for 1 h at room temperature in PBS containing 0.2% Triton X-100 and 2% BSA. The samples were incubated with primary antibodies overnight at 4 °C in blocking solution. Following 3 washes with PBS, the samples were incubated overnight at 4 °C or for 4–8 h at room temperature with secondary antibodies, phalloidin and DAPI in PBS. The samples were washed 3 times with PBS and subsequently cleared using fructose-glycerol clearing solution (25 ml glycerol, 5.3 ml dH$_2$O and 22.5 g fructose–60% glycerol and 2.5 M fructose)[62]. Alternatively, organoids and assembloids were permeabilized with 0.5–1% Triton X-100 in 1× PBS for 1 h. After, they were incubated in primary antibody in TxBuffer (0.2% gelatin 2% gelatin, 300 mM NaCl and 0.3% Triton X-100 in 1× PBS) incubated at 4 °C overnight. Following 3 washes with PBS, the samples were incubated for 2 h at room temperature with secondary antibodies in TxBuffer. The samples were washed three times with PBS and left in PBS until imaging or cleared as described above. The full list of primary and secondary antibodies used is specified in Supplementary Table 1.

### Thin section staining

For thin tissue sections (8–12 μm) and staining, livers were fixed for 2 h or overnight in 10% formalin with rolling at 4 °C and tissues incubated with 15% sucrose for 1 h, and then 30% sucrose PBS for 24-48 h, embedded into cryomolds (Sakura, 4566) with OCT compound (VWR, 361603E) and snap-frozen. Tissue blocks were cryo-sectioned on ThermoScientific CryoStar NX70 cryostat. Sections were blocked in PBS with 2% DS and 1% BSA for 2 h at room temperature, incubated with primary antibodies in 1/100-diluted blocking buffer overnight at 4 °C and with secondary antibodies + DAPI for 2 h at room temperature in 0.05% BSA PBS. Sections were mounted in Vectashield. The list of used antibodies is available in Supplementary Table 1

### Thick tissue section staining

For thick tissue sections, mice were perfused at 3.7 ml min$^{-1}$ for 10–15 min with 4% paraformaldehyde, 0.1% Tween-20 in PBS. Livers were cut in smaller pieces and post-fixed in the same solution for 24 h on a rotator at 4 °C. After, liver pieces were washed in PBS to remove fixative. For storage, liver pieces were kept in PBS at 4 C. For sectioning, livers were mounted in 4% low-melting agarose in PBS and cut into 100 μm-thick sections on a vibratome (Leica VT1200S). For deep tissue imaging, tissue sections were permeabilized with 0.5% Triton X-100 in PBS for 1 h at room temperature. The primary antibodies were diluted in Tx buffer (0.2% gelatin, 300 mM NaCl, and 0.3% Triton X-100 in PBS) and incubated for 48 h at room temperature. After washing 5× 15 min with 0.3% Triton X-100 in PBS, the sections were incubated with secondary antibodies, DAPI (1 mg ml$^{-1}$; 1:1,000) and phalloidin–Alexa Fluor 488 or 647 (Thermo Fisher Scientific; A12379 or A22287; 1:250) for another 48 h. The list of used antibodies is available in Supplementary Table 1. After washing 5× 15 min with 0.3% Triton X-100 in PBS and 3× 1 min with PBS, the optical clearing started by incubating the slices in 25% fructose for 4 h, continued in 50% fructose for 4 h, 75% fructose overnight, 100% fructose (100% wt/vol fructose, 0.5% 1-thioglycerol, and 0.1 M phosphate buffer, pH 7.5) for 6 h, and finally overnight in SeeDB solution (80.2% wt/wt fructose, 0.5% 1-thioglycerol and 0.1 M phosphate buffer)[63]. The samples were mounted and imaged in SeeDB.

### RNAscope

For RNAscope, HepOrgs were embedded into cryomolds (Sakura, 4566) with OCT compound (VWR, 361603E) and snap-frozen after fixing in 4% PFA/PBS for 30 min on ice. Tissue blocks were cryo-sectioned on ThermoScientific CryoStar NX70 cryostat (12 μm sections). The sections were stained using RNAscope Fluorescent Multiplex and RNAscope Multiplex Fluorescent V2 (Advanced Cell Diagnostics, 323100) according to the manufacturer's instructions. Probes for the target genes are in Supplementary Table 1.

### Imaging of HepOrgs, assembloids and liver tissue

For thin tissue sections (8–12 μm), RNA scope sections (12 μm) and whole-mount imaging of assembloids, images were acquired using a single photon point-scanning confocal system (Zeiss LSM 880 Inverted or Upright), with a Quasar detector with 32 spectral detection channels in the detection channels in a gallium arsenide phosphide (GaAsP) detector with 2 photomultiplier tubes and transmitted light detector. Images were acquired using a Zeiss 20× (0.8 NA) air objective or Zeiss LD LCI Plan-Apochromat 40×/1.2 DIC Imm Corr M27 immersion corrected objective. Fluorophores were excited with 405, 458, 488, 532, 561, 594, and 633 nm lasers. Images were processed using ZEN software (Zeiss), or ImageJ/Fiji.

For 3D reconstruction of bile canaliculi in organoids and assembloids, images of optically cleared organoids and assembloids were acquired with an inverted multiphoton laser-scanning microscope (Zeiss LSM 780 NLO) or a single photon point-scanning confocal system (Zeiss LSM 880 Inverted) with a Quasar detector with 32 spectral detection channels in the GaAsP detector with 2 photomultiplier tubes (Zeiss LSM 880). Images were acquired using a Zeiss LD LCI Plan-Apochromat 40×/1.2 DIC Imm Corr M27 immersion corrected objective, or a Zeiss LD LCI Plan-Apochromat 63×/1.2 DIC Imm Corr M27, with a voxel size 0.3 μm. Fluorophores were excited with 405, 488, 561, 594, and 633 nm laser lines and detected with GaAsp detectors.

Optically cleared 100-μm liver sections were imaged with an upright multiphoton laser-scanning microscope (Zeiss LSM 780 NLO) equipped with two photomultiplier tubes and one 32-channel GaAsP detector for spectral detection in the scanhead, four GaAsP non-descanned detectors for multiphoton detection, three transmitted light detectors. Liver slices were imaged twice, at low (Zeiss Plan Apo 10×/0.45 NA Air) and high resolution (63×/1.3, or 40×/1.2 Zeiss LD LCI Plan-Apochromat DIC immersion corrected objective; 0.15 or 0.3 μm voxel size), respectively. Low-resolution overviews of the large surface of liver sections were created and used to find the central and portal vein regions, and image bile canaliculi of a dedicated region. Selected regions (~300 μm × 300 μm × 100 μm; $x$, $y$, $z$) were then acquired at high resolution (0.3 μm voxel size).

For 3D visualization of bile canaliculi, bile ducts and hepatocytes, high-resolution images were processed and segmented, based on CD13, PCK and phalloidin staining, respectively, using Motion Tracking software (http://motiontracking.mpi-cbg.de) as described[31].

### High-resolution imaging with Airyscan technology

For imaging detail (for example, bile canaliculi, polarity) of liver tissue, HepOrg, and assembloids with high-resolution, Airyscan images were acquired on an inverted single photon point-scanning confocal system (Zeiss Celldiscoverer 7 with LSM 900 and Airyscan 2) using a Plan-APOCHROMAT 20×/0.95 Autocorr, Air (Zeiss), with a 1× Tubelens, and a voxel size 0.082 × 0.082 × 0.340 μm, with an image size of 179.87 × 179.87 μm. Fluorophores were excited with 405, 488, 561 and 640 nm (T10/R90) laser lines and detected with GaAsP-photomultiplier tube detectors.

### Image analysis

For quantification of organoid and assembloid morphology, nuclei and cells, custom-made pipelines in Arivis 4D software (Zeiss) were used. For organoid morphology, segmentation was based on membrane staining of organoids. At first, a median denoising filter was applied followed by setting an intensity threshold for segmentation which was

determined manually depending on membrane signal intensity. Subsequently, the morphological operations 'inclusion filling' and 'close objects' were applied. At last, segmented objects were filtered by size to match the expected morphology of the organoids of interest. In case of incomplete organoid segmentation caused by weak fluorescence signal, missing segmentation was added manually.

For cell shape, segmentation was based on membrane staining. At first, a discrete gaussian denoising filter was applied followed by two top-hat filters with specific radii. Cells were segmented using minimal intensity threshold, split sensitivity and maximal area to match the expected morphology of the cells of interest in the fluorescence images. For nuclei, segmentation was based on nuclei staining or fluorescently tagged nuclei. At first, a discrete gaussian filter was applied followed by two top-hat filters with two specific radii. Further, median and particle enhancement filters were applied.

Nuclei were segmented using diameter, probability threshold, and split sensitivity to match the expected morphology of the nuclei in the fluorescence images. In case of incomplete nuclei segmentation caused by weak fluorescence signal, missing nuclei were added manually. To determine co-localized objects, an Arivis pipeline was applied in which segmented objects of interest were imported and objects inside or intersecting with the chosen compartment by at least one voxel were obtained. This method was applied to determine the number of nuclei in cells, the number of cells in organoids and the number of co-localized nuclei. All segmentation was checked manually and corrected manually, whenever needed.

For quantification of fibrillar collagen (SHG signal) in the assembloids, max projection images of the SHG channel were used to segment the area of the fibrillar collagen by using manually selected threshold in Fiji (min value 60, max value 110). To avoid unspecific signal outside of assembloids only SHG segmented area inside of the assembloid was used for quantification. Assembloid area was segmented using the free-hand selection tool to outline the boundaries of the assembloids in the SiR-actin channel. For quantification of fibrillar collagen signal in healthy and fibrotic-like assembloids without siRNA treatment, or in *Itgb1*-knockout mesenchymal cells and their matched control assembloids, or in mesenchyme single 3D and 2D culture, whole frame or organoid intensity of SHG channel was measured for the sample and control images, matched between experiments, using the 'Integrated density' measurement function in Fiji, and displayed as RawIntDen (the sum of the values of the pixels in the image or selection) or IntDen (the product of area and mean grey value).

## Quantification of organoid shape changes over time

To quantify differences between two phenotypes, we used topological data analysis. Specifically, Marsh et al.[28] devised a temporal shape descriptor algorithm, DETECT, which extends the smooth Euler characteristic transform[64]. The DETECT algorithm processes shapes by generating a curve that captures both the geometric and topological characteristics of the shape, along with its morphological changes over time, while also being rotationally invariant. For this study, the inputs to the algorithm are the boundaries of 2D projections of each organoid at each time point, generated as described above in 'Organoid formation and assembloid morphology', and we calculate the corresponding DETECT curve (Supplementary Methods 2, appendix B). The collection of curves at all time points measures how the morphology of a particular organoid changes over time. The new metric DETECT metric developed here (described in Supplementary Methods 2, appendix C) enables us to quantify the difference between two DETECT measurements, and demonstrates its stability to small perturbations to the shape. In particular, organoids that maintain similar shapes over time yield closely aligned DETECT curves when evaluated with this metric (see Supplementary Methods 2, with details in Supplementary Fig. 2 and Supplementary Table 7, appendix B and C).

## Bile canaliculi segmentation and network features extraction

The bile canaliculi were reconstructed from high-resolution (voxel size: $0.3 \times 0.3 \times 0.3$ μm) fluorescent image stacks (~50–80 μm depth) of optically cleared HepOrg. Segmentation was performed on CD13 (for bile canaliculi) and F-actin (cell boarders) staining with phalloidin. The analysis of bile canaliculi morphology and bile canaliculi network properties was performed using a Fiji[65] script. In brief, images were imported using Bioformats[66]. Both channels (for CD13 and Phalloidin) were smoothened with a median filter and segmented independently, CD13 with the default ImageJ method, and Phalloidin[67]. The overlap between the two segmented images was kept as the basis to identify apical patches. The apical patches were then completed by a series of inflations followed by the 'fill holes' function and a series of deflations[68]. At this point a size threshold was applied on the remaining potential lumina, and the lumina touching the borders of the images were removed[68]. The resulting segmented image is then used for local thickness measurement[69], skeletonization[70] and skeleton analysis[71]. The features were then exported as a .csv file and plotted in GraphPad Prism software package (v10.0.2 and v.10.0.0). The custom script described in this manuscript can be found at GitHub (https://git.mpi-cbg.de/huch_lab/assembloid-paper).

Immunofluorescence images from several conditions were used in this analysis: Hu et al.[15], Peng et al.[16], HM-Wnt HepOrg, MM HepOrg, liver tissue; hereafter, these are referred to as 'structure' and individual bile canaliculi networks are referred to as 'network'. We analysed the features of the largest network per structure (length, junction). A network is defined as an uninterrupted continuation of bile canaliculi branches, which are interconnected (have different amount of triple and quadruple junction points). Please refer to Extended Data Fig. 1c,d for a schematic explanation of networks, branches and junctions. We analysed the features (total length = sum of all branches), and number of junctions, triple or quadruple or combined (total number) of the bile canaliculi for each of the structures. We used number of junctions as a proxy of 'connectivity'; that is, a network with more junctions is more interconnected than a network with fewer junctions. To compare between structures of different conditions, we plotted these values in dot plots, with one dot representing one structure. To plot the frequency distribution of bile canaliculi diameters of the different conditions (Fig. 2b), we took the diameters of the branches per condition and distribute them in bins of 0.25 μm and plotted them as a percentage of the total number of branches of that bin per condition (HepOrg or liver tissue).

## Time-lapse and live imaging

Organoids and assembloids images were acquired with a Leica DMIL LED (bright field only) using a Leica DF C450C camera and Leica Application Suite software (Leica) microscope. Whole-well pictures were acquired with a Leica M80 microscope using a Leica MC170 HD camera and Leica Application Suite software (Leica).

For live/dead staining a Viability/Cytotoxicity Assay Kit (Biotium, 30002) was used according to the manufacturer's protocol. In brief, 2 μM calcein and 4 μM ethidium homodimer III in PBS were added to cover the Matrigel containing organoids or mesenchyme culture, and incubated for 30 min. Then, the solution was removed and organoid or mesenchymal medium was added and cultures were imaged using a confocal microscope (Zeiss LSM 880) and processed using ImageJ/Fiji.

Time-lapse imaging of assembloid formation and cell death was carried out at 37 °C and 5% $CO_2$ using a Viventis LS1 Live microscope with a double illumination through 10× objectives and a detection 25×1.1 NA objective. MM was supplemented with SPY620-DNA stain (Spirochrome, SC401) at 1 μM final concentration. Videos were generated with ImageJ/Fiji.

The fibrous collagen signal was imaged on an upright or inverted laser-scanning confocal microscope (LSM 780 NLO equipped with

Observer Z.1 microscope stand, Zeiss). We used SHG of collagen in response to illumination with 800 nm femtosecond pulsed Chameleon Vision II Ti-Sapphire laser (Coherent). SHG signal was detected in forward propagated direction with the use of photomultiplier tube detector equipped with the bandpass filter 402/15 nm ET. For SHG of collagen fibres in live organoid cultures, W Plan-Apochromat 20×/1.0 DIC UV VIS IR water-dipping objective (Zeiss) was used. For SHG of collagen fibres in fixed organoid cultures, Plan-Apochromat 20×/0.8 objective was used. For tissue imaging, LD LCI Plan-Apochromat 40×/1.2 DIC Imm Corr M27 immersion corrected objective (Zeiss) was used. As an immersion medium for imaging samples mounted in glycerol-based mounting solution (Vectashield, Vector Laboratories) we used 80% glycerol immersion medium of refractive index n = 1,45 - Type G immersion liquid (Leica Microsystems).

## Bile acid analogue transport live imaging

We used CLF[72] to assess the functionality of bile canaliculi in HepOrg and assembloids (BSEP and MRP2 transporters). To assess both the detail of bile canaliculi and an overall CLF uptake, we performed live imaging using two imaging systems, a single photon point-scanning confocal system (Zeiss LSM 880 Inverted) for high-resolution, and Viventis LS1 lightsheet for imaging the whole structures.

For the imaging of HepOrg were released from Matrigel using Cell Recovery Solution (Corning, 354253) and re-seeded in an 8-well ibidi µ-Slide 8-well Glass Bottom coverslip (Ibidi, 80827) pre-coated with Matrigel (100%). An overlay of pure Matrigel was added and allowed to form a gel for 30 min at 37 °C. HM-Wnt medium was then added to these wells and incubated for 48 h. Medium was refreshed 24 h prior to imaging. 0.5 µM of SiR-actin (Spirochrome, SC001) was added in the culture medium 60 min before imaging. Ibidi slides were mounted into the stage (Zeiss Axio Observer. Z1 inverted stand with stage-top piezo) and allowed to equilibrate for 20 min to reduce thermal drift. XY positions were marked on Zen Black for z-stack acquisition with step size of 2.5 µm and pixel size of 0.08 µm × 0.08 µm or 0.1 µm × 0.1 µm. Live acquisition was performed on the the Zeiss LSM 880 Inverted system with the LD LCI Plan-Apochromat 63×/1.2 Imm Korr DIC M27 objective. The temperature was maintained at 37 °C with 5% humidified $CO_2$ in the sample holder chamber. All structures were imaged with 633 and 488 nm laser excitation and detection via the Fast Airyscan modality on a piezo-controlled stage at T0 prior to addition of CLF. CLF (Corning, 451041), at 10 µg ml$^{-1}$ final concentration, was added in situ, and acquisition was resumed 30 min after with acquisition every 25 min for 4.5 h. Time series hyperstacks were processed online via Airyscan processing function on Zen Black using the default settings for further analysis.

For the imaging of whole HepOrgs and assembloids, organoids were seeded in Viventis lightsheet holders in Matrigel, and overlaid with MM. HepOrgs were pre-treated for 2 days with MM before the start of imaging. CLF (Corning, 451041), at 10 µg ml$^{-1}$ final concentration was added 1 frame after the start of imaging. CMFDA[73] (also assessing BSEP and MRP2 transporters) and fluorescent phosphatidylcholine (to detect MDR2 function[74]) were used for further tests. Similarly to CLF imaging, HepOrg were imaged with CMFDA (ThermoFisher, C2925) at 5 µM final concentration, 16:0-06:0 NBD PC fluorescent lipid at 4 µM final concentration (Avanti Polar lipids, 810130P-1mg), and 5-CFDA at 5 µM final concentration (ThermoFisher, C1354), and the compounds were added 1 frame after the start of imaging. Imaging was performed at the intervals specified in the figure or figure legend. For both assembloids and HepOrg, MM was additionally supplemented with SPY620-DNA stain (Spirochrome, SC401), SPY555-DNA stain (Spirochrome, SC201) or SiR-actin (Spirochrome, SC001) at 0.5–1 µM final concentration. Time-lapse imaging was carried out at 37 °C and 5% $CO_2$ using a Viventis LS1 Live microscope with a double illumination through 10× objectives and a detection 25×1.1 NA objective. Videos were generated with ImageJ/Fiji.

## RT–qPCR

Total RNA was extracted from cells using the Arcturus PicoPure RNA Isolation Kit (Applied Biosystems, 12204-01) according to the manufacturer's protocol; including a 15 min digestion step with DNAse to remove traces of genomic DNA. The RNA (50-250 ng) was reverse-transcribed with the Moloney Murine Leukemia Virus reverse transcriptase (M-MLVRT) (Promega, M368B) or alternatively using or SuperScript III Reverse Transcriptase (Invitrogen, 18080-044) according to the manufacturer's instructions. Finally, cDNA was amplified using Fast-Start Essential DNA Green Master (Roche, 06402712001) on the LightCycler 96 machine (Roche) or using PowerUp SYBR Green Master Mix (ThermoFisher, A25741) on the Thermo Fisher QuantStudio 7 Pro. The list of primers used for RT–qPCR is provided in Supplementary Table 1. Gene expression levels were normalized to the housekeeping gene as specified in the graph axis labels.

## Cytokine array and functional assays

The supernatants (medium) from HepOrgs, homeostasis-like or fibrotic-like assembloid cultures were collected after 48 h of culturing. For albumin ELISA, bile acid measurements, and cytochrome assay, the organoids or assembloids were subsequently dissociated by TrypLE 1× treatment (10 min at 37 °C), and cell number per well was counted using a haemocytometer. The supernatant was then stored in −20 °C until analysis. For supernatant-based assays, the supernatant was used in the assay as per manufacturer's instructions—Mouse Albumin AssayMax ELISA Kit (Assaypro, EMA3201-1), Total Bile Acid Assay Kit (Colorimetric) (Cell Biolabs, STA-631) and Proteome Profiler Mouse XL Cytokine Array (R&D systems Biotechne, ARY028). For cytochrome activity measurements, the organoids were removed from Matrigel using Cell Recovery Solution (Corning, 354253) for 10 min on ice and subsequently placed in suspension 24-well plate with 0.5 ml PBS or medium in each well. Luciferin substrate was added as specified by manufacturer (P450-Glo CYP1A2 Induction/Inhibition Assay, Promega V8421; P450-Glo CYP3A4 Assay and Screening System, Luciferin IPA, Promega V9001) and left incubating with the cells 6 h in the incubator at 37 °C and 5% $CO_2$. Luciferin detection reagent was equilibrated at room temperature prior to measurement, and mixed at 1:1 ratio with supernatant from cells, incubated 20 min at room temperature and read using luminometer. CellTiter-Glo assay to measure viability was used following the manufacturer's protocol (G7570, Promega). The data was acquired on Perkin Elmer Envision 2104 for Albumin, BA, CellTiter-Glo and cytochrome activity measurement, and membranes were scanned on an iBright instrument (FL1500) for cytokine array. The unedited cytokine array scans are available in Supplementary Fig. 1.

## Bulk RNA-seq and analysis

HepOrg cultured in specified media for 1 or 2 weeks and assembloids cultured for 1 week or 2.5 weeks in their specified media were collected from Matrigel using Cell Recovery solution (354253, Corning), and then processed to RNA using PicoPure protocol, as described above. RNA was eluted with water. RNA libraries were prepared using the SMARTseq2 protocol[75] using Illumina Nextera library prep. Samples were sequenced on NovaSeq instrument, and sequencing of paired-end, 2×100-bp reads was performed at depth of 40mio reads/sample.

RNA-seq data were aligned to the mouse genome GRCm39 release 109 using STAR aligner (2.7.11b). featureCount (v2.0.6) was used to assigned reads exons, transcripts and CDS. Differential gene expression analysis was performed using the R (4.2.0) package DESeq2 (1.36.0). GSEA for bulk RNA-seq data was performed using the R package fgsea (1.22.0).

## scRNA-seq

For the scRNA-seq, hepatocytes from wild-type mice, cholangiocytes from a *Rosa26*-nTnG mouse and portal mesenchyme from a PDGFRα-H2B-GFP mouse sorted for SCA1$^+$ cells were used.

Mesenchymes, cholangiocyte organoids and HepOrgs were expanded to passage 2 for the experiment. Cells were aggregated at both healthy homeostatic and fibrotic-like ratios as described above. Hepatocytes were pre-conditioned 48 h in MM before aggregation. The assembloids were dissociated to single cells using TrypLE 1× for 10 min at 37 °C with (repeats 2 and 3) or without pre-warming TrypLE prior to dissociation. The cells were resuspended in PBS with 0.04% BSA, 0.2 U μl$^{-1}$ RNAse inhibitor (Merck, 3335399001) and 10 μM ROCK inhibitor (Y-27632, Merck/Sigma, Y0503) in BSA-coated tubes for the first replicate, additionally supplemented with DNAse (Sigma, DN25, 1:1000 from 10 mg ml$^{-1}$ stock) for the second and third biological replicate. The cells were filtered with 100 μm strainer, cell suspensions (7,000–20,000 cells) were loaded into Chromium Next GEM ChipG and processed as specified in manufacturer's instructions for 10X Chromium Next GEM Single Cell 3′ Reagent Kits v3.1 (10X Genomics). Single-cell RNA-seq libraries were prepared using Single Cell 3′ v3.1 Gel Bead kit (10X Genomics) as per manufacturer's instructions, and library amplified with 9 cycles. For the cDNA amplification, 11 cycles were used. Libraries were sequenced on NovaSeq 6000, to a depth of 30,000 reads per cell with paired-end 100 bp reads.

### scRNA-seq processing
The sequencing data was aligned to the GRCm39 (release 109) mouse genome with the count from 10X Cellranger (7.1.0). Ambient RNA was removed using Cellbender (0.3.0), the number of expected cells estimated by Cellranger and the total number of included droplets set to 25,000. Cells were retained if they had a cell probability >0.99. The sequencing data was further processed with scanpy (1.9.2[76]). Cells with fewer than 1,000 genes expressed were excluded from further analysis. We further excluded cells with more than 25 % mitochondrial read counts, cells with more than 10,000 genes expressed, and total read count over 50,000 as potential doublets. We further eliminated doublets using Scrublet[77] (https://github.com/swolock/scrublet). We further eliminated doublets using the eGFP and tdTomato marker genes in Mesenchyme and Cholangiocyte cell populations, respectively. Counts were normalized per cell and log-transformed. Variation in total gene count and percent mitochondrial gene count were regressed out before count scaling. For generating a consensus zonation score, we averaged the expression of validated pericentral or periportal markers from three different publicly available datasets[13,29,30], using the scanpy function score_genes.

### scRNA-seq dimensionality reduction and clustering
Cells were clustered using the Leiden algorithm implemented in scanpy based on the *k*-nearest neighbours (*k*NN) ggraph calculated using scanpy's neighbours function using 15 neighbours and 40 principal components. We assigned cell per cluster as one of three cell types, Hepatocytes (*Hnf4a, Apoa1, Apoc1, Hpx, Ttr, Alb, Mup20, Gpx1, Cyp3a11, Abcb4*), Mesenchyme (*Pdgfra, Pdgfrb, Des, Vim, Col1a1, Col1a2, Col3a1, Col6a2, Ly6a*) and Cholangiocytes (*Krt19, Krt7, Epcam, Ezr, Sox9, Prom1*) based on marker gene expression. All dotplots, UMAPs and correlation plots were produced with the corresponding scanpy functions. Violin plots were created using the plotnine python package. Pearson coefficient of all groups indicated in plots was calculated as the correlation between normalized counts.

### Integration with published single cell datasets
We obtained raw read count data from several previously published liver cell datasets[11–13,33–39]. Datasets were aggregated before processing using scanpy and then processed together as described above. Dimensionality reduction and clustering was performed as described above with the difference that the *k*NN ggraph for clustering was calculated using BBKNN[78]. Cell types were assigned as in the original publications used. We clustered cell types across different datasets and conditions using Pearson correlation coefficient.

### Differential gene expression and GSEA
Differential gene expression was calculated pairwise between conditions using a Wilcoxon rank-sum test. Genes were considered differentially expressed if they showed an absolute log-foldchange >1 and a Benjamini−Hochberg adjusted *P* value < 0.05. Gene set enrichment analysis was performed using the gseapy (1.0.5) package. Whole-organoid GSEA between healthy and fibrotic conditions was performed for the MSigDB hallmark (2020), WikiPathways (2019, Mouse), Reactome (2022), Elsevier Pathway Collection and GO Molecular Function (2023) gene sets. Cell type specific GSEA between conditions was performed for the MSigDB hallmark (2020) gene set. Functional identity was assessed by GSEA by comparing each cell type to all other cell types using the GO Molecular Function (2023), GO Biological Process (2023) and GO Cellular Component (2023) gene sets. For all GSEA, normalized enrichment scores (NES) and FDR-adjusted *P* values are reported.

### Cell−cell communication analysis
We used the ligand−receptor analysis framework (LIANA, 0.1.9)[79] to explore cell−cell communication via ligand−receptor expression. We used LIANAs rank_aggregation method to calculate seven different cell−cell communication metrics as well as their rank aggregation. We used the magnitude rank as interaction score. We only retained interactions with a specificity rank less than 0.05. Ligand−receptor interactions were visualized using the R (4.2.0) package ggraph (2.1.0).

### Statistical analysis
Data were analysed as detailed in figure legends and as appropriate for each experiment by using two-tailed Mann−Whitney test, Welch's *t*-test, unpaired *t*-test with Welch's correction, paired *t*-test, a Student's *t*-test, or Kruskal−Wallis test with Dunn's multiple comparisons post hoc test. *P* < 0.05 was considered statistically significant. Calculations were performed using GraphPad Prism software package (v10.0.2 and v.10.0.0). All *P* values are given in the corresponding figure legends or figures. Dispersion and precision measures (such as mean, median, s.d. and s.e.m.) are specified in the figure legends. Statistics for scRNA-seq data are described in the corresponding section.

### Reporting summary
Further information on research design is available in the Nature Portfolio Reporting Summary linked to this article.

## Data availability
The scRNA-seq and bulk RNA-seq datasets generated during this study are available at the Gene Expression Omnibus (GEO) under accession numbers GSE274971 and GSE274973. Full lists of bulk RNA-seq transcripts per million data and differentially expressed genes are provided in Supplementary Table 2 and Supplementary Table 6, respectively. GSEA terms are available in Supplementary Table 3 and Supplementary Table 5. All inferred cell−cell interactions are available in Supplementary Table 4. scRNA-seq data were aligned to the GRCm39 (release 109) mouse genome with the 10X Cellranger (7.1.0) count. Bulk RNA-seq data were aligned to the mouse genome GRCm39 release 109 using STAR aligner (2.7.11b). All other images, quantitative PCR and measurement data are presented in the manuscript, and data used to plot graphs are provided as Source Data. For GSEA, the following databases were used: whole-organoid GSEA between healthy and fibrotic conditions was performed with MSigDB hallmark (2020), WikiPathways (2019, Mouse), Reactome (2022), Elsevier Pathway Collection and GO Molecular Function (2023) gene sets. Cell-type-specific GSEA between conditions was performed with the MSigDB hallmark (2020) gene set. Functional identity was assessed by GSEA by comparing each cell type to all other cell types using the Gene Ontology (GO) molecular function (2023),

GO biological process (2023) and GO cellular component (2023) gene sets. Source data are provided with this paper.

## Code availability

The source code used for bile canaliculi segmentation and for quantifying mesenchyme nuclei is available at https://git.mpi-cbg.de/huch_lab/assembloid-paper. The macro for ball/bubbly time-lapse imaging can be found on GitHub: https://github.com/stoeter/Fiji-Tools-for-HCS/tree/master/Share/Huch_StackFilesFromKNIME.

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

**Acknowledgements** M.H., M.Z. and H.A.H. are supported by the Max Planck Gesselschaft. This work was partially supported by an EMBO YIP and by the European Union (ERC, REG_ORGatSCALE, 101088869) awarded to M.H. Views and opinion expressed are those of the author(s) only and do not reflect those of the European Union or the European Research Council. Neither the European Union nor the granting authority can be held responsible for them. This project was also partially supported by the Deutsche Forschungsgemeinschaft (DFG, German Research Foundation, 514150034) and the DFG under Germany´s Excellence Strategy EXC-2068–390729961 Cluster of Excellence Physics of Life of TU Dresden. This work was supported by the DFG Research Infrastructure NGS_CC (Meritxell Huch, project no. 51450034) as part of the Next Generation Sequencing Competence Network (DFG project 423957469). H.M.B., D.B. and H.A.H. are members of the Centre for Topological Data Analysis, which is funded by the EPSRC grant New Approaches to Data Science: Application Driven Topological Data Analysis (EP/R018472/1). H.Y., H.M.B., H.A.H. and M.H. are members of the EPSRC collaboration award EP/Z531224/1. H.A.H. is member of Mathematical Foundations of Intelligence Hub: An Erlangen programme for AI supported by EP/Y028872/1. H.Y. and H.M.B. are funded by the Ludwig Institute for Cancer Research. H.A.H. and H.Y. gratefully acknowledge funding from Leverhulme Trust Philip Leverhulme Prize PLP-2020-252, which supported this research. The authors thank J. Jarrells for assistance with FACS; S. Reinhardt and J. Bläsche at the DcGC Dresden-concept Genome Center — a core facility of the CMCB and Technology Platform of the TUD Dresden University of Technology for NGS library preparation, data production and QC analyses; F. Rost and J. Bregante for help with scRNA-seq analysis; M. Artz for help with image analysis; A. Liebert for help with 2D hepatocyte sandwich culture; J. Helppi, K. Reppe and A. Muench-Wuttke for assistance with animal breeding and hepatocyte isolation; the MPI-CBG Biomedical Services (animal facility) for animal care, transgenics and genotyping facilities for animal re-derivation and genotyping respectively; J. Peychl and R. Maraspini for imaging troubleshooting and training; R. Barsacchi and M. Stöter for high-throughput imaging and image analysis; A. Vega and I. Sebastian for assistance with image analysis pipelines; M. Bovyn for the interpretation of 2D versus 3D networks, M. Marass for insightful comments, help with editing and critical reading of the manuscript; and J. V. Iturra for critical reading of the manuscript.

**Author contributions** A.M.D., A.S. and M.H. designed the study. A.M.D. and A.S. performed most of the experiments and data analysis. A.N. performed the analysis of HepOrg shapes and siRNA knockdown experiments. A.N. and A.S. performed *Mdr2*-knockout experiments. A.M.D., A.S. and A.N. together with M.H. interpreted the results. C.L. performed the scRNA-seq and bulk RNA-seq analysis assisted by A.M.D. and A.N. for the fibrotic datasets. J.D. developed the script for bile canaliculi segmentation. H.Y. performed all the mathematical data analysis. A.L. performed some tissue staining and some HepOrg live imaging. A.C.K. performed organoid segmentation. H.Y., D.B., H.M.B. and H.A.H. developed the mathematical data analysis pipeline. S.S. assisted with hepatocyte isolation. H.M.B., H.A.H., M.Z. and M.H. supervised the work. A.M.D., A.S. and M.H. wrote the manuscript. M.H. is the lead author of this study. All authors read and commented on the manuscript.

**Funding** Open access funding provided by Max Planck Society.

**Competing interests** M.H. is inventor in several patents on organoid technology. A.S., A.M.D. are inventors in a patent on organoids. The other authors declare no competing interests.

**Additional information**
**Correspondence and requests for materials** should be addressed to Meritxell Huch.

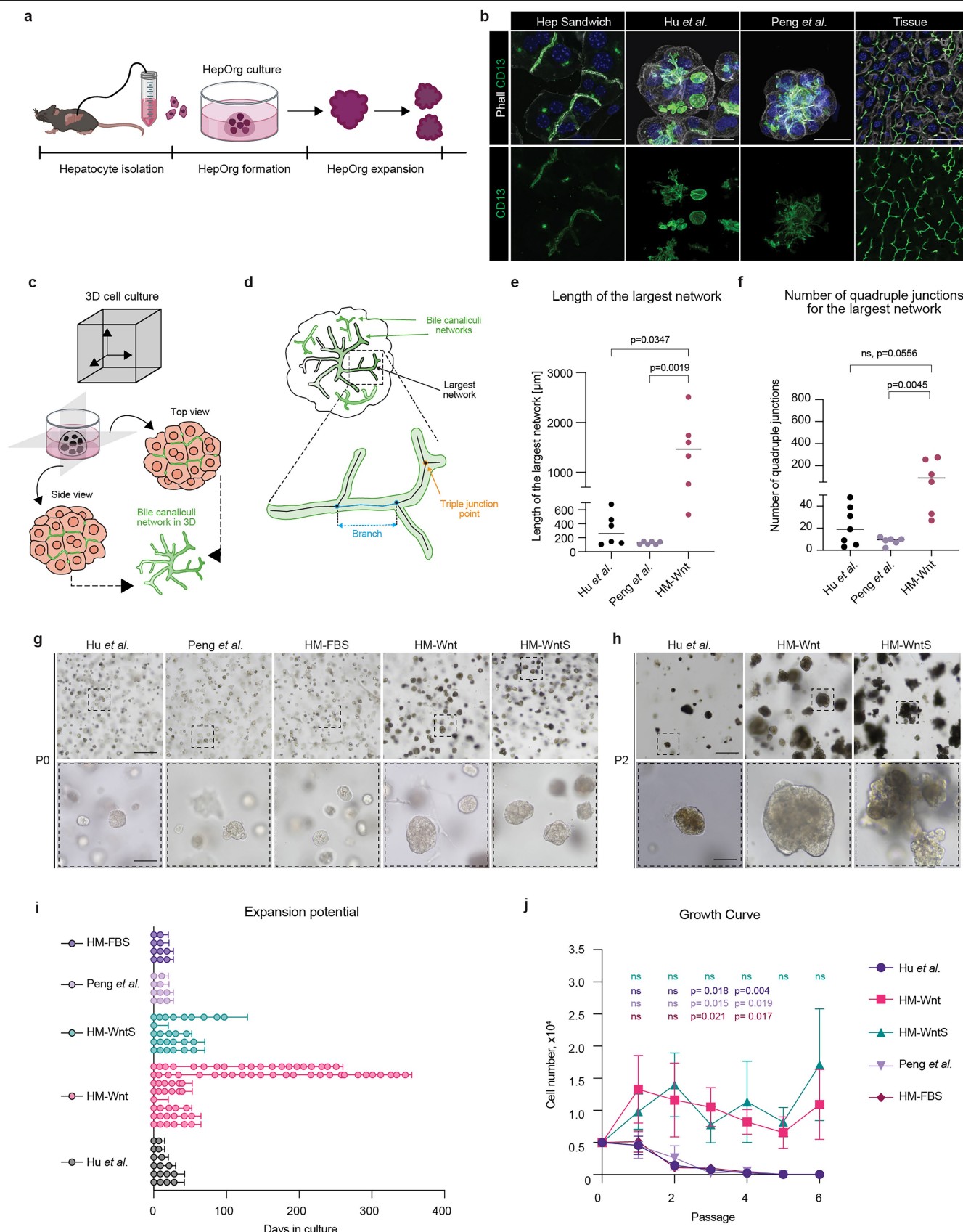

**Extended Data Fig. 1** | See next page for caption.

**Extended Data Fig. 1 | HepOrg grown in optimized medium expand long-term.** a. Experimental design. b. Immunofluorescence staining for bile canaliculi (CD13, green) and F-actin (Phall, grey) in 2D-hepatocytes cultured as sandwich culture, HepOrg cultures grown in Hu et al. media, Peng et al. media, and in liver tissue sections. Note the difference in organoid size and length of the bile canaliculi when compared to the optimized medium in Fig. 1d. DAPI stained nuclei (blue). Representative images from n = 3 independent experiments. Scale bar, 50 µm. c. Schematic illustration of the 3-dimensional (3D) nature of bile canaliculi network within HepOrg. d. Schematic representation of the different measurements used to describe the bile canaliculi network. e. Graph represents the length of the largest network in HepOrg cultured as indicated. Dot, measure per organoid. Line, mean. Kruskal-Wallis-test followed by Dunn's multiple comparison post-hoc test. f. Graph represents the number of quadruple junctions for the largest network in tissue and HepOrg cultured as indicated. Line, mean. Dot, measure per organoid. Kruskal-Wallis-test followed by Dunn's multiple comparison post-hoc test. g-h. Bright-field pictures of HepOrg at passage 0 (P0, 9 days after seeding) (g), or at passage 2 (P2, 29 days after seeding) (h), seeded as sparse culture (1000 cells/well) and cultured under the indicated media conditions. Representative images from n = 3 independent experiments. Scale bar, 500 µm; zoom-in 100 µm. i. Graph shows HepOrg expansion over time. Each line represents an independent biological replicate. Cultures were split at a 1:2 split ratio. Dot, time of passage. j. Growth curves of HepOrg grown in the indicated media. Values represent total number of cells at the indicated passage expressed as mean ± SEM from n = 4 independent biological replicates with n = 2 technical replicates. Statistics are provided between HM-Wnt and the other conditions and presented colour-coded for the condition they compare to; multiple unpaired t-tests, two-sided.

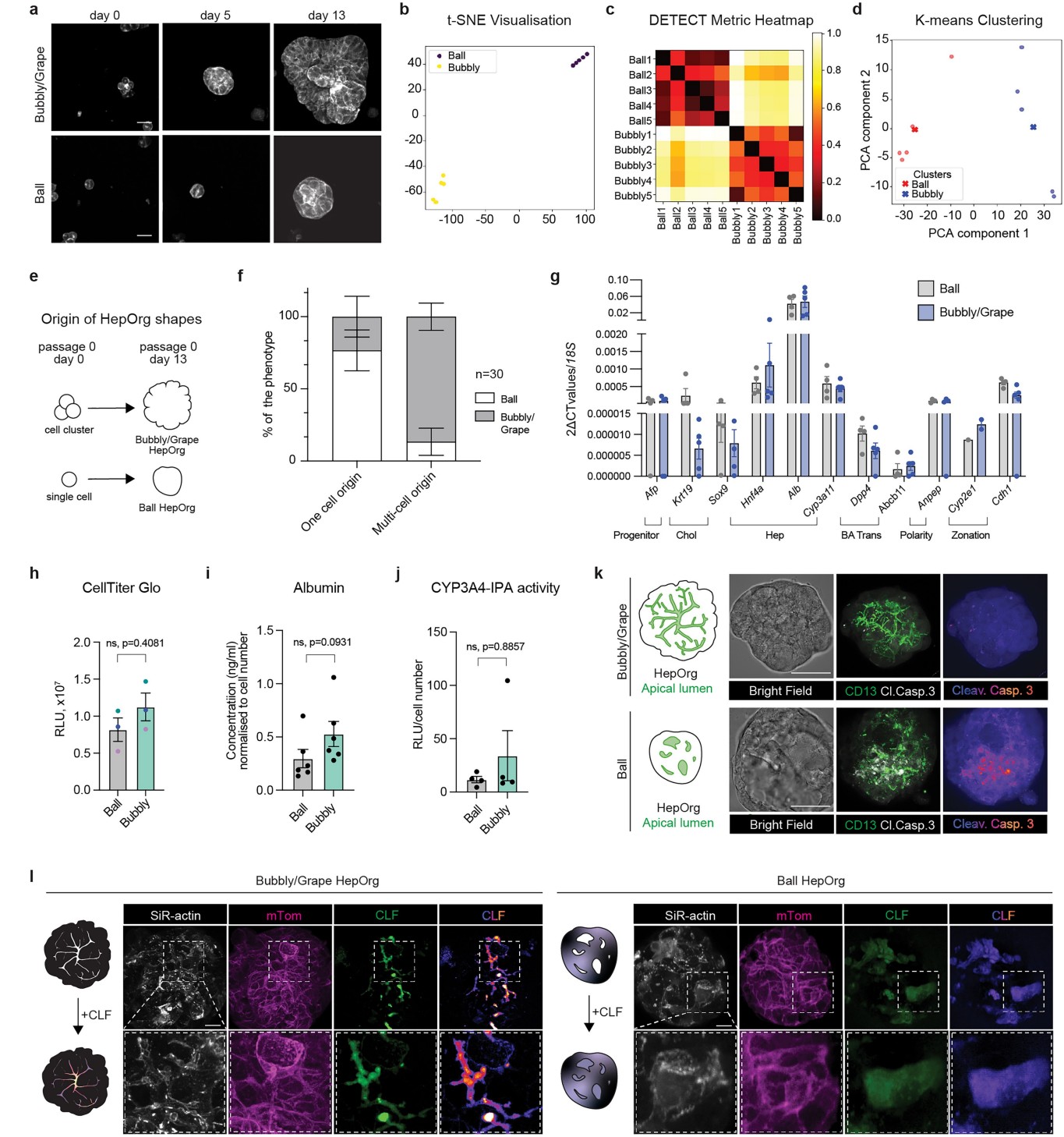

**Extended Data Fig. 2** | See next page for caption.

**Extended Data Fig. 2 | The starting number of initiating cells dictates HepOrg morphology.** a. Maximum intensity projection of images from live imaging of membrane-tdTomato HepOrg seeded as >2 hepatocyte cell clusters (top) or as single cells (bottom). Seeding density, 200 cells/µl from n = 2 independent experiments. Scale bar, 50 µm. b-d. DETECT calculations on HepOrg morphological variation integrating all time points (d0-d13). (b) t-SNE plot visualizes the DETECT results. Dot, individual organoid coloured according to shape type. (c) Heatmap of the DETECT metric distance for the DETECT results of each HepOrg pair integrating all time points (d0-d13). (d) K-means clustering applied to DETECT calculations following PCA reduction to two principal components. Dot, individual organoid; crosses, centroid of the respective clusters; colour, cluster assignment for the organoid. e-f. Analysis of the HepOrg shape according to the number of cells originating the structure. e, Schematic illustration. f, Bar graph shows the shape-type according to the initial number of cells in the structure. Results are presented as percentage from a total of n = 30 structures per experiment and expressed as mean ± SEM from n = 2 independent experiments. g. qRT-PCR expression analysis of the indicated markers in HepOrg (passage 2) hand-picked according to their ball-shape or bubbly/grape-shape morphology. Graph represents the mean ± SEM from n = 5 independent experiments for most genes, aside from Cyp2e1, tested n = 1 for ball-shape and n = 2 for bubbly-shape. Each dot is a biological replicate. The differences between ball-shape and bubbly-shape are not significant (Mann-Whitney test). Chol, cholangiocyte; Hep, hepatocyte; BAtrans, bile canaliculi transport. h. Viability assay (CellTiter-Glo) performed on ball-shape and bubbly/grape-shape HepOrg. Graph represents mean ± SEM of n = 3 biological replicates from 3 independent experiments, with dot colour denoting each independent experiment; Mann-Whitney test, two-tailed. i-j. Albumin (i) and Cytochrome activity (j) measurements of ball-shape *versus* grape-like/bubbly-shape HepOrg show non-significant but marked reduction in functionality of ball-HepOrg. Graph represents mean ± SEM of n = 5 biological replicates from 3 independent experiments; Mann-Whitney test, two-tailed. k. Brightfield and immunofluorescence images of bubbly/grape-like-shape (top) and ball-shape (bottom) HepOrg, stained for apical polarity marker CD13 (green) and apoptosis marker (cleaved caspase 3, grey; also shown in Fire LUT for easy visualization). n = 3 independent experiments. Scale bar, 50 µm. l. Still images of live cell imaging analysis of bile acid analogue uptake (CLF, green in the middle panel, fire LUT in most right panel) in membrane-tdTomato HepOrg (mTom, magenta) with bubbly/grape-shape (left) or ball-shape (right). SiR-actin (grey) labels cell borders. Note that bubbly/grape-like organoids uptake and release CLF into their bile canaliculi while ball-shape organoids accumulate it in hepatocytes. n = 2 independent experiments. Left, schematic of experimental set up. Scale bar, 20 µm.

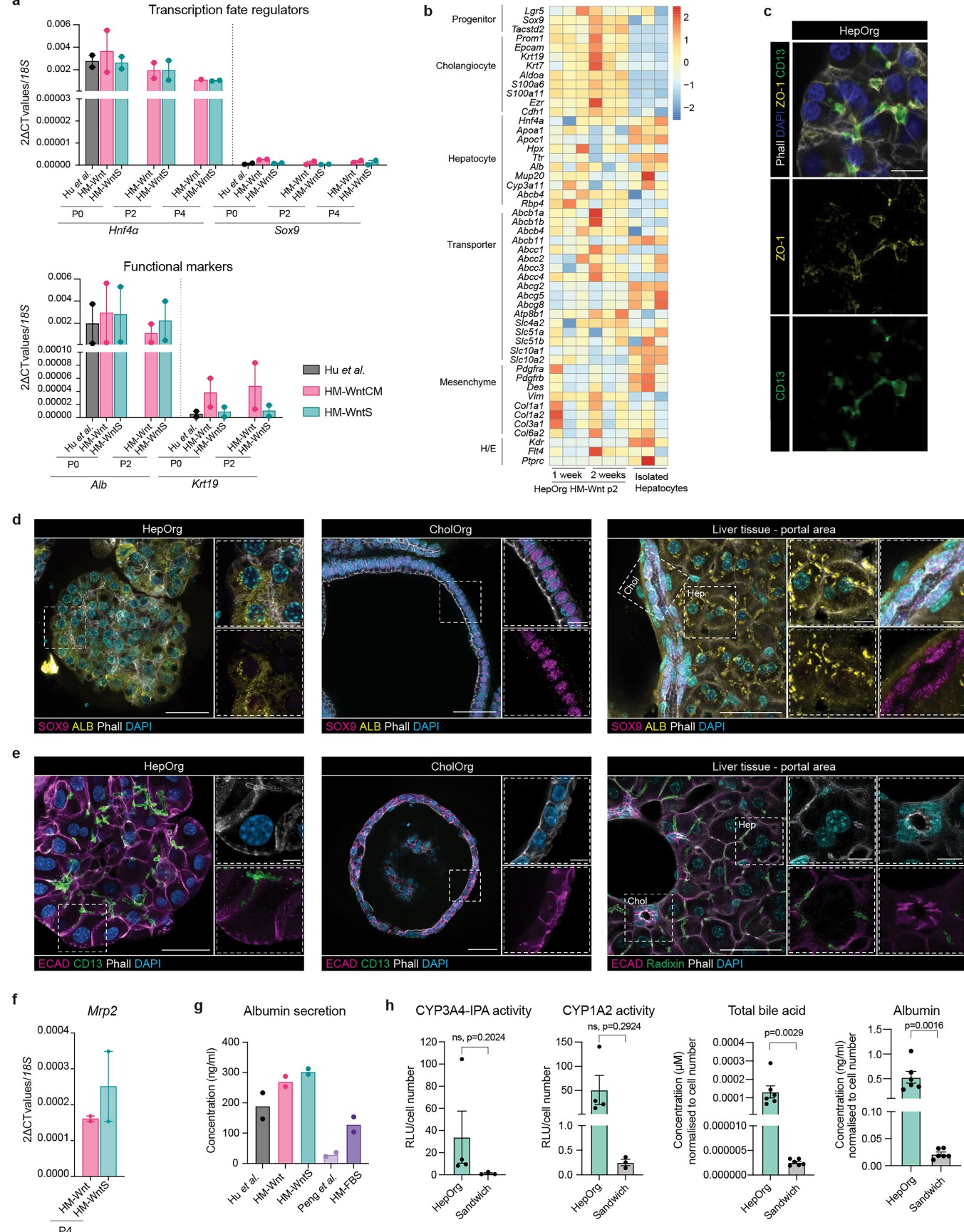

**Extended Data Fig. 3** | See next page for caption.

**Extended Data Fig. 3 | Optimised HepOrg express hepatocyte markers and present hepatocyte function similar to freshly isolated hepatocytes.** a. qRT-PCR expression analysis of selected marker genes from HepOrg grown in the indicated medium conditions over several passages (P). Graphs represent mean ± SEM of n = 2 biological replicates from n = 2 independent experiments performed as n = 2 technical replicates per experiment. b. Bulk RNA sequencing analysis of HM-Wnt media HepOrg compared to freshly isolated hepatocytes. Heatmap represents the log10(TPM + 1) (transcripts per million) values from the RNAseq for the indicated genes, which are Z-scored across the gene (n = 3 biological replicates). c. Immunofluorescence staining of the tight-junction marker ZO1 (yellow), bile canaliculi marker CD13 (green), DAPI (blue, nuclei) and Phalloidin (F-actin, grey) in HepOrg from optimized medium. Representative images of n = 2 independent experiments. Scale bar, 10 μm. d. Left and middle, immunofluorescence staining for DAPI (Cyan, nuclei), Phalloidin (F-actin, grey), hepatocyte marker albumin (ALB, yellow) and cholangiocyte marker SOX9 (magenta) in optimised HepOrg (left) and cholangiocyte organoids (CholOrg, middle) in EM medium. Right, staining in liver tissue. Representative images of n = 2 independent experiments.

Scale bar, 50 μm; zoom-in 10 μm. e. Left and middle, immunofluorescence staining for CD13 (apical, green), E-Cadherin (ECAD, basolateral, magenta), DAPI (nuclei, cyan) and F-actin (Phall, membrane, grey) in HepOrg grown in optimized medium (left) and CholOrg grown in EM medium (middle). Right, immunofluorescence staining for Radixin (apical, green) and E-Cadherin (ECAD, basolateral, magenta) in liver tissue. Representative images of n = 2 independent experiments. Scale bar, 50 μm; zoom-in 10 μm. f. qRT-PCR of multidrug resistance associated protein 2 (*Mrp2*) in HepOrg from indicated media. Graph represents mean ± SEM of n = 2 biological replicates from n = 2 independent experiments performed as n = 2 technical replicates per experiment. g. Albumin secretion of HepOrg grown in indicated medium conditions; Graph represents mean from n = 2 technical replicates. h. Cytochrome activity (HepOrg n = 4, sandwich n = 3), total bile acid measurements and Albumin secretion (n = 3, each 2 technical replicate) of HM-Wnt HepOrg (Alb and CYP3A4-IPA reproduced from HepOrg bubbly, Extended Data Fig. 2i,j) compared to hepatocyte sandwich culture show improved functionality of HepOrg. Graph represents mean ± SEM; Mann-Whitney test, two-tailed.

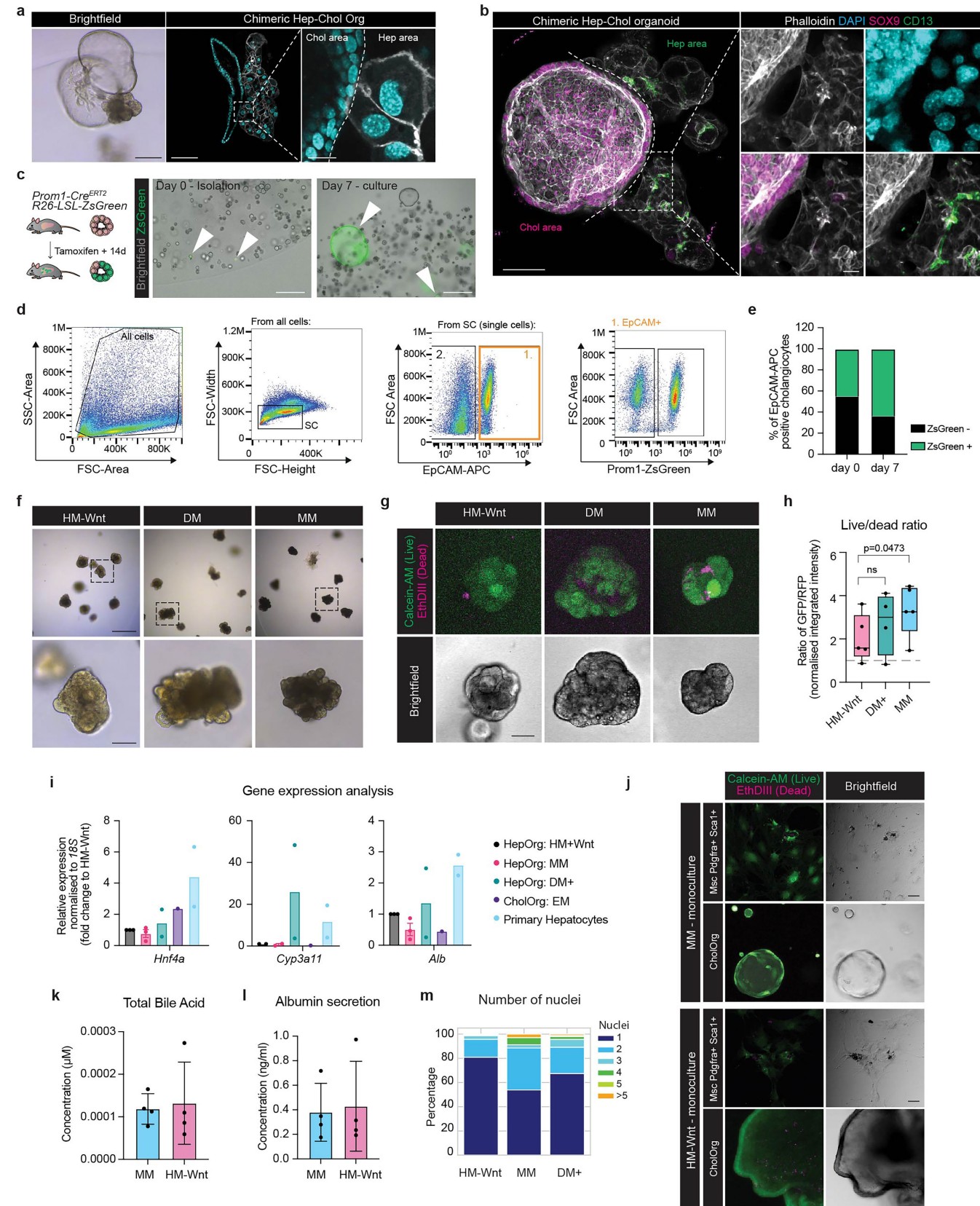

**Extended Data Fig. 4** | See next page for caption.

**Extended Data Fig. 4 | Spontaneous formation of non-physiological hepatocyte-cholangiocyte structures and identification of assembloid co-culture media.** a. Brightfield and immunofluorescence images of chimeric hepatocyte-cholangiocyte (Hep-Chol) organoid, where cholangiocyte organoids (CholOrg) spontaneously emerge in the vicinity of hepatocyte organoids (HepOrg). F-actin (Phall, grey), and nuclei (DAPI, cyan). Note the non-physiological ratio between both cell types (compare to Fig. 1a). n > 3 independent experiments. Scale bars, brightfield −100 μm, IF − 50 μm, zoom-in 10 μm. b. Immunofluorescence (IF) staining for CD13 (BC, green) and SOX9 (cholangiocytes, magenta) in chimeric Hep-Chol organoids. DAPI (nuclei, cyan) and Phalloidin (membrane, grey). Note the lack of connection between the bile canaliculi from hepatocytes and bile duct, despite the proximity of both structures. n > 3 independent experiments. Scale bar, 20 μm, zoom-in, 10 μm. c. Left, schematic of experimental approach (reproduced from ref. 20 (CC BY 4.0)). Right, representative images (n = 3 independent experiments) of seeded hepatocytes after isolation from a tamoxifen-injected *Prom1-CreERT2 x R26-LSL-ZsGreen* mouse livers after 14 days wash-out period. Note the presence of small ZsGreen-labelled cells (arrowhead, left picture), which then expand into ZsGreen-labelled cholangiocyte organoids (arrowhead, right picture). Scale bars, left 200 μm, right 500 μm. d. Sorting strategy to identify how many cholangiocytes are labelled by ZsGreen in our experiments. Representative plots of n > 2 biological replicates are shown. e. Percentage of recombined cholangiocytes from 'c'. The % of labelled cells is similar between day 0 and day 7 of culture, suggesting that the CholOrg derived from contaminating cholangiocytes in the hepatocyte isolation prep. Graph represents mean (d0 n = 3, d7 n = 2) of 2 independent experiments. f-j. Co-culture media test to obtain medium that prevented cholangiocyte and mesenchyme overgrowth, while preserving hepatocyte polarity and bile canaliculi structure. f. Representative HepOrg brightfield images at passage 1, cultured in HM-Wnt, and switched to specified media for 7 days. Scale bar, 100 μm, zoom-in, 50 μm. g. HepOrg stained with a live/dead cell dye as detailed in methods; culture at passage 1 switched to specified media for 7 days. Scale bar, 50 μm. h. Ratio of live to dead dye intensity, corresponding to pictures from g. Data is presented as box (the interquartile range, 25th and 75th percentile, with line at median) and whiskers (min, max of the data) plot from HM-Wnt n = 5; DM n = 4; MM n = 5 biological replicates. Paired t test, two-tailed. i. qRT-PCR expression analysis of selected marker genes in HM+Wnt (n = 3), MM (n = 3) and DM (n = 2) media; Graph represents mean, with ± SEM when n = 3. Freshly isolated hepatocytes (n = 2) and CholOrg (n = 1) controls are also included. j. Representative images (n = 2 independent experiments) of cholangiocyte organoids (CholOrg) and portal mesenchymal cells (Msc Pdgfra+Sca1+) monoculture, cultured in MM (top) or HM-Wnt (bottom) and stained with a live/dead fluorescent cell dye. k-l. Total bile acid (k) and albumin (l) production from HepOrg grown in HM-Wnt or MM. Data is presented as mean +/- SD from n = 4 replicates from n = 4 independent experiments. Results are expressed as μM concentration (k) or ng/ml (l) normalised to total cell number per condition; scale bar, 50 μm. m. Number of nuclei per cell in HepOrg grown in specified medium, n = 3.

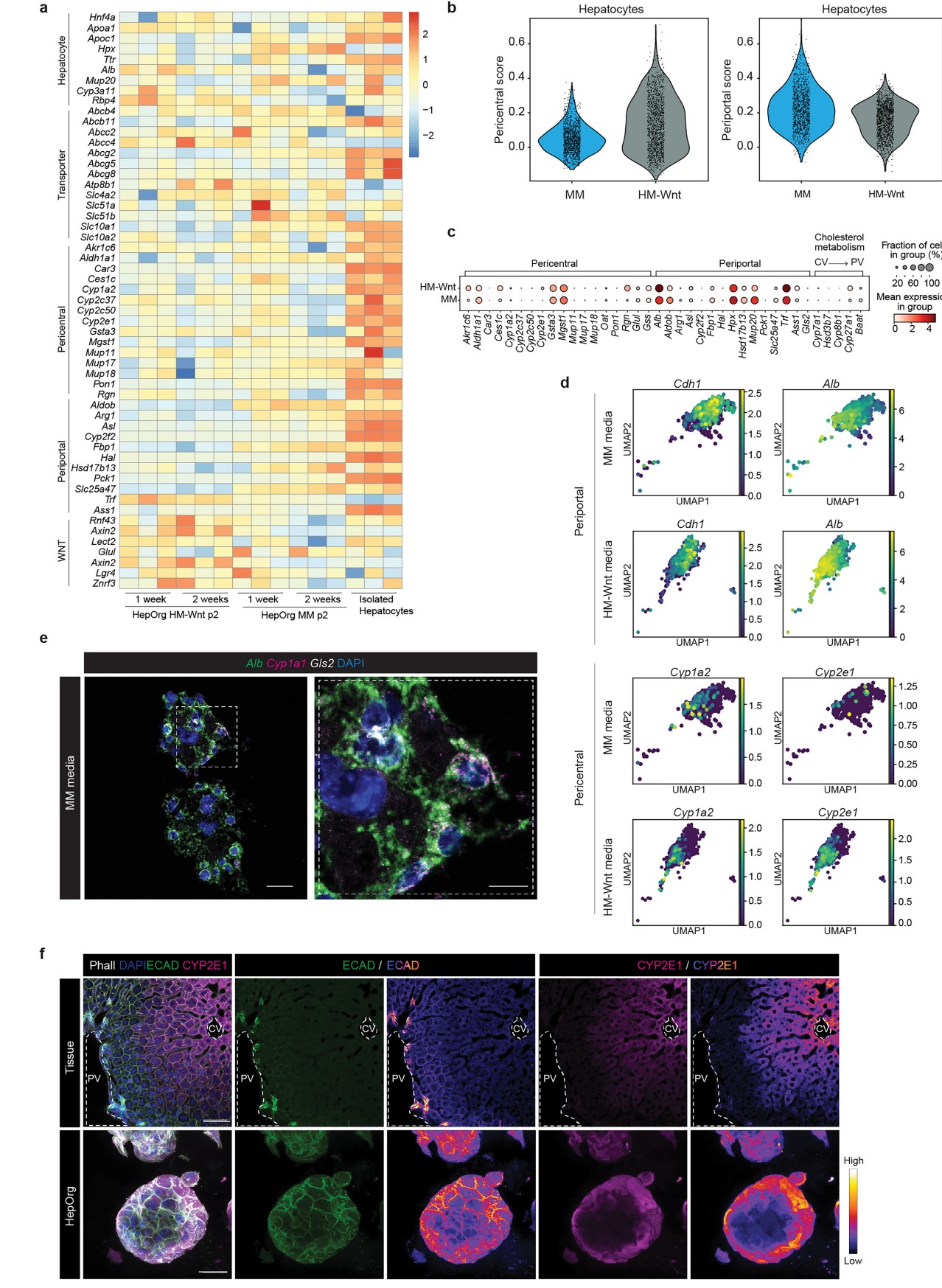

**Extended Data Fig. 5** | See next page for caption.

**Extended Data Fig. 5 | HepOrg in MM and HM-Wnt media show some degree of zonation.** a. Bulk RNA sequencing analysis of HM-Wnt and MM media HepOrg compared to freshly isolated hepatocytes. Heatmap represents the log10(TPM + 1) (transcripts per million) values from the RNAseq for the indicated genes, Z-scored across each gene (n = 3 biological replicates). For HM-Wnt, some genes are reproduced from Extended Data Fig. 3b. b. Zonation score calculated from scRNAseq data of HM-Wnt and MM media HepOrg shows that HM-Wnt-grown HepOrg are on average more pericentrally zonated, while the score is shifted periportally for MM-grown HepOrg. c. Dot plot shows gene expression from scRNAseq of HM-Wnt and MM media HepOrg for pericentral and periportal genes, as well as zonated cholesterol metabolism genes.

d. UMAP representing the hepatocytes from HM-Wnt and MM media HepOrg and expression of selected periportal (*Cdh1*, *Alb*) and pericentral (*Cyp1a2*, *Cyp2e1*) genes in the two media. e. Expression of periportal (*Gls2*, *white*) and pericentral (*Cyp1a1*, *magenta*) gene RNA visualised by RNAscope in HepOrg from MM media, representative of n = 3. *Alb* (green) and nuclei (DAPI, blue) are also shown. Scale bar, 50 μm, zoom-in, 20 μm. f. Immunofluorescence staining of periportal (E-CAD, E-cadherin, green and Fire LUT), pericentral (CYP2E1, magenta, and Fire LUT), Actin (Phalloidin, grey) and nuclei (DAPI, blue) in HepOrg cultures (P2) grown in HM-Wnt and transferred to MM for 7 days, compared to liver tissue. Representative images from 4 independent experiments are shown. Scale bar, 50 μm.

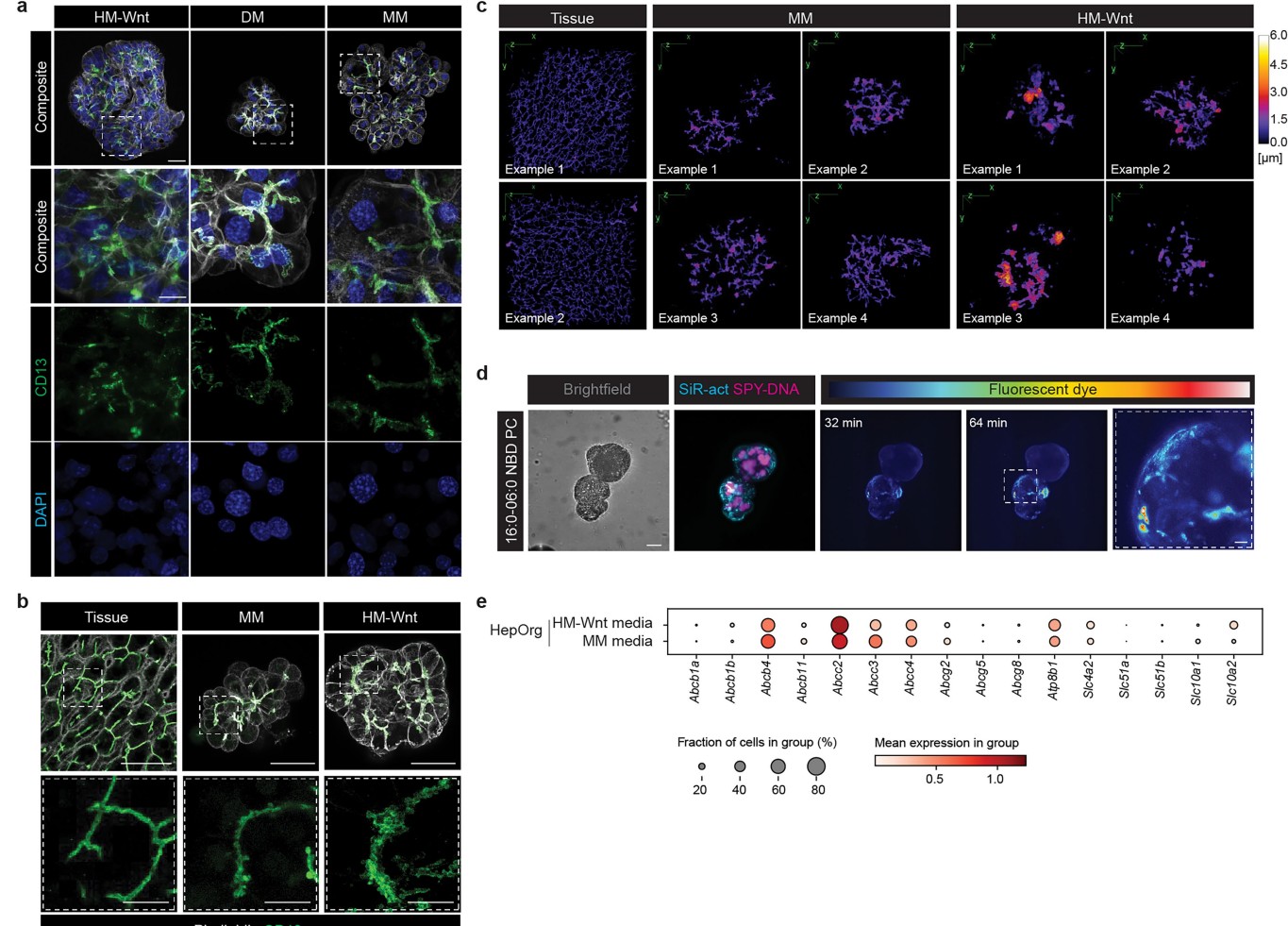

**Extended Data Fig. 6 | HepOrg in assembloid media form narrow and homogenous bile canaliculi network.** a. Immunofluorescence staining for bile canaliculi (CD13, green), nuclei (DAPI, blue) and cell borders (Phalloidin, grey) in HepOrg grown in the different media. Representative images from 3 independent experiments are shown, Scale bar, 100 μm. b. Immunofluorescence staining for bile canaliculi (CD13, green) and cell borders (Phalloidin, grey) in liver tissue (left), HepOrg cultures in MM for 7 days (middle), or HepOrg grown in HM-Wnt media (right). Representative images are shown of n = 3 independent experiments. Scale bar, 20 μm, zoom-in, 10 μm. c. Representative BC networks from healthy mouse liver tissue (left), HepOrg grown in MM media (middle), and HepOrg in HM-Wnt media (right). Colour corresponds to the mean bile canaliculi (BC) diameter in μm as indicated in intensity scale (blue, BC < 1.5 um; white, BC > 6 um). Note that BC is most homogenous in tissue, followed by HepOrg in MM media, while HepOrg in HM-Wnt media show large BC diameter variability. n = 3 independent experiments. d. Still images from time-lapse imaging analysis of fluorescent phosphatidylcholine (16:0-06:0 NBD PC) confirms functionality of MDR2 transporter. Compounds are shown in Royal LUT. Nuclei (SPY555-DNA, magenta) and actin (SiR-act, cyan) are also shown. n = 3 independent experiments. Scale bar, 50 μm. e. Dot plot shows gene expression from scRNAseq of HM-Wnt and MM media HepOrg for bile transporter genes.

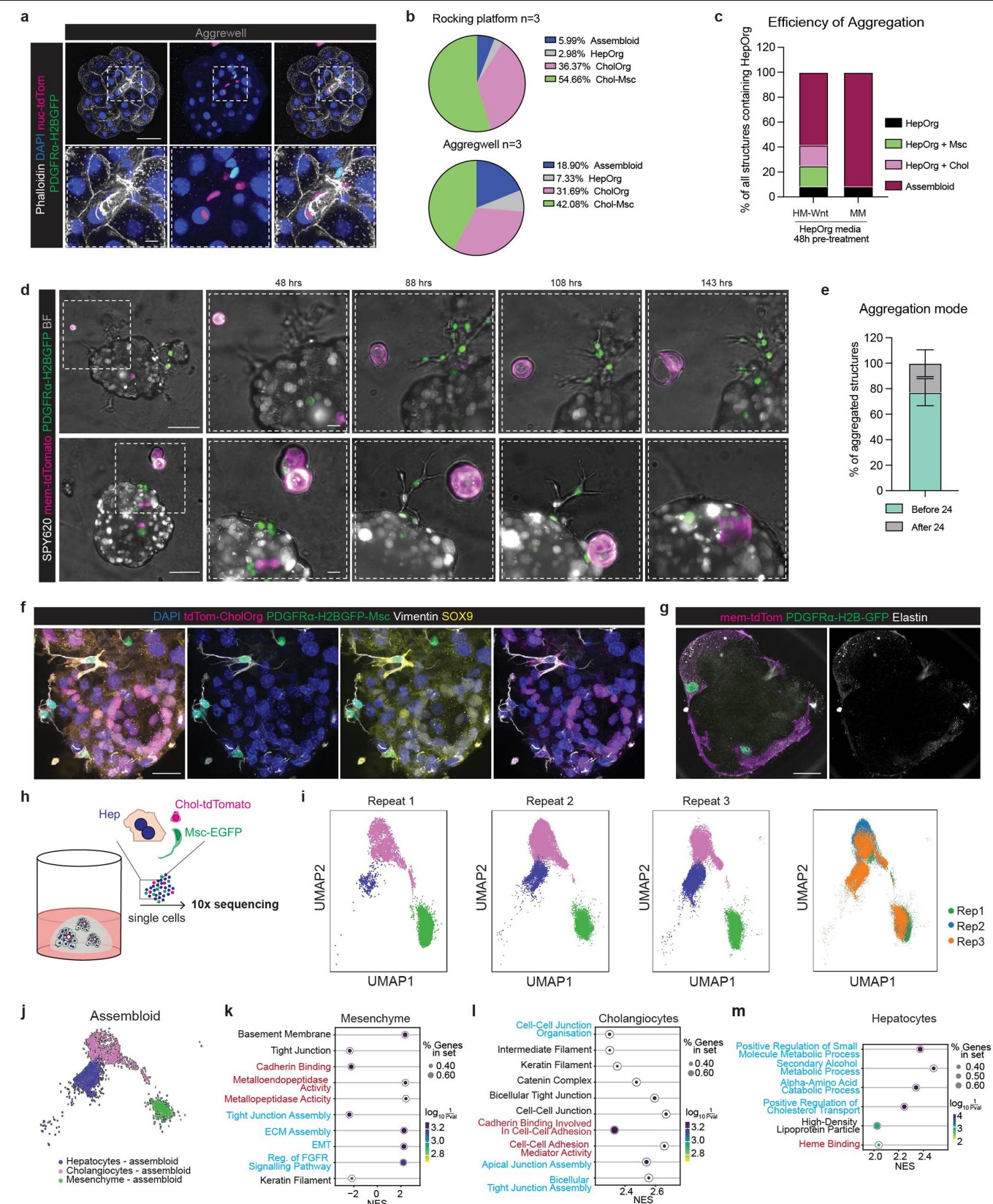

**Extended Data Fig. 7** | See next page for caption.

**Extended Data Fig. 7 | Generation and analysis of periportal assembloids.**
a. Representative immunofluorescence images of periportal assembloids generated using Aggrewell™ method. Msc (PDGFRα-H2BGFP,green), cholangiocytes (nuclear-tdTom, magenta), nuclei (DAPI, blue) and F-actin (Phalloidin, grey) are shown. n = 3 independent experiments. Scale bar, 50 μm, zoom-in, 10 μm. b-c. Aggregation efficiency of periportal assembloids. b. Aggregation efficiency compared to all structures observed (regardless whether they contained one or more cell types). Pie chart represents the mean of n = 3 biological replicates from 3 independent experiment. Results are presented as % of a specific structure respective to the total number of structures. Periportal assembloid with 3 cell types chol, hep, Msc; HepOrg, hepatocyte organoids only; Chol-Org, cholangiocyte organoid only; Chol-Msc, cholangiocyte-Msc organoid. c. Aggregation efficiency comparing conditions where HepOrg had been pre-conditioned for 48hrs prior to aggregation with the co-culture medium MM (MM) or not (HM-Wnt). Graph represents mean n = 2 biological replicates. d. Two examples of assembloid formation. Still images from time-lapse imaging analysis of periportal assembloids composed of hepatocytes, mesenchyme (nuc-GFP, green) and cholangiocytes (mem-tdTomato, magenta). Nuclei are stained with SPY620 (grey). Scale bar, 100 μm, zoom-in, 20 μm. e. Aggregation mode representing assembly before or after seeding in Matrigel. Graph represents mean ± SEM of n = 3 biological replicates from n = 3 independent experiments. f. Representative confocal images (n = 2) of periportal assembloids stained for cholangiocyte (SOX9) and mesenchymal (vimentin) markers. Nuclei are stained with DAPI (blue). Scale bar, 30 μm. g. Representative confocal images of periportal assembloids stained for portal fibroblast marker Elastin (white) marker, in combination with PDGFRα-H2BGFP endogenous signal (GFP), and with Msc membranes visualised by membrane-tdTom. (n = 7 replicates from n = 3 independent experiments). Scale bar, 50 μm. h. Schematic representation of the experimental set up. Cultures were collected at day 7 after assembly and submitted for scRNAseq analysis. i. UMAP representing three biological replicates, each visualising the proportion of mesenchymal (green), cholangiocyte (magenta) and hepatocyte (blue) cells. j. UMAP combining 3 biological replicates of scRNAseq assembloid datasets. k-m. GSEA against GO_Biological_Process_2023 (Cyan), GO_Cellular_Component_2023 (Black) and GO_Molecular_Function_2023 (Red) databases for each of the cell types in assembloids: mesenchyme (k), cholangiocytes (l) and hepatocytes (m), *vs* the other 2 cell types. NES, normalized enrichment score.

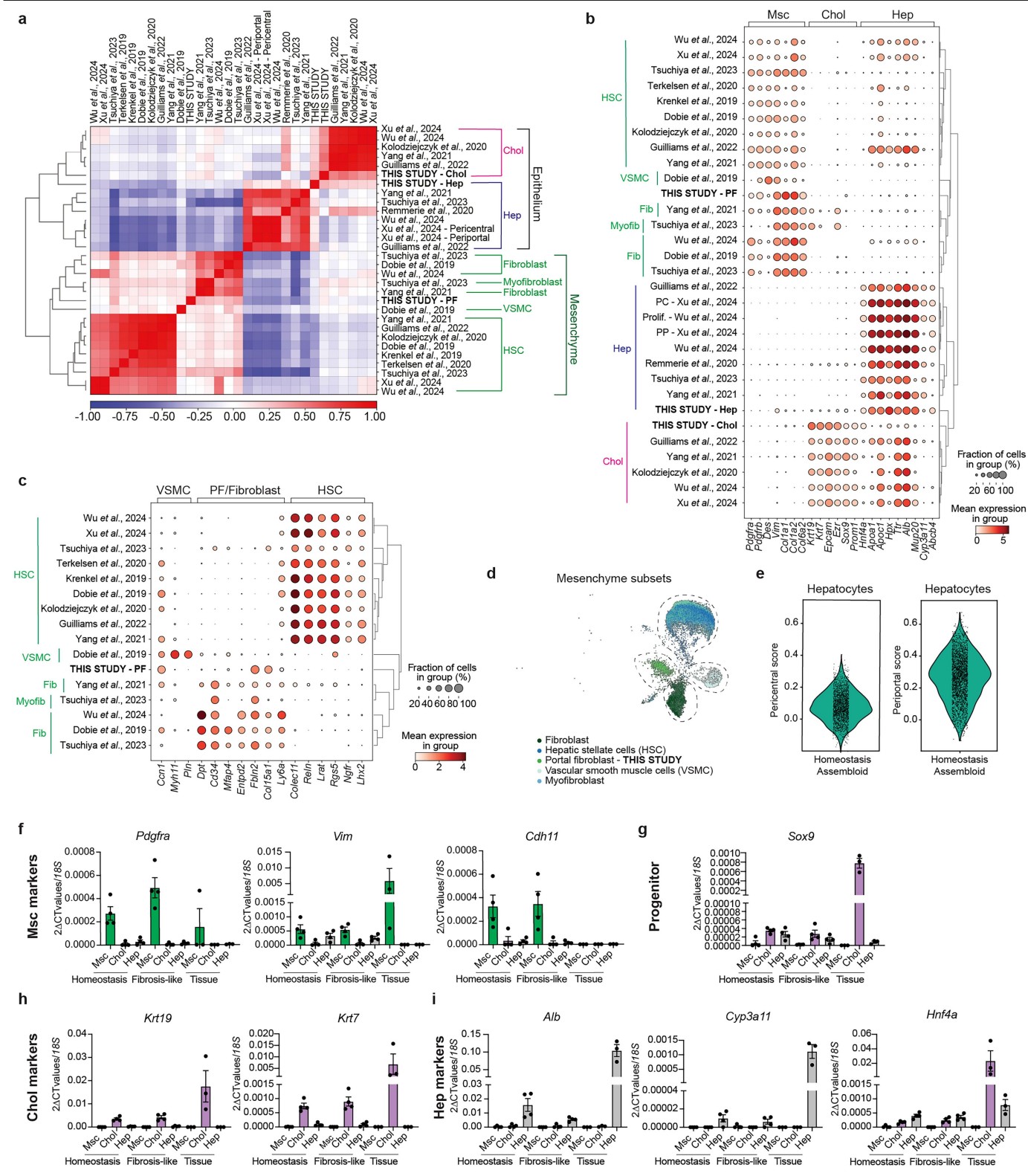

**Extended Data Fig. 8** | See next page for caption.

**Extended Data Fig. 8 | scRNAseq analysis indicates that periportal assembloids resemble in vivo liver tissue.** a. Correlation analysis of cells from healthy liver tissue datasets with assembloids (this study). b. Dot plot showing hepatocyte (Hep), cholangiocyte (Chol) and mesenchyme (Msc) markers for assembloids (THIS STUDY) and liver atlases. Note similarity between assembloid and liver tissue datasets. c. Dot plot showing hepatic stellate cells (HSC), portal fibroblasts (PFs) and vascular smooth muscle cells (VSMC) gene expression analysis of mesenchymal cell types present in liver tissue datasets compared to mesenchyme in assembloids (THIS STUDY). d. UMAP showing the mesenchymal subtypes in the different datasets: hepatic stellate cells (HSC), portal fibroblasts (PFs) and vascular smooth muscle cells (VSMC), compared to mesenchyme in assembloids (THIS STUDY). Note that the mesenchyme in assembloids clusters next to published portal fibroblasts mesenchyme. e. Zonation score calculated from scRNAseq data of assembloids shows that hepatocytes from homeostasis assembloids are on average more periportally zonated, reminiscent of hepatocytes from HepOrg grown in MM media. f-i. Analysis of Msc, cholangiocytes (Chol) and hepatocytes (Hep) cells FACS-sorted from assembloids and compared to freshly isolated cells from liver tissue. qRT-PCR expression analysis of selected marker genes for mesenchyme (f), cholangiocyte and progenitor (g-h) and hepatocyte (i) identity. Graph represents mean ± SEM from n = 4 (assembloid cells) or n = 3 (tissue cells) from 3 independent experiments.

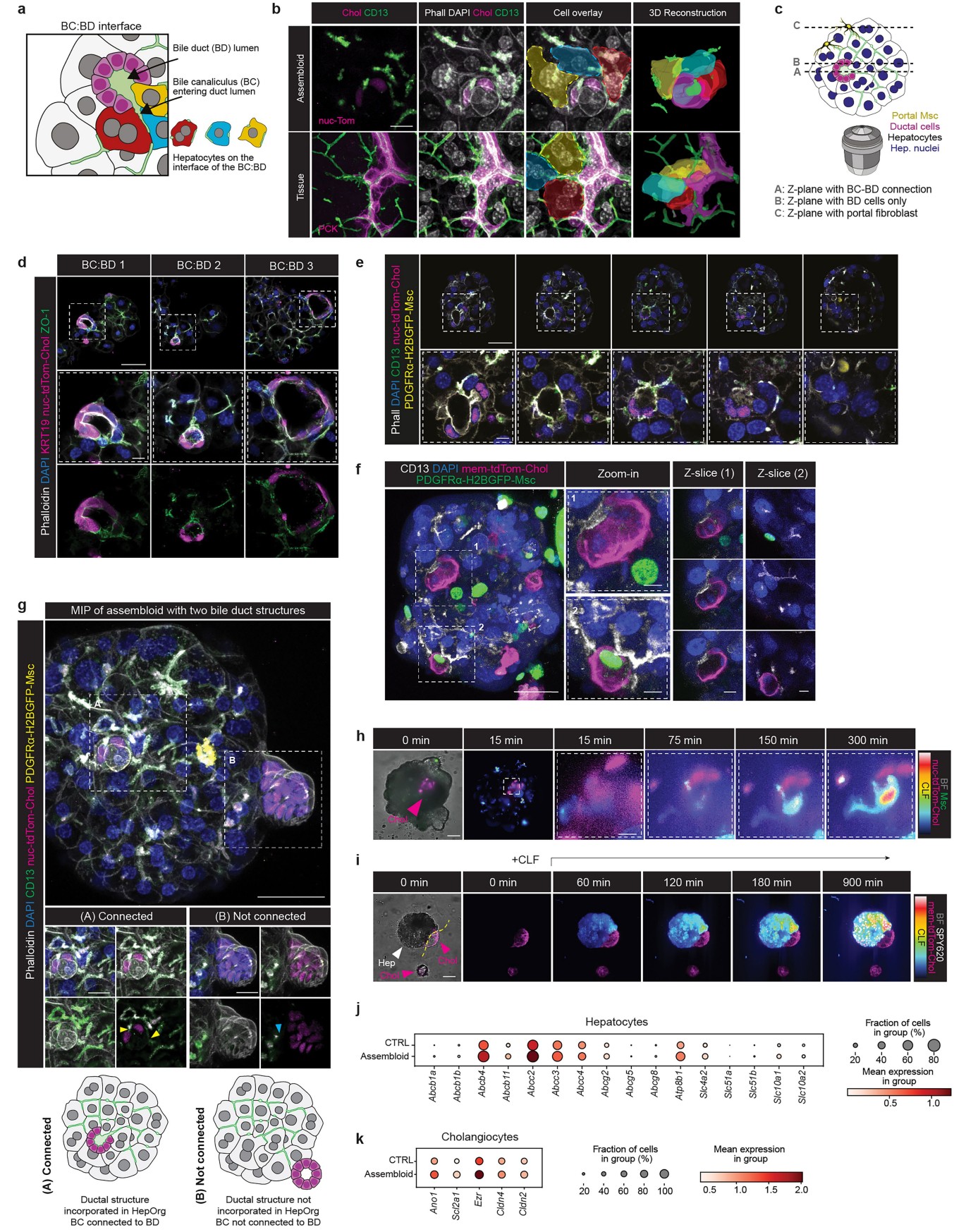

**Extended Data Fig. 9** | See next page for caption.

**Extended Data Fig. 9 | Periportal assembloids recapitulate in vivo tissue architecture and functional connection between BC from hepatocytes and bile duct cells.** a. Schematic representation of the interface between hepatocyte bile canaliculi (BC) and lumen of cholangiocyte bile duct (BD). b. Immunofluorescence staining and image reconstruction of hepatocyte-bile duct interface in assembloid (top panel), and liver tissue (bottom panel). Representative images show how the bile canaliculi (CD13, green) from hepatocytes enters into the lumen of bile duct (nuc-tdTomato, PCK, magenta). Right, 3D reconstruction where hepatocytes (red, yellow, cyan) whose bile canaliculi (green) enter the bile duct lumen (magenta) are visualised. Nuclei are counterstained with DAPI (grey). n = 5 biological experiments. Scale bar, 10 μm. c. Schematic of the imaging set up to visualise integration of the BC-BD connection with portal Msc cells. d. Immunofluorescence image of one assembloid containing three physical connections (BC:BD_1, BC:BD_2, BC:BD_3) between hepatocytes and bile duct. Connections are visualized using zonula occludens (ZO-1, green). Nuclei are counterstained with DAPI (blue). n = 6 independent experiments. Scale bar, 50 μm; zoom-in,10 μm. e. Consecutive sections of assembloids containing hepatocytes, cholangiocytes (nuc-tdTom-Chol, magenta) and portal mesenchyme (PDGFRα-H2BGFP-Msc, yellow). Left-to-right, ascending Z-stacks show 2 physical connections between bile canaliculi (CD13, green) from hepatocytes and a single bile duct lumen. Note that portal mesenchyme cells (yellow) are located in close proximity to cholangiocytes (magenta), recapitulating the architectural arrangement of the in vivo tissue (compare to Fig. 1a). Representative image of an entire assembloid (top), and detail (bottom). Nuclei is stained with DAPI (blue) and F-actin with Phalloidin (Phall, grey). n = 3 biological experiments. Scale bar, 50 μm; zoom-in, 10 μm. f. Representative immunofluorescence image (n > 3) of an assembloid containing hepatocytes, cholangiocytes (mem-tdTomato, magenta), and portal mesenchyme (PDGFRα-H2BGFP, green). Zoom-in, detail of two independent connections (1,2) between bile ducts and bile canaliculi (BC). Right, consecutive Z-stacks of Zoom-in number 1 (left) and Zoom-in number 2 (right) showing the BC entering the BD (Z-slice panels: (1) 0.79 μm, 3.95 μm and 5.53 μm; (2) 26.07 μm, 37.92 μm and 57.67 μm). Nuclei are counterstained with DAPI (blue). Scale bar, 50 μm; zoom-in,10 μm. g. Maximum intensity projection (MIP) of an assembloid with two bile duct structures, one embedded in the assembloid and connecting to the bile canaliculi network (A, yellow arrowhead) and one outside of the assembloid and not connected to the bile canaliculi (B, cyan arrowhead). Bile canaliculi (CD13, green), actin (Phalloidin, grey), nuclei (DAPI, blue) are stained, and Msc and cholangiocytes are visualised using the endogenous fluorescence from PDGFRα-H2BGFP (yellow) and nuc-tdTom (magenta), respectively. n = 3 independent experiments with n = 10 total biological replicates. Scale bar, 50 μm; zoom-in, 10 μm. h. Live imaging analysis of the uptake and flow of bile acid analogue (CLF, Fire LUT) from bile canaliculi into the lumen of bile ducts shows functional bile canaliculi-bile duct connection, n = 3. Scale bar, 50 μm. i. CLF (Fire LUT) uptake is not observed in structures with aberrant architecture where cholangiocytes (magenta, mem-tdTomato, magenta arrowhead) are not embedded in the organoid (white arrowhead), n = 3. Scale bar, 50 μm; zoom-in, 20 μm. j-k. Dot plots of scRNA gene expression for bile acid transporters (j) and cholangiocyte apical markers (k) indicate that hepatocytes (j) and cholangiocytes (k) functional marker expression are improved upon assembloid culture.

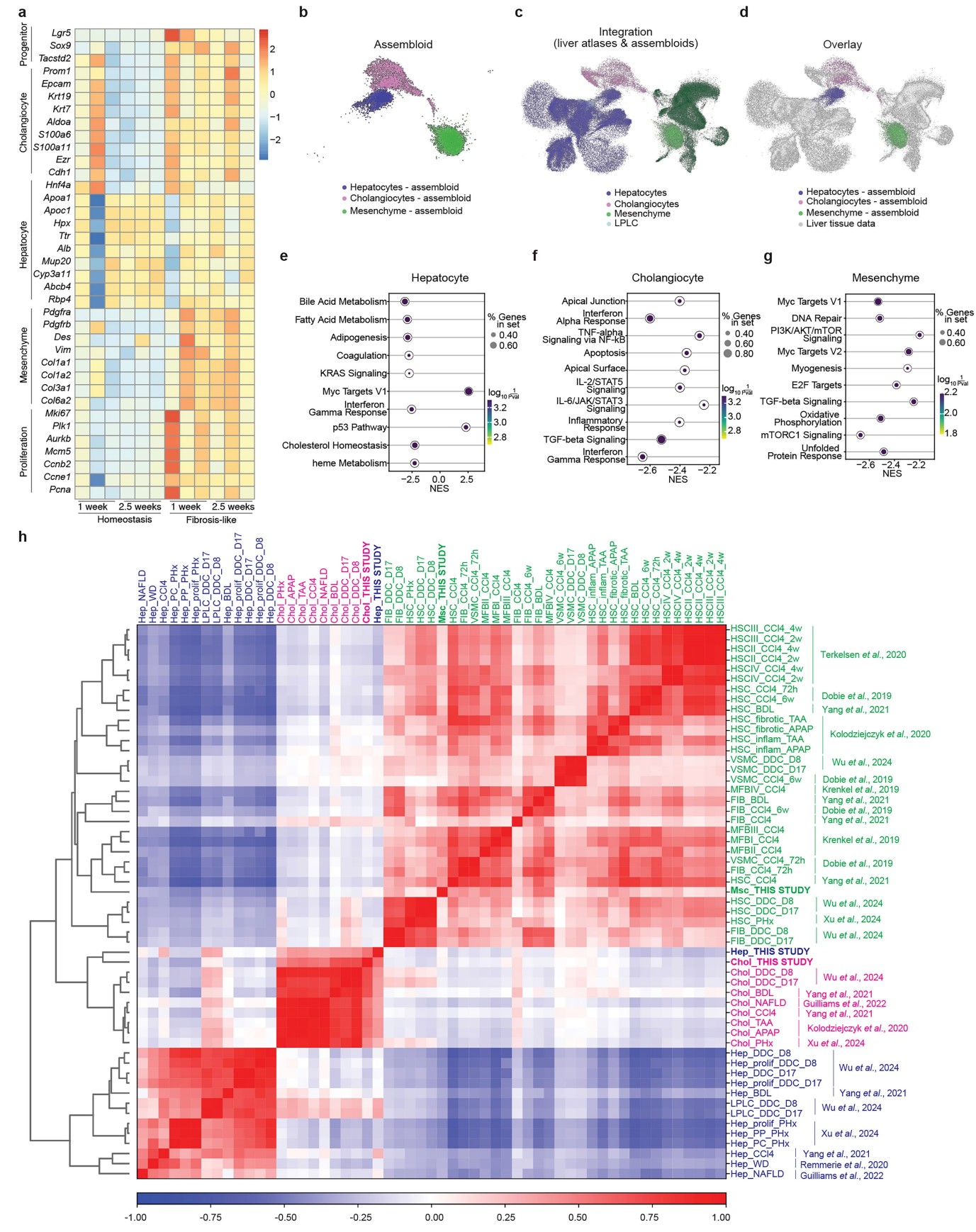

**Extended Data Fig. 10** | See next page for caption.

**Extended Data Fig. 10 | Fibrotic-like assembloids resemble in vivo fibrosis models.** a. Bulk RNA sequencing analysis of homeostasis and fibrotic-like assembloids after 7 days or 2.5 weeks of culture. Heatmap represents the log10(TPM + 1) (transcripts per million) values from the RNAseq for the indicated genes, Z-scored across each gene (n = 3 biological replicates). b. UMAP representation of fibrosis-like assembloids. c. UMAP representation of liver damage models and assembloid integration; green – Msc, magenta – cholangiocytes, blue – hepatocytes, cyan – liver progenitor-like cells (LPLC). d. UMAP representation of the liver damage model datasets (grey) compared to fibrotic-like assembloids (colour). e. GSEA of fibrosis-like and homeostasis from hepatocytes in assembloids (MSigDB_Hallmark_2020). NES, normalized enrichment score. f. GSEA of fibrosis-like and homeostasis from cholangiocytes in assembloids (MSigDB_Hallmark_2020). NES, normalized enrichment score. g. GSEA of fibrosis-like and homeostasis from mesenchyme in assembloids (MSigDB_Hallmark_2020). NES, normalized enrichment score. h. Correlation analysis of various damage models comparing liver tissue datasets with fibrosis-like assembloids (THIS STUDY).

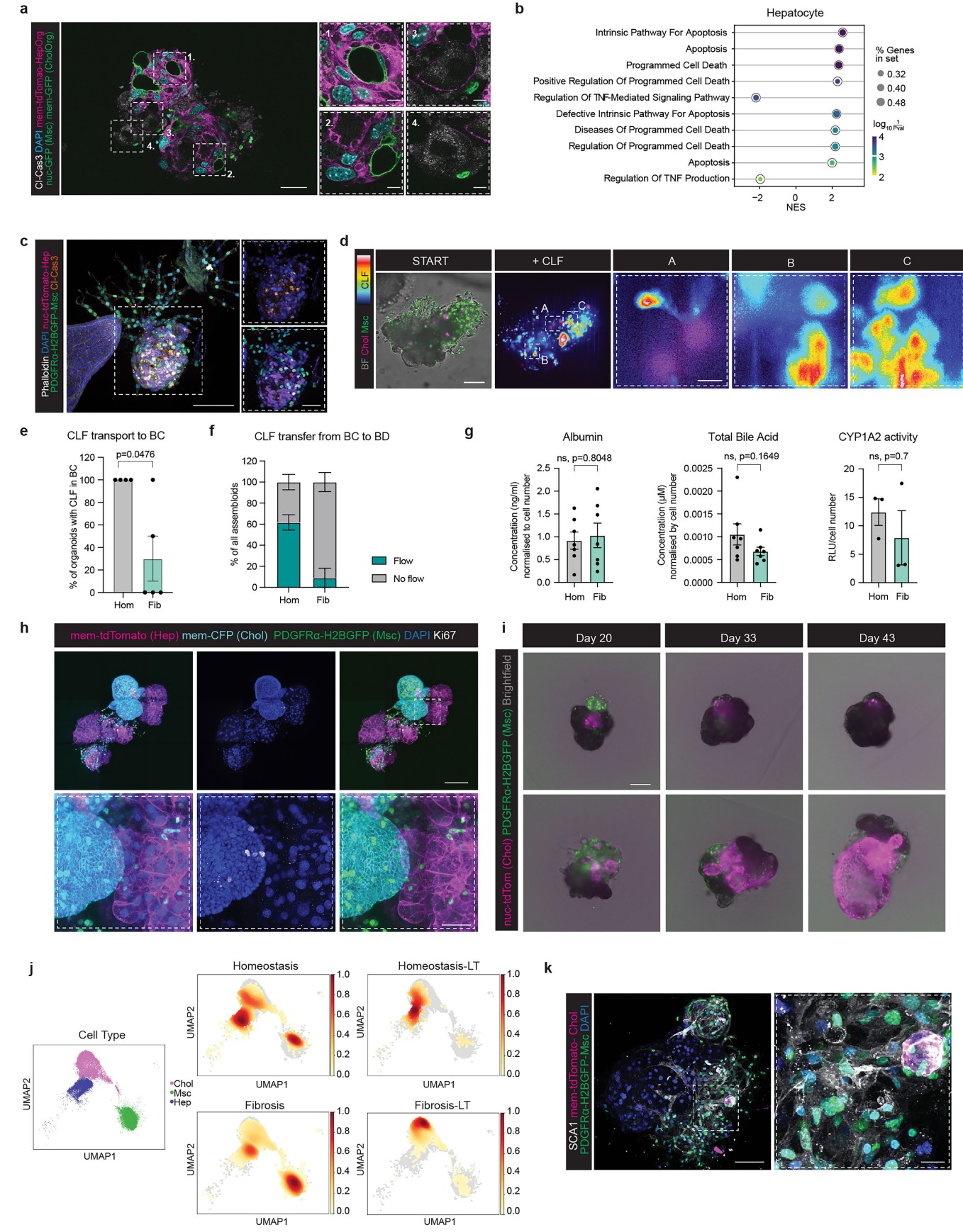

**Extended Data Fig. 11 |** See next page for caption.

**Extended Data Fig. 11 | Fibrotic-like assembloids exhibit several features of biliary fibrosis.** a. Immunofluorescence images of hepatocytes (magenta) contacted by mesenchyme (nuclear GFP, green) expressing cleaved caspase 3 (grey) and lacking the DNA staining (cyan). n > 3 independent experiments. Scale bar, 50 µm; zoom-in, 10 µm. b. GSEA of fibrosis-like versus homeostasis hepatocytes in assembloids, showing all significant terms relating to cell death. c. Freshly sorted mesenchyme also induces apoptosis in fibrotic-like ratio. Cells were stained for cleaved caspase 3 (orange), Phalloidin (membranes, white) and DAPI (nuclei, blue). Hepatocytes (nuc-tdTomato, magenta) and mesenchyme (PDGFRα-H2BGFP, green) were visualized using their endogenous fluorescent gene expression. n = 2 independent experiments. Scale bar, 100 µm; zoom-in, 50 µm. d. Live imaging of the CLF bile acid analogue uptake in fibrotic-like assembloids. n = 3 independent experiments. Scale bar, 100 µm; zoom-in, 10 µm. e. Quantification of the CLF bile acid analogue uptake and transport to bile canaliculi (BC) in assembloid structures containing 3 cell types in homeostatic (n = 4) or fibrotic-like (n = 4) conditions; Results are presented as % of organoids where CLF is detected in bile canaliculi. Graph represents mean ± SEM. Mann-Whitney test, two-tailed. f. Percentage of assembloids, where the transport of CLF from bile canaliculi (BC) to bile duct (BD) was observed. Graph represents mean ± SEM n≥3 biological replicates. g. Albumin secretion, total bile acid (n = 7) and cytochrome activity (CYP1A2, n = 3) measurements from homeostatic (Hom) or fibrotic-like (Fib) assembloids. Graph represents mean ± SEM from 3 independent experiments. Dot, biological replicate; Mann-Whitney test, two-tailed. h. Immunofluorescence images of periportal assembloids stained for proliferation marker Ki67 (white) and nuclei marker DAPI (blue), with endogenous expression of membrane CFP (mem-CFP, cyan, cholangiocytes), nuclear GFP (PDGFRα-H2BGFP, green, Msc) and membrane dTomato (mem-tdTomato, magenta, hepatocytes). n = 3 independent experiments. Scale bar, 200 µm; zoom-in, 150 µm. i. Long term culture (2.5 weeks) of periportal assembloids shows cholangiocyte expansion from nTdTom cholangiocytes (magenta). Msc (green) and brightfield (grey) are also shown. n = 3 independent experiments. Scale bar, 100 µm. j. UMAP representation of cell proportions in scRNAseq data of homeostasis and fibrosis-like assembloids, from 1 week or 2.5 week (long-term, LT) culture, showing expansion of cholangiocytes in fibrosis-like long-term condition. n = 3 independent experiments. k. Staining of fibrosis-like assembloid for portal fibroblast marker SCA1 (white) as well as all mesenchyme (PDGFRα-H2BGFP, green), cholangiocytes (mem-tdTomato, magenta) and nuclei (DAPI, blue), where cell borders are outlined by phalloidin staining (white). Scale bar, 100 µm; zoom-in, 20 µm.

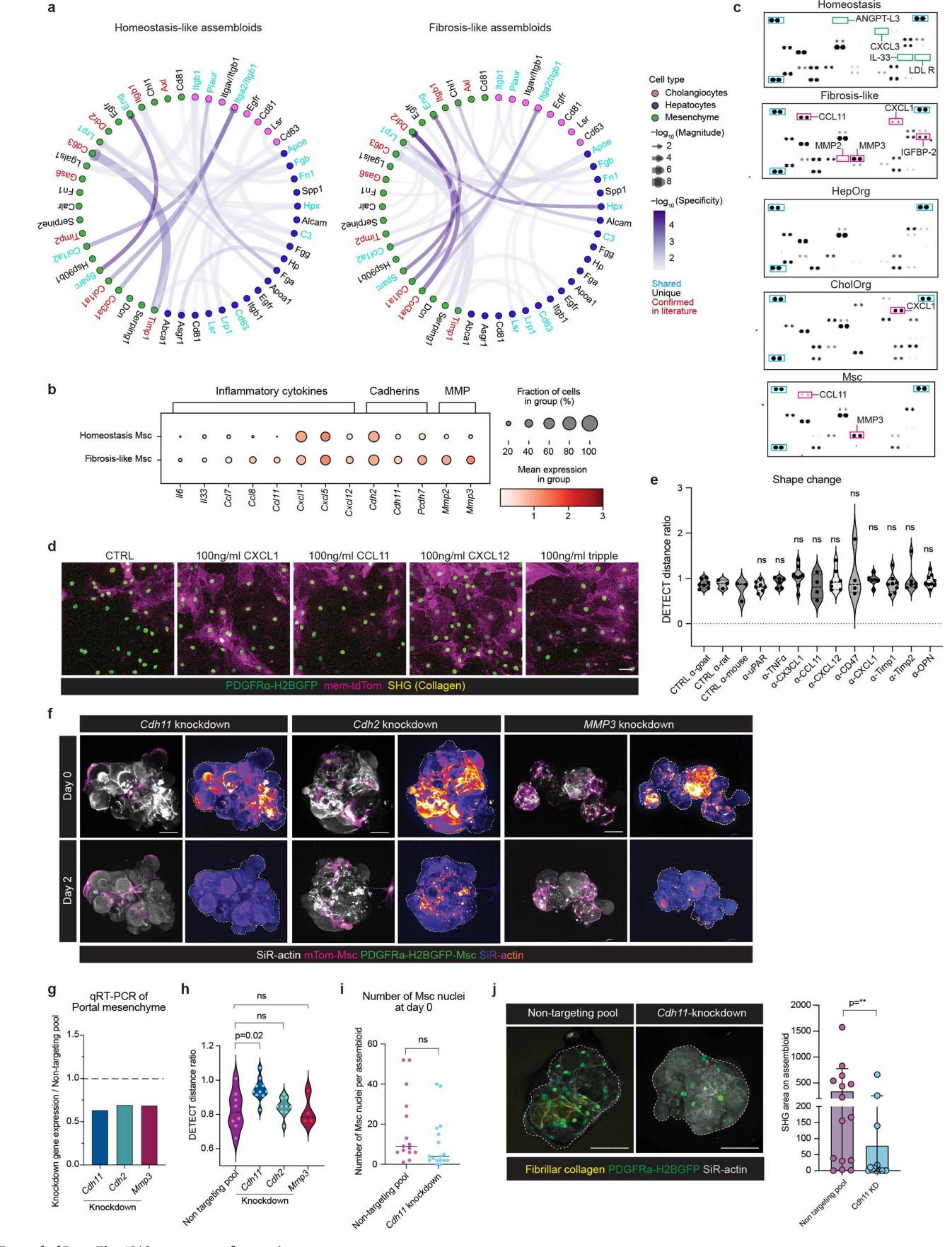

**Extended Data Fig. 12 | Assembloids as a tool to investigate cell-autonomous mechanisms in fibrosis.** a. Circular plots representing the inferred top 30 cell-cell interaction in fibrosis-like and homeostasis-like assembloids. b. Dot plot of genes upregulated in fibrotic-like *vs* homeostatic mesenchyme. c. Cytokine array of supernatant from homeostasis, and fibrotic-like assembloid culture and from hepatocyte (HepOrg) and cholangiocyte (CholOrg) organoids and portal mesenchyme, grown individually in MM media. n = 2 independent experiments. Blue boxes, hot spots showing positive control and orientation markers; magenta boxes, cytokines that are only expressed in fibrotic-like assembloids; note CXCL1 expression in cholangiocytes, while MMP3 and CCL11 are expressed by the mesenchyme. d. Immunofluorescence images for collagen deposition (SHG) in naïve mesenchyme following exposure to inflammatory cytokines for 7 days. Note that no collagen is detected. Triple denotes a combination of 100 ng/ml of each of CXCL1, CCL11 and CXCL12. Representative images of n = 3 biological replicates. Scale bar, 50 µm; zoom-in, 50 µm. e. DETECT analysis of assembloids treated with blocking antibodies against specified proteins, or matched-species control; violin plot shows DETECT distance ratios; Mann-Whitney test. Dot, individual organoids from n = 3 biological replicates. f. Still images from a live imaging analysis of fibrotic-like assembloids containing Msc (nuc-GFP, green, and membrane-tdTomato, magenta) transfected with siRNA against specified genes prior to assembly. Representative images from n = 2 independent biological replicates at day of seeding (top panel) and day 2 of culture (bottom panel). SiR-actin (membrane, grey). Fire LUT images (right panels) show examples of assembloid border segmentation based on maximum intensity projection images. Scale bar, 50 µm. g. Gene knock-down efficiency; Graph represent normalized expression of the genes in the siRNA treatment relative to the non-targeting control (dotted line). h. Violin plots showing median and quartiles of the DETECT distance ratio (d1/d0) for non-targeting pool, *Cdh11*, *Cdh2* and *Mmp3* siRNA. Dots represent values of the DETECT distance ratio of individual assembloids from n = 2 biological replicates. Kruskall-Wallis test, followed by Dunn's multiple comparisons test. i. Graph represents median from the total number of Msc cells (segmented as GFP nuclei) per assembloid at day0 (assembloid seeding). n = 2 independent experiments. Non-targeting pool, n = 15; *Cdh11-KD*, n = 17. (Mann-Whitney two tailed test, ns, not significant, p = 0,0669). j. Images show the entire assembloid presented as detail in Fig. 5d showing fibrillar collagen deposition as analysed by SHG (second harmonic generation microscopy analysis, yellow) in control (left) and *Cdh11-KD* Msc assembloids (right). SiR-actin (membrane, grey) and Msc nuclei (green). Right, quantification of SHG; Graph represents the area ± S.D. of SHG-positive signal over the area of whole assembloid, for the non-targeting pool control and the *Cdh11-KD* group at day3 (assembloid seeding). Non-targeting pool, n = 15; *Cdh11-KD*, n = 14. Mann-Whitney two tailed test, p = 0.0052. Scale bar, 100 µm.

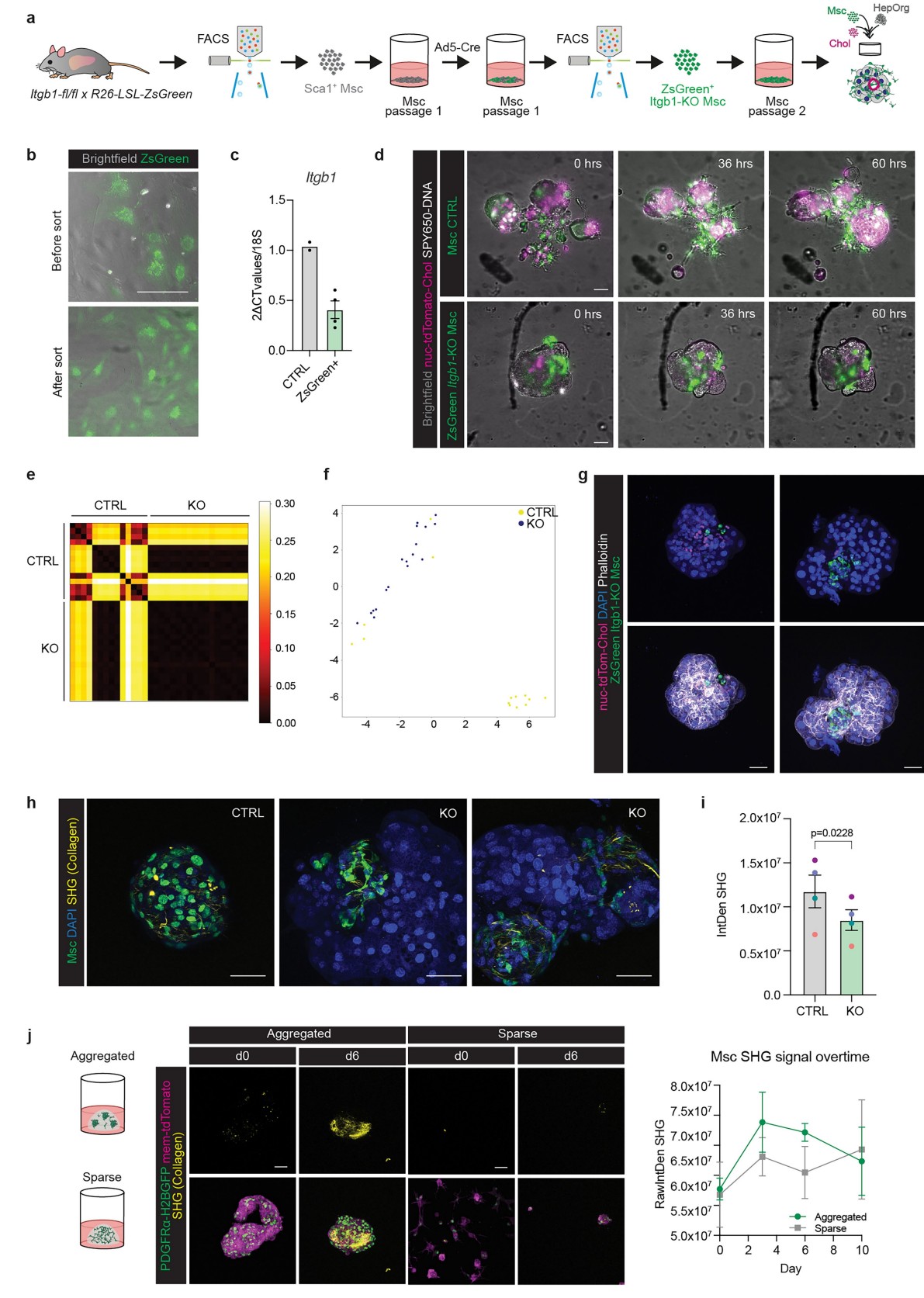

**Extended Data Fig. 13 |** See next page for caption.

**Extended Data Fig. 13 | Knockout of *Itgb1* in Msc results in reduction of fibrotic-like phenotype in assembloids.** a. Schematic of experimental set up for a-i. Reproduced from ref. 20 (CC BY 4.0). b. Representative pictures of *Itgb1-KO* cells expressing ZsGreen before and after FACS sorting. n = 2 independent experiments. Scale bar, 200 μm. c. Validation of *Itgb1-KO* (n = 4) compared to control (n = 2) by qRT-PCR against *Itgb1* gene; graph represents mean and ± SEM when n > 3. d. Live imaging analysis of fibrotic-like assembloids formed with CTRL Msc (top) and *Itgb1-KO* Msc (bottom), showing that *Itgb1-KO* prevents collapse of the structures. Representative images from n = 2 biological replicates are shown. Scale bar, 50 μm. e-f. DETECT analysis of the shape change over time. n = 2 independent experiments. (e) Heatmap of the DETECT distance metric for the DETECT results of *Itgb1-KO* Msc and CTRL-Msc containing fibrotic-like assembloids. (f) Visualisation of morphological variations of *Itgb1-KO* Msc and CTRL-Msc containing fibrotic-like assembloids using t-SNE based on DETECT calculations that integrate all time points. g. Immunofluorescence images of fibrotic-like assembloids formed by Itgb1-KO Msc (green). Representative images from n = 2 biological replicates are shown. Cholangiocytes (tdTom-nuclei, magenta) are shown, together with actin (phalloidin, grey) and nuclei (DAPI, blue) staining. Note intact organoid structure and intact hepatocyte nuclei. Scale bar, 50 μm. h. Fibrotic-like assembloids containing *Itgb1*-KO Msc (green) deposit less collagen, as visualised by second harmonic generation (SHG) imaging (yellow). Representative images are shown. Scale bar, 50 μm. i. Quantification of fibrous collagen deposition (SHG) from (h). Graph represents mean ± SEM of n = 4 biological replicates from n = 2 independent experiments. Dot, biological replicate; paired t-test, two-tailed. j. Representative images (left) and quantification (right) of collagen deposition as visualised by second harmonic generation (SHG, yellow) in cultures where 2500 Msc cells were seeded in sparse or confluent/aggregated condition (n = 4 biological replicates). Graph represents mean and ± SEM. Scale bar, 50 μm.

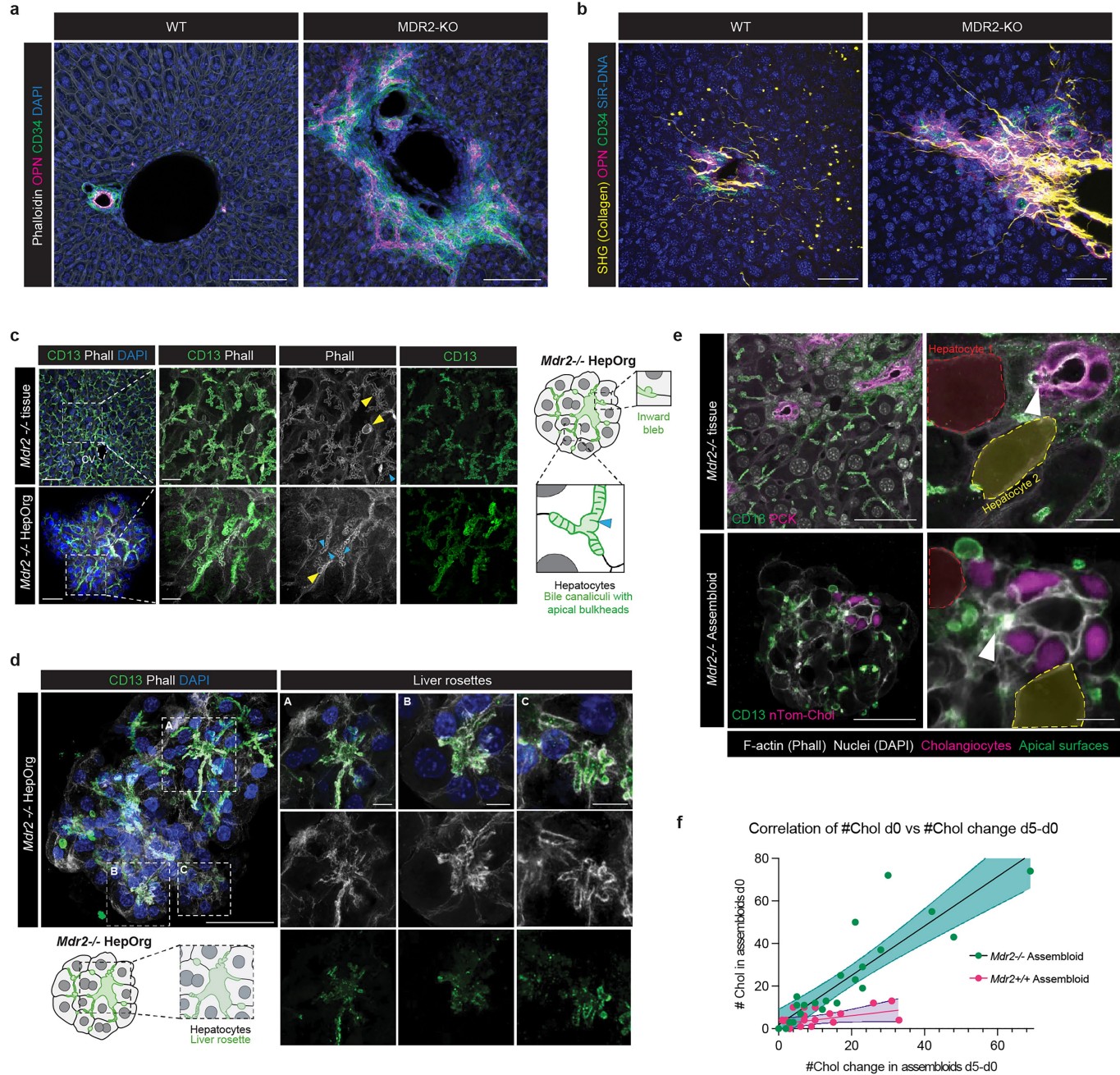

**Extended Data Fig. 14 | Mouse *MDR2*-KO hepatocyte organoids and *MDR2*-KO-hepatocyte periportal assembloids recapitulate aspects of cholestatic liver disease.** a. Immunofluorescence staining of WT control and *MDR2*-KO tissue samples showing increased areas stained for osteopontin cholangiocyte marker (OPN, magenta) and CD34 marker of portal fibroblast (green), indicative of cholangiocyte and mesenchyme expansion in this model of biliary fibrosis; nuclei (DAPI, blue) and actin cytoskeleton (Phalloidin, white) are also stained. n = 3 independent experiments. Scale bar, 100 μm. b. Immunofluorescence staining of WT control and *MDR2*-KO tissue samples showing increased areas stained for osteopontin cholangiocyte marker (OPN, magenta) and CD34 marker of portal fibroblast (green), nuclei (SiR-DNA, blue) and collagen (SHG, yellow). Scale bar, 50 μm. c. Left, Immunofluorescence staining of bile canaliculi (CD13, green) and F-actin (Phalloidin, grey) in *Mdr2-/-* liver tissue (top panel) and *Mdr2-/-* HepOrg (bottom panel). Note the similarity between the bile canaliculi morphology of the HepOrg and the tissue. Cyan arrowheads, apical bulkheads. Yellow arrowheads, inward blebs. Representative images from n = 2 (tissue), n = 3 (HepOrg) independent experiments are shown. Scale bar, 50 μm; zoom-in, 20 μm. Right, scheme showing bile canaliculi features of apical bulkheads. d. Immunofluorescence staining of bile canaliculi (CD13, green) and F-actin (Phalloidin, grey) in *Mdr2-/-* HepOrg cultures, showing emergence of hepatocyte rosettes. Representative images from n = 3 independent experiments are shown. Scale bar, 50 μm; zoom-in, 20 μm. Scheme showing bile canaliculi features of rosettes. e. Immunofluorescence staining of the interface and connection between bile canaliculi from hepatocytes and the lumen from bile duct in *Mdr2-/-* liver tissue (top) and assembloids (bottom). Hepatocytes (red, yellow) whose bile canaliculi (green) enter the bile duct lumen (magenta) are visualised. Interface between BC and BD in *Mdr2-/-* assembloid (bottom) and tissue (top) are pointed to with white arrowheads. Representative images from n = 2 biological replicates from n = 2 independent experiments are shown. Scale bar, 50 μm; zoom-in, 10 μm. f. Graph representing correlation between the initial cholangiocyte (chol) number at day 0 (d0) and day 5 (d5-d0) from n = 2 biological replicates from n = 2 independent experiments. Dots represent individual assembloids. Slopes represent simple linear regression with coloured area denoting standard error. $R^2$ = 0,759 for *Mdr2-/-* portal assembloids slope p-value p < 0,0001, and $R^2$ = 0,1423 for *Mdr2 +/+* portal assembloids, p = 0,0760.

# Reporting Summary

## Statistics

For all statistical analyses, confirm that the following items are present in the figure legend, table legend, main text, or Methods section.

| n/a | Confirmed | |
|---|---|---|
| ☐ | ☒ | The exact sample size (*n*) for each experimental group/condition, given as a discrete number and unit of measurement |
| ☒ | ☐ | A statement on whether measurements were taken from distinct samples or whether the same sample was measured repeatedly |
| ☐ | ☒ | The statistical test(s) used AND whether they are one- or two-sided<br>*Only common tests should be described solely by name; describe more complex techniques in the Methods section.* |
| ☒ | ☐ | A description of all covariates tested |
| ☐ | ☒ | A description of any assumptions or corrections, such as tests of normality and adjustment for multiple comparisons |
| ☐ | ☒ | A full description of the statistical parameters including central tendency (e.g. means) or other basic estimates (e.g. regression coefficient) AND variation (e.g. standard deviation) or associated estimates of uncertainty (e.g. confidence intervals) |
| ☐ | ☒ | For null hypothesis testing, the test statistic (e.g. *F*, *t*, *r*) with confidence intervals, effect sizes, degrees of freedom and *P* value noted<br>*Give P values as exact values whenever suitable.* |
| ☒ | ☐ | For Bayesian analysis, information on the choice of priors and Markov chain Monte Carlo settings |
| ☒ | ☐ | For hierarchical and complex designs, identification of the appropriate level for tests and full reporting of outcomes |
| ☐ | ☒ | Estimates of effect sizes (e.g. Cohen's *d*, Pearson's *r*), indicating how they were calculated |

*Our web collection on statistics for biologists contains articles on many of the points above.*

## Software and code

Policy information about availability of computer code

| | |
|---|---|
| Data collection | The following software was used for data collection: scRNAseq data - NovaSeq S4 v1.5 4XP (200cyc); Confocal Imaging: Zeiss ZEN 3.6 celldiscoverer (version 3.6.095.03000); Zen 2011 SP7 FP3 (black), version 14.0.25.201 (780 Zeiss inverted microscope); Zen Black 2011 SP7 FP3 64bit; version 14.0.27.201 (Zeiss 780 upright microscope); ZEN 2.3 SP1 FP3 (black) v14.0.25.201 (Zeiss 880 inverted microscope); ZEN 2.3 SP1 FP3 (black) v14.0.27.201 (Zeiss 880 upright microscope); Timelapse imaging - Cell Voyager Measurement System Release R1.17.05 (Yokogawa CV7000 MPO); Lightsheet imaging - Viventis Microscope and Environment v.2.0.0.2 ; Brightfield imaging - Leica Application Suite v.4.6.0 (Leica DMIL LED microscope) or v.4.13.0 (Leica M80 microscope) ; qPCR - LightCycler 96 software (Version: 1.1.0.1320) or QuantStudio7 software (QS 6/7 Pro v1.8.1); cytokine array - iBright Analysis Software v.5.2.0; Albumin ELISA and bile acid assay - Perkin Elmer Envision 2104 EnVision Manager v.1.13.3009.1401, Flow cytometry - BD FACSDiva v.8.0.2, Sony Cell Sorter Software v.3.1.2. |
| Data analysis | Confocal and light sheet imaging data was analysed with FIJI v.2.14.0/1.54f, and segmented using Arivis Vision 4D (Version: 4.1.0. Build: 16702. 20200324) or Motion Tracking v. 8.100.6 (http://motiontracking.mpi-cbg.de). Data analysis of bile canaliculi was performed using a custom script in FIJI (https://git.mpi-cbg.de/huch_lab/assembloid-paper ) . For Yokogawa timelapse imaging, data was assembled using two FIJI macros "Macro CV7000 Make Field Montages" and "Macro CV7000 Make Timelapse Movies", which are publicly available (https://github.com/stoeter/Fiji-Tools-for-HCS). Data presentation and statistical analysis performed with GraphPad Prism v10 (v10.0.2 and v.10.0.0) for non-sequencing data. For Flow cytometry data, Flowjo v.10.8.1 was used to analyse and visualise the data.<br><br>Code used for data analysis and visualisation of scRNAseq dataset included: Cellbender (0.3.0), bcl2fastq2 (2.20.0), CellRanger v.7.1.0, scanpy v.1.9.2, GSEApy v.1.0.5, plotnine v.0.12.3, R v.4.2.0, ggraph v.2.1.0 and LIANA v.0.1.9, Scrublet (https://github.com/swolock/scrublet). For bulk RNAseq analysis, featureCount (v2.0.6) was used to assigned reads exons, transcripts and CDS. Differential gene expression analysis was performed using the R (4.2.0) package DESeq2 (1.36.0). Gene set enrichment analysis for bulk RNAseq data was performed using the R |

package fgsea (1.22.0).

For manuscripts utilizing custom algorithms or software that are central to the research but not yet described in published literature, software must be made available to editors and reviewers. We strongly encourage code deposition in a community repository (e.g. GitHub). See the Nature Portfolio guidelines for submitting code & software for further information.

## Data

Policy information about availability of data

All manuscripts must include a data availability statement. This statement should provide the following information, where applicable:
- Accession codes, unique identifiers, or web links for publicly available datasets
- A description of any restrictions on data availability
- For clinical datasets or third party data, please ensure that the statement adheres to our policy

The scRNAseq and bulk RNAseq datasets generated during this study are available at Gene Expression Omnibus (GEO, https://www.ncbi.nlm.nih.gov/geo/info/seq.html) under the accession number GSE274971 and GSE274973 . The full lists of bulk RNaseq TPMs and DEG are found in Supplementary Table 2 and Supplementary Table 6, respectively. All GSEA terms are available in Supplementary Table 3 and Supplementary Table 5. All inferred cell-cell interactions are available in Supplementary Supplementary Table 4.

The scRNAseq data was aligned to the GRCm39 (release 109) mouse genome with the 10x's cellranger's (7.1.0) count. Bulk RNAseq data was aligned to the mouse genome GRCm39 release 109 using STAR aligner (2.7.11b). All other images, qPCR and measurement data are presented within the manuscript, and data used to plot graphs is provided as Source Data files for each figure.

For Gene Set Enrichment Analysis (GSEA), the following databases were used: the Whole organoid GSEA between healthy and fibrotic conditions was performed for the MSigDB hallmark (2020), WikiPathways (2019, Mouse), Reactome (2022), Elsevier Pathway Collection and GO Molecular Function (2023) gene sets; Cell type specific GSEA between conditions was performed for the MSigDB hallmark (2020) gene set. Functional identity was assessed by GSEA by comparing each cell type to all other cell types using the GO Molecular Function (2023), GO Biological Process (2023) and GO Cellular Component (2023) gene sets.

## Research involving human participants, their data, or biological material

Policy information about studies with human participants or human data. See also policy information about sex, gender (identity/presentation), and sexual orientation and race, ethnicity and racism.

| | |
|---|---|
| Reporting on sex and gender | Not applicable (N/A) - no human participants, data or biological material was used for this study. |
| Reporting on race, ethnicity, or other socially relevant groupings | N/A |
| Population characteristics | N/A |
| Recruitment | N/A |
| Ethics oversight | N/A |

Note that full information on the approval of the study protocol must also be provided in the manuscript.

# Field-specific reporting

Please select the one below that is the best fit for your research. If you are not sure, read the appropriate sections before making your selection.

☒ Life sciences      ☐ Behavioural & social sciences      ☐ Ecological, evolutionary & environmental sciences

For a reference copy of the document with all sections, see nature.com/documents/nr-reporting-summary-flat.pdf

# Life sciences study design

All studies must disclose on these points even when the disclosure is negative.

| | |
|---|---|
| Sample size | No statistical methods were used to calculate sample size and group size. For cell culture experiments we follow the recommendations from ISSCR guidelines published in Ludwig et al. 2023 (PMID: 37703820). |
| Data exclusions | scRNAseq thresholding involved excluding cells with abnormally high or low transcripts, and based on high mitochondrial gene content as is standard practice for downstream analysis of this sequencing data. We also excluded doublets to improve stringency, as described in the methods. |
| Replication | All experiments were replicated as indicated in the figure legend. |
| Randomization | Experimental procedures always involved processing control and experimental samples in a random order, rather than by condition. |

| | |
|---|---|
| Blinding | We did not perform blinding for experiments, as blinding was considered not to affect the measurement result. Whenever possible, investigators were blinded performing the experimental analysis. |

# Reporting for specific materials, systems and methods

We require information from authors about some types of materials, experimental systems and methods used in many studies. Here, indicate whether each material, system or method listed is relevant to your study. If you are not sure if a list item applies to your research, read the appropriate section before selecting a response.

## Materials & experimental systems

| n/a | Involved in the study |
|---|---|
| ☐ | ☒ Antibodies |
| ☐ | ☒ Eukaryotic cell lines |
| ☒ | ☐ Palaeontology and archaeology |
| ☐ | ☒ Animals and other organisms |
| ☒ | ☐ Clinical data |
| ☒ | ☐ Dual use research of concern |
| ☒ | ☐ Plants |

## Methods

| n/a | Involved in the study |
|---|---|
| ☒ | ☐ ChIP-seq |
| ☐ | ☒ Flow cytometry |
| ☒ | ☐ MRI-based neuroimaging |

## Antibodies

| | |
|---|---|
| Antibodies used | List of antibodies and dyes used in this study.<br>Primary antibodies:<br>Rabbit monoclonal anti-E-cadherin (24E10), Cell Signaling # 3195 RRID:AB_2291471<br>Rat monoclonal anti-E-cadherin (ECCD-2), Thermo Fisher Scientific # 131900 RRID:AB_2533005<br>Rat monoclonal anti-CD13 (ER-BMDM1), Novus # NB100-64843 RRID:AB_959651<br>Rat monoclonal anti-Cytokeratin-19 (TROMA-III), Sigma-Aldrich # MABT913 RRID:AB_2892523<br>Rabbit polyclonal anti-ZO-1, Thermo Fisher Scientific # 40-2200 RRID:AB_2533456<br>Goat polyclonal anti-Albumin, Novus # NB600-41532 RRID:AB_805588<br>Rabbit monoclonal anti-Radixin (C4G7), Cell Signaling # 2636 RRID:AB_2238294<br>Rabbit polyclonal anti-Cytokeratin, Dako # Z0622 RRID:AB_2650434<br>Rabbit monoclonal anti-Hnf4-alfa (EPR16885), Abcam # ab181604 RRID:AB_2890918<br>Rabbit monoclonal anti-SOX9 (EPR14335-78), Abcam # ab185966 RRID:AB_2728660<br>Goat anti-Osteopontin Polyclonal, R&D Systems #  AF808 RRID: AB_2194992<br>Rat anti-CD34 Monoclonal Antibody (RAM34), eBioscience™ ThermoFisher Scientific # 14-0341-85 RRID:AB_467211<br>Rabbit anti-Cleaved Caspase-3 (Asp175) monoclonal (Clone 5A1E), Cell Signaling Technology # 9664 RRID: AB_2070042<br>Rat Purified anti-mouse Ly-6A/E (Sca-1), BioLegend # 108102 RRID:AB_313339<br>Rabbit Polyclonal anti-Transcription Factor Sox-9, Millipore # AB5535 RRID:AB_2239761<br>Mouse monoclonal anti-Vimentin (clone LN-6), Merck # V2258 RRID:AB_261856<br>Rat anti-Ly-6A/E (Sca-1) monoclonal (Clone D7), Super Bright 436, ThermoFisher Scientific # 62-5981-82 RRID: AB_2637287<br>Rat anti- CD326 (EpCAM) monoclonal (Clone G8.8), APC, ThermoFisher Scientific # 17-5791-80 RRID: AB_2734965<br>Rat anti-CD11b monoclonal (Clone M1/70), PE-Cy7, BD Biosciences # 552850 RRID: AB_394491<br>Rat anti-CD31 monoclonal (Clone 390), PE-Cy7, BD Biosciences # 561410 RRID: AB_10612003<br>Rat anti-CD45 monoclonal (Clone 30-F11), PE-Cy7, BD Biosciences # 552848 RRID: AB_394489<br>Rabbit anti-Elastin (polyclonal IgG), Cedarlane Laboratories Limited #CL55041AP RRID: AB_10061195<br>Rabbit polyclonal anti-Glutamine Synthetase (GS), Sigma-Aldrich # G2781 RRID: AB_259853<br>Rabbit polyclonal anti-Cytochrome P450 2E1 (CYP2E1), Abcam # ab28146 RRID: AB_2089985<br>Rabbit anti-Ki67 monoclonal (Clone SP6), ThermoFisher Scientific #RM 9106 S1 RRID:AB_2341197<br><br>Secondary antibodies<br>Donkey anti-Rat IgG (H+L) Highly Cross-Adsorbed Secondary Antibody, Alexa Fluor 488, ThermoFisher Scientific # A-21208 RRID:AB_2535794<br>Donkey anti-Rat IgG (H+L) Highly Cross-Adsorbed Secondary Antibody, Alexa Fluor Plus 555, ThermoFisher Scientific # A48270 RRID:AB_2896336<br>Donkey anti-Rat IgG (H+L) Highly Cross-Adsorbed Secondary Antibody, Alexa Fluor 594, ThermoFisher Scientific # A-21209 RRID:AB_2535795<br>Goat anti-Rat IgG (H+L) Cross-Adsorbed Secondary Antibody, Alexa Fluor 647, ThermoFisher Scientific # A-21247 RRID:AB_141778<br>Goat anti-Rabbit IgG (H+L) Cross-Adsorbed Secondary Antibody, Alexa Fluor 532, ThermoFisher Scientific # A-11009 RRID:AB_2534076<br>Donkey anti-Rabbit IgG (H+L), highly cross-adsorbed, CF™ 633 antibody, Merck # SAB4600132<br>Donkey anti-Rabbit IgG (H+L) Highly Cross-Adsorbed Secondary Antibody, Alexa Fluor 647, ThermoFisher Scientific # A-31573 RRID:AB_2536183<br>Donkey anti-Mouse IgG (H+L) Polyclonal Antibody, Alexa Fluor 647, ThermoFisher Scientific # A-31571 RRID:AB_162542<br>Donkey anti-Goat IgG (H+L) Cross-Adsorbed Secondary Antibody, Alexa Fluor 594, ThermoFisher Scientific # A-11058 RRID:AB_2534105<br>Donkey anti-Goat IgG (H+L) Cross-Adsorbed Secondary Antibody, Alexa Fluor 647, ThermoFisher Scientific # A-21447 RRID:AB_2535864<br>Donkey Anti-Rat IgG (H+L), Highly Cross-Adsorbed, CF 568, Biotium # 20092 RRID:AB_10855000 |

Dyes
SiR-Actin, Spirochrome # SC001
Corning® Cholyl-Lysyl-Fluorescein (CLF), Corning # 451041
SPY620-DNA, Spirochrome # SC401
DAPI, BD Biosciences # BD564907
Phalloidin 647, ThermoFisher Scientific # A22287
Phalloidin 488, ThermoFisher Scientific # A12379
CellTracker™ Green CMFDA Dye (Chloromethylfluorescein diacetate), ThermoFisher Scientific #C2925
16:0-06:0 NBD PC fluorescent lipid (phosphatidylcholine), Avanti Polar lipids, Merck #810130P-1mg
5-CFDA, AM (5-Carboxyfluorescein Diacetate, Acetoxymethyl Ester), ThermoFisher Scientific #C1354
SPY555 DNA, Spirochrome #SC201

Blocking antibodies
Normal Goat IgG Control R&D Systems, Biotechne #AB-108-C RRID:AB_354267
Normal Rat IgG Control (Azide Free), R&D Systems, Biotechne #6-001-F RRID:AB_2616570
Mouse IgG1 Isotype Control, R&D Systems, Biotechne #MAB002 RRID:AB_357344
Mouse uPAR Antibody R&D Systems, Biotechne #AF534-SP RRID:AB_2165351
Human/Mouse TNF-alpha Antibody, R&D Systems, Biotechne #AF410-SP RRID:AB_354479
Mouse CX3CL1/Fractalkine Chemokine Domain Antibody, R&D Systems, Biotechne #AF472-SP RRID:AB_2276839
Mouse CCL11/Eotaxin Antibody, R&D Systems, Biotechne #AF420-SP RRID:AB_354486
Human/Mouse CXCL12/SDF-1 Antibody, R&D Systems, Biotechne #MAB310-SP RRID:AB_2276927
BD Pharmingen™ Purified Rat Anti-Mouse CD47 Clone miap301 unlabelled Antibody, BD Biosciences #555297 RRID:AB_395713
Human/Mouse CXCL12/SDF-1 Antibody, R&D Systems, Biotechne #MAB310-SP RRID:AB_2276927
Mouse TIMP-1 Antibody, R&D Systems, Biotechne #AF980-SP RRID:AB_355759
Human/Mouse TIMP-2 Antibody, R&D Systems, Biotechne #AF971-SP RRID:AB_355752
Mouse Osteopontin/OPN Antibody, R&D Systems, Biotechne #AF808-SP RRID:AB_2194992
Mouse CXCL1/GRO alpha /KC/CINC-1 Antibody, R&D Systems, Biotechne #AF-453-SP RRID:AB_354495

| Validation | All antibodies are commercially available, and were validated for specificity and application by manufacturers listed here and specified below, or published for the application and species before. Additional information on validation can be found on the manufacturer's websites. Antibodies were used at concentrations suggested in previous published methodologies (https://star-protocols.cell.com/protocols/2730, https://www.cell.com/cell-stem-cell/fulltext/S1934-5909(21)00287-3) or titrated in-house, with the concentration specified in Supplementary Information. |

Primary antibodies
anti-E-cadherin (24E10), Cell Signaling # 3195 - WB, IHC, IF, Flow Cytometry; Species reactivity - Mouse, Human;
anti-E-cadherin (ECCD-2),Thermo Fisher Scientific # 131900 - WB, IHC;  Species reactivity - Mouse;
anti-CD13 (ER-BMDM1), Novus # NB100-64843 - Flow cytometry, IHC; Species reactivity - Mouse;
anti-Cytokeratin-19 (TROMA-III), Sigma-Aldrich # MABT913 - EM, IF, IHC, IP, WB; Species reactivity - Mouse, Human;
anti-ZO-1, Thermo Fisher Scientific # 40-2200 - WB, IHC, IF; Species reactivity - Mouse, Human, Dog, Rat;
anti-Albumin, Novus # NB600-41532 R- WB, ELISA, ICC/F, IHC; Species reactivity - Mouse;
anti-Radixin (C4G7) ,Cell Signaling # 2636 - WB; Species reactivity - Mouse, Human, Rat, Monkey;
anti-Cytokeratin, Dako # Z0622 - information from manufacturer's website not available, product discontinued;
anti-Hnf4-alfa (EPR16885), Abcam # ab181604 - IP, ChIP, WB, IHC, ChIC; Species reactivity - Mouse, Human, Rat;
anti-SOX9 (EPR14335-78), Abcam # ab185966 - WB, IHC, IF, Flow cytometry; Species reactivity - Mouse, Human, Rat;
anti-Osteopontin, R&D Systems # AF808 - ELISA, WB, IHC, IF, Neutralisation; Species reactivity - Mouse;
anti-CD34 (RAM34), eBioscience™ ThermoFisher Scientific # 14-0341-85 - IHC, Flow cytometry; Species reactivity - Mouse;
anti-Cleaved Caspase-3 (Asp175)(Clone 5A1E), Cell Signaling # 9664 - WB, IP, IHC, IF, Flow cytometry; Species reactivity - Mouse, Human, Rat, Monkey;
anti-Ly-6A/E (Sca-1), BioLegend # 108102 - Flow cytometry, WB, IP, IF, IHC; Species reactivity - Mouse;
anti-Transcription Factor Sox-9, Millipore # AB5535 -WB, IHC; Species reactivity - Mouse, Human, Rat, Chicken;
anti-Vimentin (clone LN-6), Merck # V2258  - IHC, IP, IF, WB; Species reactivity - Mouse, Human, Pig, Sheep, Bovine, Rabbit, Feline, Rat;
anti-Ly-6A/E (Sca-1) (Clone D7), Super Bright 436, ThermoFisher Scientific # 62-5981-82 - Flow cytometry; Species reactivity - Mouse;
anti- CD326 (EpCAM) (Clone G8.8), APC, ThermoFisher Scientific # 17-5791-80 - Flow cytometry; Species reactivity - Mouse;
anti-CD11b (Clone M1/70), PE-Cy7, BD Biosciences # 552850 - Flow cytometry; Species reactivity - Mouse, Human;
anti-CD31 (Clone 390), PE-Cy7, BD Biosciences # 561410 - Flow cytometry; Species reactivity - Mouse;
anti-CD45 (Clone 30-F11), PE-Cy7, BD Biosciences # 552848 - Flow cytometry; Species reactivity - Mouse;
anti-Elastin (polyclonal IgG), Cedarlane Laboratories Limited #CL55041AP - information from manufacturer's website not available, product discontinued;
anti-Glutamine Synthetase (GS), Sigma-Aldrich # G2781 - WB, IHC; Species reactivity - Rat;
anti-Cytochrome P450 2E1 (CYP2E1), Abcam # ab28146 - WB, IF; Species reactivity - Mouse, Human, Rat, Rabbit;
anti-Ki67 (Clone SP6), ThermoFisher Scientific #RM 9106 S1 - IHC; Species reactivity - Human;

Secondary antibodies
Donkey anti-Rat IgG (H+L) AF-488, ThermoFisher Scientific # A-21208 - IHC, IF; Species reactivity - Rat;
Donkey anti-Rat IgG (H+L) AF-Plus-555, ThermoFisher Scientific # A48270 - IF; Species reactivity - Rat;
Donkey anti-Rat IgG (H+L) AF-594, ThermoFisher Scientific # A-21209 - IHC, IF; Species reactivity - Rat;
Goat anti-Rat IgG (H+L) AF-647, ThermoFisher Scientific # A-21247 - WB, IP, IHC, IF; Species reactivity - Rat;
Goat anti-Rabbit IgG (H+L) AF-532, ThermoFisher Scientific # A-11009 - WB, IF; Species reactivity - Rabbit;
Donkey anti-Rabbit IgG (H+L), CF™ 633, Merck # SAB4600132 - Flow cytometry, IHC, IF; Species reactivity - Rabbit;
Donkey anti-Rabbit IgG (H+L) AF-647, ThermoFisher Scientific # A-31573 - WB, IHC, IF; Species reactivity - Rabbit;
Donkey anti-Mouse IgG (H+L) AF-647, ThermoFisher Scientific # A-31571 - Flow cytometry, IF; Species reactivity - Mouse;

Donkey anti-Goat IgG (H+L) AF-594, ThermoFisher Scientific # A-11058 - Flow cytometry, IHC, IF; Species reactivity - Goat;
Donkey anti-Goat IgG (H+L) AF-647, ThermoFisher Scientific # A-21447 - IHC, IF; Species reactivity - Goat;
Donkey anti-Rat IgG (H+L) CF 568, Biotium # 20092 - Flow cytometry, IHC, IF, WB; Species reactivity - Rat;

Blocking antibodies
Goat IgG Control R&D Systems, Biotechne #AB-108-C  - Flow cytometry, IF; Species reactivity - Goat;
Rat IgG Control (Azide Free) R&D Systems, Biotechne #6-001-F - Control applications  - no other validation specified; Species reactivity - Rat;
Mouse IgG1 Control R&D Systems, Biotechne #MAB002 - Flow cytometry; Species reactivity - Mouse;
Mouse uPAR Antibody R&D Systems, Biotechne #AF534-SP - Blocade of Receptor-Ligand Interaction, CyTOF, Flow cytometry, WB, IHC; Species reactivity - Mouse;
Human/Mouse TNF-alpha Antibody R&D Systems, Biotechne #AF410-SP - CyTOF, ELISA capture, IHC,IF, Flow cytometry, Neutralisation, WB; Species reactivity - Mouse, Human;
Mouse CX3CL1/Fractalkine Chemokine Domain Antibody, R&D Systems, Biotechne #AF472-SP - Flow cytometry, IHC, IF, Neutralisation, WB; Species reactivity - Mouse;
Mouse CCL11/Eotaxin Antibody, R&D Systems, Biotechne #AF420-SP - ELISA capture, IHC, Neutralisation, WB; Species reactivity - Mouse;
Human/Mouse CXCL12/SDF-1 Antibody, R&D Systems, Biotechne #MAB310-SP  - Neutralisation; Species reactivity - Mouse, Human;
Rat Anti-Mouse CD47 (miap301) Antibody, BD Biosciences #555297  - Flow cytometry, IF, IHC;  Species reactivity - Mouse;
Mouse TIMP-1 Antibody, R&D Systems, Biotechne #AF980-SP -  IHC, IP, Neutralisation, WB; Species reactivity - Mouse;
Human/Mouse TIMP-2 Antibody, R&D Systems, Biotechne #AF971-SP  - ELISA, WB; Species reactivity - Mouse, Human;
Mouse Osteopontin/OPN Antibody, R&D Systems, Biotechne #AF808-SP - ELISA Capture, IHC, IF, Neutralisation, WB; Species reactivity - Mouse;
Mouse CXCL1/GRO alpha /KC/CINC-1 Antibody R&D Systems, Biotechne #AF-453-SP - Neutralisation, WB;  Species reactivity - Mouse.

# Eukaryotic cell lines

Policy information about cell lines and Sex and Gender in Research

| | |
|---|---|
| Cell line source(s) | The mesenchymal cells and organoid lines are all primary material derived from isolated livers by investigators in this study. |
| Authentication | N/A |
| Mycoplasma contamination | Mycoplasma contamination was regularly tested on all cell and organoid laboratory lines throughout this study, using MycoAlert® Mycoplasma Detection Kit (Lonza #LT07-118). |
| Commonly misidentified lines (See ICLAC register) | No ICLAC lines were used in this study. |

# Animals and other research organisms

Policy information about studies involving animals; ARRIVE guidelines recommended for reporting animal research, and Sex and Gender in Research

| | |
|---|---|
| Laboratory animals | Mouse experiments were performed in accordance with the German animal welfare legislation and in strict pathogen-free conditions in the animal facility of the MPI-CBG. Protocols were approved by the Institutional Animal Welfare Officer (Tierschutzbeauftragter), and all necessary licenses were obtained from the regional Ethical Commission for Animal Experimentation of Dresden, Germany (Tierversuchskommission, Landesdirektion Dresden). The MPI-CBG's laboratory animal housing is exclusively barrier housing. All mice are kept in IVC systems (individually ventilated cages) under a 12:12-hour light/dark cycle. The animal room temperature is between 20 and 24 C and the relative humidity is 55±10%. Both are subject to constant monitoring. Sterile food and water were given ad libitum. Healthy adult mice (8-25 weeks of age) of both sexes were used for experiments. For MDR2 experiments, 8 weeks of age were used. WT, C57/Bl6 mice, Rosa26-mTmG, Rosa26-nTnG, PDGFRα-H2B-GFP, PDGFRα-H2B-GFP x Rosa26-mTmG, Prom1-CreERT2 x R26-LSL-ZsGreen, Itgb1-fl/fl x R26-LSL-ZsGreen or MDR2-KO mice were used for experiments. Rosa26-mTmG [Gt(ROSA)26Sortm4(ACTB-tdTomato,-EGFP)Luo/J] and Rosa26-nTnG [B6;129S6-Gt(ROSA)26Sortm1(CAG-tdTomato*,-EGFP*)Ees/J] were obtained from the Jackson Laboratory (JAX). The PDGFRα-H2B-GFP ([B6.129S4-Pdgfratm11(EGFP)Sor/J] was described previously and obtained from Prof. Magdalena Zernicka-Goetz. The PDGFRα-H2B-GFP x Rosa26-mTmG was generated by crossing the PDGFRα-H2B-GFP with the Rosa26-mTmG obtained and described above. The MDR2-knock out line [FVB.129P2-Abcb4tm1Bor/J] was described before. The R26-LSL-ZsGreen B6.Cg-Gt(ROSA)26Sortm6(CAG-ZsGreen1)Hze/J) was obtained from JAX. Itgb1-fl/fl (B6;129-Itgb1tm1Efu/J) was described previously, and obtained from Prof. Magdalena Zernicka-Goetz. Itgb1-fl/fl x R26-ZsGreen was generated by crossing the Itgb1-fl/fl with the R26-LSL-ZsGreen. The Prom1-CreERT2 x R26-LSL-ZsGreen was generated by crossing the Prom1-CreERT2 (B6N;129S-Prom1tm1(cre/ERT2)Gilb/J) -described previously and obtained from Prof Richard Gilbertson- with the R26-LSL-ZsGreen. The Confetti mouse (Gt(ROSA)26Sortm1(CAG-Brainbow2.1)Cle/J) was described before and recombined in vitro to obtain mCFP cholangiocyte organoids. Mice were bred onto a C57/B6 background. |
| Wild animals | This study did not involve wild animals. |
| Reporting on sex | Both male and female mice were used in this study indistinctively. |
| Field-collected samples | This study did not involve field-collected samples. |

| Ethics oversight | Mouse experiments were performed in accordance with the German animal welfare legislation and in strict pathogen-free conditions in the animal facility of the MPI-CBG. Protocols were approved by the Institutional Animal Welfare Officer (Tierschutzbeauftragter), and all necessary licenses were obtained from the regional Ethical Commission for Animal Experimentation of Dresden, Germany (Tierversuchskommission, Landesdirektion Dresden). |
|---|---|

Note that full information on the approval of the study protocol must also be provided in the manuscript.

# Flow Cytometry

## Plots

Confirm that:

☒ The axis labels state the marker and fluorochrome used (e.g. CD4-FITC).

☒ The axis scales are clearly visible. Include numbers along axes only for bottom left plot of group (a 'group' is an analysis of identical markers).

☒ All plots are contour plots with outliers or pseudocolor plots.

☒ A numerical value for number of cells or percentage (with statistics) is provided.

## Methodology

| Sample preparation | For the analysis of cholangiocytes from hepatocyte isolation and HepOrg culture, primary hepatocytes and other liver cells were isolated from mice as described in Methods Section. The cells were then strained with 100 µM strainer, washed one time with AdDMEM/F12 (ThermoFisher, 12634010) medium containing 1% HEPES (ThermoFisher, #15630-056), 1% Penicillin/Streptomycin (ThermoFisher, #15140-122), Glutamax (ThermoFisher, #35050-068), spun 5min at 100g and stained 30min with following antibody against EpCAM, conjugated to APC (CD326 (EpCAM) Monoclonal Antibody (G8.8), APC, eBioscience, ThermoFisher, #17-5791-80). Before sorting, cells were washed one time with above medium, spun 5min at 100g and resuspended in the same medium for sorting. In case of cholangiocytes from HepOrg culture, all cells from the culture were collected in above media, spun 5min at 200g and dissociated by incubation with TrypLE Express (ThermoFisher, #12605010) for 5 min at 37°C, before being strained and stained as outlined above. |
|---|---|
| Instrument | Sony MA900 Multi-Application Cell Sorter, FacsAria Fusion Cell Sorter |
| Software | BD FACSDiva v.8.0.2, Sony Cell Sorter Software v.3.1.2, Flowjo v.10.8.1 |
| Cell population abundance | The final sorted population was above 90% of the total detected events. Single cells were 75-81% of the final sorted population. EpCAM+ population was 0.08% of total events in the hepatocyte isolation prep. EpCAM+ population was 19-47% of all single cells in the hepatocyte organoid culture prep. The proportions of ZsGreen negative and positive fractions are specified in Extended Data Figure 4e. The mode of 4-way purity was always selected for cell sorts. |
| Gating strategy | The starting population was determined based on the FSC-A and SSC-A statuses, and single cells were gated based on FSC-H and FSC-W intensities. All single cells were gated on positive staining for the EpCAM-APC antibody, as determined by the negative unstained control. Then, EpCAM-positive population was gated on ZsGreen positive or negative fluorescence, as determined by negative control with ZsGreen expression. |

☒ Tick this box to confirm that a figure exemplifying the gating strategy is provided in the Supplementary Information.

