## [Peer Review File · Nature]

Mouse liver assembloids model periportal architecture and biliary fibrosis

Corresponding Author: Professor Meritxell Huch

Version 0:

Reviewer comments:

Referee #1

(Remarks to the Author)

In the manuscript entitled “Functional periportal liver assembloids recapitulate the mesoscale hepatic architecture and model biliary fibrosis in vitro” Dowbaj and colleagues aim to create periportal hepatic architecture consisting of polarized hepatocytes, cholangiocytes and mesenchymal cells. Activation of Wnt signaling in culture improved hepatocyte organoids. In order to resemble the different cell types found in the periportal area, the authors then added cholangiocytes and mesenchymal cells at predefined ratios to the culture, each labeled with its own fluorescent marker. Cellular assembly and connections are comparable to native tissue. Interestingly, increasing the number of mesenchymal cells with increase in fibrotic response genes, mostly based on homotypic cell-cell interactions. Overall, this is a very impressive manuscript representing a technical breakthrough, providing multiple high-quality imaging data and single cell transcriptomics, amongst others, to provide deep phenotypic analysis of these assembloids. The insight into disease mechanisms, however, may be limited, given the lack of mechanistic interrogation.

Major Criticisms:

The use of cholyl-L-lysyl-fluorescein to study biliary transport and visualize bile ducts is great, but it should be noted that the transporter responsible is ABCC2, meaning that it only represents one group of bile salt / biliary transporters. This limitation should be highlighted, given that particularly MDR2/ABCB4 loss has been shown as a genetic model of cholestatic liver disease.

The authors claim they generated peri-portal assemblies. It is interesting that historically it has been known that the peri-central hepatocytes experience increased Wnt activity. Can the authors explain why the hepatocytes express a peri-portal gene expression pattern in the presence of high Wnt levels.

The authors claim a novel mechanism “to activate the myofibroblastic state in portal fibroblasts independently of an immune activation” simply by increasing the number of portal fibroblasts 10-fold and increasing their cell-cell interactions. Why this is an interesting hypothesis, prior work has shown that the mechanics of the cellular environment (stiffness, etc.) can affect myofibroblast activity, so this may not be a novel observation. This finding may explain to some degree the irreversibility of fibrosis observed in the clinic but may not really be related to true causal/mechanistic relationships between portal fibroblasts and the occurrence of fibrosis. Relatedly, the question is whether addition of stellate cells may change the growth patterns/responses of this assembly.

Minor Criticisms:

Line 42/43 refers to “cholestatic liver diseases, ranging from primary biliary cholangitis (PBC) to stones, amongst others” is too vague, and clinically incorrect: cholestatic liver disease can have genetic or immune origin, and is worthy of being modeled in vitro. Gall stone disease, however, is different. The cited references are also not directly reflective of this statement.

Figure 3, panel d – x-axis label “DC” should probably mean “CC” for cholangiocytes or “BC” for biliary cells?

Referee #2

(Remarks to the Author)

This elegant study comes from leaders in liver organoid modeling and covers an important gap as the majority of current

studies are either done in animal systems, 2D culture or short term liver slice systems - all with grave disadvantages. The authors show an improved model in which optimization and different ratios of mesenchymal cells result in larger organoid with functional bile canaliculi and drainage as well as a induction of fibrogenesis. Despite the elegance of the system and the superb images, the study also has weaknesses that need to be addressed:

1. Although the model is elegant, many parts of the manuscript remain at a descriptive levels with a lack of functional testing of the many proposed ligand-receptor interactions (Col1a1-Ddr2, Col3a1-Ddr2, Gas6-Axl, Lgals-Intgb1, Timp2-Itgb1, Timp1-Cd63, Col1a1-Ddr1, Col3a1-Ddr1, Cxcl12-Itgav, Col6a1-Sdc1, Apoa1-Lrp1. In 2023, ligand-receptor interactions should not only be displayed but also functionally investigated.
2. Likewise, the authors suggest a role for homotypic (Msc-Msc) interactions in driving fibrosis, but. do not provide functional evidence for the concept. The authors do not show a role for any of the proposed mediators (Il6,Il33, Ccl11, Cxcl5, Cxcl1 and Cxcl12m Mmp2, Mmp3) and the corresponding receptors. Moreover, a previous study (Wang et al, Sci Transl Med 2023) suggested neurotrophic receptor tyrosine kinase 3-neurotrophin 3 interactions as autocrine/"homotypic interaction" driver of HSC activation and fibrosis. Hence, this concept is not entirely novel. Did the authors determine this pathway in their model? Likewise, the alternative hypothesis of suppressive mediators from epithelial compartments that are being diluted out in the setting of increased Msc is not discussed or experimentally tested. Please see also comment on apoptotic hepatocytes below.
3. While the authors clearly achieved their goal of developing a model with physiological bile canaliculi, they did not demonstrate sufficiently the relevance of this system for modeling liver physiology and pathophysiology. The manuscript does not investigate estng what consequences the presence of bile canaliculi have on detoxification and metabolite transport as well as cell-cell interactions and development of hepatocyte dysfunction and fibrogenesis. What happens to bile salts after they are draining into the ducts – are there any readouts that can detect beneficial consequences of this? Does hepatocyte gene expression and function change as a result?
4. Metabolic functions of the liver - undoubtedly one its most important roles in organismal health - are largely ignored in the manuscript. Proper liver architecture is key to metabolism and detoxification – and relies on spatially organized metabolic activities, referred to as zonation. It is not clear whether the employed model reflects zoned liver functions – at minimum, this should be shown for the cholesterol-bile acid pathway, which has a key role in cholestatic liver disease and is strongly zoned.
5. The correlation between the model of this study and published scRNA-seq datasets (Fig.4i) appears rather low with the exception of the correlation between hepatocytes from this model and hepatocytes from the Guilliams study. Especially the correlation to PF scRNA-seq data from the other studies is very low and lower than the correlation with VSMCs. Moreover, since these are periportal assembloids there should be an additional analysis comparing hepatocytes to portal (Zone 1) and pericentral (Zone 3) hepatocytes from previous scRNA-seq datasets – this might make the results stronger and more meaningful.
6. It is widely believe that signals from inflammatory cells, in particular macrophages, have a key role in the fibrogenic process. These cells are missing in the current model, which represents potentially a very important gap in regards to studying fibrogenesis.
7. Evidence for cleaved caspase 3 and hepatocyte is weak (there is no quantification comparing homeostatic- and fibrotic-like assembloids) and there is no functional investigation (e.g. using caspase inhibitors) for confirmation and for determination of the role of apoptotic hepatocyte death in fibrogenesis in this model system. How doe Msc induce hepatocyte death? Furthermore, the presence of apoptotic hepatocytes puts into question the proposed key role of homotypic Msc-Msc interactions.
8. It would be helpful if this system could be further validated and if practical applications of this systems could be demonstrated e.g. drug screening for biliary fibrosis.

Referee #3

(Remarks to the Author)

This manuscript by Dowbaj et al. describes a new organoid model composed of adult hepatocytes, cholangiocytes and liver mesenchymal cells, which the authors use to model biliary fibrosis in vitro. The authors also optimize the current hepatocyte organoid models to generate organoids with physiological / functional bile canaliculi networks, and assemble these with cholangiocytes and portal fibroblasts to generate periportal assembloids.

A substantial amount of the data presented in this manuscript is descriptive in nature, and much more quantitation (with statistical analysis) is required to underpin the findings and conclusions drawn. Furthermore, more data is required to convince the reader that the assembloid system provides a platform that will provide step changes in our understanding of liver physiology and disease – for example disease modelling / perturbations with this system leading to conceptual advances in our understanding of liver disease pathogenesis.

Specific comments:

1. Line 90-93 and Supp Fig 1b: How does CD13 immunostaining indicate insufficient BC and BD formation? This should be clarified for the reader.
2. The authors state that their system exhibits a 'physiologically relevant' BC network and size, however there is no data presented to measure what the physiologically relevant / in vivo baseline is to allow direct comparison.

3. Line 98 and Fig1c: what is the 'expansion potential', what does this measure? The authors also state increased size, cell number and organoid formation efficiency but make no statistical comparisons and present no quantitative data on this, and instead representative images are shown. Comprehensive quantitation and statistical analyses should be performed.

4. Line 462: the methods state organoid efficiency presented as percentage of seeded cells, however in Supp fig 1e data is presented as number of HepOrgs/1000 seeded hepatocytes. Can the authors explain this discrepancy?

5. Supp fig 1f: can the authors comment on the large drop in cell number (almost to baseline levels) at passage 4 in the HM-Wnt sample.

6. Line 105, line 468, Supp fig 2a,b: bubbly and ball shaped morphology. This is a qualitative measurement not a quantitative measurement. The authors need to assess cell shape and morphology with a quantitative approach using cell segmentation software such as Imaris or Cellpose.

7. Line 111: quantification of the longer thinner BC shapes between the bubbly and ball structures should be performed. Similar to data showing bile acid secretion presented later in the manuscript, this observation should be quantitated to assess functionality in the bubbly vs ball shapes.

Also is there any way the authors can quantitate the size of the biliary network, ie, quantitate the number of nodes and edges?

8. Line 440: the authors enriched for bubbly-HepOrg by enriching for cell clusters – please describe how this enriching was done.

9. Line 122, Fig 1d: This should be quantified and compared between groups.

10. Line 124 Fig 1f, Supp Fig 2e: Can this CLF functionality be quantified and compared between the two groups? Eg areas of intense staining (in the BC) vs the more diffuse staining seen in the ball structures.

11. Line 129: It is not clear whether the cholangiocyte-like structures are due to differentiation of hepatocytes or from existing cholangiocytes/BDs in the media aggregating with hepatocytes. This should be explored in more detail.

12. Line 131-133: Again these data should be quantified - for example, the smoothness/sphericity of the chol-like structures, and if smooth is this then suggestive of lack of outward connections?

13. Line 140 Fig 2a,b: Authors state 'even more improved' hepatocyte polarity but I do not see any quantification or comparison between groups other than a representative image.

Further to this there is comment of the MM supporting cholangiocytes and mesenchyme without their outgrowth – is this within the HepOrg culture? ie do these HepOrgs contain multiple cell types?

14. Line 145 Fig 2c, Suppl 6a: Quantification of BC diameter is performed and the authors state this is significant, however I cannot see any analyses linked with these data.

15. Line 149, 671 Fig 2d: regarding the analyses of BC:BC connectivity – this would benefit from a schematic outlining, especially for the liver tissue samples, what areas and how these measurements were taken.

16. Given these data and experimental approaches are geared towards forming organoids which aim to recapitulate the peri-portal region in vivo, can the authors use immunofluorescence staining of hepatocyte zonation markers to investigate whether the hepatocytes are similar to zone 1 hepatocytes in vivo and whether these hepatocyte markers are influenced by the different culture conditions?

Furthermore, the authors show qPCR for genes associated with mature hepatocyte function however it would be helpful to also include genes associated with portal and central hepatocyte function to investigate whether these hepatocytes are portal, central, or a mix of both phenotypes.

17. Fig 3c: does this graph show the number of assembloids vs those that have not assembled (eg HepOrg)? This is not clear in the figure legend. Also statistical comparisons should be shown here.

18. Line 169: linked with this figure states 70-90% efficiency however this is not what the graphs shows, ie assembloid-shaker is average of 70% with a range between 50-85%. Linked figures Supp 7b,c do not clarify this, and furthermore Supp Fig 7c compares HM vs MM, but my understanding is that the optimal media had already been chosen by this point - please clarify these points.

Further, in Fig 3c the aggrewell (non-shaker) seems to have equal efficiency but less variability, why did the authors not use this approach?

19. Line 171: The authors state the majority of assembloids formed within 48hrs - please show these data including quantification.

20. Line 174: the authors state the assembloids retain the cell proportions observed in vivo in tissue, however no comparative data is provided regarding this, or comparison with previously published data where in vivo measurements

have been performed.

21. Fig 3b: Tissue - what does the white arrowhead refer to? Fig 3f: these appear to be different stages of maturity? What does the star mean, is this the lumen?

22. Given that the cell composition in Fig 3d shows that hepatocytes are the dominant cell type in assembloids, can the authors state why this is not recapitulated in their scRNAseq data (Fig 4g,h)? The low number of hepatocytes might impact on the analysis shown in Fig 4b,e,i. The authors should discuss this limitation in these datasets.

23. Line 191: the authors cannot rely solely on scRNAseq (mRNA-level) data to confirm that the mesenchymal component in their in vitro system resembles portal fibroblasts. Immunostaining with various markers would be useful additional confirmation.

24. Line 181, fig 4b: The expression of hepatocyte genes across the board suggests a wash of hepatocyte genes in the dataset. Can the authors try using an algorithm such as SoupX to see if this helps with this pervading signal (if they believe this to be technical). In contrast, if the authors feel this represents a real finding, can they explain why all cell types are expressing hepatocyte genes. It is likely this is a typical hepatocyte gene wash which is commonly observed in this type of data, however the authors need to examine this / comment on it further.

Furthermore, can the authors comment on the numbers of cells as there appears to be a distinct lack of hepatocytes compared to the overall abundance that is observed in the assembloids. It is also not stated how many assembloid preps were sequenced, and bar charts showings numbers / ratios of cells contributed by each sample would also be helpful for the reader.

25. Fig5a shows a representative image from n=6 experiments. These data should be quantified.

26. Line 200-202: do these images of tissue and assembloid simply show that hepatocytes surround the BD? This 'recapitulation' of tissue needs to be more comprehensively characterized, and furthermore labelling in the image and figure legend needs to be clearer.

27. Line 230-234 and associated figures: The authors should quantify these data with comparisons to homeostasis. Cell death and proliferation (for example using Ki67 IHC) of all cells in the homeostasis and fibrotic-like conditions should also be quantified. Is there any cell death or proliferation in the cholangiocyte and mesenchymal lineages?

Similarly in Figs 10e, 11a, 11b, 8f - are these differences statistically significant?

28. Supp fig 11b: what is healthy vs unhealthy? In the figure legend and main results section homeostasis and fibrotic-like are the two conditions listed. Also are fig 11b and 11c the same data? If so, should these be in a different order?

29. Line 228 and Supp Fig 10c: DC are not significantly increased in fibrosis-like state, but this is not what is stated in the results text.

30. Line 242-245: The authors should analyse the preceding data in greater depth to allow a more comprehensive and in-depth comparison with in vivo cholestatic liver injury.

31. Is there a fibrotic phenotype when the cholangiocyte numbers increase/change relative to hepatocytes?

32. All videos would benefit from labelling on the video itself, not just in the figure legend. Figure legends are incomplete, for example information on what the arrowheads and stars refer to. All immunostaining images would benefit from clearer labelling to help guide the reader.

Referee #4

(Remarks to the Author)

Dowbaj et al., Functional periportal assembloids...

The authors digested the livers of adult mice enzymatically, cultured them as "organoids" using a Matrigel-based culture technique, and characterized the cultured mouse cells by imaging and image analysis, RNA-seq, a functional assay with a fluorescent bile salt analogue (CLF), and the analysis of cytokines, albumin and bile acids in the culture medium. The authors report that the organoids "recapitulate the mesoscale hepatic architecture", "model biliary fibrosis in vitro", "recapitulate the architecture and cellular interactions of the native periportal region of the liver tissue", "periportal assembloids are fully functional", "generate a physiological and functional bile canaliculi network", functional bile canaliculi connect to bile ducts, "the periportal assembloids recapitulate the architectural arrangement and gene expression of the tissue". Moreover, the authors state that the organoids "are reminiscent of the defects observed in cholestatic injury and in mouse models of biliary/cholestatic fibrosis". Finally, the authors conclude that "our periportal assembloid model, where architecture and cell-cell interactions are recapitulated both at the meso- and cellular scale, represents the first system where to collectively study bile canaliculi formation, bile drainage and cholestatic liver disease".

Comments

(1) It is not correct that this is the first study showing that primary liver cells form functional bile canaliculi in vitro and that bile canaliculi link to bile ducts. This has been reported in several publications. An example is Tanimizu et al, 2021 (doi: 10.1038/s41467-021-23575-1). These authors reported „the generation of a hepatobiliary tubular organoid (HBTO) using mouse hepatocyte progenitors and cholangiocytes. Hepatocytes form the bile canalicular network and secrete metabolites into the canaliculi, which are then transported into the biliary tubular structure.” The image below (Fig. 1C) of Tanimizu et al , 2021, shows the network of bile canaliculi that are formed by the apical membrane of hepatocytes (white) and are linked to a bile duct that is formed by cholangiocytes (green):

• These authors (Tanimizu et al , 2021) also used a Matrigel-based culture technique and applied the same bile salt analogue (CLF) to study secretion of CLF into the bile canalicular network in vitro and the functional link between bile canaliculi and bile ducts. It should be considered that Tanimizu et al. represent only an example showing a functional canalicular network; it should be considered that such culture techniques are already routinely applied to study if pharmaceutical candidate compounds reduce the secretion of bile acids (which is an undesired effect in drug development). Published examples are: DOI: 10.3791/60507-v; DOI: 10.3791/60507; [https://doi.org/10.1016/0270-9139\(95\)90371-2](https://doi.org/10.1016/0270-9139(95)90371-2); DOI: 10.1007/s00204-015-1575-9) (but many more studies exist).

(2) This reviewer does not agree to the statement that the “Periportal assembloids model cholestatic liver fibrosis in the absence of immune compartment” (headline in results, page 6) or “model biliary fibrosis in vitro” (as written in the title). It is well-known that in cholestatic liver disease (and the resulting periportal fibrosis) bile ducts proliferate (named ductular reaction) during the entire disease process initially forming a tighter duct network around portal veins and later infiltrating into the liver lobule (DOI: 10.1002/hep.30150; and others). Together with the new ducts also periportal fibroblasts proliferate. This key feature is not shown in the present study (although cholangiocytes were included into the culture system) and it is therefore difficult to understand why the authors claim that the liver assembloids “model cholestatic liver fibrosis”.

(3) The authors claim that the organoids “recapitulate the mesoscale hepatic architecture” (Title and several similar statements in the results and discussion). This statement may be misinterpreted. Previously, some authors used the term mesoscale hepatic architecture for structures ranging from the micron to the millimeter level (doi: 10.7150/thno.71718). The millimeter level includes the liver lobule (radius 200-400 μm). However, in their organoids the authors did not see any lobules that consist of a central vein, periportal fields (containing branches of the portal vein and liver artery) and sheets of hepatocytes forming sinusoids from the periphery to the central vein. A key feature of these sheets of hepatocytes is their zonation. Often used markers of zonation are, e.g, expression of glutamine synthetase and certain cytochrome P450 isoenzymes preferentially in the lobular center but not in the periphery. Since the authors did not show any lobular zonation, it may appear overstretched to conclude that the “assembloids recapitulate the mesoscale hepatic architecture”.

(4) A further critical conclusion is that “assembloids ... recapitulate the ... gene expression of the tissue...”. However, several previous studies have shown that hepatocytes in culture up and downregulate hundreds of genes by large factors compared to the in vivo situation. Therefore, many authors use freshly isolated liver cells or fresh liver tissue as controls for comparison with cultured liver cells in RNA-seq or other gene expression studies. The authors of the present study did not include such controls. Their comparison is therefore based on literature data which limits the accuracy.

Version 1:

Reviewer comments:

Referee #1

(Remarks to the Author)

In this revised manuscript, Dowbaj and colleagues, present extensive new data from new lines of experiments to address the reviewers' previous criticisms and concerns. While novel mechanistic biologic insight remains limited, this manuscript presents a true experimental tour-de-force and will serve as a landmark resource and greatly advance the field. I have no further major criticisms or concerns.

Referee #2

(Remarks to the Author)

The authors have added a large amount of data and provided a carefully written point-by-point response. They satisfactorily addressed a number of the points raised in the previous review, including bile acid metabolism, improved comparisons to scRNA-seq datasets and integration of some functional studies. However, several of the major weaknesses have not been convincingly resolved and large parts of the study remain descriptive and mechanisms are not investigated in sufficient depth. The study will not convince the field that the assembloid model indeed represents a major advance that will change how liver disease mechanisms can be studied and how new therapeutic approaches can be tested.

1. The authors have integrated some functional studies but many of ligand-receptor pairs are only superficially investigated. The majority of the proposed central interactions have either not been validated or are not studied in sufficient detail and scientific novelty is limited.

1a. The authors did not validate their findings on Cdh11 and Itgb1 in vivo - thus it is not clear how useful the assembloid models is to determine key drivers of liver disease. The author should similar roles for Cdh11 and Itgb1 in vivo, including decreased fibrosis and other effects..

1b. The heading states "Periportal assembloids as a tool to investigate cellular and molecular mechanisms of cholestatic biliary fibrosis" but the studies on Itgb1 and Cdh11 do not really define cellular and molecular mechanism. Importantly, the authors have not proven that Cdh11 mediates a homotypic MSC-MSC interaction as (i) other cells are present in the assembloids and (ii) Cdh11 preferentially undergoes heterophilic interactions with Cdh8 (which seems to be also highly expressed in hepatocytes - the authors knocked out Cdh2 but also not in different cell types; knocking out Cdh8, Cdh11 and possibly other interaction partners in various cell types would be important). Likewise, the investigation of Itgb1 is also superficial. What is the responsible ligand? What is the mechanism? Can the authors demonstrate that these are homotypic MSC-MSC interactions?

1d. Many other ligand-receptor pairs such as Col1a1-Ddr2, Gas6-Axl, Lgals-Itgb1 or Timp1-Cd63 were not studied (Cdh11 has been implicated in fibrogenesis in many organs including the liver, hence the novelty is limited).

1e. Interactions between other cell types were not studied. In addition to absent effects of chemokine likely due to the lacking integration of immune cells (see comment below), interactions of mesenchyme with hepatocytes (e.g. leading to cell death) or cholangiocytes and interactions between hepatocytes and cholangiocytes were not studied.

2. The authors have not followed up on the suggestion to use their system for screens (a main advantage of the described model over in vivo models). Without the integration of such experiments, the impact that the paper and model will have on the community is only going to be moderate. As discussed before, this could have been a drug screen, RNAi or CRISPR screen, with subsequent validation of the data an in vivo model of biliary fibrosis.

3. The lack of immune cells remains a weakness. While hepatic stellate cells and endothelial cells are also missing (their close interactions suggest that they should be integrated at the same time), the focus on periportal fibrosis justifies the sole integration of portal fibroblasts. However, some of the questions/experiments of the current study are directly affected by the absence of Kupffer cells/macrophages or other immune populations - e.g. the absent effects of cytokine treatment or blocking antibodies against these cytokines (CXCL12, CCL11, CXCL1, CX3CL1) may be due to the absence of leukocytes in the assembloid model as many of receptors are largely on immune cells. Furthermore, Kupffer cells are an important part of zone 1 as they are also zoned and their protection from gut-derived bacteria is largely in this zone (Gola et al, Nature 2021).

4. The mechanisms how fibroblast drive hepatocyte death and whether this is an artifact of the system remain unclear. The authors added data using caspase inhibitors but the mode of hepatocyte death and the impact of cell death on fibrosis and other cell types remain vague (cell death is a main driver of liver disease progression).

Referee #3

(Remarks to the Author)

I would like to congratulate the authors on an outstanding and comprehensive body of work which will advance the field. In particular, the large amount of new data added in revision has significantly enhanced the manuscript. However, I would like the authors to consider my further specific comments which are listed below:

1) Original reviewer comment:

'Regarding point 11: Line 129: It is not clear whether the cholangiocyte-like structures are due to differentiation of hepatocytes or from existing cholangiocytes/BDs in the media aggregating with hepatocytes. This should be explored in more detail.'

Author rebuttal states: 'However, the reviewer has a valid point indicating that this could also be also due to cholangiocytes transdifferentiating from hepatocytes. While we believe studying transdifferentiation would be a subject of a full new project in itself, we have attempted to address the question of the reviewer by generating hepatocyte organoid cultures from an induced Prom1- CreRT2xfl-fl-ZsGreen mouse line'.

- I do think the question of hepatocyte / cholangiocyte transdifferentiation in this system is an important one, particularly given the recent publication in Nature demonstrating epithelial plasticity in human chronic liver disease (Acquisition of epithelial plasticity in human chronic liver disease, Nature 2024 Jun;630(8015):166-173), and that the authors are promoting this as a new model system that will help increase our understanding of human liver disease pathogenesis, and also represent a very useful system for drug testing in models of human liver disease.

To this end, in the revised manuscript, the authors have used a Prom1-CreRT2xfl-fl-ZsGreen reporter mouse line to investigate whether cholangiocytes give rise to cholangiocyte organoids in their system – they conclude that 'Cholangiocyte organoids in HM-Wnt media arise from cholangiocytes in the hepatocyte isolation prep'. However, I don't think the authors

can conclude this for the following reasons:

a) Prom1 is not specific for cholangiocytes, as it also labels mouse and human hepatocytes (eg Hepatocyte-specific Prominin-1 protects against liver injury-induced fibrosis by stabilizing SMAD7. PMID: 36038590). Therefore, Prom1 CreERT2 will trace both cholangiocytes and hepatocytes, not just cholangiocytes.

b) Furthermore, the cholangiocyte recombination efficiency is only 45%, which still does not help answer this question as some of the Zs Green negative cholangiocyte-like structures may have either derived from unlabelled (non-recombined) cholangiocytes or hepatocytes.

To answer this question, the authors could employ a fairly straightforward approach using IV injection of AAV8-TBG-Cre in Zs green reporter mice. This would label ~99-100% of hepatocytes green at the start of the experiment, and then these green hepatocytes could be traced throughout the organoid timecourse. This would give a clear readout of whether the cholangiocyte-like structures are due to differentiation of hepatocytes or from existing cholangiocytes/BDs in the media aggregating with hepatocytes.

2) Lines 435-438: 'Additionally, CDH11 knockdown also resulted in a marked reduction in collagen deposition and a reduction in hepatocyte cell death in assembloids cultured with 10-fold excess mesenchyme (Fig.6e and Suppl. Fig.15 j-k), which now resembled homeostatic control assembloids'.

- I could not see any quantitation / graphs presented here regarding SHG assessment of collagen and hepatocyte death – quantitation should be presented.

Referee #4

(Remarks to the Author)

Referee #4 to the manuscript Dowbaj et al., Functional periportal assembloids...

Comments to authors and editors

General summary

The authors have not convincingly addressed my comments. The manuscript remains a description of a three-dimensional cell culture system (named HepaOrg). This cell culture system recapitulates some features of liver tissue *in vivo*, such as formation of bile canaliculi into which a bile acid analogue is secreted and also polarized cells can be maintained for longer periods; however, these features have already been reported in previous publications using similar cell culture systems. The claim that the organoids “are reminiscent of the defects observed in cholestatic injury and in mouse models of biliary/cholestatic fibrosis” is not convincing, because critical histological features of these diseases are not seen in the presented images of the cell culture system. Moreover, the HepaOrg system does not recapitulate critical structural and functional features of a liver lobule (the smallest functional unit of the liver) and does not show typical features of liver tissue such as zoned expression of specific genes (meaning that expression levels differ in a specific pattern between the centre and the periphery of the lobules). Experiments to understand the molecular mechanisms responsible for tissue organization were not performed in this study.

Next, I address in more detail the four comments to which the authors have responded.

Comment 1: (“It is not correct that this is the first study showing that primary liver cells form functional bile canaliculi *in vitro* and that bile canaliculi link to bile ducts.”)

The authors agreed to my comment that they have not convincingly emphasized the novelty of their study. However, also the revised version does not convincingly show novelty. Before discussing the authors’ arguments in detail, I would like to give some examples what would be versus is not sufficiently novel to justify publication in a top journal. Having an organoid (or any culture system) that recapitulates a part of liver tissue, such as a group of liver lobules or at least one liver lobule would be of relevant novelty. However, this is clearly not achieved by the culture system presented in the present study. Only bile canaliculi or even bile canaliculi that connect to bile ducts (as demonstrated in the present work) are not novel and have already been reported previously. Also, liver cells that maintain polar features over a longer period of time are not novel. The authors now explain their novelty as a combination of 7 features (long term passaging while retaining bile canaliculi; the bile canalicular diameter is between 1.5 and 2 µm; no cholestatic features; elongated interconnected bile canaliculi; functional connection of bile canaliculi to bile ducts; integration of fibroblasts; no immune system involvement). However, these features are not novel and several previous publications reached already a similar phenotype. It is also not novel that bile canaliculi in culture can have a similar diameter as *in vivo* (1-2 µm), since there are several publications where this has been shown (together with some functional features, such as transport of bile acids). It is also not sufficient that the authors used a 3D (instead of a 2D) system, since there are already dozens of publications with 3D “organoids” published. The authors included an additional experiment showing that deoxycholic acid led to bile canaliculi dilution and other morphological changes. However, several previously published studies have shown the influence of bile acids leading to altered (e.g. widened) bile canaliculi. For example doi.org/10.1016/j.bcp.2012.07.008 write: “Cells cultured for 3 days in the presence of UDCA, the bile canaliculi lumen was enlarged and the bile canaliculi ...”. In conclusion, the novelty remains absent or moderate.

Comment 2: (“This reviewer does not agree to the statement that the “Periportal assembloids model cholestatic liver fibrosis ...”). Also in the revised version the authors do not show characteristic features of fibrosis and ductular reaction which are

seen in vivo. In vivo, a very characteristic formation of new ducts (appearing as new tubes and a denser network of tubes) is seen. This is not seen in Fig R30. Please consider that “cholangiocyte cell expansion” is not the same as ductular reaction.

Comment 3: (It is not convincing that the organoids “recapitulate the mesoscale hepatic architecture...”). The authors agree that their culture system does not recapitulate a liver lobule. The problem I see is that features at a smaller scale (bile canaliculi, polar cells, interactions between cell types, response of hepatocytes to Wnt) have already been reported in previous publications. The data based on which the authors claim “zonation” (Fig. R2e, f) are not convincing. In vivo, e.g., CYP2E1 is expressed in the central part of the liver lobule. However, this is not the case for the organoids (HepOrg), where CYP2E1 occurs predominantly the peripheral cells and less in the center. However, also (suspiciously) ECAD is also less expressed in the centre. In contrast, glutamine synthetase (GS) is expressed only in an inner ring of hepatocytes around the central vein in vivo. However, in the organoid (HepOrg) GS appears unspecifically in the periphery and in the centre. It remains unclear, why the authors think that they obtained zonation similar as in vivo.

Comment 4: (This reviewer finds it unclear if “assembloids ... recapitulate the ... gene expression of the liver tissue”). I am aware of the supplemental figure but would criticize that the procedure to compare cells from organoids (HepOrg) to primary (isolated) hepatocytes is not adequate. For the analysis of RNAseq data, I would expect (among others) a comparison of cells from HepOrg and primary hepatocytes with a list of the differentially expressed genes, including fdr-adjusted p-values and fold-changes. A statement such as “our hepatocytes cluster with publicly available hepatocytes” is not sufficient since advanced biostatistical methods for quantification of similarity are available; but one would begin with the basics, especially a list of differentially expressed genes.

Version 2:

Editorial note: An additional referee (Referee #5) was recruited to assess the authors' response to the second round of review. This referee only shared comments with the Editors. The referee was satisfied with the authors' response.

POINT-BY-POINT RESPONSE to the REVIEWER'S COMMENTS:

Referee#1:

In the manuscript entitled "Functional periportal liver assembloids recapitulate the mesoscale hepatic architecture and model biliary fibrosis in vitro" Dowbaj and colleagues aim to create periportal hepatic architecture consisting of polarized hepatocytes, cholangiocytes and mesenchymal cells. Activation of Wnt signaling in culture improved hepatocyte organoids. IN order to resemble the different cell types found in the periportal area, the authors then added cholangiocytes and mesenchymal cells at predefined ratios to the culture, each labeled with its own fluorescent marker. Cellular assembly and connections are comparable to native tissue. Interestingly, increasing the number of mesenchymal cells with increase in fibrotic response genes, mostly based on homotypic cell-cell interactions. Overall, this is a very impressive manuscript representing a technical breakthrough, providing multiple high-quality imaging data and single cell transcriptomics, amongst others, to provide deep phenotypic analysis of these assembloids. The insight into disease mechanisms, however, may be limited, given the lack of mechanistic interrogation.

A: We thank the reviewer for the positive comments regarding our findings, advance and quality of our manuscript. We now provide some insights into the mechanisms driving the phenotype observed in the fibrotic-like assembloids, which we hope the reviewer will find interesting.

Major Criticisms:

The use of cholyl-L-lysyl-fluorescein to study biliary transport and visualize bile ducts is great, but it should be noted that the transporter responsible is ABCC2, meaning that it only represents one group of bile salt / biliary transporters. This limitation should be highlighted, given that particularly MDR2/ABCB4 loss as been shown as a genetic model of cholestatic liver disease.

A: We thank the reviewer for the suggestion.

The reviewer is correct in noting that our analysis of the bile salt transport was limited in the previous version of the manuscript (Previous Fig.1f and Suppl. Fig.2e), where we had employed cholyl-L-lysyl-fluorescein (CLF) to confirm the functionality and activity of 2 transporters: MRP2/ABCC2, from the ABCC subfamily and also the rate limiting step bile acid transporter BSEP/ABCB11, from the ABCB subfamily, which also transports CLF through the canalicular membrane ¹.

Six ABC transporters belonging to the ABCC, ABCB and ABCG subfamilies are located in the canalicular membrane. These can be classified into the conjugate transporter ABCC2/MRP2, the drug efflux pumps (ABCB1/MDR1 and ABCG2) and three transporters involved in bile formation: the bile acid exporter ABCB11/BSEP, the phosphatidylcholine transporter ABCB4/MDR2, and the cholesterol transporter ABCG5/G8².

In this revised version, we have expanded on our previous analysis. We have first analysed the expression of the 6 known canalicular membrane ABC transporters in our new bulk RNAseq data from HepOrg and freshly isolated hepatocytes, and our scRNAseq data from hepatocytes derived from assembloids. We have consistently and readily observed expression of BSEP (Abcb11) and MDR2 (Abcb4). ABCG5/G8 transporters were expressed, albeit at low levels. We have also detected expression of the transporter MRP2 (Abcc2), confirming our previous analysis. Conversely, we have not detected the drug transporters MDR1A (Abcb1a) or MDR1B (Abcb1b), in agreement with the results from fresh isolated hepatocytes and previous publications indicating that Mdr1 transporters are lowly expressed in mouse, contrary to human hepatocytes ³. These results are presented in New Suppl. Fig.7a, Suppl. Fig.8d, Suppl. Fig.12g and in Fig. R1a-c below.

Next, we have investigated the function of these transporters using live imaging. To assess the function of MDR2 we have employed a commercially available fluorescent phosphatidylcholine (PC)⁴. To ensure that this was specific, we generated HepOrg from WT and mutant *Mdr2*^{-/-} mice. The results show transport to BC in the WT but not in mutant *Mdr2*^{-/-} organoids (New Fig.2e, Suppl. Video 5 and below Fig.R1d).

We were unable to find specific analogues that would be transported specifically by either MRP2/Abcc2 or BSEP/Acbc11, therefore we have attempted to confirm our previous CLF results using another fluorescent analogue (CMFDA), which is also taken up by hepatocytes but intracellularly converted to 5-chloromethylfluorescein (CMF), which is excreted to the canalicular membrane specifically by BSEP and MRP2⁵. CMFDA was readily taken up by HepOrg and converted to CMF that was secreted into the bile canaliculi of hepatocyte organoids (New Fig.2e, Suppl. Video 4, also Figure R1e below). In addition, we have also tested the function of ATP-binding cassette (ABC) transporters in general by using 5-CFDA (5-carboxyfluorescein diacetate), which additionally allows us to test the functionality of esterases in hepatocyte organoids. 5-CFDA is nonfluorescent but in hepatocytes is converted into fluorescent carboxyfluorescein (CF) by intracellular esterases, and transported out of the cells via ATP-binding cassette (ABC) transport proteins from the hepatocyte apical membrane^{6,7}. We observed fluorescent signal in HepOrg, which varied between cells. Some cells accumulated CF, while others excreted it (New Suppl. Fig.3i 5-CFDA bubbly panel, Suppl. Video 1, Fig.R1f below). This indicates that some hepatocytes have functional esterase activity and functional ABC transport.

Taken together, these results confirm that our improved Hepatocyte Organoid model exhibits several functional bile acid/salt transporters, including BSEP, MRP2, MDR2 and, more broadly, ABC transporters.

These results are now presented in the new Fig.2e and new Suppl. Videos 1 and 3-5, and are summarized below in Fig. R1 for the reviewer's assessment.

Figure R1

Figure R1: Hepatocyte organoids exhibit functional bile acid transporters. **a.** Bulk RNA sequencing analysis of HM-Wnt and MM media HepOrg compared to freshly isolated hepatocytes. Heatmap represents the TPM (transcripts per million) values from the RNAseq for the indicated genes ($n = 3$ biological replicates). **b.** Dot plot shows gene expression from scRNAseq of HM-Wnt, MM media and assembloid hepatocytes for transporter genes. **c.** UMAP representing expression of various bile acid transporters in hepatocytes from assembloids. **d.** Still images from time-lapse imaging analysis of HepOrg showing functional transport of phosphatidylcholine (PC, 16:0-06:0 NBD PC) in WT but not MDR2-KO HepOrg; Fluorescent PC is shown in Royal LUT. Nuclei (SPY555-DNA, magenta) and actin (SiR-A, cyan) are also shown. Scale bar, 50 μ m. **e.** Still images from time-lapse imaging analysis of HepOrg with bile acid analogue CMFDA (Royal LUT). Nuclei (SPY555-DNA, magenta) and actin (SiR-A, cyan) are also shown. Scale bar, 50 μ m. **f.** Still images from time-lapse imaging analysis of HepOrg with bile acid analogue 5-CFDA (Royal LUT). Nuclei (SPY555-DNA, magenta) and actin (SiR-A, cyan) are also shown. Schematic represents metabolism of 5-CFDA in hepatocytes. Scale bar, 50 μ m.

The authors claim they generated peri-portal assemblies. It is interesting that historically it has been known that the peri-central hepatocytes experience increased Wnt activity. Can the authors explain why the hepatocytes express a peri-portal gene expression pattern in the presence of high Wnt levels.

A: We thank the Reviewer for this valuable comment. Wnt activity in the liver exhibits a gradient along the central-to-portal vein axis, with elevated levels in the pericentral region, as the reviewer correctly indicated, and reduced activity in the portal region. This is thought to be due in part to the varying expression of Wnt ligands across the axis. While WNT9b and WNT2 are pericentrally zoned^{8,9}, the canonical WNT3 ligand (which is the one present in our expansion and assembloid media) is found across the portal-central vein axis and not reported to be zoned¹⁰. This is consistent with the expression of the various Wnt targets. We and others have shown that the liver-specific Wnt targets glutamine synthetase (GS) or Cyp2e1 are exclusively expressed in the central region. In contrast, Wnt targets that are not liver specific but found across all tissues, like Znr3 or Lgr4, extend in their expression beyond the central region all the way to the portal area^{11,12}.

In our system, we grow and expand the HepOrg in HM-WNT CM, which comprises 30% of WNT3a medium, 15% RSPO1, and 3 μ M CHIR9902, a GSK-3 inhibitor that enhances Wnt signalling activity. Therefore, this combination of Wnt3a, RSPO1 and CHIR9902 results in high Wnt signalling activity in our expansion medium. For the periportal assembloids, however, we utilize the co-culture medium (MM media), which contains 30% WNT CM but is devoid of the other Wnt pathway modulators, thereby reducing the overall strength of Wnt signalling activation in the cultures, as we have now confirmed by gene expression of classical Wnt targets (See Fig.R1a above (WNT), and Fig.R2a below, for the reviewer only). We had reasoned that this reduction would be sufficient to induce periportal gene expression in the assembloids.

We have now evaluated the extent of periportal and pericentral gene expression in our hepatocyte organoids in the different conditions as well as in periportal assembloids. This involved analysing RNAseq data from HepOrg in different conditions, scRNAseq data and carrying out additional RNAscope and immunofluorescence analysis.

*From the new RNAseq data we found that assembloids and HepOrg in MM medium show higher levels of some hepatocyte genes that are considered to be periportal zoned (e.g. Hpx, Mup20, Aldob) as well as the periportal zoned cholesterol-bile acid pathway genes (Baat and Acaa1b), but also some pericentral genes such as Glul (GS) or Axin2, both also expressed in HM-WntCM, as expected (New Suppl. Fig. 7 and Fig.R2 below). These observations suggested that the cultures exhibited some degree of zonation. To assess that, we evaluated our scRNAseq data. To facilitate comparison of the expression of pericentral and periportal region genes in our scRNAseq, we have generated a consensus signature of pericentral and periportal genes. This was achieved by combining the expression of published and validated pericentral or periportal markers from three different publicly available datasets¹³⁻¹⁵, and using this as a pericentral or periportal score. **The results demonstrate that hepatocyte organoids cultured in high Wnt conditions exhibit a more pericentral score, while HepOrg transferred to MM co-culture medium acquire some periportal features and present a periportal score.** We have validated these results by RNAscope and immunofluorescence analysis for some portal (Gls2 – RNA, ALB, ECAD - protein) and central (Cyp1a1 – RNA, CYP2E1, GS -protein) gene markers (New Fig.2a and Suppl. Fig.7) Notably, the Immunofluorescence analysis indicates that the medium switch promotes some degree of zonation as we detect, within the same structure, some cells expressing portal markers while others express some central markers.*

Taken together these results indicate that our strategy of reducing the strength of the Wnt activation in the assembloids medium, by removing RSPO1 and CHIR9902 but retaining Wnt3a ligand, enables the hepatocytes to express a more periportal gene signature. We have included these

new results in New Fig.2 and Suppl. Fig7 in the revised version of the manuscript, also pasted below for the reviewer's assessment, Figure R2).

Figure R2

Figure R2 HepOrg are partially zoned, and increase periportal gene expression in MM medium. **a.** Dotplot representing gene expression of WNT pathway genes across conditions in hepatocytes from scRNAseq (For reviewer only). **b.** Periportal and pericentral zonation score calculated from consensus gene expression from Halpern et al., Ben Moshe et al., and Wu et al. show higher periportal gene expression in MM media devoid of Wnt factors, when compared to Wnt-enriched HM-Wnt media hepatocytes. **c.** Dotplot representing gene expression of periportal, pericentral and zoned cholesterol genes across conditions in hepatocytes from scRNAseq. **d.** Expression of periportal (*Gls2*) and pericentral (*Cyp1a1*) gene RNA visualised by RNAscope in HepOrg from MM media. **e.-f.** Immunofluorescence staining of periportal (ALB, Albumin, green **e**) and (E-CAD, E-cadherin, green, **f**) and pericentral (*GS*, Glutamine synthetase, magenta, **e**) and (*CYP2E1*, magenta, **f**) in liver tissue (top) or in HepOrg cultures grown in HM-Wnt and cultured in MM media for 7 days (bottom). Specific stainings are also shown in Fire LUT for the ease of viewing. Representative images are shown.

The authors claim a novel mechanism “to activate the myofibroblastic state in portal fibroblasts

independently of an immune activation” simply by increasing the number of portal fibroblasts 10-fold and increasing their cell-cell interactions. Why this is an interesting hypothesis, prior work has shown that the mechanics of the cellular environment (stiffness, etc.) can affect myofibroblast activity, so this may not be a novel observation.

A: We thank the reviewer for this suggestion, which significantly strengthens the manuscript. In the previous version of the manuscript, we presented a novel complex cellular model whereby the sole increase in the number of starting portal mesenchyme allowed us to model many aspects of biliary fibrosis, including gene expression, hepatocyte death, ductal cell expansion, collagen deposition, and bile flow obstruction. However, as the reviewer correctly pointed out, we had not explored the mechanistic underpinnings of the fibrotic phenotype observed.

Since the only difference between the homeostatic and fibrotic assembloids is the number of portal fibroblasts (Msc) at the starting time point, and in this revised version we now show that the tested cytokines from Msc secretome seem to have little direct impact on the phenotype observed (see new Suppl. Fig.15d-e and Fig.R3 below), we then have hypothesized that either (i) the imbalance in the cell-cell interactions, tilted towards an increase in the homotypic Msc-Msc interactions, or Msc-epithelial interactions or (ii) the cell-ECM interactions could explain the phenotype observed. To assess that, we have investigated our scRNAseq for potential cell adhesion, cell-ECM or ligand-receptor interactions that would be increased in the fibrotic assembloids compared to the homeostatic ones, focusing on the Msc-interactions that were also present in in vivo models of fibrosis (New Fig.5d and Suppl. Fig.15a). We found that the expression of the cell adhesion molecule Cdh11 was increased in fibrotic-like Msc, and the ligand-receptor interactions involving the collagen receptor ITGB1, the ECM modulators TIMP1 and TIMP2 amongst others (Supplementary Extended Dataset 4) significantly changed in intensity and magnitude. We selected some of these genes (Supplementary Extended Dataset 1), to perform a small knockdown screen in assembloids cultured in fibrotic conditions (10-fold excess Msc). As a first screening readout, we have discriminated shape changes over time, as fibrotic assembloids are more compacted compared to assembloids with physiological Msc numbers (see previous Fig.6 and Suppl. Fig.10a-b, now new Fig.5 and Suppl. Fig.13a-b). We have applied the shape descriptor algorithm DETECT (Detecting Temporal Shape Changes with the Euler Characteristic Transform)¹⁶ and developed the DETECT metric, to quantitatively and statistically compare spatio-temporal changes in morphology of organoids under different conditions. We found that inhibiting Msc-Msc cell adhesion by Cadherin-11 (CDH11)-knockdown consistently prevented the fibrotic-like phenotype in assembloids, even in the presence of 10-fold more mesenchyme. CDH11 knockdown also resulted in a marked reduction in collagen deposition and a reduction in hepatocyte cell death in assembloids cultured with 10-fold excess mesenchyme, which now resembled homeostatic control assembloids. Notably, this was specific to CDH11, as CDH2 knockdown did not rescue the fibrotic phenotype, although both can mediate Msc-Msc interactions (New Fig.6a-e and Suppl. Fig.15f-k and Fig.R4). Similar results were obtained when knocking out the collagen receptor Integrin beta 1 (ITGB1) in Msc prior to assembly (New Suppl. Fig.16 and Fig.R5). None of the other molecules tested were able to prevent the change in organoid shape or rescue the fibrotic phenotype, including none of the selected ECM modulators or the cytokines tested (New. Suppl. Fig.15e).

Interestingly, the two positive hits in our study, namely CDH11 and ITGB1, are molecules involved in the Msc-interactions with the environment, either other Msc cells or the ECM. We speculate a potential scenario whereby excess portal mesenchyme increases the cell adhesion between Msc-Msc cells, which could potentially increase the fibrogenic response of the Msc, increase collagen deposition, and finally also subsequently Msc-ECM interactions. This combined would result in the fibrotic phenotype observed. Supporting this hypothesis, we have consistently observed that monocultures of portal fibroblasts grown at high density in 3D, but not in sparse culture conditions in 3D, express markers of fibrogenic activation and deposit fibrillar collagen (New Suppl. Fig.18 and Fig.R6), in agreement with

previous publications that show that CDH11 expression increases with high seeding density¹⁷, and that increase in CDH11-CDH11 interactions induces collagen production¹⁸.

While our results show a clear direct effect whereby preventing Msc-Msc interactions through CDH11-knockdown or Msc-ECM interactions via ITGB1-knockout prevents the fibrotic phenotype caused by excess mesenchyme, we want to acknowledge that we do not investigate other mechanisms namely mechanics, as pointed by the reviewer, or the whole fibrotic secretome (we only study some of the cytokines). In addition, how increasing Msc-Msc interactions translates into the exact intracellular molecular effectors that instruct hepatocyte cell death and duct cell expansion remains unknown. We believe that the downstream mechanisms behind involve a complex cascade of exceedingly complex events. During the fibrotic reaction, a myriad of biochemical signals as well as biophysical properties (e.g. mechanics) change dynamically. These individually, as well as collectively, impact the organ at the tissue, cellular and molecular scales to culminate in the emergence of the fibrotic phenotype. The in-depth analysis of these complex cascades is extremely interesting but constitutes a full project in itself that could take years to complete, and we would like to address in future studies.

In this revised version of the manuscript, we have included these new results in the new Fig.6 and Suppl. Fig.15-16 and 18 and also toned down our statements regarding mechanisms of fibrosis, limited our conclusions to the observed results, and speculated on the potential implications in the discussion. These are also pasted here for the Reviewer as Fig.R3, R4, R5 and R6.

Figure R3

Figure R3: Inflammatory cytokines do not have induce a fibrotic-like phenotype on mesenchyme. **a.** Schematic of experimental setup. Freshly isolated and sorted Msc cells are plated in the presence or absence of cytokines. Second harmonic generation (SHG) imaging for collagen fibres is performed after indicated number of days. **b.** Immunofluorescence images show lack of collagen deposition (SHG) in naïve mesenchyme following exposure to inflammatory cytokines for 7days.

Figure R4

Figure R4: Knockdown of CDH11 in Msc prior to assembly results in reduction of fibrotic-like phenotype in assembloids by altering Msc-Msc interactions. **a.** Schematic illustration of siRNA-mediated knock down experiments in portal mesenchyme; mTom PDGFRa-H2B-GFP-Msc incubated with targeting or non-targeting siRNAs were combined with WT HepOrg and mCFP Chol to make assembloids. Following assembloid generation, structures were analysed looking at hepatocyte apoptosis and fibrillar collagen deposition. **b.** Live imaging of the periportal liver assembloids where portal fibroblasts were transfected with either non-targeting pool of siRNA (left) or Cdh11 siRNA (right). Top panel depicts day 0. Bottom panel depicts periportal assembloids on day 2 of the live imaging. Membranes of assembloids are visualized with SiR-actin (grey), nuclei of portal fibroblast with PDGFRa-H2B-GFP (green), membranes of the portal fibroblasts with Tomato signal (magenta). Fire LUT images present examples of the segmentation of the assembloid borders (middle panels) or mesenchyme membranes (left panels) on maximum intensity projection images. Cdh11 knockdown assembloids present bubbly/grape shape with lower number of portal fibroblasts compared to the control group. Scale bar 50µm. **c.** Schematic of the DETECT distance ratio metric (d1/d0) (see methods for details). If the bubbly/shape assembloid remains that shape, the values DETECT distance ratio metric =1 (top panel), while if it transitions to small circle shape the metric would be less than 1 (bottom panel). **d.** Violin plots showing DETECT distance ratio for the non-targeting pool and Cdh11 KD group. Cdh11 KD assembloids significantly higher DETECT distance ratio compared to and non-targeting pool. Mann Whitney test $p=0,0008$ with Mann Whitney U - 42. Dots, individual assembloids from $n=2$ independent experiments. **e.** Second harmonic generation microscopy images of the fixed periportal assembloids showing prevention of the fibrillar collagen (yellow) deposition in Cdh11 knockdown assembloids (right) compared to the control group (left). Membranes of assembloids are visualized with SiR-actin (grey) while nuclei of portal fibroblast with PDGFRa-H2B-GFP (green). Representative images from $n=2$ biological replicates.

Figure R5

Figure R5: Knockout of ITGB1 in Msc results in reduction of fibrotic-like phenotype in assembloids. **a.** Schematic of experimental set up. Mesenchyme was isolated from *Itgb1-fl/fl* x *ZsGreen* mice, and grown to passage 1 in standard 2D-culture, then subject to Ad5-Cre viral transduction to induce cassette recombination, resulting in a mixed Msc culture of ZsGreen-negative *Itgb1*-WT cells and ZsGreen-positive *Itgb1*-KO cells. Msc were subsequently sorted for the expression of ZsGreen, and used to generate assembloids. **b.** Representative pictures of *Itgb1*-KO cells expressing ZsGreen before and after FACS sorting. **c.** Validation of *Itgb1*-KO by qRT-PCR against *Itgb1* gene. **d.** Live imaging analysis of CTRL Msc and *Itgb1*-KO Msc containing fibrotic-like assembloids, showing that *Itgb1*-KO prevents collapse of the structures. Representative images from *n*=2 biological replicates are shown. Scale bar, 50 μ m. **e.** Heatmap of the DETECT distance metric for the DETECT results of *Itgb1*-KO Msc and CTRL-Msc containing fibrotic assembloids. **f.** Visualisation of morphological variations of *Itgb1*-KO Msc and CTRL-Msc containing fibrotic assembloids using t-SNE based on DETECT calculations that integrate all time points. **g.** Immunofluorescence images of fibrotic-like assembloids formed by *Itgb1*-KO Msc (green). Representative images from *n*=2 biological replicates are shown. Msc (green), cholangiocytes (tdTom-nuclei, magenta) are shown, together with actin (phalloidin, grey) and nuclei (DAPI, blue) staining. Note intact hepatocyte nuclei; scale bar, 50 μ m. **h.** Fibrotic-like assembloids containing *Itgb1*-KO Msc (green) deposit less collagen, as visualised by second harmonic generation (SHG) imaging (yellow). Representative images are shown; scale bar, 50 μ m. **i.** Quantification of fibrous collagen deposition (SHG) from (i). Graph represents mean \pm SEM of *n*=4 biological replicates from *n*=2 independent experiments, with each dot denoting biological replicate; paired *t*-test. *p*=0.0228.

Figure R6

Figure R6: Induction of Msc-Msc direct contacts partially mimics fibrotic-like phenotype observed in fibrotic-like assembloids. **a.** Brightfield images of 2500 Msc cells seeded in sparse or confluent/aggregated condition showing $n=2$ independent biological replicates (right). Schematic representation of the experiment, left. Scale bar, 500 μm . **b-c.** Representative images (**b**) and quantification (**c**) of collagen fibre deposition as visualised by second harmonic generation (SHG) in sparse and confluent/aggregated condition. **d.** Gene expression analysis (qRT-PCR) of sparse and confluent/aggregated Msc for selected genes. Graph represents mean \pm SEM of $n=9$ biological replicates from $n=3$ independent experiments, with each dot denoting biological replicate; Mann-Whitney test.

This finding may explain to some degree the irreversibility of fibrosis observed in the clinic but may not really be related to true causal/mechanistic relationships between portal fibroblasts and the occurrence of fibrosis.

A: This is a very interesting point raised here by the Reviewer. We agree that our experiments cannot determine whether the observed phenotype would be the initiation or progression towards irreversibility of fibrosis.

Interestingly, one piece of new data that might point towards the answer to this question is that in our model we did not detect mesenchymal cell expansion. This is similar to the proliferative status of mesenchymal cells in all published models of liver fibrosis we have studied (6 different fibrosis models from 6 different studies). Only CCl4-damage models, with hepatotoxic fibrosis driven by hepatic stellate cells, show increased proliferation of mesenchyme. The other models, in particular the DDC and BDL models, both modelling biliary fibrosis similar to our assembloids, show similar levels of mesenchyme proliferation to our model (Fig.R7 below, for Reviewer only). A potential plausible explanation could be that at the time fibrosis is established, mesenchymal cell proliferation is terminated. Our model, where we add an excess number of mesenchymal cells from the start, might be reflecting this advanced stage, mimicking the state of irreversibility of fibrosis, rather than initial steps in the fibrotic response. This hypothesis is something we would like to investigate in future studies. Therefore, we present these results below for the Reviewer only, and also discuss this concept in the manuscript (lines 538-539). Would the reviewer consider that these results are necessary to be included in the manuscript, we would be happy to follow his/her advice on that.

Figure R7

Figure R7: Expression of *Mki67* gene in different models of fibrosis comparing to the *Msc* on our fibrotic-like assembloids (THIS STUDY). HSC, hepatic stellate cell, grey; FIB, fibroblast, green; VSMC, vascular smooth muscle cell, blue. PHx, partial hepatectomy; DDC; CCI4; TAA; APAP; BDL, bile duct ligation – all specify the type of damage applied to mice. Where appropriate, time of damage is also specified: 17d, 17 days; 8d, 8 days; 72h, 72 hours; 2w, 2 weeks; 3w, 3 weeks; 4w, 4 weeks. The publication is specified on the left.

Relatedly, the question is whether addition of stellate cells may change the growth patterns/responses of this assembly.

A: As the reviewer points out, hepatic stellate cells (HSCs) are known drivers of fibrosis, especially hepatotoxic-driven fibrosis. However, in our manuscript we focused on reconstructing the periportal region of the liver lobule. The periportal region around the ductal tree is mostly populated by portal fibroblasts, with little presence of hepatic stellate cells (HSCs), as shown in our previous work and the work of others¹⁹⁻²¹. Since we have focused our work on recapitulating only this area of the liver, we did not consider including HSCs at the time of the original submission. Furthermore, in contrast to hepatotoxic fibrosis, HSCs play a less prominent role in biliary fibrosis (e.g.²²), although, as the reviewer rightly points out, they are also recruited to the diseased areas after activation of portal fibroblasts^{22,23}. Therefore, in light of the reviewer's comment, we set out to investigate the role of HSCs in our assembloids.

We have attempted to isolate and expand HSCs in culture to achieve sufficient numbers to incorporate into our culture system. Unfortunately, our efforts to expand HSCs in culture have been unsuccessful so far. We developed a FACS strategy to isolate PDGFR α +SCA1+/- HSCs from PFs. We took advantage of the bona fide portal fibroblast marker CD34²⁰, and confirmed that all CD34- cells were HSCs, as they expressed significantly higher levels of Reelin. We attempted to expand these cells prior to assembloid culture. Unfortunately, the HSC fraction (CD34-) did not grow in vitro except in one case (n=1 of n=5 independent experiments), which precluded its further analysis. In contrast, SCA1+ portal fibroblasts (PDGFR α +SCA1+CD34+) were easily expanded until P5 (n=5 experiments). Similarly, SCA1-portal

fibroblasts (PDGFR α +SCA1-CD34+) could also be expanded, but only to P1 and with much lower efficiency (only 2 of 5 experiments) (Fig.R8 below, for the Reviewer only).

Given (1) the low yield of all sub-fractionated populations, (2) our inability to expand CD34- cells and (3) lack of access to the *Lrat-CRE* mouse, which specifically marks the HSC population, we cannot currently perform HSC experiments with assembloids. We believe that the sorting and expansion of HSC would require time beyond the scope of this revision and is a separate project that we hope to pursue in the lab in the future.

We have discussed this caveat in the revised version of the manuscript, lines 545-548.

Figure R8

Figure R8: Hepatic stellate cells (HSC) do not expand in vitro (adapted from ¹⁹). **a.** Sorting strategy to isolate HSC from the liver. After gating on single cells, the hematopoietic and endothelial cells are excluded using CD11b, CD45 and CD31 antibodies. Then, the mesenchyme is selected gated on PDGFR α -H2BGFP marker, followed by separation into SCA1+ and SCA1- cells, which are further gated on CD34+ and CD34-. **b.** Expression of markers in the different sub-populations (PDGFR α +SCA1+CD34- (red), PDGFR α +SCA1+CD34+ (yellow), PDGFR α +SCA1-CD34+ (teal), PDGFR α +SCA1-Cd34- (purple)). **c.** HSC cells do not expand in culture; passaging number for the different mesenchyme fractions; open circle denotes passage while line corresponds to no further expansion; black circle denotes using all of the cells for a co-culture experiment.

Minor Criticisms:

Line 42/43 refers to “cholestatic liver diseases, ranging from primary biliary cholangitis (PBC) to stones, amongst others” is too vague, and clinically incorrect: cholestatic liver disease can have genetic or immune origin, and is worthy of being modeled in vitro. Gall stone disease, however, is different. The cited references are also not directly reflective of this statement.

A: We apologise for this inaccuracy from our side. We have now re-written this paragraph and added appropriate references.

Figure 3, panel d – x-axis label “DC” should probably mean “CC” for cholangiocytes or “BC” for biliary cells?

A: Corrected, now in New Fig.3d.

Referee #2 (Remarks to the Author):

This elegant study comes from leaders in liver organoid modeling and covers an important gap as the majority of current studies are either done in animal systems, 2D culture or short term liver slice systems - all with grave disadvantages. The authors show an improved model in which optimization and different ratios of mesenchymal cells result in larger organoid with functional bile canaliculi and drainage as well as an induction of fibrogenesis. Despite the elegance of the system and the superb images, the study also has weaknesses that need to be addressed:

A: We thank the reviewer for the positive comments and noting that our findings cover an important gap in the field. We hope that this improved manuscript has now corrected the previous weaknesses, and thanks to all the reviewer comments, the quality has improved significantly.

1. Although the model is elegant, many parts of the manuscript remain at a descriptive levels with a lack of functional testing of the many proposed ligand-receptor interactions (Col1a1-Ddr2, Col3a1-Ddr2, Gas6-Axl, Lgals-Intgb1, Timp2-Itgb1, Timp1-Cd63, Col1a1-Ddr1, Col3a1-Ddr1, Cxcl12-Itgav, Col6a1-Sdc1, Apoa1-Lrp1. In 2023, ligand-receptor interactions should not only be displayed but also functionally investigated.
2. Likewise, the authors suggest a role for homotypic (Msc-Msc) interactions in driving fibrosis, but do not provide functional evidence for the concept. The authors do not show a role for any of the proposed mediators (Il6, Il33, Ccl11, Cxcl5, Cxcl1 and Cxcl12m Mmp2, Mmp3) and the corresponding receptors.

A: We thank the reviewer for raising these very important points here, investigation of which significantly strengthened the manuscript. Because both points refer to the experimental support for how excess of fibroblasts induces fibrotic-like state in the absence of an immune niche, we answer both points together below.

In the previous version of the manuscript, we presented a novel complex cellular model whereby the sole increase in the number of starting portal mesenchyme resulted allowed us to model many aspects of biliary fibrosis, including gene expression, hepatocyte death, modest ductal cell expansion, collagen deposition, and bile flow obstruction. We had described many changes in cell-cell interactions, specially within the Msc-Msc interactions, including ligand-receptor interactions as well as secretion of different cytokines. However, as the reviewer correctly pointed out, we had not explored the mechanism underpinning the fibrotic phenotype observed.

To functionally test some of the cell interactions, we first checked our scRNAseq data. Given that we have expanded our scRNAseq analysis with additional biological replicates, we have first re-analysed this new scRNAseq dataset, focusing on the Msc-interactions that were also present in in vivo models of fibrosis^{14,20,21,24-30}. Overall, the new scRNAseq yielded similar results and confirmed many of the previous changes in cell-cell interactions. We found that the expression of the cell adhesion molecule Cdh11 was increased in fibrotic-like Msc, and the ligand-receptor interactions involving the collagen receptor ITGB1, the ECM modulators TIMP1 and TIMP2, as well as the cytokines/ligands Cxcl12, Ccl11 and Cxcl1 were increased in fibroblasts from fibrotic-like assembloids (see Supplementary Dataset 4 for details). For the secreted cytokines, we had validated these findings by cytokine array analysis, which confirmed the secretion of CCL11, MMP2 and MMP3 by the mesenchymal cells from fibrotic-like assembloids, but not from homeostatic assembloids (Previous Fig.6i-j and Suppl. Fig.14h).

Following from these results, to functionally test some of the cell interactions, we first tested the hypothesis that the increase in the levels of cytokines released could explain the fibrotic phenotype of the fibrotic-like Msc. Inhibiting selected secreted cytokines (CXCL1, CX3CL1, CCL11, CXCL12) did not

rescue the fibrotic phenotype (New Suppl. Fig.15e and Fig.R9c). Similarly, addition CXCL1, CCL11 or CXCL12 did not change the Msc phenotype (New Suppl. Fig.15d-e and Fig.R9). These results suggested that the paracrine signalling from Msc was secondary to the fibrotic-like phenotype observed, at least for the cytokines tested. Then, we next hypothesized that either (i) the imbalance in the cell-cell interactions, tilted towards an increase in the homotypic Msc-Msc interactions, or Msc-epithelial interactions or (ii) the cell-ECM interactions (like the Col3a1 or Spp) could explain the phenotype observed. We selected some of these genes to perform a small knockdown screen in assembloids cultured in fibrotic conditions (10-fold excess Msc). As a first screening readout, we discriminated shape changes over time, as fibrotic assembloids are more compacted compared to structures with physiological Msc numbers. We applied the shape descriptor algorithm DETECT (Detecting Temporal Shape Changes with the Euler Characteristic Transform)¹⁶, and developed the DETECT metric, to quantitatively and statistically compare spatio-temporal changes in morphology of organoids under different conditions. We found that inhibiting Msc-Msc cell adhesion by Cadherin-11 (CDH11)-knockdown consistently prevented the fibrotic-like phenotype in assembloids, even in the presence of 10-fold more mesenchyme (New Fig.6 and Suppl. Fig.15, and Fig.R4). Specifically, assembloids did not change shape over time compared to controls, which significantly compacted over time. Additionally, CDH11 knockdown also resulted in a marked reduction in collagen deposition and a reduction in hepatocyte cell death in assembloids cultured with 10-fold excess mesenchyme (New Fig.6 and Suppl. Fig.15, and Fig.R4), which now resembled homeostatic control assembloids. Notably, this was specific to CDH11, as CDH2 knockdown did not rescue the fibrotic phenotype, although both can mediate Msc-Msc interactions (New Suppl. Fig.15). Similar results were obtained when knocking out the collagen receptor ITGB1 in Msc (New Suppl. Fig.16 and Fig.R5). None of the other ligand-receptor or ECM molecules tested were able to prevent the change in organoid shape or rescue the fibrotic phenotype (Fig. R9).

Interestingly, the two positive hits in our study, namely CDH11 and ITGB1, are molecules involved in the Msc-interactions with the environment, either other Msc cells or the ECM. We speculate a potential scenario whereby excess portal mesenchyme increases the cell adhesion between Msc-Msc cells, which could potentially increase the fibrogenic response of the Msc, increase collagen deposition, and finally also subsequently Msc-ECM interactions. This combined would result in the fibrotic phenotype observed. Supporting this hypothesis, we have consistently observed that monocultures of portal fibroblasts grown at high density in 3D, but not in sparse culture conditions in 3D, express markers of fibrogenic activation and deposit fibrillar collagen (New Suppl. Fig.18 and Fig.R6), in agreement with previous publications that show that CDH11 expression increases with high seeding density¹⁷, and that increase in CDH11-CDH11 interactions induces collagen production¹⁸.

While our results show a clear direct effect whereby preventing Msc-Msc interactions through CDH11-knockdown or Msc-ECM interactions via ITGB1-knockout prevents the fibrotic phenotype caused by excess mesenchyme, we want to acknowledge that we do not investigate other mechanisms or the whole fibrotic secretome (we only study some of the cytokines). In addition, how increasing Msc-Msc interactions translates into the exact intracellular molecular effectors that instruct hepatocyte cell death and duct cell expansion remains unknown. We believe that the downstream mechanisms behind involve a complex cascade of exceedingly complex events. During the fibrotic reaction, a myriad of biochemical signals as well as biophysical properties (e.g. mechanics) change dynamically. These individually, as well as collectively, impact the organ at the tissue, cellular and molecular scales to culminate in the emergence of the fibrotic phenotype. The in-depth analysis of these complex cascades is extremely interesting but constitutes a full project in itself that could take years to complete, and we would like to address in future studies.

In this revised version of the manuscript, we have included these new results in the new Fig.6 and Suppl. Fig.15-16 and also toned down our statements regarding mechanisms of fibrosis, limited our

conclusions to the observed results, and speculated on the potential implications in the discussion. These are also pasted below for the Reviewer as Fig. R9, R3, R4, R5 and R6.

Figure R9

Figure R9: Ligand-receptor and Msc-Msc interactions that do not result in a phenotype. Various cell-cell interactions tested do not rescue the fibrotic-like phenotype. DETECT analysis of assembloids treated with blocking antibodies against specified proteins, or matched-species control; violin plot shows DETECT distance ratios; Mann-Whitney test.

Figure R3

Figure R3: Inflammatory cytokines do not have induce a fibrotic-like phenotype on mesenchyme. **a.** Schematic of experimental setup. Freshly isolated and sorted Msc cells are plated in the presence or absence of cytokines. Second harmonic generation (SHG) imaging for collagen fibres is performed after indicated number of days. **b.** Immunofluorescence images show lack of collagen deposition (SHG) in naïve mesenchyme following exposure to inflammatory cytokines for 7days.

Figure R4

Figure R4: Knockdown of CDH11 in Msc prior to assembly results in reduction of fibrotic-like phenotype in assembloids by altering Msc-Msc interactions. **a.** Schematic illustration of siRNA-mediated knock down experiments in portal mesenchyme; mTom PDGFRa-H2B-GFP-Msc incubated with targeting or non-targeting siRNAs were combined with WT HepOrg and mCFP Chol to make assembloids. Following assembloid generation, structures were analysed looking at hepatocyte apoptosis and fibrillar collagen deposition. **b.** Live imaging of the periportal liver assembloids where portal fibroblast were transfected with either non-targeting pool of siRNA (left) or Cdh11 siRNA (right). Top panel depicts day 0. Bottom panel depicts periportal assembloids on day 2 of the live imaging. Membranes of assembloids are visualized with SiR-actin (grey), nuclei of portal fibroblast with PDGFRa-H2B-GFP (green), membranes of the portal fibroblasts with Tomato signal (magenta). Fire LUT images present examples of the segmentation of the assembloid borders (middle panels) or mesenchyme membranes (left panels) on maximum intensity projection images. Cdh11 knockdown assembloids present bubbly/grape shape with lower number of portal fibroblasts compared to the control group. Scale bar 50um. **c.** Schematic of the DETECT distance ratio metric (d1/d0) (see methods for details). If the bubbly/shape assembloid remains that shape, the values DETECT distance ratio metric =1 (top panel), while if it transitions to small circle shape the metric would be less than 1 (bottom panel). **d.** Violin plots showing DETECT distance ratio for non-targeting pool and Cdh11 KD group. Cdh11 KD assembloids significantly higher DETECT distance ratio compared to and non-targeting pool. Mann Whitney test $p=0,0008$ with Mann Whitney U - 42. Dots, individual assembloids from $n=2$ independent experiments. **e.** Second harmonic generation microscopy images of the fixed periportal assembloids showing prevention of the fibrillar collagen (yellow) deposition in Cdh11 knockdown assembloids (right) compared to the control group (left). Membranes of assembloids are visualized with SiR-actin (grey) while nuclei of portal fibroblast with PDGFRa-H2B-GFP (green). Representative images from $n=2$ biological replicates.

Figure R5

Figure R5: Knockout of ITGB1 in Msc results in reduction of fibrotic-like phenotype in assembloids. a. Schematic of experimental set up. Mesenchyme was isolated from *Itgb1-fl/fl* x *ZsGreen* mice, and grown to passage 1 in standard 2D-culture, then subject to Ad5-Cre viral transduction to induce cassette recombination, resulting in a mixed Msc culture of *ZsGreen*-negative *Itgb1*-WT cells and *ZsGreen*-positive *Itgb1*-KO cells. Msc were subsequently sorted for the expression of *ZsGreen*, and used to generate assembloids. **b.** Representative pictures of *Itgb1*-KO cells expressing *ZsGreen* before and after FACS sorting. **c.** Validation of *Itgb1*-KO by qRT-PCR against *Itgb1* gene. **d.** Live imaging analysis of CTRL Msc and *Itgb1*-KO Msc containing fibrotic-like assembloids, showing that *Itgb1*-KO prevents collapse of the structures. Representative images from *n*=2 biological replicates are shown. Scale bar, 50 μ m. **e.** Heatmap of the DETECT distance metric for the DETECT results of *Itgb1*-KO Msc and CTRL-Msc containing fibrotic assembloids. **f.** Visualisation of morphological variations of *Itgb1*-KO Msc and CTRL-Msc containing fibrotic assembloids using t-SNE based on DETECT calculations that integrate all time points. **g.** Immunofluorescence images of fibrotic-like assembloids formed by *Itgb1*-KO Msc (green). Representative images from *n*=2 biological replicates are shown. Msc (green), cholangiocytes (tdTom-nuclei, magenta) are shown, together with actin (phalloidin, grey) and nuclei (DAPI, blue) staining. Note intact hepatocyte nuclei; scale bar, 50 μ m. **h.** Fibrotic-like assembloids containing *Itgb1*-KO Msc (green) deposit less collagen, as visualised by second harmonic generation (SHG) imaging (yellow). Representative images are shown; scale bar, 50 μ m. **i.** Quantification of fibrous collagen deposition (SHG) from (i). Graph represents mean \pm SEM of *n*=4 biological replicates from *n*=2 independent experiments, with each dot denoting biological replicate; paired t-test.

Figure R6

Figure R6: Induction of Msc-Msc direct contacts partially mimics fibrotic-like phenotype observed in fibrotic-like assembloids. **a.** Brightfield images of 2500 Msc cells seeded in sparse or confluent/aggregated condition showing $n=2$ independent biological replicates (right). Schematic representation of the experiment, left. Scale bar, 500 µm. **b-c.** Representative images (**b**) and quantification (**c**) of collagen fibre deposition as visualised by second harmonic generation (SHG) in sparse and confluent/aggregated condition. **d.** Gene expression analysis (qRT-PCR) of sparse and confluent/aggregated Msc for selected genes. Graph represents mean \pm SEM of $n=9$ biological replicates from $n=3$ independent experiments, with each dot denoting biological replicate; Mann-Whitney test.

Moreover, a previous study (Wang et al, Sci Transl Med 2023) suggested neurotrophic receptor tyrosine kinase 3-neurotrophin 3 interactions as autocrine/"homotypic interaction" driver of HSC activation and fibrosis. Hence, this concept is not entirely novel. Did the authors determine this pathway in their model?

A: We thank the reviewer for bringing up to our attention this important report by Wang and colleagues from the Friedman lab, showing an activation of autocrine loop in hepatic stellate cells. To investigate the impact of neurotrophin-3 (NTF3) / neurotrophic receptor tyrosine kinase-3 (NTRK3) in portal fibroblasts interactions in our model, we first checked the expression and interaction of NTF3-NTRK3 in our scRNAseq data and in publicly available datasets. We did not detect the receptor Ntrk3 in portal fibroblasts (shown below for Reviewer only, Fig.R10), although the ligand was present in the datasets. This suggest that for portal fibroblasts this autocrine/homotypic interaction might not play a role, contrary to HSCs as observed by Wang et al.

Figure R10

Figure R10: No expression of NTF3-NTRK3 axis in portal fibroblasts from various datasets. Dotplot showing expression of *Ntf3* and *Ntrk3* genes in fibroblasts from liver cell atlases^{20,21,24,26,27,30}, not all datasets discriminate between portal and central vein fibroblasts. While the ligand *Ntf3* is expressed in majority of fibroblasts, the receptor *Ntrk3* gene is not.

Likewise, the alternative hypothesis of suppressive mediators from epithelial compartments that are being diluted out in the setting of increased Msc is not discussed or experimentally tested. Please see also comment on apoptotic hepatocytes below.

A: We thank the reviewer for noticing this oversight from our side. This is an extremely valid point that we have not addressed at the time of the initial submission.

To answer this question, we have re-analysed the cell-cell interactions with the new replicates of scRNAseq, paying special attention to the different interactions between epithelium and mesenchyme. We have found that some potential mediators could be diluted between homeostasis and fibrotic-like conditions, demonstrating as disappearance of cell-cell interactions. Cholangiocytes in fibrotic-like conditions lost several interactions with mesenchyme, for e.g. integrin (*Itgb1*) interaction with mesenchyme galectin (*Lgals1*), and *Cd63* interaction with mesenchymal *Timp1*. Cholangiocytes also lost an interaction with other cholangiocytes, using *Fn1-Plaur* axis. The same was true for hepatocytes, which lost *Serp1 G1 (Serping1)* interaction with mesenchymal *Lrp* (LDL Receptor Related Protein 1) as well as *C3* interaction with *Cd81* of Msc. Hepatocytes to hepatocyte signalling was also altered in strength and magnitude, involving e.g. *Abca1-Apoe* transporter-apolipoprotein interaction, or *Asgr1-Hp* interaction. We have included these results here for the Reviewer below (Fig.R11 and Suppl. Fig.15a) However, because we do not validate these results, as it would require complicated knock in strategies, in the resubmitted version of the manuscript we have opted to only discuss this point.

Figure R11

Figure R11: Loss and gain of cell-cell interactions in fibrotic-like assembloids. Circular plots representing the inferred top 30 cell-cell interaction in fibrosis-like and homeostasis-like assembloids, obtained from scRNAseq data analysis Plot shows interactions that are reported in literature and shared between homeostasis and fibrosis-like assembloids (red), shared but not mentioned in literature (cyan) or unique to one model (black) selected from the top 100 significant interactions from both models.

3. While the authors clearly achieved their goal of developing a model with physiological bile canaliculi, they did not demonstrate sufficiently the relevance of this system for modeling liver physiology and pathophysiology.

A: We apologise for not having sufficiently highlighted the relevance of our model system and we thank the reviewer for this comment, which we believe helps us strengthen the message of our manuscript. We clarify this below:

Regarding the physiological point: *Hepatocytes are uniquely polarized cells, with one or more apical poles contributing to several bile canaliculi of directly opposing hepatocytes. Hepatocyte polarity is essential for hepatocyte function and loss of polarity results in hepatocyte damage^{31,32}. In our previous version we have only done side-by-side comparisons of our model to the in vivo tissue but we have not contextualized its relevance to study liver physiology compared to existing models.*

For this revision, we have now done side-by-side comparisons between the 2D sandwich culture of adult mouse hepatocytes, previously published model of 3D HepOrg³³, our improved 3D HepOrg cultures, and adult mouse liver tissue. Our improved HepOrg present bile canaliculi with a 1.5-2 μm diameter, which is within the range of the in vivo tissue baseline (1-2 μm)^{34,35}. In addition, bile canaliculi form a network in vivo, which is highly interconnected in 3D space and is fully functional - it readily transports fluorescent bile acid analogues to the apical bile canaliculi domain. These architectural features are not seen in previously published HepOrg methods, which present significantly shorter and barely connected BC (New Fig1. And Suppl. Fig.1). Also, despite some of these BC features are seen in hepatocytes cultured in 2D sandwich culture, hepatocytes in 2D sandwich culture lose their polarity within days, and bile canaliculi transport can no longer be studied in these models, as previously reported^{31,36-38}.

In addition, recent work has shown that bile canaliculi remodel during development. In the developing liver, bulkhead structures, which are extensions of the apical membrane sealed by tight junctions in the lumen between two adjacent hepatocytes, emerge in both human and mouse hepatocytes but are inexistent in adult healthy liver tissue³⁸. Interestingly, the data acquired in the course of this revision shows that our improved HepOrg present apical bulkheads when grown in expansion medium HM-Wnt, but bulkheads are absent when we grow the cultures in co-culture MM medium. These results are in agreement with the notion that the expanding cultures more closely resemble a developing, proliferative tissue, while the HepOrg in the co-culture MM-medium acquire more features of an adult tissue, including a narrower BC without bulkheads (Fig.R12 for reviewer's assessment below). On the contrary, hepatocytes in 2D-sandwich culture always present bulkheads, as previously reported³⁷(Fig.R12).

Overall, the above comparisons confirm that our 3D model is physiologically closer to the in vivo scenario in terms of structure and function compared with previous methods, and moreover that it can recapitulate developmental processes occurring in vivo.

Figure R12

Figure R12: Optimized HepOrg cultures exhibit physiological bile canaliculi size and network comparable to that of native liver tissue. **a.** Immunofluorescence staining of bile canaliculi (CD13, green) and F-actin (Phall, grey) in HepOrg cultures grown in HM-Wnt media (left) or in MM for 7 days (middle). Right, mouse healthy liver tissue control. Note the similarity between the bile canaliculi of HepOrg in MM (middle) and tissue (right). Representative images are shown. Scale bar, 50 μm. Magnification, 10 μm. **b.** Immunofluorescence staining for bile canaliculi (CD13, green) and F-actin (phalloidin, grey) of hepatocytes in 2D-sandwich culture, and HepOrg cultures grown in Hu et al., and Peng et al. media, as indicated. Note the difference in organoid size and length of the bile canaliculi when compared to the optimized medium in (a). Nuclei are stained with DAPI (blue). Representative immunofluorescence images are shown (n=3 independent experiments). Scale bar, 50 μm.

[Redacted text]

(Fig.R13 for the Reviewer only).

Similarly, HepOrg generated from MDR2 knockout mouse, a well-characterized model for primary sclerosing cholangitis⁴³, showed the presence of dilated canaliculi, inward blebs, and accumulation of liver rosettes, all of them observed in Mdr2^{-/-} liver tissue but absent in control tissue and HepOrg derived from WT littermates (New Fig.6 and Suppl. Fig.17, below presented in Fig.R14 for the reviewer). Additionally, by combining these Mdr2^{-/-} HepOrg with WT cholangiocytes and WT mesenchymal cells to generate chimeric Mdr2^{-/-};WT assembloids, we now provide the proof-of-concept that the assembloid system allows the study of cell-autonomous mechanisms in biliary fibrosis (New Fig.6 and Fig.R14).

These new data further validate that our system can be used to model liver disease, in addition to fibrosis. In the new version of the manuscript, we have added the Mdr2 results but not the DCA data, as we feel that including it could dilute the main message of the paper, and we decided to only show

them to the reviewer. Nevertheless, if the reviewer and editor think this would be a useful addition, we would be willing to add them to the manuscript.

Taken together, our results indicate that the presence of a physiological BC is critical when we aim to model physiological liver tissue and liver pathophysiology, specifically cholestatic liver disease, which cannot be modelled well in the published methods as they present some features of cholestasis at the baseline. Critically, our results also show that the past focus of organoid models on using gene expression, albumin secretion or enzymatic activity as a readout of function is not sufficient to guarantee physiological bile canaliculi. We hope that these new additions, combined with our previous analyses, now better illustrate the significance, importance and novelty of our findings, and clarifies this point for the Reviewer.

[Redacted figure]

Figure R13: Modelling acute cholestatic response in optimised hepatocyte organoids using DCA. **a.** Schematic illustration showing hallmarks of cholestasis namely, liver rosette combined with dilated BC and apical bulkheads. The effect of deoxycholic acid (DCA) overload-induced cholestatic response in HepOrg is shown. Bile canaliculi (green), hepatocytes (orange). **b.** Maximum intensity projection of F-actin (magenta) and CD13 (green) in DMSO-treated and DCA-treated hepatocyte organoids. HepOrg treated with 200 μ M DCA show canalicular dilation. Representative images are shown. Scale bar 50 μ m, magnification 10 μ m. **c.** Still images of a representative live cell imaging analysis of the bile canaliculi expansion upon DCA treatment. Note the increase of BC size in DCA-treated organoids. Cell borders were visualized following SiR-actin labelling (grey); bile canaliculi are seen as enrichment in SiR-actin intensity. Both control (DMSO, top panel) and DCA-treated (bottom panel) hepatocyte organoids were incubated with CLF to visualise bile canaliculi at the beginning (12h post-treatment) and at the

end of imaging (25h post-treatment). **d.** High-resolution microscopy images of individual bile canaliculi in liver tissue (left panel) and HepOrg (right panel). Horizontal panels show a single bile canaliculus or rosette at multiple z-planes to appreciate the detail of lumina compared between tissue and HepOrg. Bile canaliculi in control HepOrgs treated with DMSO (top panel, right) show remarkable similarities to Sham-operated liver tissue (top panel, left). Hepatocytes accumulate apical bulkheads upon bile duct ligation (BDL) *in vivo* and DCA treatment *in vitro*. Note the presence of hepatocyte rosette tissue in hepatocyte organoids. Sham and DMSO controls show the absence of bulkheads. Both tissue and hepatocyte organoids are stained for F-actin (grey). Scale bar, 2 μ m.

Figure R14

Figure R14: Modelling cholestasis and biliary fibrosis using optimised hepatocyte organoids from *Mdr2*^{-/-} mice. **a.** Schematic illustration showing the generation of *Mdr2*^{+/+} and *Mdr2*^{-/-} HepOrg. Hepatocytes were isolated and cultured in the same conditions. Compared to WT (top), the *Mdr2*^{-/-} HepOrg (bottom) show dilated bile canaliculi with bulkheads. Representative IF image of *Mdr2*^{-/-} HepOrg is shown. DAPI (blue), Phalloidin (grey), CD13 (green). Scale bar 50 μ m, 20 μ m, 10 μ m. **b.** Immunofluorescence staining of bile canaliculi (CD13, green) and F-actin (Phalloidin, grey) in *Mdr2*^{-/-} HepOrg cultures (bottom panel) compared with *Mdr2*^{-/-} liver tissue (top panel). Note the similarity in the bile canaliculi morphology between HepOrg and tissue. Cyan arrowheads indicate apical bulkheads, and yellow arrowheads indicate inward blebs, in both tissue and HepOrg. Representative images are shown. Scale bar, 50 μ m; magnification, 20 μ m. **c.** Immunofluorescence staining of bile canaliculi (CD13, green) and F-actin (Phalloidin, grey) in *Mdr2*^{-/-} HepOrg cultures, showing emergence of hepatocyte rosettes. Representative images are shown. Scale bar, 50 μ m; magnification, 20 μ m. **d.** Schematic illustration showing the starting material for periportal assembloid formation. *Mdr2*^{-/-} assembloids are made

from *MDR2*^{-/-} HepOrg, WT cholangiocytes, and WT Msc. *MDR2*^{+/+} are made from *MDR2*^{+/+} HepOrg, WT cholangiocytes, and WT Msc. Cholangiocytes are labelled with nuclear tdTomato, Msc with nuclear PDGFR α -H2B-GFP, and hepatocytes are unlabelled. The same number of HepOrg, DCs and Msc is added in both conditions. **e.** Live imaging showing changes in *MDR2*^{-/-} and *MDR2*^{+/+} PAs during first five days of the assembloid formation. Hepatocyte organoids (brightfield, grey), ductal cells (nTom, magenta). Representative images from n=2 independent experiments at 24, 48 and 132h are shown. Scale bar, 50 μ m. **f.** Graph representing the change in the ductal cell (DC) number between the day 5 (d5) and day 0 (d0) after generating assembloids, using either *MDR2*^{+/+} or *MDR2*^{-/-} HepOrg. Graph represents mean \pm SEM of n=2 biological replicates from n=2 independent experiments. Each dot represents change in total number of DC from d0-d5 of an individual portal assembloid, *MDR2*^{+/+} assembloids (n=23), and *MDR2*^{-/-} assembloids (n=22). ****, $P < 0.0001$ using a two-tailed Mann-Whitney test including all technical replicates from both experiments.

The manuscript does not investigate estng what consequences the presence of bile canaliculi has on detoxification and metabolite transport as well as cell-cell interactions and development of hepatocyte dysfunction and fibrogenesis.

A: In the previous version of the manuscript (Suppl. Fig.2), we had demonstrated that organoids grown from isolated single cells generally grow as ‘ball-like’ structures with aberrant bile canaliculi, and aberrant bile acid transport, while organoids grown from clusters of 2 or more cells (doublets, triplets) would generally form bubbly/grape-like structures with physiological bile canaliculi and physiological bile acid transport (Suppl. Fig.2e). In this revised manuscript version, we have taken advantage of this difference to systematically study the impact of physiological bile canaliculi on metabolite transport, hepatocyte function and fibrogenesis. The results are presented below, according to the different assays used.

1) impact of BC on metabolism and metabolite transport

We have used the fluorescent bile acid analogues CMFDA (5-Chloromethylfluorescein diacetate) and 5-CFDA (5-carboxyfluorescein diacetate) to test the functionality of esterases and their metabolite transport to bile canaliculi in organoids with aberrant (ball) and physiological (bubbly/grape-like) bile canaliculi. The fluorescent bile acid analogue CMFDA is intracellularly converted to 5-chloromethylfluorescein (CMF) by esterases, which is then excreted through the canalicular membrane specifically by BSEP and MRP2⁵. We observed fluorescent CMF signal in the bile canaliculi in bubbly/grape-like shaped but not in ball shaped HepOrg structures, indicative of esterase activity and active MRP2 and/or BSEP transport through the apical membrane. Similarly, the nonfluorescent 5-CFDA is converted into fluorescent carboxyfluorescein (CF) by intracellular esterases, and subsequently transported out of the cells via ATP-binding cassette (ABC) transport proteins from the hepatocyte apical membrane^{6,7}. We observed esterase activity, measured as intracellular fluorescent signal, in some cells within HepOrg with bubbly shape and bile canaliculi structure (detected by concurrent live SiR-Actin staining), while ball-shaped organoids presented reduced esterase activity. This data is now presented in the new Suppl. Fig.3i, also pasted below for the reviewer as Fig.R15.

Taken together, these results now show the beneficial consequences of a physiological bile canaliculi network on metabolism and metabolite transport.

2) Impact of BC on Hepatocyte Function

To investigate the consequences of the presence of bile canaliculi on hepatocyte function, we performed total bile acid, albumin, and cytochrome activity assay analysis on both types of HepOrg structures (ball vs bubbly shaped). All the functional tests showed that ‘ball/aberrant BC’ organoids compared to ‘bubbly/normal BC’ HepOrg present reduced functionality, albeit not statistically significant.

In addition, we observed increased hepatocyte death (cleaved caspase-3 positive staining) in structures with aberrant (ball) but not in structures with physiological (bubbly/grape-like) bile canaliculi. This is

consistent with the published findings that intracellular retention of bile acids impacts hepatocyte function³¹. This data is now presented in the new Suppl. Fig. 3, also pasted below (Fig.R15).

3) Impact of BC on Fibrogenesis

To investigate the impact of altered bile canaliculi (BC) on fibrogenesis, we have made assembloids using WT cholangiocytes and WT portal fibroblasts, and hepatocyte organoids derived from *Mdr2*^{-/-} mouse livers, which display aberrant BC architecture and similar BC cholestatic features to native *Mdr2*^{-/-} tissue – specifically: bulkheads, inward blebs and rosettes (see New Fig.6 and Suppl. Fig.17, and Fig.R14 above). Notably, assembloids formed by *Mdr2*^{-/-} hepatocytes, but not WT controls, exhibit ductal cell expansion, resembling the ductular proliferation observed in vivo, in liver tissue from *Mdr2*^{-/-} mice. These results indicate that aberrant BC architecture and bile acid obstruction directly contribute to the fibrotic response, namely to the ductular reaction observed in cholestatic liver disease. The mechanism behind this observation is very interesting and something we would like to investigate in future studies.

In summary, the results indicate that the presence of physiological bile canaliculi architecture and 3D bile canaliculi network can improve hepatocyte function, bile acid transport and metabolic function, at least for the metabolites tested. By manipulating a system that retains physiological features at baseline, we are now able to faithfully model features of the cholestatic response in vitro, observing the appearance of cholestatic hallmarks such as inward BC blebs and hepatocyte rosettes, akin to the features that are found in mouse models of cholestatic liver disease and human disease³¹. Finally, our results also highlight the utility of the assembloid model to underscore cell-autonomous mechanisms of liver fibrosis.

Figure R15

Figure R15: Impact of bile canaliculi on metabolite transport and hepatocyte function. **a.** Still images from time-lapse imaging analysis of ball-shaped and bubbly/grape-like-shaped HepOrg with bile acid analogues CMFDA and 5-CFDA (Royal LUT) show reduction in bile acid transport and metabolism in the ball-shaped structures.

Nuclei (SPY555-DNA, magenta) and actin (SiR-A, cyan) are also shown. Scale bar, 50 μ m. **b.** Brightfield and immunofluorescence images of bubbly/grape-like-shape (top) and ball-shape (bottom) HepOrg, stained for apical polarity marker CD13 (green) and apoptosis marker (cleaved caspase 3, grey; also shown in Fire LUT for ease of viewing). Scale bar, 50 μ m. **c.** Viability assay (CellTiter Glo) performed on ball-shape and bubbly/grape-shape HepOrg. Graph represents mean \pm SEM of $n=3$ biological replicates from $n=3$ independent experiments, with dot colour denoting each independent experiment; Mann-Whitney test. **d-g.** Albumin (e), Cytochrome activity (f-g) and total bile acid (d) measurements of ball-shape versus grape-like/bubbly-shape HepOrg show non-significant but marked reduction in functionality of ball-HepOrg. Graph represents mean \pm SEM of $n=5$ biological replicates from $n=3$ independent experiments; Mann-Whitney test.

What happens to bile salts after they are draining into the ducts – are there any readouts that can detect beneficial consequences of this? Does hepatocyte gene expression and function change as a result?

A: It is well documented that bile acid signalling can benefit bile duct cell function through the activation of farnesoid X receptor (FXR) and G protein-coupled bile acid receptor 1 (GPBAR1 or TGR5). This benefit includes protection against bile acid-induced toxicity, modulation of inflammation, and secretion of bile components that impact the overall liver function.

To assess the beneficial effects of BD-BC connections, as a proxy for bile salts draining, and given that when we observe cholangiocytes embedded within an assembloid we obtain 100% of BD-BC connection, we have now explored our scRNAseq data to investigate relevant changes in cholangiocyte and hepatocyte genes expression upon assembloid culture. We have compared gene expression of hepatocyte and cholangiocyte cells from assembloids (where there is BD-BC connection) and hepatocytes from HepOrg or cholangiocytes derived from cholangiocyte organoids (not assembloids), but cultured in the same media conditions.

We observed improved expression of several transporter genes (e.g. Abcb4, Abcc2, Abcc3) as well as metabolism genes (e.g. Cyp27a1, Cyp7b1, Baat, Hsd17b4) in hepatocytes derived from assembloids compared to hepatocytes derived from HepOrg (in the absence of ductal cells, i.e. devoid of BD-BC connection). Similarly, Cholangiocytes derived from assembloids (with connection) also showed improved expression of apical polarity markers such as Ezr and Ano1 indicative of increased maturation.

These results, now presented in new Suppl. Fig.12g-h and below in Fig.R16, indicate that having a BD-BC connection improves the maturation of hepatocytes and cholangiocytes. However, while highly indicative, we want to acknowledge that our studies cannot prove that the enhanced gene expression of transporter genes is a direct consequence of the bile acid drainage. That would require complicated experiments where we would need to block bile acid production in hepatocytes from assembloids, and we feel that, although very interesting, this is beyond the scope of this manuscript.

Figure R16

Figure R16: Effect of bile canaliculi-duct connection on cholangiocyte and hepatocyte gene expression. a-b. Functionality of cholangiocytes (a) and hepatocytes (b) is slightly improved upon assembloid culture, as visualised the dot plot of scRNAseq gene expression of transporters. Cells from assembloids are compared to a same media matched control (CTRL).

4. Metabolic functions of the liver - undoubtedly one its most important roles in organismal health - are largely ignored in the manuscript. Proper liver architecture is key to metabolism and detoxification – and relies on spatially organized metabolic activities, referred to as zonation. It is not clear whether the employed model reflects zoned liver functions

A: We thank the reviewer for this comment that has helped increasing the quality of our manuscript. We have evaluated the extent of periportal and pericentral gene expression in our hepatocyte organoids in the different conditions as well as in assembloids. This involved analysing RNAseq data from HepOrg cultured in different conditions, scRNAseq data from assembloids and carrying out additional RNAscope and immunofluorescence analysis.

*From the new RNAseq data we found that assembloids and HepOrg in MM medium show higher levels of some hepatocytes genes that are considered to be periportal zoned (e.g. Hpx, Mup20, Aldob) as well as the periportal zoned cholesterol-bile acid pathway genes (Baat and Acaa1b), but also some pericentral genes such as Glul (GS) or Axin2, both also expressed in HM-WntCM, as expected (New Suppl. Fig. 7 and Fig.R2 below). These observations suggested that the cultures exhibited some degree of zonation. To assess that, we evaluated our scRNAseq data. To facilitate comparison of the expression of pericentral and periportal region genes in our scRNAseq, we have generated a consensus signature of pericentral and periportal genes. This was achieved by combining the expression of published and validated pericentral or periportal markers from three different publicly available datasets ¹³⁻¹⁵, and using this as a pericentral or periportal score. **The results demonstrate that hepatocyte organoids cultured in high Wnt conditions exhibit a more pericentral score, while HepOrg transferred to MM co-culture medium acquire some periportal features and present a periportal score.** We have validated these results by RNAscope and immunofluorescence analysis for some portal (Gls2 – RNA, ALB, ECAD - protein) and central (Cyp1a1– RNA, CYP2E1, GS -protein) gene markers (New Fig.2a and Suppl. Fig.7 and presented below Fig.R2 below). Notably, the Immunofluorescence analysis indicates that the medium switch promotes some degree of zonation as we detect, within the same structure, some cells expressing portal markers while others express some central markers.*

Figure R2

Figure R2 *HepOrg* are partially zoned, and increase periportal gene expression in MM medium. **a.** Dotplot representing gene expression of WNT pathway genes across conditions in hepatocytes from scRNAseq (For reviewer only). **b.** Periportal and pericentral zonation score calculated from consensus gene expression from Halpern et al., Ben Moshe et al., and Wu et al. show higher periportal gene expression in MM media devoid of Wnt factors, when compared to Wnt-enriched HM-Wnt media hepatocytes. **c.** Dotplot representing gene expression of periportal, pericentral and zoned cholesterol genes across conditions in hepatocytes from scRNAseq. **d.** Expression of periportal (*Gls2*) and pericentral (*Cyp1a1*) gene RNA visualised by RNAscope in *HepOrg* from MM media. **e.-f.** Immunofluorescence staining of periportal (ALB, Albumin, green **e**) and (E-CAD, E-cadherin, green, **f**) and pericentral (*GS*, Glutamine synthetase, magenta, **e**) and (*CYP2E1*, magenta, **f**) in liver tissue (top) or in *HepOrg* cultures grown in HM-Wnt and cultured in MM media for 7 days (bottom). Specific stainings are also shown in Fire LUT for the ease of viewing. Representative images are shown.

– at minimum, this should be shown for the cholesterol-bile acid pathway, which has a key role in cholestatic liver disease and is strongly zoned.

A: We have now explored the genes relating to this pathway both using RNAseq and scRNAseq. We found that some of the hepatocytes in HepOrg as well as in assembloids express some of the cholesterol-bile acid pathway genes including Cyp7a1, and Cyp7b1 (both involved in cholesterol biosynthesis), Cyp27a1 (biosynthesis of bile acids), Baat (formation of bile acid-amino acid conjugates), Slc275a (bile acid re-conjugation and recycling), as well as the ATP-binding cassette (ABC) transporters mediating secretion of bile acids (Abcb11) and phospholipids (Abcb4), amongst others⁴⁴⁻⁴⁶. The results are presented in Fig.R2 and Fig.R17 below.

Combined, the immunofluorescence analysis, RNAscope, RNAseq and scRNAseq gene expression analysis for both zonation markers as well as cholesterol-bile acid genes, our data indicates that HepOrg and periportal assembloid models present some degree of zonation. These new data are presented in the new Fig.2a and Suppl. Fig.7, and also pasted here as Fig.R2 and Fig.R17.

Figure R17

Figure R17: Expression of cholesterol pathway genes in hepatocytes from HepOrg and assembloids. Dotplot of genes from the cholesterol metabolism pathway which are zoned from portal to central axis, in scRNAseq of hepatocytes from HepOrg in HM-Wnt media, HepOrg in MM media, homeostatic and fibrotic-like assembloids.

5. The correlation between the model of this study and published scRNA-seq datasets (Fig.4i) appears rather low with the exception of the correlation between hepatocytes from this model and hepatocytes from the Guilliams study. Especially the correlation to PF scRNA-seq data from the other studies is very low and lower than the correlation with VSMCs

A: We thank the reviewer for pointing this out, which made us realize that there was a mistake on our previous analysis. Now, as part revision we have extended our scRNAseq analysis not only by increasing our sample size and number of replicates, but also by comparing our data to more publicly available datasets – some of which have been published since our first submission. The results show that the portal fibroblast used in our periportal assembloids closely correlate with other portal fibroblasts previously published^{20,27,30}, while HSCs and VSMCs cluster separately. These results are in line with the immunofluorescence staining for specific portal markers (PDGFR α /SCA1 and PDGFR α /elastin). The results are presented in New Suppl. Fig.9g, Suppl. Fig.13n also pasted below for the reviewer as Fig.R18.

Figure R18

Figure R18: Periportal assembloids contain portal fibroblast mesenchymal subtype. **a.** Correlation analysis of healthy cell types comparing liver tissue datasets with assembloids. **b.** Representative confocal images of periportal assembloids stained for portal fibroblast marker Elastin (white) marker, in combination with PDGFR α -H2BGFP endogenous signal, and with Msc membranes visualised by membrane-tdTomato. Scale bar, 50 μ m. **c.** Immunofluorescence images staining of fibrosis-like assembloid for portal fibroblast marker SCA1 (white), as well as all mesenchyme (PDGFR α -H2BGFP, green), cholangiocytes (mem-tdTomato, magenta) and nuclei (DAPI, blue), where cell borders are outlined by phalloidin staining (white). Scale bar, 100 μ m, 20 μ m.

Moreover, since these are periportal assembloids there should be an additional analysis comparing hepatocytes to portal (Zone 1) and pericentral (Zone 3) hepatocytes from previous scRNA-seq datasets – this might make the results stronger and more meaningful.

A: As detailed above, we have now used scRNAseq, immunofluorescence, RNAseq and RNAscope analysis to thoroughly evaluate the extent of the zonation in our model. We show that HepOrg as well as hepatocytes from assembloids cultured in our assembloid medium (MM media) present some degree of zonation with some cells expressing periportal and others pericentral markers within the same structure (see Fig. R2 and response to point 4 above).

6. It is widely believe that signals from inflammatory cells, in particular macrophages, have a key role in the fibrogenic process. These cells are missing in the current model, which represents potentially a very important gap in regards to studying fibrogenesis.

A: We agree with the Reviewer, that the addition of immune cells would be a great way to further improve modelling of fibrosis in vitro, as they are essential in this disease's pathology. However, we believe that optimization of immune cell inclusion in the culture to be out of scope for this publication, and is the subject of intensive future studies in our lab. We have discussed this caveat on the manuscript, lines 482-485 and 545-548 .

7. Evidence for cleaved caspase 3 and hepatocyte is weak (there is no quantification comparing homeostatic- and fibrotic-like assembloids)

*A: In addition to cleaved caspase 3 staining, in our original submission we observed a substantial increase in cell death, as evidenced by the significant disruption of (hepatocyte) cell integrity in live imaging movies, as well as disappearance of DNA signal, which suggests DNA fragmentation and cell death, in addition to positive cleaved caspase3 staining in fibrosis-like assembloids but not in homeostatic ones (Previous Suppl. Fig.10d-e and Video 15). We have quantified the number of apoptotic cells in assembloids, and observed a significant increase in apoptotic hepatocytes in fibrotic-like assembloids compared to homeostatic assembloids. These results are presented in New Fig.5, Suppl. Fig.13-14, and also below for the Reviewer as **Fig.R19a-d**.*

and there is no functional investigation (e.g. using caspase inhibitors) for confirmation and for determination of the role of apoptotic hepatocyte death in fibrogenesis in this model system.

A: We fully agree with the reviewer, we cannot rule out the presence of other mechanisms inducing cell death. We are sorry for the misunderstanding; our initial intent was not to state that apoptosis was the primary cell death program and that it played a functional role in the fibrotic process, but merely that fibrogenesis induced cell death.

In our scRNAseq analysis we found enrichment in cell-death terms, like "apoptosis" and "intrinsic pathway for apoptosis" in fibrotic-like vs homeostasis-like assembloids. However, we not only find apoptotic terms but also terms related to general "programmed cell death", suggestive that not only apoptosis but other cell-death mechanisms could be involved. Nevertheless, as suggested by the reviewer, we attempted to inhibit apoptosis in fibrotic-like assembloids with an apoptosis inhibitor Z-VAD-FMK. However, we have not observed a reduction hepatocyte cell death, as assembloids still progressed as usual and deposited collagen (See the below Fig.19e-i, for Reviewers only). We have also attempted to test other forms of cell death and tested phosphor-MLKL staining as a sign of necroptosis, but the results are inconclusive (See the below Fig.R19e-i, for Reviewers only). We have now clarified in the text that cell death occurs via apoptosis and other mechanisms. It would be interesting to characterize whether all cell death programs (autophagy, ferroptosis, pyroptosis...) have a functional role. We have considered this for future studies.

Figure R19

Figure R19: Confirmation of hepatocyte death in fibrotic-like assembloids. **A**. Immunofluorescence staining (left) of homeostatic-like and fibrotic-like assembloids for cleaved caspase 3 staining (orange); mesenchyme (PDGFRα-H2BGFP, green), cholangiocytes (mem-tdTomato, magenta), actin (Phalloidin, white) and nuclei (DAPI, blue) are also shown. Quantification of hepatocytes with cleaved caspase 3 positive staining (right). Violin plot shows distribution of values from $n=3$ independent biological replicates. Mann-Whitney test. Scale bar, 100 μm, 50 μm and 25 μm. **b**. GSEA of fibrosis-like versus homeostasis hepatocytes in assembloids, showing all significant terms relating to cell death. **c**. Immunofluorescence images of hepatocytes contacted by mesenchyme expressing cleaved caspase 3 (grey) and lacking the DNA staining (cyan); scale bar, 50 μm, inserts 10 μm. **d**. Time-lapse imaging stills of periportal organoids with high mesenchymal ratio and stained with a DNA dye (SPY620, white); note bursts of DNA signal; scale bar, 50 μm. **e**. Time-lapse imaging stills of periportal assembloids treated with 20 μm Z-VAD-FMK apoptosis inhibitor, or DMSO control. DNA dye (SPY620, white); cholangiocytes (nuclear tdTomato, magenta), and Msc (PDGFRα-H2B-GFP, green) are shown. Scale bar, 50 μm. **f**. DETECT analysis of assembloids treated with treated with 20 μm Z-VAD-FMK apoptosis inhibitor, or DMSO control.; violin plot shows DETECT distance ratios, each dot representing an organoid from $n=5$ biological replicates; Mann-Whitney test. **g-h**. Caspase inhibitor-treated fibrotic-like assembloids deposit less collagen, as visualised by second harmonic generation (SHG) imaging (g), but the change is not statistically significant. Bar graph shows average intensity quantification of SHG signal in $n=5$ biological replicates. Mann-Whitney test. Representative images are shown; scale bar, 50 μm. **i**. Phospho-MLKL (Ser345) staining in fibrotic-like assembloids shows diffuse, unspecific signal. Scale bar, 100 μm.

How do Msc induce hepatocyte death? Furthermore, the presence of apoptotic hepatocytes puts into question the proposed key role of homotypic Msc-Msc interactions.

A: As indicated above in response to this reviewer's points 1 and 2, our data shows that hepatocyte death is, at least in part, mediated by increased Msc-Msc contact, since hepatocyte death and assembloid structure can be rescued when inhibiting cell-cell/ECM contacts by knockdown of the Msc-Msc interaction through CDH11 and Msc-ECM adhesion by ITGB1.

While these results show a clear direct effect, we want to acknowledge that we do not investigate the exact intracellular molecular effectors behind that observation, i.e., we still do not know the intracellular signals that instruct hepatocytes to initiate cell death. We believe that this mechanistic insight would be extremely interesting but the subject of a full project itself that we would like to address in future studies.

We have included a section in the discussion to highlight this caveat of the manuscript (lines 533-541).

8. It would be helpful if this system could be further validated and if practical applications of this systems could be demonstrated e.g. drug screening for biliary fibrosis.

A: We find this reviewer's suggestion very appealing and interesting, however, in our personal experience, having set up screenings for drug targets for cancer organoids in the past⁴⁷, we are aware that a drug screening for targets against fibrosis is a project on its own. We believe our reductionistic model to be complementary to animal models of fibrosis and we sincerely hope that other labs will embrace and utilize the system to perform drug screenings for biliary fibrosis, as the reviewer suggests. We have highlighted this potential application in our manuscript (discussion line 560-566).

Referee #3 (Remarks to the Author):

This manuscript by Dowbaj et al. describes a new organoid model composed of adult hepatocytes, cholangiocytes and liver mesenchymal cells, which the authors use to model biliary fibrosis in vitro. The authors also optimize the current hepatocyte organoid models to generate organoids with physiological / functional bile canaliculi networks, and assemble these with cholangiocytes and portal fibroblasts to generate periportal assembloids.

A substantial amount of the data presented in this manuscript is descriptive in nature, and much more quantitation (with statistical analysis) is required to underpin the findings and conclusions drawn. Furthermore, more data is required to convince the reader that the assembloid system provides a platform that will provide step changes in our understanding of liver physiology and disease – for example disease modelling / perturbations with this system leading to conceptual advances in our understanding of liver disease pathogenesis.

A: We thank the reviewer for these comments. We now provide detailed quantitation and statistical analysis to all the figures presented in the manuscript as detailed below.

In light of this comment, we realize that we might not have contextualized well our findings and the importance of our model. Therefore, before addressing the specific comments of this Reviewer, we would like to give a broader answer below as to why our model represents a big and not only incremental advance in liver organoid models, and how it can help conceptual advances in liver disease pathogenesis.

1) When compared to existing HepOrg models, our improved system retains physiological bile canaliculi size and network after passages, as we now quantitatively demonstrate below. Conceptually, we show that physiological 3D cell orientation and spatial organization translate to improved organ function over previous 2D models, and can therefore better reflect features of organ dysfunction.

2) By combining this optimized HepOrg with cholangiocytes and mesenchyme, we obtain a periportal assembloid model that connects bile canaliculi with the bile duct. We demonstrate that the existence of this drainage from bile into bile duct facilitates hepatocyte and cholangiocyte maturation. In addition, by manipulating the number of mesenchymal cells, we provide the very first in vitro model of biliary fibrosis that could be used as a platform to test drugs against this disease.

3) To our knowledge, our model is the first one in the field where bile canaliculi network is maintained long term, and where it is possible to study cholestatic liver disease, since the HepOrg structures are not cholestatic at baseline. By manipulating the bile canaliculi structure, our model provides a first available platform to test drugs not only for preventing fibrosis but also to reverse cholestasis; giving a conceptual edge for our system over previous ones.

5) Finally, by combining WT cholangiocytes and portal mesenchyme with mutant hepatocytes defective in bile canaliculi transport (Mdr2^{-/-} HepOrg), we provide evidence that our model allows identifying cell-autonomous mechanisms involved in the fibrotic response. To our knowledge, our model is the first in vitro model that allows underscoring the impact of a cholestatic bile canaliculi in the pathogenesis of biliary fibrosis driven by Mdr2 mutations.

Specific comments:

1. Line 90-93 and Supp Fig 1b: How does CD13 immunostaining indicate insufficient BC and BD formation? This should be clarified for the reader.

A: Hepatocytes are uniquely polarized cells, with one or more apical cell membrane sides contributing to several bile canaliculi of directly opposing hepatocytes ³¹. Aminopeptidase N (CD13) is a transmembrane protein located at the apical surface of polarised epithelial cells ⁴⁸. As such, it is extensively used in the literature to mark hepatocyte's apical surfaces and because it is evenly distributed throughout the lumen, it is a reliable marker for BC segmentation ^{35,49-55}.

The reviewer refers here to our results from the previous version of the manuscript, where, using these markers, we had first investigated the presence of BC in previously reported hepatocyte organoid models (line 90). Despite the fact that in those models hepatocytes are polarized, we did not find evidence for a sufficient formation of a bile canaliculi network when analysing the bile canaliculi of these structures (previous Suppl. Fig1).

*As part of the response to point 9 from this reviewer (Rev_2_point_9: Line 122, Fig 1d: This should be **quantified and compared between groups**), we have now formally quantified the length of the BC network (the sum of all branches), and the number of junctions (a proxy for branching and connectivity) of these published models. As expected, the quantification confirmed what we had previously qualitatively observed: the BC is significantly shorter and minimally branched in the published Hu et al. and Peng et al. methods compared to our optimized and improved HepOrg method.*

*These results are now presented in New Fig.1d and Suppl. Fig.1a-f, and also pasted below for the Reviewer assessment as **Fig.R20**.*

Figure R20

Figure R20: Comparison of bile canaliculi between HepOrg models. *a.* Schematic illustration of 3D bile canaliculi structure of HepOrg and schematic representation of measurements in the 3D bile canaliculi network. *b.* Number of triple junctions for the largest network in organoids and tissue. *c.* Length of the largest network comparing HepOrg in different media. *d.* Representative bright field images of HepOrg at passage 1 (P1) cultured under the indicated media conditions: Hu et al., Peng et al., control media (HM) supplemented with FBS and HM-Wnt and HM-WntS. Scale bar, 500 μm (top), 100 μm (bottom).

2. The authors state that their system exhibits a ‘physiologically relevant’ BC network and size, however there is no data presented to measure what the physiologically relevant / in vivo baseline is to allow direct comparison.

A: We respectfully believe the reviewer might have missed the data presented in original submission’s Figure 2c-e, where we included the results of BC diameter and BC connectivity (network) from in vivo tissue (green) compared to HepOrg grown in HM-Wnt media (pink) or MM media (blue). We have now clarified this better in the figure legend and also included the statistics in the Figure and Figure legend (New Fig.1 and Suppl. Fig1). Note that our improved HepOrg system presents values similar to the ones from the in vivo tissue, indicating that indeed the system exhibits physiological bile canaliculi size and network.

3. Line 98 and Fig1c: what is the ‘expansion potential’, what does this measure? The authors also state increased size, cell number and organoid formation efficiency but make no statistical comparisons and

present no quantitative data on this, and instead representative images are shown. Comprehensive quantitation and statistical analyses should be performed.

A: We believe here the reviewer might have misunderstood the data presented in this previous version Fig.1c. In the dot curve over time, each dot indicates one split at a 1:2 split ratio, meaning that from 1 well of organoids we are seeding 2 wells at the indicated time point. As example, for HM-Wnt the number of splits = 19 (passage 19) or what equals to 2^{19} . On the contrary, for the control HM-FBS this is only 1 split, 2^1 . In Suppl. Figure 1e-f we presented the organoid formation efficiency and the quantification of the cell numbers over ~6 splits. The reviewer is right, though, when they indicate that there are no statistics to these two graphs. We have now included the statistics for these analyses. These results are now presented in the new Fig.1f and Suppl. Fig.1j.

4. Line 462: the methods state organoid efficiency presented as percentage of seeded cells, however in Supp fig 1e data is presented as number of HepOrgs/1000 seeded hepatocytes. Can the authors explain this discrepancy?

A: We thank the reviewer for noticing this editing error in our side. We have now edited the methods and legend text to “organoid formation efficiency is presented as total number of organoid structures grown per 1000 seeded cells”.

5. Supp fig 1f: can the authors comment on the large drop in cell number (almost to baseline levels) at passage 4 in the HM-Wnt sample.

A: We thank the Reviewer for noticing this discrepancy. There was a typo in the file used for making the curve, we have corrected that and also performed an additional biological replicate to strengthen this data, which are now presented in the new Suppl.Fig.1j of the manuscript and shown here for the ease of the Reviewer in Fig.R21 below. The raw data is all presented in the Source Data file.

Figure R21

Figure R21: Growth curves of HepOrg grown in the indicated media. Values indicate mean \pm SEM of $n=4$ independent biological replicates performed. Number of cells at a given passage is shown.

6. Line 105, line 468, Supp fig 2a,b: bubbly and ball shaped morphology. This is a qualitative measurement not a quantitative measurement. The authors need to assess cell shape and morphology with a quantitative approach using cell segmentation software such as Imaris or Cellpose.

A: We thank the Reviewer for this suggestion which has significantly improved the quality of our manuscript. At the time of submission, there were no publicly available algorithms that would accurately quantify shape differences between 3D structures. Classic shape descriptors such as area or circularity can only capture partial differences between 3D structures. Cell segmentation, suggested by

the reviewer, is very useful to identify cells but does not provide shape of a multicellular structures in 3D, neither will it quantify differences between shapes.

Nonetheless, we appreciate that the reviewer had a very valid point and we have aimed to address this in the course of this revision. Therefore, in this revised version of the manuscript, we have employed topological data analysis (TDA), a field within mathematics that studies the shape and structure of data using topological metrics, to quantify differences between two phenotypes. Specifically, the group of Heather Harrington ¹⁶ devised a temporal shape descriptor algorithm, DETECT (Detecting Temporal Shape Changes with the Euler Characteristic Transform), which extends the smooth Euler characteristic transforms⁵⁶. This algorithm processes shapes by generating a curve that captures both the geometric and topological characteristics of the shape. In addition, by doing the same analysis for all time points, the resulting collection of curves provides information about the morphological changes over time, while also being rotationally invariant (New Suppl. Fig.2b-c). For this study, the inputs to DETECT are the boundaries of 2D projections of each organoid at each time point which calculate the corresponding DETECT curves. The collection of curves at all time points measures how the morphology of a particular organoid changes over time.

We next needed to be able to compare these quantifications from DETECT curves. To this end, we propose a metric on these DETECT curves to quantify the morphological difference up to rotations and reflections between two shapes over time that is also stable to small perturbations of input shapes. Under this metric, organoids that maintain similar shapes over time yield aligned DETECT curves.

Now, thanks to the comment of this reviewer, we employ this DETECT metric all over our manuscript to quantitatively compare spatio-temporal changes in morphology of organoids under different conditions, and visualize the outcomes with heatmaps, t-SNE dimensionality reduction and/or violin plots.

Regarding the specific question of the Reviewer on bubbly vs ball, as shown in the revised Fig.1 (also pasted below for the reviewer assessment as Fig.R22), organoids are distinctly categorized into two groups, which separates them by condition in either ball or bubbly-shaped, aligning perfectly with visual observations. Similarly, the distance metric used to measure the difference between two DETECT calculation results predicts that distances between organoids within each condition (bubbly or ball) are significantly less than those between different conditions, indicating statistically significant differences between the two groups. Further details about the Silhouette scores for the k-mean clustering, the DETECT algorithm and the corresponding metric can be found in Appendix B-C.

Figure R22

Figure R22: DETECT analysis of ball and bubbly/grape-like organoid shapes. *a.* Maximum intensity projection of still images from live imaging analysis of HepOrg derived from membrane-tdTomato mice, seeded as single (bottom) or >2 hepatocyte cell clusters (top), seeded at 200 cells/ μ l density. Hepatocyte organoids (HepOrg) acquire a coarse (bubbly/grape-like, top panel) or a smooth (ball-like, bottom panel) morphology. Representative pictures from $n=2$ independent experiments are shown. *b.* Illustration of the DETECT (Detecting Temporal Shape Changes with the Euler Characteristic Transform) pipeline. The algorithm's input takes segmented boundaries of organoids and computes the ECT (Euler Characteristic Transform) as collection of curves. These curves are then used to calculate the DETECT curve. The DETECT curves can be used as feature vectors for downstream visualisation algorithms and clustering algorithms, or to compute the pairwise DETECT metric heatmap. *c.* Visualisation of morphological variations of p0 HepOrg using t-SNE based on DETECT calculations integrating all time points (d0-d13), with points coloured according to the shape type of the corresponding organoid. *d.* Heatmap of the DETECT metric distance for the DETECT results of each pair of p0 HepOrg integrating all time points (d0-d13). *e.* Results of K-means clustering applied to DETECT calculations post-PCA reduction to two principal components. Each dots represents an organoid, labelled red for ball-shaped, and blue for bubbly/grape-like-shaped. Crosses indicate the centroids of the respective clusters, with colours denoting the cluster assignment.

7. Line 111: quantification of the longer thinner BC shapes between the bubbly and ball structures should be performed

A: In our previous version, we had shown, that ball and bubbly-like HepOrg both express similar levels of hepatocyte markers. We had also shown, using F-actin and CD13 staining, that “ball-shaped” HepOrg presented barely any bile canaliculi (BC). On the contrary, tissue and “bubbly” HepOrg structures presented long and thin BC (see new Suppl. Fig.3). Since ball-like structures do not consistently present well developed BC, but only some heterogenous apical lumina which remain round and wide, quantifying these shapes would not represent an accurate representation of BC features. Therefore, we now present additional images for the reviewer to illustrate the heterogeneity and

differences between BC in organoids with bubbly-like shape compared to organoids with ball-shape. Also, we provide schematics that clarify this point in the new Suppl. Fig.3, also pasted below.

We want to stress here that for all the experiments concerning assembloids we enrich for bubbly-like structures and we discard ball-like structures. Nonetheless, we thought that for transparency, it was important to report the presence of these structures, but since these ball-shaped structures cannot be leveraged to replicate bona-fide assembloids, we do not feel it would be relevant for the main message of our manuscript to further characterize these. Nevertheless, following the Reviewers' comments, we did perform additional functional characterization of these shapes below in new Suppl. Fig.3, which we pasted below for the reviewer as Fig.R23.

Figure R23

Figure R23: HepOrg with ball-like morphology present aberrant bile canaliculi and reduced functionality. **a.** Viability assay (CellTiter Glo) performed on ball-shape and bubbly/grape-shape HepOrg. Graph represents mean \pm SEM of $n=3$ biological replicates from $n=3$ independent experiments, with dot colour denoting each independent experiment; Mann-Whitney test. **b-e.** Albumin (c), Cytochrome activity (d-e) and total bile acid (b) measurements of ball-shape versus grape-like/bubbly-shape HepOrg show non-significant but marked reduction in functionality of ball-HepOrg. Graph represents mean \pm SEM of $n=5$ biological replicates from $n=3$ independent experiments; Mann-Whitney test. **f.** Maximum intensity projections (MIP) of immunofluorescence staining in two examples of ball-shaped HepOrg, which were identified based on brightfield images (most left). CD13 (green) and F-actin (Phalloidin, grey) staining was used to visualise apical lumina. Note that staining revealed aberrant apical

lumina, either their reduced presence (top panel) or rounded and enlarged shapes (bottom panel). Inserts show detail of aberrant apical lumina, outlined with yellow dashed lines and indicated by yellow arrowheads. Adjacent schematics illustrate the differences between two examples. Representative images of $n=3$ independent experiments are shown. Scale bar, $50\ \mu\text{m}$, magnification, $10\ \mu\text{m}$. **g.** Brightfield and immunofluorescence images of bubbly/grape-like-shape (top) and ball-shape (bottom) HepOrg, stained for apical polarity marker CD13 (green) and apoptosis marker (cleaved caspase 3, grey; also shown in Fire LUT for ease of viewing). **h.** Still images of a representative live cell imaging analysis of the bile acid analogue (CLF, green LUT in the middle panel, fire LUT in most right panel) uptake in HepOrg derived from membrane-tdTomato (mTom, magenta) mice with bubbly/grape-shape (top) or ball-shape (bottom). Cell borders were visualized following SiR-actin labelling (grey). Note that bubbly/grape-like organoids uptake CLF and release it into their bile canaliculi while ball-shape organoids accumulate CLF in non-bile canaliculi structures. Schematic of the experimental set up is shown on the left of the immunofluorescence panels. Scale bar, $20\ \mu\text{m}$. **i.** Still images from time-lapse imaging analysis of ball-shaped and bubbly/grape-like-shaped HepOrg with bile acid analogues CLF, CMFDA and 5-CFDA (Royal LUT) show reduction in bile acid transport and metabolism in the ball-shaped structures. Nuclei (SPY555-DNA, magenta) and actin (SiR-A, cyan) are also shown. Scale bar, $50\ \mu\text{m}$.

Similar to data showing bile acid secretion presented later in the manuscript, this observation should be quantitated to assess functionality in the bubbly vs ball shapes.

A: We believe the reviewer might have missed the data presented in the previous version's Suppl. Fig.2e and Suppl. Video 1 regarding the functionality of the BC in ball vs bubbly HepOrg structures. In this figure we had already shown that in ball structures the bile acid analogue CLF (used as a proxy for functional bile transport) is accumulated inside of the hepatocytes, indicating loss of function of the bile transport and a cholestatic phenotype, consistent with aberrant or absent BC. On the contrary, bubbly-like structures readily secrete CLF to their apical domain, consistent with a functional bile acid transport and a thin, long and connected BC.

In this revised version of the manuscript, we complement this result with another bile acid analogue, CMFDA, which is metabolized by esterase and excreted by the cells by the MRP2 and/or BSEP transporter. The results obtained confirm once more, that ball structures show defects in bile canaliculi transport compared to bubbly-like HepOrg structures (see new Suppl. Fig.3h-i and Fig.R23 above).

Together, our results indicate that ball HepOrg present good expression of hepatocyte markers but exhibit defects in BC architecture and function, indicating that expression of hepatocyte markers is not sufficient to guarantee a physiological and functional BC network. This is highly relevant to our study, because to connect bile canaliculi with bile duct requires that both structures should present physiological architecture, as we had already illustrated in Suppl. Fig.4 of the previous version of the manuscript.

To clarify and illustrate this point better, we now provide schematics for the Reviewer and future readers to better understand BC architecture and the outcome of the functional experiments, and how these are proxies for functional BC. We also provide better-quality videos and images to illustrate this point as supplementary videos (Videos 1-2).

Also is there any way the authors can quantitate the size of the biliary network, ie, quantitate the number of nodes and edges?

A: As mentioned above, since ball-shaped HepOrg barely presented "real" bile canaliculi (BC) structures, but primarily heterogeneous apical lumina, we cannot analyse any features of their bile canaliculi network.

8. Line 440: the authors enriched for bubbly-HepOrg by enriching for cell clusters – please describe how this enriching was done.

A: As described in the results from previous version's Suppl. Fig.2 (line 108), the majority of ball-shaped structures start when hepatocytes are seeded from single cells, while hepatocyte seeded as doublets generate bubbly/grape-like structures (23% vs 87% efficiency) (previous Suppl. Fig. 2b). Therefore, to enrich for these bubbly/grape-like hepatocyte organoids, we aimed to start the cultures having doublets or cell clusters. As expected, some ball-like structures also arise in these conditions, as it is impossible to achieve 100% of the cells as doublets. For majority of experiments, we remove ball-structures by hand-picking out undesired ball morphologies, as described already in the methods section of the original submission.

9. Line 122, Fig 1d: This should be quantified and compared between groups.

A: As mentioned above to the response to point 1 of this Reviewer, we have now formally quantified the length of the BC and the number of junctions. These results are now presented in the new Fig.1 and Suppl. Fig.1, and also above in Fig.R20, as part of the response to point 1 of this Reviewer.

10. Line 124 Fig 1f, Supp Fig 2e: Can this CLF functionality be quantified and compared between the two groups? Eg areas of intense staining (in the BC) vs the more diffuse staining seen in the ball structures.

A: This point is similar to the point 7 raised by this reviewer. We apologise for the lack of clarity in our original presentation. As mentioned in the reply to point 7, we believe that the reviewer has not fully appreciated the assay in the original submission's Fig.1f and Suppl. Fig2e. In these figures, we showed that in ball structures the bile acid analogue CLF is taken up by hepatocytes but is not secreted, instead it is retained in the cells, appearing as diffuse staining. On the contrary, bubbly-like structures readily uptake and secrete CLF to their apical domain. We now take advantage of the Fire-LUT pseudo-colour lookup table to illustrate the intensity of staining in the BC (see Fig.R23 - point7).

11. Line 129: It is not clear whether the cholangiocyte-like structures are due to differentiation of hepatocytes or from existing cholangiocytes/BDs in the media aggregating with hepatocytes. This should be explored in more detail.

A: We thank the reviewer for this valid point. We observed that in the hepatocyte expansion condition (HM-Wnt) cholangiocyte organoid structures sometimes spontaneously emerge. We had interpreted these as cholangiocyte contamination from the hepatocyte prep, which is enriching for hepatocytes but can also contain other cell types. However, the reviewer has a valid point indicating that this could also be also due to cholangiocytes transdifferentiating from hepatocytes. While we believe studying transdifferentiation would be a subject of a full new project in itself, we have attempted to address the question of the reviewer by generating hepatocyte organoid cultures from an induced Prom1-CreRT2xfl-fl-ZsGreen mouse line^{57,58}. In this mouse model, induction with tamoxifen in an adult mouse specifically recombines cholangiocytes to express ZsGreen with a 45% efficiency. This strategy allows us to detect cholangiocyte "contamination" due to presence of cholangiocytes in the original prep and not transdifferentiation. The results clearly indicate that the majority of the contaminating cholangiocyte organoids (~70%) are ZsGreen-labelled, in agreement with the labelling efficiency, indicating that these arise from contaminating cholangiocytes in our hepatocyte prep before seeding. These results are shown in new Suppl. Fig.4 and below as Fig.R24 for the reviewer assessment.

Figure R24

Figure R24: Cholangiocyte organoids in HM-Wnt media arise from cholangiocytes in the hepatocyte isolation prep. **a.** Schematic of experimental approach. *Prom1-Cre^{ERT2}xfl-fl-ZsGreen* transgenic mouse line is used to induce specific cholangiocyte labelling with ZsGreen protein in the adult animal, prior to isolation of hepatocytes. By injection of tamoxifen, cholangiocytes are labelled by fluorescent protein ZsGreen and this process has been shown to occur with close to 80% percent efficiency^{57,58}. We have then allowed for the tamoxifen washout period of 14 days, to exclude any effects relating to the injection and tamoxifen in murine system, followed by standard hepatocyte isolation, as described in our methods. **b.** Specific labelling of cholangiocytes by the *Prom1-CreRT2xfl-fl-ZsGreen* tamoxifen injection (Green), co stained by cholangiocyte marker osteopontin (OPN, red). Nuclei are also stained (DAPI, blue). Scale bar, 50 μ m. Adapted from⁵⁷. **c.** Pictures of seeded hepatocytes after isolation indicate presence of small ZsGreen-labelled cells (arrowhead, left picture), which then expand into ZsGreen-labelled cholangiocyte organoids (arrowhead, right picture). Scale bar, 500 μ m, 200 μ m. **d.** Sorting strategy to identify how many of cholangiocytes are labelled by ZsGreen in our experiments. After single cell selection, the cells are gated on EpCAM-APC, which is a marker of cholangiocytes and further gated on Prom1-ZsGreen to identify the percentage of recombined cells. **e.** Percentage of recombined cholangiocytes correlates between day 0 and day 7 of culture, indicating expansion of cholangiocyte organoid from the contaminating cholangiocytes in the hepatocyte isolation prep.

12. Line 131-133: Again these data should be quantified - for example, the smoothness/sphericity of the chol-like structures, and if smooth is this then suggestive of lack of outward connections?

A: This line 131-133 refers to the results shown in Suppl. Fig.4 where we observed no connection between the cholangiocyte and hepatocyte structures, when cholangiocytes arise spontaneously in culture. In the manuscript we opted to show this data to be thorough and transparent, and for the field to be able to replicate our results. We did not aim this result as a quantitative metric. The quantification is an absence of connection in 100% of the structures that emerge as chimeras in our HM-Wnt medium. Absence of connection in this case is expected given that: (1) bile duct organoids are not embedded within the hepatocyte structure, but next to it, and (2) cholangiocyte structures have far too large lumens (50-100 μ m) compared to the BC diameter (1-2 μ m), which would prevent any physiological connection.

13. Line 140 Fig 2a,b: Authors state 'even more improved' hepatocyte polarity but I do not see any quantification or comparison between groups other than a representative image.

A: We apologise for the inaccuracy in our language here. What we meant in this sentence is that while hepatocytes are well polarized when grown in the expansion medium HM-Wnt (previous Fig.1e), and possess a BC network, these BC are still ~5 μm in diameter, which is 2-3 μm wider compared to tissue (see previous Fig. 2c). Instead, the BC size upon culture in MM media decreases to 1-2.5 μm , which is within the tissue range (see previous Fig.2c and 2d). However, this specific sentence is now removed as it was redundant with the sentence below where we had explained that quantification.

It now reads as follows:

"Remarkably, image analysis and reconstruction demonstrated that HepOrg cultured in MM presented a significantly more homogenous and physiological bile canaliculi (BC) diameter that was within the tissue's physiological range of 1.5-2.5 μm ^{34,35}, thinner than the 2-7 μm diameter of the cultures in our optimized expansion medium HM-Wnt (Fig.2b-e, Suppl. Fig.8a-b). Notably, the BC network connectivity was further improved, resembling that of the native liver tissue (Fig.2f)."

Remarkably, image analysis and reconstruction demonstrated that HepOrgs cultured in MM presented a significantly more homogenous and physiological BC diameter that was within the tissue's physiological range of 1.5-2.5 μm ^{34,35}, thinner than the 2-7 mm diameter of the cultures in our optimized expansion medium HM-Wnt

Further to this there is comment of the MM supporting cholangiocytes and mesenchyme without their outgrowth – is this within the HepOrg culture? ie do these HepOrgs contain multiple cell types?

A: No, HepOrg cultures do not contain multiple cell types. We believe the reviewer is referring to previous Suppl. Fig.5f, which we hope to clarify here. Once we had a medium that would enable growth of HepOrg with a BC network (MM or HM-Wnt), we tested the suitability of these media to support the other cells needed for periportal assembloids. For that, in previous Suppl. Fig.5f, we cultured only cholangiocytes or only mesenchyme in either HM-Wnt or MM. Using live/dead staining, we observed that MM supports both mesenchyme and cholangiocytes, while HM-Wnt results in excessive cholangiocyte growth (compare panels CholOrg in MM and HMWnt for the size of the cholangiocyte organoids). Therefore, when preparing assembloids, we opted to use MM medium since: (1) it enables the formation of better BC in HepOrg (previous Fig.2b-e), (2) it allows culturing the 3 cells types, and (3) it prevents excessive cholangiocyte proliferation.

14. Line 145 Fig 2c, Suppl 6a: Quantification of BC diameter is performed and the authors state this is significant, however I cannot see any analyses linked with these data.

A: We apologise for this omission; we have now provided the statistics for this panels.

15. Line 149, 671 Fig 2d: regarding the analyses of BC:BC connectivity – this would benefit from a schematic outlining, especially for the liver tissue samples, what areas and how these measurements were taken.

A: We now provide a schematic of how the quantification for the BC:BC connectivity is done (see Fig.R20).

Regarding image acquisition for HepOrg, we have already previously mentioned in the method section that we used high-resolution (voxel size: 0.3x0.3x0.3 μm) fluorescent image stacks (~50-80 μm depth) of optically cleared HepOrg (see previous methods line 619 now line 1025-1026). Segmentation was performed on CD13 (for bile canaliculi) and F-actin (cell borders) staining with phalloidin. Voxel size

was kept small (0.3 μ m) because bile canaliculi sizes are between 0.5-3 μ m. For tissue, the information on voxel size was given in line 629, now new line 1025. The Reviewer is asking for the information about which areas of the liver tissue the samples were imaged from. Briefly, we used 100 μ m-thick liver sections from C57BL/6, prepared and stained as previously described in Methods under “Thick tissue section staining”. From each tissue section, we chose 6 regions (150x150 μ m), 2 close to PV, 2 close to CV, and 2 in the middle between CV and PV. We took images from 3 different areas to have a better coverage of the tissue. Tissues were stained with CD13, Phalloidin, DAPI and Pan-cytokeratin (PCK). We distinguished PV area based on PCK signal, which specifically labels ductal cells. When taking these images, we saved ROIs of taken regions, which we added to complement the schematic outlining in Fig.R25. Regarding image processing, and segmentation, the same script was used as for the analysis of HepOrg as described in the methods section.

The details of the areas of liver tissue used for the quantification are below for the Reviewer only as Fig.R25.

Figure R25

Figure R25: Liver tissue section and areas selected for imaging bile canaliculi based on CD13 positive signal, and exclusion of PCK.

16. Given these data and experimental approaches are geared towards forming organoids which aim to recapitulate the peri-portal region in vivo, can the authors use immunofluorescence staining of hepatocyte zonation markers to investigate whether the hepatocytes are similar to zone 1 hepatocytes in vivo and whether these hepatocyte markers are influenced by the different culture conditions? Furthermore, the authors show qPCR for genes associated with mature hepatocyte

function however it would be helpful to also include genes associated with portal and central hepatocyte function to investigate whether these hepatocytes are portal, central, or a mix of both phenotypes.

A: We thank the reviewer for this comment. We have now evaluated the extent of periportal and pericentral gene expression in our hepatocyte organoids and assembloids in the different conditions. This involved analysing RNAseq data from HepOrg cultured in different conditions, scRNAseq data from assembloids and carrying out additional RNAscope and immunofluorescence analysis.

*From the new RNAseq data we found that assembloids and HepOrg in MM medium show higher levels of some hepatocytes genes that are considered to be periportal zoned (e.g. Hpx, Mup20, Aldob) as well as the periportal zoned cholesterol-bile acid pathway genes (Baat and Acaa1b), but also some pericentral genes such as Glul (GS) or Axin2, both also expressed in HM-WntCM, as expected (New Suppl. Fig. 7 and Fig.R2 below). These observations suggested that the cultures exhibited some degree of zonation. To assess that, we evaluated our scRNAseq data. To facilitate comparison of the expression of pericentral and periportal region genes in our scRNAseq, we have generated a consensus signature of pericentral and periportal genes. This was achieved by combining the expression of published and validated pericentral or periportal markers from three different publicly available datasets ¹³⁻¹⁵, and using this as a pericentral or periportal score. **The results demonstrate that hepatocyte organoids cultured in high Wnt conditions exhibit a more pericentral score, while HepOrg transferred to MM co-culture medium acquire some periportal features and present a periportal score.** We have validated these results by RNAscope and immunofluorescence analysis for some portal (Gls2 – RNA, ALB, ECAD - protein) and central (Cyp1a1 – RNA, CYP2E1, GS -protein) gene markers (New Fig.2a and Suppl. Fig.7). Notably, the Immunofluorescence analysis indicates that the medium switch promotes some degree of zonation as we detect, within the same structure, some cells expressing portal markers while others express some central markers.*

Combined, the immunofluorescence analysis, RNAscope, RNAseq and scRNaseq gene expression data indicates that HepOrg and periportal assembloids exhibit some degree of zonation. Now, in the new version of the manuscript, we have included this new data in the new Fig.2 and Suppl. Fig.7, also pasted below as Fig.R2.

Figure R2

Figure R2 HepOrg are partially zoned, and increase periportal gene expression in MM medium. **a.** Dotplot representing gene expression of WNT pathway genes across conditions in hepatocytes from scRNAseq (For reviewer only). **b.** Periportal and pericentral zonation score calculated from consensus gene expression from Halpern et al., Ben Moshe et al., and Wu et al. show higher periportal gene expression in MM media devoid of Wnt factors, when compared to Wnt-enriched HM-Wnt media hepatocytes. **c.** Dotplot representing gene expression of periportal, pericentral and zoned cholesterol genes across conditions in hepatocytes from scRNAseq. **d.** Expression of periportal (*Gls2*) and pericentral (*Cyp1a1*) gene RNA visualised by RNAscope in HepOrg from MM media. **e-f.** Immunofluorescence staining of periportal (ALB, Albumin, green **e**) and (E-CAD, E-cadherin, green, **f**) and pericentral (GS, Glutamine synthetase, magenta, **e**) and (CYP2E1, magenta, **f**) in liver tissue (top) or in HepOrg cultures grown in HM-Wnt and cultured in MM media for 7 days (bottom). Specific stainings are also shown in Fire LUT for the ease of viewing. Representative images are shown.

17. Fig 3c: does this graph show the number of assembloids vs those that have not assembled (eg HepOrg)? This is not clear in the figure legend. Also statistical comparisons should be shown here.

A: We apologise for the lack of clarity on our presentation. Indeed, the graph indicates in blue the % of assembloids that contain the 3 cells (~70%) while in grey the hepatocyte organoids that failed to incorporate the 3 cells (HepOrg or HepOrg that have incorporated 1 of the other cells). We have clarified that in the legend and also provided statistical analysis for this panel.

18. Line 169: linked with this figure states 70-90% efficiency however this is not what the graphs shows, ie assembloid-shaker is average of 70% with a range between 50-85%.

Linked figures Supp 7b,c do not clarify this, and furthermore Supp Fig 7c compares HM vs MM, but my understanding is that the optimal media had already been chosen by this point - please clarify these points.

A: This line 169 referred to the combined results presented in previous Fig.3c and Suppl.Fig.7c, and the text presented in lines 550 from the previous version. In shaker and aggrewell, the average efficiency of assembly is 70%, with a variability ranging from 50-85% efficiency in shaker (previous Fig.3c). As detailed in the text, in Suppl. Fig.7c (Now Suppl. Fig.9c), we improved the formation efficiency to 90% by pre-treating HepOrg for 48hrs with MM media prior to assembly. We have edited the text, figure and methods sections to better reflect this as follows in lines 257-260:

Line 257-260: " Both methods attained multicellular structures with high efficiency, an average of 70% of the total number of hepatocyte organoids, which was further improved to nearly 90% when the HepOrg were pre-conditioned for 48 hrs prior to aggregation in the co-culture medium (Fig.3b-c and Suppl. Fig. 9a-c, and methods)."

Further, in Fig 3c the aggrewell (non-shaker) seems to have equal efficiency but less variability, why did the authors not use this approach?

A: As the Reviewer noted, the aggrewell was less variable for the assembloid formation. However, in aggrewell, the assembloids tend to acquire a ball-shape structure compared to rocking platform/shaker. Therefore, we have opted for using a shaker method. We have clarified that in the text:

Line 260-262: "We selected rocking platform aggregation for subsequent experiments given that aggrewell tended to generate more ball-like structures (see methods for details)."

19. Line 171: The authors state the majority of assembloids formed within 48hrs - please show these data including quantification.

A: We respectfully believe that the Reviewer has not appreciated the data in previous Fig.3e and Suppl. Fig.7e (and associated Suppl. Video 2 and Suppl. Video 3), where we show the formation of assembloids occurs within the first 48hrs after formation. The graph in previous Suppl. Fig. 7d quantifies this, and shows that the majority of assembloids are already formed with 3 cell types at 48hrs post seeding, but they can also recruit additional cells later, once embedded in matrigel.

20. Line 174: the authors state the assembloids retain the cell proportions observed in vivo in tissue, however no comparative data is provided regarding this, or comparison with previously published data where in vivo measurements have been performed.

A: This information was given in the original submission (line 526-529). We have now transferred this section to the main text to clarify this point.

Line 249-252: “To define the numbers of the different cell types needed in periportal assembloids, we took advantage of our previous studies on the homeostatic proportion of hepatocytes and cholangiocytes (97% vs 3%⁵⁹, as well as of the proportions of portal mesenchymal: ductal cells in the mouse postnatal liver (3:10 ratio¹⁹).”

21. Fig 3b: Tissue - what does the white arrowhead refer to? Fig 3f: these appear to be different stages of maturity? What does the star mean, is this the lumen?

A: We apologise for missing this in the original submission. The arrowheads in Fig.3b refer to binucleated hepatocytes, while the star in Fig.3f refers to the cholangiocyte lumen, as correctly identified by the Reviewer. We have now added this information to the figure legends.

22. Given that the cell composition in Fig 3d shows that hepatocytes are the dominant cell type in assembloids, can the authors state why this is not recapitulated in their scRNAseq data (Fig 4g,h)? The low number of hepatocytes might impact on the analysis shown in Fig 4b,e,i. The authors should discuss this limitation in these datasets.

A: The Reviewer has rightly spotted that. In our previous version of the manuscript, we had an under-representation of hepatocytes in our dataset. We believe that to be caused by the poor recovery of hepatocytes following dissociation. We have now improved the dissociation method and can recover more hepatocytes, which are delicate, big and on the edge of 10x chip size limit. With this improved method we have now performed 2 more biological replicates of scRNAseq, which we have combined with our previous analysis. We now observed an improved recovery of hepatocytes that better aligns with the cellular proportions observed in vitro, shown below in Fig.R26 and in new Suppl. Fig.10b.

Figure R26

Figure R26: Improved hepatocyte recovery in the new scRNAseq datasets. a. UMAP representation of the three scRNAseq repeats. **b.** Proportion of cells in each of the scRNAseq biological replicates, separated into homeostasis and fibrosis-like datasets.

23. Line 191: the authors cannot rely solely on scRNAseq (mRNA-level) data to confirm that the mesenchymal component in their in vitro system resembles portal fibroblasts. Immunostaining with various markers would be useful additional confirmation.

A: We believe the reviewer overlooked our data from previous Suppl. Fig.11d, where in addition to all the scRNAseq data, we validate the nature of our portal fibroblast (PFs) with specific portal fibroblast marker SCA1^{19,20}). Nonetheless, to answer to this reviewer question, we have now performed additional staining for other genes known to be PF-specific in conjunction with PDGFRA (Elastin), which we now include in the revised version of the manuscript and below (Fig.R18).

Figure R18

Figure R18: Periportal assembloids contain portal fibroblast mesenchymal subtype. a. Correlation analysis of healthy cell types comparing liver tissue datasets with assembloids. **b.** Representative confocal images of periportal assembloids stained for portal fibroblast marker Elastin (white) marker, in combination with PDGFR α -H2BGFP endogenous signal, and with Msc membranes visualised by membrane-tdTom. Scale bar, 50 μ m. **c.** Immunofluorescence images staining of fibrosis-like assembloid for portal fibroblast marker SCA1 (white), as well as all mesenchyme (PDGFR α -H2BGFP, green), cholangiocytes (mem-tdTomato, magenta) and nuclei (DAPI, blue), where cell borders are outlined by phalloidin staining (white). Scale bar, 100 μ m, 20 μ m.

24. Line 181, fig 4b: The expression of hepatocyte genes across the board suggests a wash of hepatocyte genes in the dataset. Can the authors try using an algorithm such as SoupX to see if this helps with this pervading signal (if they believe this to be technical). In contrast, if the authors feel this represents a real finding, can they explain why all cell types are expressing hepatocyte genes. It is likely this is a typical hepatocyte gene wash which is commonly observed in this type of data, however the authors need to examine this / comment on it further.

A: We thank the Reviewer for this suggestion. We have run CellBender on our datasets, which similarly to SoupX, removes the unspecific wash of hepatocyte genes in other cell types. The results are shown in Fig.R27 below.

Figure R27

Figure R27: Using CellBender to remove unspecific gene wash from scRNAseq samples_a. Dot plot showing homeostasis assembloid cell gene expression before (a) and after (b) CellBender implementation.

Furthermore, can the authors comment on the numbers of cells as there appears to be a distinct lack of hepatocytes compared to the overall abundance that is observed in the assembloids. It is also not stated how many assembloid preps were sequenced, and bar charts showing numbers / ratios of cells contributed by each sample would also be helpful for the reader.

A: Regarding the number of assembloid preps sequenced at the time of original submission, the dataset only included 1 biological replicate. As mentioned in the answer to the point 22 of this same reviewer, we have now increased this to n=3 biological replicates. As mentioned above, our improved dissociation method has enabled us now to improve the recovery of number of hepatocytes as illustrated in Fig.R26 above. We have now re-done all our analysis with the 3 different replicates. Notably, we observe similar results between the biological replicates.

25. Fig5a shows a representative image from n=6 experiments. These data should be quantified.

A: We have quantified the number of successful connections between BC of hepatocytes and cholangiocyte duct-like structures. Briefly, once there are cholangiocytes within the hepatocyte organoid structure, there is a connection between BC and BD, i.e., 100 percent of efficiency. We have written that in the text now and it is presented in new Suppl. Fig.12f and Fig.R28 below to illustrate the concept.

Figure R28

Figure R28: Connection of bile duct (BC) to bile canaliculi occurs when BD is embedded within the structure. Maximum intensity projection (MIP) of assembloid with two bile duct structures, one integrated with the bile canaliculi network (A) and one on the outside of assembloid, not connected (B). Bile canaliculi (CD13, green), actin (Phalloidin, grey), nuclei (DAPI, blue) are stained, and Msc (PDGFRα-H2BGFP, yellow) and cholangiocytes (nuc-tdTomato, magenta) are also visualised. Scale bar, 50 μm; magnification, 20 μm.

26. Line 200-202: do these images of tissue and assembloid simply show that hepatocytes surround the BD? This 'recapitulation' of tissue needs to be more comprehensively characterized, and furthermore labelling in the image and figure legend needs to be clearer.

A: The reviewer refers here to lines 200-202 (previous Fig.5a-b, Suppl. Fig.9a-c and Suppl. Videos 4-7) that detail the extensive characterization of the BC-BD connection. We are sorry that we did not clarify the extent of tissue recapitulation, as the position of hepatocytes relative to the BD is not the only feature that is conserved between our organoid and the tissue, but there are several key characteristics, such as the functional connection between BC and BD. The data in previous Figure 5 and Suppl. Fig.9 (new Fig.4 and Suppl. Fig.12), including the serial confocal sections (previous Fig.5b, Suppl. Fig.9b-c), the reconstruction (previous Fig.5a and Suppl. Fig.9a) and the videos of the reconstruction (previous Suppl. Video 4-7) clearly show that the BC connects to the bile duct. Altogether, the previous version of the manuscript presented five different examples of BC-BD connection. In addition, these results combined with the CLF experiments, where we observed the fluorescent bile acid analogue flowing from the BC into the BD (previous Fig.5c, Suppl. Fig.9d), are the functional proof of that connection. Finally, the new data showing increase on gene expression of mature genes for both cholangiocytes and hepatocytes also show that this connection is functional (new Suppl. Fig.12h-g).

We believe that altogether, combined with an efficiency of connection of 100% (see point 25 of this reviewer), provides compelling evidence of the connection being functional and also beneficial for the overall fitness of the cells in a multicellular structure. If the reviewer or editor considers this not to be

sufficiently characterized, we would be happy to hear potential suggestions and/or provide additional examples of connection.

27. Line 230-234 and associated figures: The authors should quantify these data with comparisons to homeostasis. Cell death and proliferation (for example using Ki67 IHC) of all cells in the homeostasis and fibrotic-like conditions should also be quantified. Is there any cell death or proliferation in the cholangiocyte and mesenchymal lineages?

A: We provide now quantified the cleaved caspase 3 data comparing the homeostasis and fibrotic-like assembloids (New Fig.5g also below as Fig.R29a). From the scRNAseq data we observe increased proliferation in cholangiocytes (1.5-fold change) and in hepatocytes (2-fold change) when comparing the same cells in homeostasis or fibrotic assembloids. Additionally, we performed Ki67 staining of assembloids, observing positive staining in fibrotic-like cholangiocytes (New Fig.14h also below as Fig.R29b-d).

Regarding the proliferation of mesenchyme, we did not detect mesenchymal cell expansion (1-fold change). This is similar to the proliferative status of mesenchymal cells in all published models of liver fibrosis we have studied (6 different fibrosis models from 6 different studies). Only CCl4-damage models, with hepatotoxic fibrosis driven by hepatic stellate cells, show increased proliferation of mesenchyme. The other models, in particular the DDC and BDL models, both biliary fibrosis type of models similar to our assembloids, show similar levels of proliferation as our fibrotic-like mesenchyme (Fig.R7 for Reviewer only). A potential explanation for the lack of Msc expansion could be that at the time of fibrotic-like assembloid day 7 and DDC/BDL damage models of fibrosis, the fibrotic phenotype is already established, and mesenchymal cell proliferation is terminated. Our model, where we add an excess number of mesenchymal cells might be reflecting this advanced stage, mimicking the state of irreversibility of fibrosis, rather than initial steps in the fibrotic response. This hypothesis is something we would like to investigate in future studies. Therefore, we present these results below for the Reviewer only, and also discuss this concept in the manuscript (lines 538-539). Would the reviewer consider that these results are necessary to be included in the manuscript, we would be happy to follow his/her advice on that.

Figure R29

Figure R29: Cell death and proliferation in periportal assembloids. **a.** Immunofluorescence staining (left) of homeostatic-like and fibrotic-like assembloids for cleaved caspase 3 staining (orange); mesenchyme (PDGFR α -H2BGFP, green), cholangiocytes (mem-tdTomato, magenta), actin (Phalloidin, white) and nuclei (DAPI, blue) are also shown. Quantification of hepatocytes with cleaved caspase 3 positive staining (right). Violin plot shows distribution of values from n=3 independent biological replicates. Mann-Whitney test. Scale bar, 100 μ m, 50 μ m and 25 μ m. **b.** Immunofluorescence images of long-term culture (2.5 weeks) of periportal assembloids, staining for proliferation marker Ki67 (white). Endogenous expression of membrane CFP (mem-CFP, cyan, cholangiocytes), nuclear GFP (PDGFR α -H2BGFP, green, Msc) and membrane dTomato (mem-tdTomato, magenta, hepatocytes) is shown, with nuclei (DAPI, blue) also stained. Scale bar,

200 μm , inserts 50 μm . **c.** Mki67 expression from homeostasis and fibrotic-like assembloid cells from scRNAseq analysis. (c) Percentage of cells expressing Mki67 in each cell and condition subtype. (d) Fold change of fibrotic-like to homeostatic like Mki67 gene expression.

Similarly in Figs 10e, 11a, 11b, 8f - are these differences statistically significant?

A: We thank Reviewer for this suggestion, which strengthens the results of this manuscript. We have now quantified collagen deposition (SHG intensity) and the amount of apoptotic signal (cleaved Caspase 3 staining), shown here in Fig.R19 (and new Fig.5g) above for cleaved caspase 3, and in new Fig.5h for second harmonic generation. We have also performed statistical analysis on these results, which clearly indicate that the differences are statistically significant. These are now presented in the corresponding figures and explained in the figure legends accordingly.

28: Supp fig 11b: what is healthy vs unhealthy? In the figure legend and main results section homeostasis and fibrotic-like are the two conditions listed. Also are fig 11b and 11c the same data? If so, should these be in a different order?

A: We apologise for this typo. We have corrected “healthy” to homeostatic-like and “unhealthy” to fibrotic-like. We have also inverted the order of previous Suppl. Fig.11b&c (now Suppl. Fig.14d-e), as suggested by the Reviewer.

29. Line 228 and Supp Fig 10c: DC are not significantly increased in fibrosis-like state, but this is not what is stated in the results text.

A: We thank the Reviewer for spotting this mistake and apologise for having missed to add this data in the original submission. As mentioned in point 27, our new analysis of the scRNAseq data reveals that ~15% of the ductal cells are proliferating in the fibrotic assembloids (Fig.R29 above). Additionally, we include homeostatic and fibrotic periportal organoids cultured for longer time, which also exhibit cholangiocyte expansion only under fibrotic but not homeostatic conditions (new Suppl. Fig.14i-j and Fig.R30 below for reviewer’s assessment).

Figure R30

Figure R30: Long term culture of fibrotic-like assembloids results in expansion of the cholangiocyte compartment. **a.** Long-term culture (2.5 weeks) of periportal assembloids shows cholangiocyte expansion (magenta) in some structures. PDGFR α -H2BGFP Msc (green) and brightfield (grey) are also shown; scale bar, 100 μm . **b.** UMAP representation of cell proportions in scRNAseq data of homeostasis and fibrosis-like assembloids, from 1 week or 2.5-week (long-term, LT) culture, showing expansion of cholangiocytes in fibrosis-like long-term condition.

30. Line 242-245: The authors should analyse the preceding data in greater depth to allow a more comprehensive and in-depth comparison with in vivo cholestatic liver injury.

A: As requested by the reviewer, we have now compared our biliary fibrosis model to other models of biliary fibrosis as well as to the Mdr2^{-/-} cholestatic liver model. We observe that, similar to these cholestatic biliary fibrosis models, fibrotic assembloids exhibit:

1) correlation with the gene expression of the BDL (as we had already shown in the previous version) and also other publicly available fibrosis models (Fig. 5c-d, Suppl. Fig.13e-h and Suppl. Fig 14i-n).

2) present bile flow obstruction, as we had shown using CLF imaging in vivo, in previous Suppl. Fig.9d, and Suppl. Video 13 (now new Suppl. Fig.14d-f and Suppl. Video 17). Note that this is a hallmark of any cholestatic liver disease^{23,31}.

3) significant increase in collagen deposition (Fig. 5h and the non-targeting control in Fig.6 and Suppl. Fig.15).

4) significant increase in the number of ductal cells, reminiscent of ductular reaction observed in all cholestatic liver injuries in vivo (Fig. 5f, Suppl. Fig. 14h-k).

5) hepatocellular death (Fig. 5g and the non-targeting control in Suppl. Fig.15j).

None of these features are observed in homeostatic assembloids.

31. Is there a fibrotic phenotype when the cholangiocyte numbers increase/change relative to hepatocytes?

A: As shown in Suppl. Fig.4 of the original submission, just an overgrowth of cholangiocytes in culture does not cause fibrotic-like phenotype in hepatocytes, as the hepatocytes in those cultures appear very healthy with well-defined nuclei and cell borders. However, the presence of cholangiocytes in expanding hepatocyte organoids in HM-Wnt medium is not desirable as, in that case, contrary to when in assembloid medium (MM), cholangiocyte organoids continue to grow and overtake the cultures.

32. All videos would benefit from labelling on the video itself, not just in the figure legend. Figure legends are incomplete, for example information on what the arrowheads and stars refer to. All immunostaining images would benefit from clearer labelling to help guide the reader.

A: We apologise for this oversight and thank the Reviewer for this comment. We have now added all the fluorophore colour-codes to the videos themselves, as well as edited the video legends to make them clearer. We have also made several schematics (new Fig.1 Suppl. Fig1, Fig.2, Fig.4, Fig.6, Suppl. Fig16-18) and better labelled the figures to help guide the reader.

Referee #4 (Remarks to the Author):

Dowbaj et al., Functional periportal assembloids...

The authors digested the livers of adult mice enzymatically, cultured them as “organoids” using a Matrigel-based culture technique, and characterized the cultured mouse cells by imaging and image analysis, RNA-seq, a functional assay with a fluorescent bile salt analogue (CLF), and the analysis of cytokines, albumin and bile acids in the culture medium. The authors report that the organoids “recapitulate the mesoscale hepatic architecture”, “model biliary fibrosis in vitro”, “recapitulate the architecture and cellular interactions of the native periportal region of the liver tissue”, “periportal assembloids are fully functional”, “generate a physiological and functional bile canaliculi network”, functional bile canaliculi connect to bile ducts, “the periportal assembloids recapitulate the architectural arrangement and gene expression of the tissue”. Moreover, the authors state that the organoids “are reminiscent of the defects observed in cholestatic injury and in mouse models of biliary/cholestatic fibrosis”. Finally, the authors conclude that “our periportal assembloid model, where architecture and cell-cell interactions are recapitulated both at the meso- and cellular scale, represents the first system where to collectively study bile canaliculi formation, bile drainage and cholestatic liver disease”.

A: We thank the reviewer for reading our paper and providing insightful comments.

Comments

(1) It is not correct that this is the first study showing that primary liver cells form functional bile canaliculi in vitro and that bile canaliculi link to bile ducts. This has been reported in several publications. An example is Tanimizu et al, 2021 (doi: 10.1038/s41467-021-23575-1). These authors reported „the generation of a hepatobiliary tubular organoid (HBTO) using mouse hepatocyte progenitors and cholangiocytes. Hepatocytes form the bile canalicular network and secrete metabolites into the canaliculi, which are then transported into the biliary tubular structure.” The image below (Fig. 1C) of Tanimizu et al , 2021, shows the network of bile canaliculi that are formed by the apical membrane of hepatocytes (white) and are linked to a bile duct that is formed by cholangiocytes (green). These authors (Tanimizu et al , 2021) also used a Matrigel-based culture technique and applied the same bile salt analogue (CLF) to study secretion of CLF into the bile canalicular network in vitro and the functional link between bile canaliculi and bile ducts. It should be considered that Tanimizu et al. represent only an example showing a functional canalicular network; it should be considered that such culture techniques are already routinely applied to study if pharmaceutical candidate compounds reduce the secretion of bile acids (which is an undesired effect in drug development). Published examples are: DOI: 10.3791/60507-v; DOI: 10.3791/60507; [https://doi.org/10.1016/0270-9139\(95\)90371-2](https://doi.org/10.1016/0270-9139(95)90371-2); DOI: 10.1007/s00204-015-1575-9) (but many more studies exist).

A: From this comment we realize we had done a poor job in emphasizing the novelty of our study. We thank the reviewer for pointing this out and apologise for not having duly highlighted the fundamental differences between our study and the previous work of others. Because the reviewer cites studies with hepatocytes cultured in 2D-sandwich culture and Tanimizu’s studies in particular, we have split this answer in 2 parts (Part 1: comparison to 2D cultures and Part 2: Tanimizu).

Part 1) Novelty of our HepOrg culture system compared to published hepatocyte sandwich culture methods:

In the previous version of the manuscript, we had shown that our improved HepOrg model presents functional bile canaliculi (previous Fig.1 and Fig.2). However, we had not compared the BC of our HepOrg method to the gold-standard method for culturing hepatocytes from primary tissues, the “2D

hepatocyte sandwich culture” cited by the reviewer⁶⁰⁻⁶². Therefore, in this revised version we compare our model to the 2D-sandwich cultures suggested by the reviewer (Fig.R31). We also compare our system to the other 3D-hepatocyte organoids published by Hu et al., and Peng et al., 2018, to better highlight the novelty, significance and impact of the bile canaliculi in our model.

1.1) Novelty regarding bile canaliculi network properties: the bile canaliculi network is a bile transport network. Mathematically, transport networks are **spatial networks, which are fundamentally different in 2D compared to 3D networks in space**⁶³. Similarly, the function of flow networks is also highly dependent on their structure⁶⁴, which is fundamentally different between 2D and 3D networks. Consequently, the bile canaliculi network is also fundamentally different between the sandwich culture (2D) and our HepOrg culture (3D) (Compare Fig.2c and Suppl. Fig.8b with Suppl. Fig.1b, also below in Fig. R31a-b). **In our improved HepOrg the network is growing and expanding its complexity as hepatocytes are dividing.** This is reminiscent to what has been published regarding bile canaliculi formation during liver tissue development and is **fundamentally different to how hepatocytes in 2D sandwich culture eventually form bile canaliculi**, as it has been extensively recognized in the literature^{31,65}.

1.2) Novelty regarding the ability to maintain the polarity of hepatocytes and bile canaliculi properties for extended time in culture: As rightly pointed by the reviewer and extensively studied in the literature, in sandwich culture hepatocytes form functional bile canaliculi³⁶. However, primary hepatocytes require time (3–5 days) to fully polarize and develop metabolic functions in culture, after which the usage of this system is limited to 1–2 weeks. Therefore, **2D-sandwich hepatocytes are not suitable for long-term studies**^{66,67}. Instead, **our HepOrg model expands hepatocytes as hepatocyte organoids that retain bile canaliculi properties for several passages in culture** (Suppl. Fig. 1g-j, also in Fig.R31c below). Only hepatocytes grown in 3D as HepOrg from Hu et al. have been expanded long-term. However, as we had shown in our previous version, these organoids present short and barely connected bile canaliculi (Fig.1e).

1.3) Novelty regarding the similarity to in vivo bile canaliculi physiology: Our improved HepOrg present bile canaliculi with a **1.5-2 μm diameter, which is within the range of the in vivo tissue baseline** (Fig.2b-d and Suppl. Fig.8b also in Fig.R31d-e). In addition, the BC form a network, which is highly interconnected in 3D space and is fully functional - it readily transports fluorescent bile acid analogues to the apical bile canaliculi domain (previous Fig.1, now new Fig.2e). Despite some of these bile canaliculi features are seen in hepatocytes cultured in **2D sandwich culture, those hepatocytes lose their polarity within days and bile canaliculi transport can no longer be studied in these models**, as previously reported by many groups^{36,31,37,38}. More importantly, and as published by others³⁷, **hepatocytes in 2D-sandwich cultures present features of cholestasis, contrary to our improved HepOrg** (see point below 1.4).

1.4) Novelty regarding modelling of cholestatic response (pathophysiology) Polarity defects lead to aberrant bile canaliculi and subsequent cholestasis, which results in bile acid retention within hepatocytes, and cellular damage^{31,32}. In cholestatic liver tissue hepatocytes arrange around enlarged canalicular lumens, termed cholestatic liver cell rosettes³⁹. Also, in cholestasis bile canaliculi present “inward blebs”, which are deformations of the canalicular membrane⁴⁰. Finally, in cholestasis bulkhead structures appear. Bulkheads are extensions of the hepatocyte’s apical membrane sealed by tight junctions that appear in development, are lost in the adult healthy liver, and re-appear in mouse and human cholestatic liver disease^{37,38,41}. **Our improved HepOrg do not present any of these cholestatic features, namely rosettes, inward blebs or apical bulkheads** (Fig.R31f). On the contrary, hepatocytes in 2D-sandwich culture readily present bulkheads, as previously reported³⁷. This makes **our improved HepOrg model superior and also enables the study of bile canaliculi changes in**

cholestatic liver disease, which is not possible at all with the previous 2D-methods cited by the reviewer.

[Redacted text]

(Fig.R13 for Reviewer only). Similarly, HepOrg generated from hepatocytes derived from MDR2 knockout mouse, a well-characterized model for primary sclerosing cholangitis ⁴³, showed the presence of dilated canaliculi, inward blebs, and accumulation of liver rosettes, all of them observed in Mdr2^{-/-} liver tissue but absent in control tissue and HepOrg derived from WT littermates (new Fig.6f and Suppl. Fig.17 and below as Fig.R14a-c). Additionally, by combining these Mdr2^{-/-} HepOrg with WT cholangiocytes and WT mesenchymal cells we now provide the proof-of-concept that our assembloid system allows the study of cell-autonomous mechanisms in biliary fibrosis (new Fig.6 and Fig.R14d-f).

Taken together, our results indicate that the presence of a physiological BC is critical when we aim to model physiological liver tissue and liver pathophysiology, specifically cholestatic liver disease, which cannot be modelled well in the 2D-published methods because they present some features of cholestasis already at the baseline. In the new version of the manuscript, we have added the Mdr2 results but not the DCA data, as we feel that including it could dilute the main message of the paper, and we decided to only show them to the reviewer. Nevertheless, if the reviewer thinks this would be a useful addition, in concordance with the editor, we would be willing to add them in the main manuscript or in the supplementary information.

*For all the above, our improved HepOrg and assembloids models **represent a way to transition from the 2D-models used in pharmaceutical screenings cited by the reviewer to a more physiological 3D-models, where to study cholestasis-inducing drugs in a more physiologically-relevant context.** They could represent novel platforms for:*

- (1) toxicology testing of drugs that induce cholestasis, which was not possible before because the bile canaliculi are already aberrant at baseline in the previous models,*
- (2) test drugs against cholestasis, which can be modelled in HepOrg model as we show in the Fig.R14,*
- (3) investigate cell-autonomous mechanisms of cholestatic liver disease and biliary fibrosis. We show a proof-of-concept experiment in new Fig 6g-h.*

We now present this new data in the new Fig.1-2 Suppl. Fig.1 and 8, and Fig.6 and Suppl. Fig.17 of the revised version of the manuscript, also pasted below as Fig.R31 and Fig.R14, together with Fig.R13 for the Reviewer only.

Figure R31

Figure R31: Comparison of 3D HepOrg models to 2D hepatocyte collagen sandwich culture. a. Immunofluorescence staining of bile canaliculi (CD13, green) and F-actin (Phall, grey) in Sandwich hepatocyte culture, HepOrg cultures grown in Hu *et al.* media, HM-Wnt media or in MM media. Right, mouse healthy liver tissue control. Note the similarity between the bile canaliculi of HepOrg in MM and tissue (right). Representative images are shown. Scale bar, 50 μm . Magnification, 10 μm . **b.** Schematic illustration showing difference in bile canaliculi spatial arrangement comparing 2D and 3D system. **c.** Growth curves of HepOrg grown in the indicated media. Bile canicular length is improved in HepOrg cultured in HM-Wnt media. **d.** Left to right, immunofluorescence staining (most left) and image reconstruction and analysis of bile canaliculi from HM-Wnt (top), MM (middle) HepOrg, and liver tissue (bottom) stained for the bile canaliculi marker CD13 (green), nuclei (DAPI), and F-actin (Phall, grey). 3D reconstruction shows bile canalicular networks in 3D. Middle panel shows a detail of the reconstructed BC network. Right panel shows individual networks. Colour, individual interconnected network. Most right shows the skeleton of the BC network. Note that in MM the networks are longer and more interconnected. Scale bar, 20 μm . **e.** Graph represents the distribution of BC diameters in HepOrg grown in HM-Wnt (pink) and MM (blue), compared to liver tissue (green). The data is plotted as histogram. The curve represents the Kernel Density Estimate (KDE) line, used here to estimate the probability density function of a

continuous random variable, offering a smooth curve that represents the distribution of data points. **f.** HM-Wnt HepOrg recapitulate features of expanding/developing tissue, showing emergence of bile canaliculi bulkheads. In contrast, MM HepOrg show thin bile canaliculi without bulkheads. Immunofluorescence staining of bile canaliculi (CD13, green) and F-actin (Phall, grey).

[Redacted figure]

Figure R13: Modelling acute cholestatic response in optimised hepatocyte organoids using DCA. **a.** Schematic illustration showing hallmarks of cholestasis namely, liver rosette combined with dilated BC and apical bulkheads. The effect of deoxycholic acid (DCA) overload-induced cholestatic response in HepOrg is shown. Bile canaliculi (green), hepatocytes (orange). **b.** Maximum intensity projection of F-actin (magenta) and CD13 (green) in DMSO-treated and DCA-treated hepatocyte organoids. HepOrg treated with 200 μ M DCA show canalicular dilation. Representative images are shown. Scale bar 50 μ m, magnification 10 μ m. **c.** Still images of a representative live cell imaging analysis of the bile canaliculi expansion upon DCA treatment. Note the increase of BC size in DCA-treated organoids. Cell borders were visualized following SiR-actin labelling (grey); bile canaliculi are seen as enrichment in SiR-actin intensity. Both control (DMSO, top panel) and DCA-treated (bottom panel) hepatocyte organoids were incubated with CLF to visualise bile canaliculi at the beginning (12h post-treatment) and at the end of imaging (25h post-treatment). **d.** High-resolution microscopy images of individual bile canaliculi in liver tissue (left panel) and HepOrg (right panel). Horizontal panels show a single bile canaliculus or rosette at multiple z-planes to appreciate the detail of lumina compared between tissue and HepOrg. Bile canaliculi in control HepOrg treated with DMSO (top panel, right) show remarkable similarities to Sham-operated liver tissue (top panel, left). Hepatocytes accumulate apical bulkheads upon bile duct ligation (BDL) in vivo and DCA treatment in vitro. Note the presence of hepatocyte rosette tissue in hepatocyte organoids. Sham and DMSO controls show the absence of bulkheads. Both tissue and hepatocyte organoids are stained for F-actin (grey). Scale bar, 2 μ m.

Figure R14

Figure R14: Modelling cholestasis and biliary fibrosis using optimised hepatocyte organoids from *Mdr2*^{-/-} mice. **a.** Schematic illustration showing the generation of *Mdr2*^{+/+} and *Mdr2*^{-/-} HepOrg. Hepatocytes were isolated and cultured in the same conditions. Compared to WT (top), the *Mdr2*^{-/-} HepOrg (bottom) show dilated bile canaliculi with bulkheads. Representative IF image of *Mdr2*^{-/-} HepOrg is shown. DAPI (blue), Phalloidin (grey), CD13 (green). Scale bar 50 μ m, 20 μ m, 10 μ m. **b.** Immunofluorescence staining of bile canaliculi (CD13, green) and F-actin (Phalloidin, grey) in *Mdr2*^{-/-} HepOrg cultures (bottom panel) compared with *Mdr2*^{-/-} liver tissue (top panel). Note the similarity in the bile canaliculi morphology between HepOrg and tissue. Cyan arrowheads indicate apical bulkheads, and yellow arrowheads indicate inward blebs, in both tissue and HepOrg. Representative images are shown. Scale bar, 50 μ m; magnification, 20 μ m. **c.** Immunofluorescence staining of bile canaliculi (CD13, green) and F-actin (Phalloidin, grey) in *Mdr2*^{-/-} HepOrg cultures, showing emergence of hepatocyte rosettes. Representative images are shown. Scale bar, 50 μ m; magnification, 20 μ m. **d.** Schematic illustration showing the starting material for periportal assembloid formation. *Mdr2*^{-/-} assembloids are made from *Mdr2*^{-/-} HepOrg, WT cholangiocytes, and WT Msc. *Mdr2*^{+/+} are made from *Mdr2*^{+/+} HepOrg, WT cholangiocytes, and WT Msc. Cholangiocytes are labelled with nuclear tdTomato, Msc with nuclear PDGFRa-H2B-GFP, and hepatocytes are unlabelled. The same number of HepOrg, DCs and Msc is added in both conditions. **e.** Live imaging showing changes in *Mdr2*^{-/-} and *Mdr2*^{+/+} PAs during first five days of the assembloid formation. Hepatocyte organoids (brightfield, grey), ductal cells (nTom, magenta). Representative images from n=2 independent experiments at 24, 48 and 132h are shown. Scale bar, 50 μ m. **f.** Graph representing the change in the ductal cell (DC) number between the day 5 (d5) and day 0 (d0) after generating assembloids, using either *Mdr2*^{+/+} or *Mdr2*^{-/-} HepOrg. Graph represents mean \pm SEM of n=2 biological replicates from n=2 independent experiments. Each dot represents change in total number of DC from d0-d5 of an individual portal assembloid, *Mdr2*^{+/+} assembloids (n=23), and *Mdr2*^{-/-} assembloids (n=22). ****, P < 0.0001 using a two-tailed Mann-Whitney test including all technical replicates from both experiments.

Part 2 Novelty compared to Tanimizu and colleague's work

An example is Tanimizu et al, 2021 (doi: 10.1038/s41467-021-23575-1). These authors reported „the generation of a hepatobiliary tubular organoid (HBTO) using mouse hepatocyte progenitors and cholangiocytes. Hepatocytes form the bile canalicular network and secrete metabolites into the canaliculi, which are then transported into the biliary tubular structure.” The image below (Fig. 1C) of Tanimizu et al , 2021, shows the network of bile canaliculi that are formed by the apical membrane of hepatocytes (white) and are linked to a bile duct that is formed by cholangiocytes (green). These authors (Tanimizu et al , 2021) also used a Matrigel-based culture technique and applied the same bile salt analogue (CLF) to study secretion of CLF into the bile canalicular network in vitro and the functional link between bile canaliculi and bile ducts.

We fully agree with the reviewer that the work of Tanimizu et al. ⁶⁸is interesting. Of note, in our earlier version of the manuscript we had acknowledged this earlier work from Tanimizu's lab (previous reference 21 line 65, now in lines 74-78 and 490-493). However, while interesting, our study is fundamentally different to that of Tanimizu and colleagues in that:

*2.1) **In the work by Tanimizu et al., the primary cells are cultured in a 2D sandwich hepatocyte culture with the modification of the ECM composition.** It allows formation of structures, but they remain **restricted to 2D** as they grow between layers of hydrogel (Collagen/Matrigel). As stated above, culturing hepatocytes in 2D does not allow them to expand, imposes constraints in the architecture of the 3D bile canaliculi network and its similarity to the native tissue, and is potentially cholestatic (present bulkheads). In fact, the close inspection of the figures presented in Tanimizu's manuscript indicate that the bile canaliculi are wider than in vivo baseline (>2 μm). See Fig.2a and Suppl. Fig.4 of the Tanimizu et al. manuscript. Notably, Tanimizu's 2D-Hepatocyte cultures show bulkhead formation (Fig.2b from ⁶⁸), suggestive of dilated bile canaliculi and cholestatic phenotypes (Fig.R32 for the Reviewer only).*

*2.2) **Tanimizu's model uses 2D-cultures** of 'small hepatocytes' combined with ductal cells. Our periportal **assembloids are 3D**, contain **hepatocytes and cholangiocytes from hepatocyte and cholangiocyte organoids** respectively, and **additionally incorporate portal fibroblasts**, which is novel and different to Tanimizu's work, and allows us to model biliary fibrosis. To our knowledge, this is the first in vitro model that incorporates these 3 cell types of the periportal region from adult liver tissue.*

*2.3) The HBTO from Tanimizu et al. shows fluorescent bile acid analogue transfer from bile canaliculi to bile duct, as rightly pointed by the reviewer. However, when examining the publication in detail, we noted that the timescales of this transfer are measured in hours, from 2 to 12 hours (Fig.3a- b, Suppl. Fig.9 and methods from ⁶⁸). Instead, in our assembloids this transfer occurs in a timescale of minutes, as we had shown in the previous version of the manuscript. This resembles the native tissue timescale for the flow of bile from bile canaliculi to bile duct as described previously ⁶⁹, and indicates that **our assembloid model better recapitulates the physiological link between hepatocyte's bile canaliculi and the bile duct with physiological timescales of bile acid transport.** Below we present these differences between Tanimizu's and our study, to facilitate the reviewer's assessment (Fig.R33 for the Reviewer only).*

Taken together, from points 1 and 2 we demonstrate that the HepOrg model and the periportal assembloid model presented in our study, both represent significantly improved in vitro models compared to the existing methods because:

- (1) HepOrg expand long term with serial passaging, while retaining bile canaliculi function*
- (2) HepOrg and periportal assembloids possess physiological bile canaliculi with diameter similar to the tissue (1.5-2 μm) and physiological apical bile acid transport*

- (3) the bile canaliculi do not show cholestatic signature (absence of bulkheads) and hence allow modelling cholestatic response in hepatocytes,
- (4) HepOrg and periportal assembloids present an elongated and interconnected bile canaliculi network in 3D,
- (5) assembloids show transfer of fluorescent bile analogue from BC to BD at physiological timescales
- (6) the assembloids incorporate portal fibroblasts, which allows us to model aspects of biliary fibrosis.
- (7) as stated above, we prove periportal assembloids can be used as a tool to investigate contribution of each of the cells (hepatocytes, cholangiocytes, mesenchyme) to biliary fibrosis and to cholestatic disease, without the co-funding factors induced by the immune system (New Fig.6a-d).

To our knowledge, while the existing methods fulfil some of the criteria outlined above, our HepOrg and assembloids are the first model to fulfil them all. We hope that all these results combined together better illustrate the significance, importance and novelty of our findings, and clarifies this point for the Reviewer.

We have edited the manuscript accordingly to better reflect these differences between the previous models and our study (lines 490-507).

Figure R32

Figure R32: Tanimizu’s model compared to HepOrg and homeostatic assembloids. a. Adapted from Fig. 1c from Tanimizu et al., 2021. Legend reads as follows: “tdTomato+ hepatocytes maintain their lineage to form a continuous luminal network with cholangiocytes. Optical cross-sections along broken lines in the square four panels are shown in the far-right panels. The luminal network among tdTomato+ hepatocytes (upper right

panel) is connected to that of the *tdTomato*-CK19+ biliary structure (lower right panel) at the boundary (middle right panel). The luminal network is recognized by F-actin bundles visualized using phalloidin. The immunostaining with phalloidin (white), anti-CK19 antibody (green), and Hoechst 33342 (blue) was repeated three times, independently. Two fields were examined in each sample. Two out of six areas were further used to collect serial optical sections for 3D reconstruction. The representative images are shown in this figure. Bars represent 40 μm ." Note that we have highlighted, with blue arrowheads, several apical bulkheads and dilated bile canaliculi, normally observed in 2D-cultured hepatocytes, and also present in Tanimizu's model. Please compare these images with Figure R31b and R31d and to R13d DMSO control. **b-c.** HepOrg (b) or assembloids (c) do not present apical bulkheads at baseline.

Figure R33

Figure R33: Transfer of the Bile acid analogue CLF in Tanimizu's model compared to homeostatic assembloids. **a.** Adapted from Fig 3b from Tanimizu et al., 2021 Legend reads as follows: "Transport of bile acids in HBTO. CLF taken up by hepatocytes is transported into the biliary network. The organoids were incubated in the presence of CLF for 30 min. After five washes, images were taken at 30 min (panels 1 and 2) and at 6 hours (panels 3 and 4). Broken lines indicate the boundary between HEP and BD. Representative images are shown in this figure. Bars represent 100 μm ." **b.** Live imaging analysis of the uptake and flow of the bile acid analogue cholyl-L-lysyl-fluorescein (CLF, Fire LUT) in homeostatic periportal assembloids shows functional transport of bile salts from bile canaliculi into the lumen of the bile duct lined by cholangiocytes (*mem-tdTomato*, magenta). Representative images from $n=3$ independent experiments are shown. Scale bar, 50 μm . Note that in Tanimizu's experiment the transfer of the bile analogue CLF takes 6h. In assembloids, this transfer has already occurred at the first imaging interval 14 min.

(2) This reviewer does not agree to the statement that the "Periportal assembloids model cholestatic

liver fibrosis in the absence of immune compartment” (headline in results, page 6) or “model biliary fibrosis in vitro” (as written in the title). It is well-known that in cholestatic liver disease (and the resulting periportal fibrosis) bile ducts proliferate (named ductular reaction) during the entire disease process initially forming a tighter duct network around portal veins and later infiltrating into the liver lobule (DOI: 10.1002/hep.30150; and others).

A: In the previous version of the manuscript, we had shown that our periportal assembloids, when cultured with excess portal mesenchyme, modelled aspects of cholestatic biliary fibrosis namely, hepatocyte death, ECM deposition and bile flow obstruction (see previous Fig.6 and Suppl. Fig10-14). Also, in the previous Suppl. Fig.10c we had already alluded to the fact that there was ductal cell expansion in the fibrotic, but not in the homeostatic assembloids. However, the reviewer is right in that we have not formally quantified that cholangiocyte expansion in detail. In this revised version of the manuscript, we have revisited that data, as well as included new added data in the scRNAseq. We now show that cholangiocytes significantly increase in number in fibrotic-like assembloids, reminiscent of ductular reaction present in biliary fibrosis.

In addition, to further study whether the fibrotic assembloids resemble the fibrotic tissue, we have now cultured the assembloids for up to 1 month (4 weeks). In those longer-cultured assembloids, we readily observed expansion of cholangiocytes in fibrotic but not homeostatic assembloids at 2.5 weeks of culture. These results are now presented in the new Suppl. Fig.14 (pasted here as Fig.R30 for the reviewer).

Finally, we have also generated assembloids from hepatocytes derived from Mdr2^{-/-} mice, a well-studied model of biliary fibrosis. In these assembloids the hepatocytes are mutant but the ductal and mesenchymal cells are derived from WT mice. Notably, in these assembloids ductal cells also proliferate, resembling the in vivo liver tissue of Mdr2^{-/-} mice, demonstrated now in new Fig.6 also Fig.R14 above.

Figure R30

Figure R30: Long term culture of fibrotic-like assembloids results in expansion of the cholangiocyte compartment. *a. Long-term culture (2.5 weeks) of periportal assembloids shows cholangiocyte expansion (magenta) in some structures. PDGFR α -H2BGFP Msc (green) and brightfield (grey) are also shown; scale bar, 100 μ m. **b.** UMAP representation of cell proportions in scRNAseq data of homeostasis and fibrosis-like assembloids, from 1 week or 2.5-week (long-term, LT) culture, showing expansion of cholangiocytes in fibrosis-like long term condition.*

Together with the new ducts also periportal fibroblasts proliferate. This key feature is not shown in the present study (although cholangiocytes were included into the culture system) and it is therefore difficult to understand why the authors claim that the liver assembloids “model cholestatic liver fibrosis”.

A: We thank the Reviewer for this comment, which has made us improve our manuscript. While we consistently observe ductal cell expansion in our fibrotic assembloids, we did not detect mesenchymal cell expansion. This is in line with the proliferative status of mesenchymal cells in all published models of fibrosis we have studied, which do not show mesenchymal cell expansion (6 different fibrosis models from 6 different studies). Only CCl4-damage models, with hepatotoxic fibrosis driven by hepatic stellate cells, show increased proliferation of mesenchyme. The other models, in particular the DDC and BDL models, both models of biliary fibrosis similar to our assembloids, do not show increased proliferation of mesenchyme (Fig.R7 for reviewer). A potential explanation for this result could be that at the time of analysis fibrosis is already established, and mesenchymal cell proliferation is terminated. Our model, where we add an excess number of mesenchymal cells, might be reflecting this advanced stage, mimicking the state of irreversibility of fibrosis, rather than initial steps in the fibrotic response. This hypothesis is something we would like to investigate in future studies. We discuss this concept in the manuscript, lines 527-541.

Therefore, in light of the data presented in new Fig.5 and Fig.6, and associated Suppl. Fig.13-18, we believe that we provide enough evidence to support our claims that cholestatic liver fibrosis can be modelled in our system. In particular we show that our fibrotic assembloids present (i) similar expression patten as in vivo models of biliary fibrosis, (ii) hepatocyte death, (iii) cholangiocyte cell expansion, (iv) bile flow obstruction, (v) collagen deposition, and now (vi) the limited proliferative status of mesenchyme as in vivo models of biliary fibrosis.

Figure R7

Figure R7: Expression of Mki67 gene in different models of fibrosis comparing to the Msc on our fibrotic-like assembloids (THIS STUDY). HSC, hepatic stellate cell, grey; FIB, fibroblast, green; VSMC, vascular smooth muscle cell, blue. PHx, partial hepatectomy; DDC; CCl4; TAA; APAP; BDL, bile duct ligation – all specify the type of damage applied to mice. Where appropriate, time of damage is also specified: 17d, 17 days; 8d, 8 days; 72h, 72 hours; 2w, 2 weeks; 3w, 3 weeks; 4w, 4 weeks. The publication is specified on the left.

(3) The authors claim that the organoids “recapitulate the mesoscale hepatic architecture” (Title and several similar statements in the results and discussion). This statement may be misinterpreted. Previously, some authors used the term mesoscale hepatic architecture for structures ranging from the micron to the millimeter level (doi: 10.7150/thno.71718). The millimeter level includes the liver lobule (radius 200-400 μm). However, in their organoids the authors did not see any lobules that consist of a central vein, periportal fields (containing branches of the portal vein and liver artery) and sheets of hepatocytes forming sinusoids from the periphery to the central vein.

A: We thank the Reviewer for bringing this issue to our attention. As rightly pointed by the reviewer, our structures are around 200 μm in size, and it was never our intention (and we did not claim) to imply the recapitulation of the whole liver lobule in the system.

In fact, the exact definition of mesoscale is ‘Of medium size or extent; between microscale and macroscale’⁷⁰. As a result, mesoscale could mean different things to different people. The accepted definition is that the biological mesoscale begins at length scales larger than the molecular machines, and extends up to the dimensions of an individual cell⁷¹. In our study, with the word “mesoscale”, we wanted to illustrate the intermediate scale between the cellular scale and macroscopic tissue scale, for which we thought the term was appropriate.

We have now specified what we mean with “mesoscale” in the introduction (line 71-72), to avoid confusion, and we acknowledge that the definition is not universal. However, in case the reviewer and the editor consider this term inappropriate, we would be happy to remove this term from the manuscript and follow the reviewer’s and editor’s advice on this point.

A key feature of these sheets of hepatocytes is their zonation. Often used markers of zonation are, e.g, expression of glutamine synthetase and certain cytochrome P450 isoenzymes preferentially in the lobular center but not in the periphery. Since the authors did not show any lobular zonation, it may appear overstretched to conclude that the “assembloids recapitulate the mesoscale hepatic architecture”.

A: We have now evaluated the extent of periportal and pericentral gene expression in our hepatocyte organoids in the different conditions. This involved analysing RNAseq data from HepOrg cultured in different conditions, scRNAseq data from assembloids and carrying out additional RNAscope and immunofluorescence analysis.

*From the new RNAseq data we found that assembloids and HepOrg in MM medium show higher levels of some hepatocytes genes that are considered to be periportal zoned (e.g. Hpx, Mup20, Aldob) as well as the periportal zoned cholesterol-bile acid pathway genes (Baat and Acaa1b), but also some pericentral genes such as Glul (GS) or Axin2, both also expressed in HM-WntCM, as expected (New Suppl. Fig. 7 and Fig.R2 below). These observations suggested that the cultures exhibited some degree of zonation. To assess that, we evaluated our scRNAseq data. To facilitate comparison of the expression of pericentral and periportal region genes in our scRNAseq, we have generated a consensus signature of pericentral and periportal genes. This was achieved by combining the expression of published and validated pericentral or periportal markers from three different publicly available datasets¹³⁻¹⁵, and using this as a pericentral or periportal score. **The results demonstrate that hepatocyte organoids cultured in high Wnt conditions exhibit a more pericentral score, while HepOrg transferred to MM co-culture medium acquire some periportal features and present a periportal score.** We have validated these results by RNAscope and immunofluorescence analysis for some portal (Gls2 – RNA, ALB, ECAD - protein) and central (Cyp1a1 – RNA, CYP2E1, GS -protein) gene markers (New Fig.2a and Suppl. Fig.7 and also pasted below Fig.R2) Notably, the Immunofluorescence analysis indicates that the medium switch promotes some degree of zonation as we detect, within the same structure, some cells expressing portal markers while others express some central markers.*

Figure R2

Figure R2 *HepOrg and assembloids are partially zoned with some pericentral and periportal gene expression. a.* Dotplot representing gene expression of WNT pathway genes across conditions in hepatocytes from scRNAseq (For reviewer only). *b.* Periportal and pericentral zonation score calculated from consensus gene expression from Halpern et al., Ben Moshe et al., and Wu et al. show higher periportal gene expression in MM media devoid of Wnt factors, when compared to Wnt-enriched HM-Wnt media hepatocytes. *c.* Dotplot representing gene expression of periportal, pericentral and zoned cholesterol genes across conditions in hepatocytes from scRNAseq. *d.* Expression of periportal (*Gls2*) and pericentral (*Cyp1a1*) gene RNA visualised by RNAscope in HepOrg from MM media. *e-f.* Immunofluorescence staining of periportal (*ALB*, Albumin, green *e*) and (*E-CAD*, E-cadherin, green, *f*) and pericentral (*GS*, Glutamine synthetase, magenta, *e*) and (*CYP2E1*, magenta, *f*) in liver tissue (top) or in HepOrg cultures grown in HM-Wnt and cultured in MM media for 7 days (bottom). Specific stainings are also shown in Fire LUT for the ease of viewing. Representative images are shown.

(4) A further critical conclusion is that “assembloids ... recapitulate the ... gene expression of the tissue...”. However, several previous studies have shown that hepatocytes in culture up and downregulate hundreds of genes by large factors compared to the in vivo situation. Therefore, many authors use freshly isolated liver cells or fresh liver tissue as controls for comparison with cultured

liver cells in RNA-seq or other gene expression studies. The authors of the present study did not include such controls. Their comparison is therefore based on literature data which limits the accuracy.

A: We respectfully believe that the reviewer has missed our data presented in previous Suppl. Fig.5b, where we compare the expression of freshly isolated hepatocytes to the expression of hepatocyte organoids grown in different conditions. In addition, we had also compared our scRNAseq to publicly available liver cell atlas that contain the expression of freshly isolated liver cells, as the reviewer indicates.

In the course of this revision, we have included new samples and additional new recently published liver atlas datasets. This new data confirmed the results from the previous version of the manuscript, and now, once more our hepatocytes, cholangiocytes and mesenchyme cluster with publicly available hepatocyte, cholangiocyte and portal fibroblast data from in vivo atlases (Fig.R18). In addition to gene expression analysis, throughout the manuscript we also compare the hepatocyte polarity and 3D-tissue architecture to the one in the tissue with a series of immunofluorescence staining, which all of them confirm the similarity of the organoids to the tissue (see Fig.1g, Fig.3b, Fig.4i, Fig.5a, Suppl. Fig.3b&c, Suppl. Fig.9a, and Suppl. Fig11e-f from previous version).

In this revised version of the manuscript, we have extended on these analyses and performed RNAseq for hepatocytes, and additional qPCRs, always comparing hepatocyte organoids to the freshly isolated hepatocytes (new Suppl. Fig.7a and Fig.R34 for reviewer's assessment). Also, since we had only compared fresh isolated cells to hepatocyte organoids but not cells from assembloids, we have now extended these studies and performed additional qPCRs comparing hepatocytes, cholangiocytes and mesenchyme cells sorted from assembloids, to the freshly isolated hepatocytes, cholangiocytes and mesenchyme. Altogether, the results confirm that the cultured hepatocyte organoids and cultured cells from the assembloids resemble fresh isolated hepatocytes, cholangiocytes and portal mesenchyme, in terms of their gene expression profiles (new Suppl. Fig.11a-c and Fig.R34 for the reviewer below).

Finally, to complement the response to the reviewer, we have also performed functional assays for albumin secretion, total bile acid analysis, and cytochrome assays, comparing to hepatocyte controls grown in sandwich hepatocyte culture, as requested by the Reviewer⁶⁰⁻⁶², demonstrated in new Suppl. Fig.4h and FigR34 below.

We believe that all these comparisons (qPCR, scRNAseq, RNAseq, IF and functional assays) proof that the HepOrg and Assembloids described in our manuscript resemble and recapitulate major features of the in vivo cells in vitro.

Figure R34

Figure R34: Comparison of HepOrg and periportal assembloid cells to freshly isolated cells from liver tissue. a. Bulk RNA sequencing analysis of HM-Wnt and MM media HepOrg compared to freshly isolated hepatocytes. Heatmap represents the TPM (transcripts per million) values from the RNAseq for the indicated genes ($n = 3$ biological replicates). **b-d.** Analysis of Msc, cholangiocytes and hepatocytes FACS-sorted from assembloids, compared to freshly isolates cells from liver tissue, qRT-PCR expression analysis of selected marker genes for mesenchyme (*Pdgfra*, *Vim*, *Cdh11*), cholangiocyte (*Krt19*, *Krt7*), progenitor (*Sox9*) and hepatocyte (*Alb*, *Cyp3a11*, *Hnf4a*) identity. Graph represents mean \pm SEM from $n=4$ (assembloid) or $n=3$ (tissue) biological replicates from at least $n=3$ independent experiments. **e.** Albumin secretion, Cytochrome activity and total bile acid measurements of HM-Wnt HepOrg compared to hepatocyte sandwich culture show improved functionality of HepOrg. Graph represents mean \pm SEM of $n>3$ biological replicates from $n= 3$ independent experiments, with each dot denoting biological replicate; Mann-Whitney test.

References

- 1 Barber, J. A. *et al.* Quantification of Drug-Induced Inhibition of Canalicular Cholyl-L-Lysyl-Fluorescein Excretion From Hepatocytes by High Content Cell Imaging. *Toxicol Sci* **148**, 48-59, doi:10.1093/toxsci/kfv159 (2015).
- 2 Kroll, T., Prescher, M., Smits, S. H. J. & Schmitt, L. Structure and Function of Hepatobiliary ATP Binding Cassette Transporters. *Chem Rev* **121**, 5240-5288, doi:10.1021/acs.chemrev.0c00659 (2021).
- 3 Cui, Y. J., Cheng, X., Weaver, Y. M. & Klaassen, C. D. Tissue distribution, gender-divergent expression, ontogeny, and chemical induction of multidrug resistance transporter genes (*Mdr1a*, *Mdr1b*, *Mdr2*) in mice. *Drug Metab Dispos* **37**, 203-210, doi:10.1124/dmd.108.023721 (2009).

- 4 Wustner, D., Mukherjee, S., Maxfield, F. R., Muller, P. & Herrmann, A. Vesicular and nonvesicular transport of phosphatidylcholine in polarized HepG2 cells. *Traffic* **2**, 277-296, doi:10.1034/j.1600-0854.2001.9o135.x (2001).
- 5 Brecklinghaus, T. *et al.* The hepatocyte export carrier inhibition assay improves the separation of hepatotoxic from non-hepatotoxic compounds. *Chem Biol Interact* **351**, 109728, doi:10.1016/j.cbi.2021.109728 (2022).
- 6 Lin, C. J. *et al.* Direct visualization of functional heterogeneity in hepatobiliary metabolism using 6-CFDA as model compound. *Biomed Opt Express* **7**, 3574-3584, doi:10.1364/Boe.7.003574 (2016).
- 7 Murray, J. W., Han, D. & Wolkoff, A. W. Hepatocytes maintain greater fluorescent bile acid accumulation and greater sensitivity to drug-induced cell death in three-dimensional matrix culture. *Physiol Rep* **2**, doi:ARTN e12198 10.14814/phy2.12198 (2014).
- 8 Russell, J. O. & Monga, S. P. Wnt/beta-Catenin Signaling in Liver Development, Homeostasis, and Pathobiology. *Annu Rev Pathol* **13**, 351-378, doi:10.1146/annurev-pathol-020117-044010 (2018).
- 9 Wang, B., Zhao, L., Fish, M., Logan, C. Y. & Nusse, R. Self-renewing diploid Axin2(+) cells fuel homeostatic renewal of the liver. *Nature* **524**, 180-185, doi:10.1038/nature14863 (2015).
- 10 Brosch, M. *et al.* Epigenomic map of human liver reveals principles of zoned morphogenic and metabolic control. *Nat Commun* **9**, 4150, doi:10.1038/s41467-018-06611-5 (2018).
- 11 Belenguer, G. *et al.* RNF43/ZNRF3 loss predisposes to hepatocellular-carcinoma by impairing liver regeneration and altering the liver lipid metabolic ground-state. *Nat Commun* **13**, 334, doi:10.1038/s41467-021-27923-z (2022).
- 12 Planas-Paz, L. *et al.* The RSPO-LGR4/5-ZNRF3/RNF43 module controls liver zonation and size. *Nat Cell Biol* **18**, 467-479, doi:10.1038/ncb3337 (2016).
- 13 Halpern, K. B. *et al.* Single-cell spatial reconstruction reveals global division of labour in the mammalian liver. *Nature* **542**, 352-356, doi:10.1038/nature21065 (2017).
- 14 Wu, B. *et al.* A spatiotemporal atlas of cholestatic injury and repair in mice. *Nat Genet* **56**, 938-952, doi:10.1038/s41588-024-01687-w (2024).
- 15 Ben-Moshe, S. *et al.* The spatiotemporal program of zonal liver regeneration following acute injury. *Cell Stem Cell* **29**, 973-989 e910, doi:10.1016/j.stem.2022.04.008 (2022).
- 16 Marsh, L. Z. F. Y. Q. X. L. X. B. H. M. H. H. A. Detecting Temporal shape changes with the Euler Characteristic Transform. *Transactions of Mathematics and Its Applications In Press* (2024).
- 17 Alimperti, S., You, H., George, T., Agarwal, S. K. & Andreadis, S. T. Cadherin-11 regulates both mesenchymal stem cell differentiation into smooth muscle cells and the development of contractile function. *J Cell Sci* **127**, 2627-2638, doi:10.1242/jcs.134833 (2014).
- 18 Row, S., Liu, Y. Y., Alimperti, S., Agarwal, S. K. & Andreadis, S. T. Cadherin-11 is a novel regulator of extracellular matrix synthesis and tissue mechanics. *J Cell Sci* **129**, 2950-2961, doi:10.1242/jcs.183772 (2016).
- 19 Cordero-Espinoza, L. *et al.* Dynamic cell contacts between periportal mesenchyme and ductal epithelium act as a rheostat for liver cell proliferation. *Cell Stem Cell* **28**, 1907-1921 e1908, doi:10.1016/j.stem.2021.07.002 (2021).

- 20 Dobie, R. *et al.* Single-Cell Transcriptomics Uncovers Zonation of Function in the Mesenchyme during Liver Fibrosis. *Cell Rep* **29**, 1832-1847 e1838, doi:10.1016/j.celrep.2019.10.024 (2019).
- 21 Williams, M. *et al.* Spatial proteogenomics reveals distinct and evolutionarily conserved hepatic macrophage niches. *Cell* **185**, 379-396 e338, doi:10.1016/j.cell.2021.12.018 (2022).
- 22 Wells, R. G. The portal fibroblast: not just a poor man's stellate cell. *Gastroenterology* **147**, 41-47, doi:10.1053/j.gastro.2014.05.001 (2014).
- 23 Pinzani, M. & Luong, T. V. Pathogenesis of biliary fibrosis. *Biochim Biophys Acta Mol Basis Dis* **1864**, 1279-1283, doi:10.1016/j.bbadis.2017.07.026 (2018).
- 24 Kolodziejczyk, A. A. *et al.* Acute liver failure is regulated by MYC- and microbiome-dependent programs. *Nat Med* **26**, 1899-1911, doi:10.1038/s41591-020-1102-2 (2020).
- 25 Krenkel, O., Hundertmark, J., Ritz, T. P., Weiskirchen, R. & Tacke, F. Single Cell RNA Sequencing Identifies Subsets of Hepatic Stellate Cells and Myofibroblasts in Liver Fibrosis. *Cells* **8**, doi:10.3390/cells8050503 (2019).
- 26 Xu, J. *et al.* A spatiotemporal atlas of mouse liver homeostasis and regeneration. *Nat Genet* **56**, 953-969, doi:10.1038/s41588-024-01709-7 (2024).
- 27 Yang, W. *et al.* Single-Cell Transcriptomic Analysis Reveals a Hepatic Stellate Cell-Activation Roadmap and Myofibroblast Origin During Liver Fibrosis in Mice. *Hepatology* **74**, 2774-2790, doi:10.1002/hep.31987 (2021).
- 28 Terkelsen, M. K. *et al.* Transcriptional Dynamics of Hepatic Sinusoid-Associated Cells After Liver Injury. *Hepatology* **72**, 2119-2133, doi:10.1002/hep.31215 (2020).
- 29 Remmerie, A. *et al.* Osteopontin Expression Identifies a Subset of Recruited Macrophages Distinct from Kupffer Cells in the Fatty Liver. *Immunity* **53**, 641-657 e614, doi:10.1016/j.immuni.2020.08.004 (2020).
- 30 Tsuchiya, Y. *et al.* Fibroblast growth factor 18 stimulates the proliferation of hepatic stellate cells, thereby inducing liver fibrosis. *Nat Commun* **14**, 6304, doi:10.1038/s41467-023-42058-z (2023).
- 31 Gissen, P. & Arias, I. M. Structural and functional hepatocyte polarity and liver disease. *J Hepatol* **63**, 1023-1037, doi:10.1016/j.jhep.2015.06.015 (2015).
- 32 Treyer, A. & Musch, A. Hepatocyte polarity. *Compr Physiol* **3**, 243-287, doi:10.1002/cphy.c120009 (2013).
- 33 Hu, H. *et al.* Long-Term Expansion of Functional Mouse and Human Hepatocytes as 3D Organoids. *Cell* **175**, 1591-1606 e1519, doi:10.1016/j.cell.2018.11.013 (2018).
- 34 Morales-Navarrete, H. *et al.* Liquid-crystal organization of liver tissue. *Elife* **8**, doi:10.7554/eLife.44860 (2019).
- 35 Morales-Navarrete, H. *et al.* A versatile pipeline for the multi-scale digital reconstruction and quantitative analysis of 3D tissue architecture. *Elife* **4**, doi:10.7554/eLife.11214 (2015).
- 36 Fu, D., Wakabayashi, Y., Ido, Y., Lippincott-Schwartz, J. & Arias, I. M. Regulation of bile canalicular network formation and maintenance by AMP-activated protein kinase and LKB1. *J Cell Sci* **123**, 3294-3302, doi:10.1242/jcs.068098 (2010).
- 37 Mayer, C. *et al.* Apical bulkheads accumulate as adaptive response to impaired bile flow in liver disease. *EMBO Rep* **24**, e57181, doi:10.15252/embr.202357181 (2023).
- 38 Bebelman, M. P. *et al.* Hepatocyte apical bulkheads provide a mechanical means to oppose bile pressure. *J Cell Biol* **222**, doi:10.1083/jcb.202208002 (2023).

- 39 Nagore, N., Howe, S., Boxer, L. & Scheuer, P. J. Liver-Cell Rosettes - Structural Differences in Cholestasis and Hepatitis. *Liver* **9**, 43-51 (1989).
- 40 Gupta, K. *et al.* Actomyosin contractility drives bile regurgitation as an early response during obstructive cholestasis. *Journal of Hepatology* **66**, 1231-1240, doi:10.1016/j.jhep.2017.01.026 (2017).
- 41 Belicova, L. *et al.* Anisotropic expansion of hepatocyte lumina enforced by apical bulkheads. *J Cell Biol* **220**, doi:10.1083/jcb.202103003 (2021).
- 42 Delzenne, N. M., Calderon, P. B., Taper, H. S. & Roberfroid, M. B. Comparative hepatotoxicity of cholic acid, deoxycholic acid and lithocholic acid in the rat: in vivo and in vitro studies. *Toxicol Lett* **61**, 291-304, doi:10.1016/0378-4274(92)90156-e (1992).
- 43 Smit, J. J. *et al.* Homozygous disruption of the murine mdr2 P-glycoprotein gene leads to a complete absence of phospholipid from bile and to liver disease. *Cell* **75**, 451-462, doi:10.1016/0092-8674(93)90380-9 (1993).
- 44 Dawson, P. A., Lan, T. & Rao, A. Bile acid transporters. *J Lipid Res* **50**, 2340-2357, doi:10.1194/jlr.R900012-JLR200 (2009).
- 45 Halilbasic, E., Claudel, T. & Trauner, M. Bile acid transporters and regulatory nuclear receptors in the liver and beyond. *J Hepatol* **58**, 155-168, doi:10.1016/j.jhep.2012.08.002 (2013).
- 46 Dikkers, A. & Tietge, U. J. Biliary cholesterol secretion: more than a simple ABC. *World J Gastroenterol* **16**, 5936-5945, doi:10.3748/wjg.v16.i47.5936 (2010).
- 47 Broutier, L. *et al.* Human primary liver cancer-derived organoid cultures for disease modeling and drug screening. *Nat Med* **23**, 1424-1435, doi:10.1038/nm.4438 (2017).
- 48 Wang, L. T., Rajah, A., Brown, C. M. & McCaffrey, L. CD13 orients the apical-basal polarity axis necessary for lumen formation. *Nat Commun* **12**, 4697, doi:10.1038/s41467-021-24993-x (2021).
- 49 Meyer, K. *et al.* Bile canaliculi remodeling activates YAP via the actin cytoskeleton during liver regeneration. *Mol Syst Biol* **16**, e8985, doi:10.15252/msb.20198985 (2020).
- 50 Segovia-Miranda, F. *et al.* Three-dimensional spatially resolved geometrical and functional models of human liver tissue reveal new aspects of NAFLD progression. *Nat Med* **25**, 1885-1893, doi:10.1038/s41591-019-0660-7 (2019).
- 51 Itoh, M., Terada, M. & Sugimoto, H. The zonula occludens protein family regulates the hepatic barrier system in the murine liver. *Biochim Biophys Acta Mol Basis Dis* **1867**, 165994, doi:10.1016/j.bbadis.2020.165994 (2021).
- 52 Ghallab, A. *et al.* Interruption of bile acid uptake by hepatocytes after acetaminophen overdose ameliorates hepatotoxicity. *J Hepatol* **77**, 71-83, doi:10.1016/j.jhep.2022.01.020 (2022).
- 53 Liu, C. *et al.* Expression of aminopeptidase N in bile canaliculi: a predictor of clinical outcome in biliary atresia and a potential tool to implicate the mechanism of biliary atresia. *J Surg Res* **100**, 76-83, doi:10.1006/jsre.2001.6205 (2001).
- 54 Röcken, C., Licht, J., Roessner, A. & Carl-McGrath, S. Canalicular immunostaining of aminopeptidase N (CD13) as a diagnostic marker for hepatocellular carcinoma. *J Clin Pathol* **58**, 1069-1075, doi:10.1136/jcp.2005.026328 (2005).
- 55 Lian, W. N. *et al.* Targeting of aminopeptidase N to bile canaliculi correlates with secretory activities of the developing canalicular domain. *Hepatology* **30**, 748-760, doi:DOI 10.1002/hep.510300302 (1999).

- 56 Crawford, L., Monod, A., Chen, A. X., Mukherjee, S. & Rabadán, R. Predicting Clinical Outcomes in Glioblastoma: An Application of Topological and Functional Data Analysis. *J Am Stat Assoc* **115**, 1139-1150, doi:10.1080/01621459.2019.1671198 (2020).
- 57 Aloia, L. *et al.* Epigenetic remodelling licences adult cholangiocytes for organoid formation and liver regeneration. *Nat Cell Biol* **21**, 1321-1333, doi:10.1038/s41556-019-0402-6 (2019).
- 58 Zhu, L. *et al.* Prominin 1 marks intestinal stem cells that are susceptible to neoplastic transformation. *Nature* **457**, 603-607, doi:10.1038/nature07589 (2009).
- 59 Prior, N. *et al.* Lgr5(+) stem and progenitor cells reside at the apex of a heterogeneous embryonic hepatoblast pool. *Development* **146**, doi:10.1242/dev.174557 (2019).
- 60 Reif, R. *et al.* Bile canalicular dynamics in hepatocyte sandwich cultures. *Arch Toxicol* **89**, 1861-1870, doi:10.1007/s00204-015-1575-9 (2015).
- 61 Arterburn, L. M., Zurlo, J., Yager, J. D., Overton, R. M. & Heifetz, A. H. A morphological study of differentiated hepatocytes in vitro. *Hepatology* **22**, 175-187 (1995).
- 62 Korelova, K., Jirouskova, M., Sarnova, L. & Gregor, M. Isolation and 3D Collagen Sandwich Culture of Primary Mouse Hepatocytes to Study the Role of Cytoskeleton in Bile Canalicular Formation In Vitro. *J Vis Exp*, doi:10.3791/60507 (2019).
- 63 Modes, C. D., Magnasco, M. O. & Katifori, E. Extracting Hidden Hierarchies in 3D Distribution Networks. *Phys Rev X* **6**, doi:ARTN 031009 10.1103/PhysRevX.6.031009 (2016).
- 64 Karschau, J. *et al.* Resilience of three-dimensional sinusoidal networks in liver tissue. *PLoS Comput Biol* **16**, e1007965, doi:10.1371/journal.pcbi.1007965 (2020).
- 65 LeCluyse, E. L., Audus, K. L. & Hochman, J. H. Formation of extensive canalicular networks by rat hepatocytes cultured in collagen-sandwich configuration. *Am J Physiol* **266**, C1764-1774, doi:10.1152/ajpcell.1994.266.6.C1764 (1994).
- 66 Zeigerer, A. *et al.* Functional properties of hepatocytes in vitro are correlated with cell polarity maintenance. *Exp Cell Res* **350**, 242-252, doi:10.1016/j.yexcr.2016.11.027 (2017).
- 67 Turncliff, R. Z., Tian, X. & Brouwer, K. L. Effect of culture conditions on the expression and function of Bsep, Mrp2, and Mdr1a/b in sandwich-cultured rat hepatocytes. *Biochem Pharmacol* **71**, 1520-1529, doi:10.1016/j.bcp.2006.02.004 (2006).
- 68 Tanimizu, N. *et al.* Generation of functional liver organoids on combining hepatocytes and cholangiocytes with hepatobiliary connections ex vivo. *Nat Commun* **12**, 3390, doi:10.1038/s41467-021-23575-1 (2021).
- 69 Vartak, N. *et al.* Intravital Dynamic and Correlative Imaging of Mouse Livers Reveals Diffusion-Dominated Canalicular and Flow-Augmented Ductular Bile Flux. *Hepatology* **73**, 1531-1550, doi:10.1002/hep.31422 (2021).
- 70 Laughlin, R. B., Pines, D., Schmalian, J., Stojkovic, B. P. & Wolynes, P. The middle way. *Proc Natl Acad Sci U S A* **97**, 32-37, doi:10.1073/pnas.97.1.32 (2000).
- 71 Sear, R. P., Pagonabarraga, I. & Flaus, A. Life at the mesoscale: the self-organised cytoplasm and nucleoplasm. *BMC Biophys* **8**, 4, doi:10.1186/s13628-015-0018-6 (2015).

We would like to thank the Editor and each of the Reviewers for their time and effort providing insightful feedback, which has greatly helped strengthen our manuscript. We have carefully addressed their comments raised during this second round of review and are confident that the manuscript has improved as a result.

The second round of comments are presented in regular font, followed by our responses in blue and *italics*, with new data and edits to the main text in blue, *italics and highlighted in grey*. In the manuscript, the new edits are presented in highlighted in grey and with track changes. Also in the manuscript, the sections highlighted in grey without track changes correspond to answers to the reviewer already present in the earlier version of the manuscript.

Before the point-by-point response to the Reviewers, we would like to summarize the key changes to the manuscript following this second revision:

1. We further clarified the advance of our model over previously published 2D and 3D models in the text, namely:

- the presence of physiological bile canaliculi network in 3D (*Lines 78-81; 197-199 and 406-414*)
- the inclusion of 3 cell types (cholangiocytes, hepatocytes and portal mesenchyme (*Lines 74-81*))
- the recapitulation of many aspects of the complex hepatic architecture *in vitro* (*Lines 399-407*)
- a platform to discover cell autonomous contributions to biliary fibrosis and cholestatic liver disease, previously inaccessible in existing epithelial-only organoid models or full-knockout models, thanks to the ability to manipulate individual cells in the system (*Lines 333-336; 385-387 and 420-431 and 447-454*).

2. We have clearly specified limitations of this study in the text:

- We show only proof-of-principle experiments with genetic MDR2 and ITGB1 knockout, or siRNA-silencing of Cdh11 (*Line 425-432*), and highlight that more studies are needed to decipher all the pathways contributing to cholestasis and biliary fibrosis (*Lines 432-434*).
- We clearly state that the system is still missing the endothelial and immune compartment, as well as other liver mesenchymal subsets, all of which would be needed to fully model the liver in a dish (*Lines 264, 331, 341, 417-419*).
- We state that while many hepatocytes die of apoptosis in the system, we cannot exclude other types of cell death (*Lines 435-440*).
- We have discussed the zonation limitations of our system, which is too small to achieve full liver lobule zonation (*Lines, 415-417*).
- We discuss the limitation of cholangiocyte tracing in our system. (*Lines 147-149*)

3. We provide additionally the following data:

- Single bar graph showing the quantification of the second harmonic generation (SHG) images from main Figure 6e (see Figure R6 from this letter, Extended Data Fig. 12j) (requested by Reviewer_3).
- Spreadsheets of differentially expressed genes with the corresponding *p-values* for all the bulk RNAseq present in the paper (Supplementary Dataset 6) (requested by Reviewer_4).

Taken together, our study presents a novel organoid model with physiological bile canaliculi and three cell types recapitulating complex liver architecture of the periportal region, which allows unique opportunity to investigate cell-autonomous processes in the development of disease phenotypes.

POINT-BY-POINT RESPONSE to the REVIEWER'S COMMENTS:

Referee #1 (Remarks to the Author):

In this revised manuscript, Dowbaj and colleagues, present extensive new data from new lines of experiments to address the reviewers' previous criticisms and concerns. While novel mechanistic biologic insight remains limited, this manuscript presents a true experimental tour-de-force and will serve as a landmark resource and greatly advance the field. I have no further major criticisms or concerns.

*We thank the Reviewer for reading, **appreciating the advance and considering our manuscript a landmark resource for the field.***

Referee #2 (Remarks to the Author):

The authors have added a large amount of data and provided a carefully written point-by-point response. They satisfactorily addressed a number of the points raised in the previous review, including bile acid metabolism, improved comparisons to scRNA-seq datasets and integration of some functional studies. However, several of the major weaknesses have not been convincingly resolved and large parts of the study remain descriptive and mechanisms are not investigated in sufficient depth. The study will not convince the field that the assembloid model indeed represents a major advance that will change how liver disease mechanisms can be studied and how new therapeutic approaches can be tested.

We thank the Reviewer for acknowledging that we have addressed a number of points from the previous round.

*We are extremely surprised by the change of mind and tone of this Reviewer regarding the advance of our model. Basically, **the statements regarding the advance are contradictory between the first and second round of revision.** In the **first** round of revision, the Reviewer said:*

*“This elegant study comes from leaders in liver organoid modelling and **covers an important gap** as the majority of current studies are either done in animal systems, 2D culture or short-term liver slice systems - all with grave disadvantages. **The authors show an improved model** in which optimization and different ratios of mesenchymal cells result in larger organoid with functional bile canaliculi and drainage as well as an induction of fibrogenesis”.*

*Now, in this **second** round the Reviewer says:*

*“**The study will not convince the field** that the assembloid model indeed represents a major advance”.*

Given that not a single paper has been published showing a liver assembloid model, we cannot understand the change of mind of the Reviewer regarding the advance of our model and we can only think that the Reviewer has misinterpreted our goal with the new experiments regarding the fibrotic mechanism.

*It was **not our intention to describe novel biological insights underlying a new fibrotic mechanism**, which would require in vivo validation and many years of work. Instead, we hoped to show that our system can be used as a platform to study fibrosis. In the first round the Reviewer stated that the manuscript is just “descriptive” and not showing “functional assays”. In the revised version we provided a proof-of-concept that tests can be done with siRNA and blocking antibodies, as well as knockout cells, that all demonstrate the applicability of our new assembloid structure to investigate cellular and molecular mechanisms involved in fibrosis, in addition to the ability to manipulate cells in our system in a cell-autonomous manner.*

We have re-written the manuscript to further clarify this point in this new revised version of the manuscript (copied here for the Reviewer’s assessment):

Lines 34-36: “By generating chimeric assembloids between mutant and wild-type cells, or after gene knockdown, we show proof-of-concept that our system is amenable to investigating both gene function and cell-autonomous mechanisms in liver disease.”

Lines 83-84: “Additionally, we provide proof-of-concept that liver assembloids can be used to investigate molecular as well as cell-autonomous mechanisms in liver disease.

Lines 343-344: “Hence, we next tested the proof-of-concept whether assembloids could be exploited to investigate cell-cell and cell-ECM interactions in fibrosis”

Lines 423-424: “provide proof-of-concept that assembloid cells can be used as platform for mechanistic discovery and to investigate cell-autonomous mechanisms in cholestatic liver fibrosis.”

Line 447-457: “Our system represents a tractable *in vitro* tool where to investigate the dynamics of cholestatic biliary fibrosis and elucidate the exact contribution of each of the different niche cells to the initiation and/or progression of the disease. Noteworthy, this is not possible in current epithelial-only liver organoids models, neither in iPSC derived organoid models where the different populations co-develop from a single iPSC clone. In addition, the possibility to generate chimeric assembloids from different cell types also enables underscoring cell-autonomous phenotypes from full mutant mice, without the need of generating complex cell-specific conditional alleles. We envision that in the future, once translated to human cells, periportal assembloids will eventually represent a way to transition from the 2D-models used in pharmaceutical screenings to more physiological 3D- models, to study drug efficacy and toxicity in a more physiologically relevant context.”

1. The authors have integrated some functional studies but many of ligand-receptor pairs are only superficially investigated. The majority of the proposed central interactions have either not been validated or are not studied in sufficient detail and scientific novelty is limited.

Once more, our intention was not to uncover a novel molecular mechanism but the proof of concept that functional studies can be performed in our model. A full well-done mechanistic study is a project per se and would be out of the scope of this manuscript. Would require years and a detailed analysis, backed by in vivo data. Instead, here we provide a model fit to investigate molecular mechanisms in vitro, and demonstrate that this model can be used with that purpose, targeting known players that have been previously tested also in vivo, such as Itgb1 or Cdh11 – both cited by us in previous manuscript version by Ref. 92-95.

We did perform a small screen (14 ligand or receptor molecules) in assembloids, which we reported in the revised version of the manuscript. Additionally, we tested several other molecules using siRNA silencing, but unfortunately, as the actual knockdown was not very efficient, we excluded them from the final revised version (namely, Itga11, Mmp2, Mmp3, Mia3). Nevertheless, this small screen was sufficient to observe a rescue of fibrotic phenotype with 2 candidates, Cdh11 (forming homotypic Msc-Msc interactions) and Itgb1 (forming interaction with ECM). Both implicated in vivo (as cited in Ref. 91 - 93 of the manuscript), which confirms that our model is valid for these types of experiments, as outlined in line 512 of the previous version of the manuscript. We have now discussed this point in:

Lines 424-427: “In our small-scale proof-of-principle screen, we identified CDH11 and ITGB1 as specific mesenchymal contributors to fibrotic-like phenotype. Both of them are known to be upregulated in fibrosis⁹¹⁻⁹³, which validates our proof-of-concept knockdown/knockout experiments.

We would like to stress that this is the first time this type of complex in vitro model has been developed, and leveraging it to find novel players in fibrosis in this manuscript seems, to us, out of scope.

*To investigate ligand-receptor pairs we also performed experiments with blocking antibodies targeting the assembloid as a whole, at the time of assembly, and throughout the growth of the structure. **This point also answers the question below in point 1b** from this Reviewer, which the Reviewer has missed. Additionally, for the 2 positive hits *Cdh11* and *Itgb1*, we show a cell-specific silencing or knockout selective for mesenchymal cells. Therefore, **in contrast to the statement of the Reviewer, we were able to find interesting** interactions in a cost- and time-effective manner compared with in vivo experiments, and without the need for animal models, which represents a substantial advance of the organoid models in general.*

1a. The authors did not validate their findings on *Cdh11* and *Itgb1* in vivo - thus it is not clear how useful the assembloid models is to determine key drivers of liver disease. The author should similar roles for *Cdh11* and *Itgb1* in vivo, including decreased fibrosis and other effects..

*As extensively discussed above, we did **not aim to claim that *Cdh11* or *Itgb1* are novel players in fibrosis, but that its perturbation, in the form of knockdown or knockout, serves as a proof-of-concept** experiment to showcase how the system can be used to study cell-autonomous mechanisms in liver fibrosis. This cannot currently be achieved with any other in vitro 3D system.*

*As outlined by this same Reviewer in point 1d, and us in the manuscript Ref. 91-93 (PMID:35774522 and PMID: 30509494), *Cdh11* and *Itgb1* had already been implicated in liver fibrosis. By having acknowledged that in the manuscript, we showed we did not intend to claim novelty but instead, we were using this to show a proof-of-principle experiment (as outlined in line 512 of the previous manuscript version) or support the concept that **silencing candidate genes only in portal fibroblasts (prior to mixing them with epithelial cells) is feasible and allows the investigation of cell-autonomous processes in liver**. Overall, we feel that stating that *Cdh11* is not novel in liver fibrosis as a weakness of our manuscript is a misrepresentation of what we tried to achieve with this experiment, as **we did not intend to reveal new biological insights** regarding *Cdh11*.*

*Now the Reviewer asks for an in vivo knockdown of *Cdh11*, but the general role for *Cdh11* in fibrosis has been proven by others (Ref. 92 and 93 of our manuscript, PMID:35774522 and PMID: 30509494), and we struggle to see a point in repeating this experiment, **as the novelty lies in knocking down genes in a complex vitro system, we do not aim to uncover here new players in fibrosis, which would require a substantial follow-up validation in vivo that would take years.***

Moreover, validating the specific role of Msc-Msc Cdh11 homotypic interactions or Msc-ECM interactions **in vivo cannot be technically done**, since all cells are at all times in contact with other cells or the ECM, and discriminating the impact of heterotypic cell-contact vs homotypic cell contacts is, at present, impossible. **This experiment can currently be designed only in an ex-vivo system like ours, where inhibition can be detangled from the interaction with other cells by inhibiting gene function before assembly into a complex organoid.**

We have now re-written the manuscript to emphasise that point in the text with more clarity, which is presented below for the Reviewer's assessment and can be found in the revised manuscript lines:

Lines 420-424: "Paradoxically, while recapitulating native tissue architecture is essential, too much complexity could represent an obstacle when we aim to investigate the precise role of specific cell types to a particular aspect of the (patho-)physiology of a tissue, **without other confounding factors.** We provide proof-of-concept that assembloid cells can be used as platform for mechanistic discovery and to investigate cell-autonomous mechanisms in cholestatic liver fibrosis."

1b. The heading states "Periportal assembloids as a tool to investigate cellular and molecular mechanisms of cholestatic biliary fibrosis" but the studies on Itgb1 and Cdh11 do not really define cellular and molecular mechanism. Importantly, the authors have not proven that Cdh11 mediates a homotypic MSC-MSC interaction as (i) other cells are present in the assembloids and (ii) Cdh11 preferentially undergoes heterophilic interactions with Cdh8 (which seems to be also highly expressed in hepatocytes - the authors knocked out Cdh2 but also not in different cell types; knocking out Cdh8, Cdh11 and possibly other interaction partners in various cell types would be important).

We respectfully disagree with the Reviewer here: as described in Brasch et al., Cell Reports 23, 1840–1852, 2018 (PMID: 29742438), Cdh11 interacts with Cdh11, Cdh24, and Cdh8 with high affinity. Any other interaction is of low affinity. Nevertheless, the Reviewer has a point in considering that Cdh8 could be another partner.

However, **Cdh8 is not expressed in the liver at all** according to human protein atlas and the mouse liver cell atlas, and we confirmed it in our own data (see **Figure R1 for the Reviewer only**, below). Similarly, Cdh24 in liver is only expressed in endothelial cells. Therefore, from these 3 Cadherins (Cdh11, Cdh24 and Cdh8), **only Cdh11 is expressed in our system and only in portal fibroblasts** as we showed in Suppl. Fig 15b in the manuscript.

Therefore, in our system, Cdh11 can only make homophilic interactions (i.e., Cdh11-Cdh11) in mesenchyme. This, together with the fact that we inhibit it exclusively in mesenchymal cells prior to assembly, proves that we are disrupting Msc-Msc interactions exclusively.

Given that inhibiting Msc-Msc interactions through Cdh11 prevents the fibrotic phenotype of assembloids, we believe that our manuscript shows that assembloids can be used as a tool to study cellular and molecular mechanisms.

We have edited the manuscript to clarify that point for the reader, in

Lines 428-431: Given that CDH11 is only expressed in mesenchyme in our system and none of the other known CDH11 interactors for CDH11 are present either, these results suggest that Msc-adhesion could be the conveyers of the fibrotic phenotype observed

Figure R1

Figure R1 for Reviewer only: *Cdh8* expression in mouse and human liver tissue.

a. Expression of *Cdh8* in liver cell atlas (Guilliams *et al.*, 2018, Cell); **b.** Expression of *Cdh8* in human protein atlas (brain and CNS only). **c.** Expression of *Cdh8* in periportal assembloid model (not expressed).

Likewise, the investigation of *Itgb1* is also superficial. What is the responsible ligand? What is the mechanism? Can the authors demonstrate that these are homotypic MSC-MSC interactions?

To our knowledge, the Itgb1 ligands are all ECM molecules. Indeed, in our scRNAseq dataset, we have found several ligands for Itgb1, however, unfortunately, they all constitute ECM components, which would make it very challenging to investigate, because to grow/be cultured organoids require a matrix consisting of many components including collagen, laminin, entactin and others. At present, we are unaware of a method to deplete the ECM and still be able to grow organoids. Potentially, generating rationally designed matrices with physiological compositions would in principle help to dissect the roles of specific ECM molecules. This approach is certainly enticing, and the organoid field would benefit by such systems; unfortunately, to the best of our knowledge, there are no systems fit for this purpose.

1d. Many other ligand-receptor pairs such as Col1a1-Ddr2, Gas6-Axl, Lgals-Itgb1 or Timp1-Cd63 were not studied (Cdh11 has been implicated in fibrogenesis in many organs including the liver, hence the novelty is limited).

*We are very confused by this comment. The Reviewer lists Timp1 and Itgb1 as not studied but these were presented in previous Suppl. Figure 15e and Suppl. Figure 16 (now Extended Data Fig. 13). As above, the Reviewer says it is known that Cdh11 contributes to liver fibrosis, and we find the same interaction to be important in fibrotic-like assembloids, which validates **our proof-of-concept experiment**. Once more, we were not claiming to gain novel biological insights related to Itgb1 or Cdh11 in fibrosis, but instead used it only as a proof of concept. We have clarified that in the text, as described above.*

1e. Interactions between other cell types were not studied. In addition to absent effects of chemokine likely due to the lacking integration of immune cells (see comment below), interactions of mesenchyme with hepatocytes (e.g. leading to cell death) or cholangiocytes and interactions between hepatocytes and cholangiocytes were not studied.

*As pointed below, we understand that the Reviewer is not satisfied with our analysis of the cytokines (previous Suppl. Figure 15c-d, now Extended Data Fig. 13) and considered that these “may be due to the absence of leukocytes in the assembloid model as many of receptors are largely on immune cells”. However, **we analysed these cytokines as per this same Reviewer’s request in point 6 of the first revision. Immune cells were not present in the first submission either.** We believe that asking for an experiment in the first round and then highlighting the inherent limitations of the requested experiments in the second round seems an unfair criticism. Please also see the comment below to point 3.*

In general, reliably including immune cells in organoids has been a challenge in the field for many years now, that has been only recently tackled by only a few laboratories in the world, albeit not in liver assembloids nor liver organoids (such as <https://www.nature.com/articles/s41586-024-07791-5>, PMID: 39143209). We acknowledge that adding immune cells would benefit our system, and we hope that in the future this will be implemented, but we would like to highlight that this is true for virtually all other organoid models being published, and the lack of these cells does not invalidate the usefulness of this system.

2. The authors have not followed up on the suggestion to use their system for screens (a main advantage of the described model over in vivo models). Without the integration of such experiments, the impact that the paper and model will have on the community is only going to be moderate. As discussed before, this could have been a drug screen, RNAi or CRISPR screen, with subsequent validation of the data an in vivo model of biliary fibrosis.

Unfortunately here we are once again confused. In the first submission, this comment came across as a nice addition, that could be discussed. Citing the Reviewer: “would be helpful”, not that it is essential.

*During the first revision we had actually screened 14 receptor-ligand pairs, and showed that screening in our system is possible, in principle. Setting up a genome-wide siRNA or CRISPR screen would require a time frame that is simply not compatible with a major revision, especially considering that it would require further validation in vivo, or in human cells, which would constitute a completely new project per se. Therefore, we feel it **is not fair to point to the lack of a fully-fledged high-throughput screen as a “weakness” in this second round of revision, if this was not made as a clear request in the first place.** Currently, we do not think that it is in the scope of this manuscript to set up a whole-genome HTS, but it could be a nice suggestion for a future project, even though this would require at least a couple of years for proper optimization and in vivo validation.*

3. The lack of immune cells remains a weakness. While hepatic stellate cells and endothelial cells are also missing (their close interactions suggest that they should be integrated at the same time), the focus on periportal fibrosis justifies the sole integration of portal fibroblasts. However, some of the questions/experiments of the current study are directly affected by the absence of Kupffer cells/macrophages or other immune populations - e.g. the absent effects of cytokine treatment or blocking antibodies against these cytokines (CXCL12, CCL11, CXCL1, CX3CL1) may be due to the absence of leukocytes in the assembloid model as many of receptors are largely on immune cells. Furthermore, Kupffer cells are an important part of zone 1 as they are also zoned and their protection from gut-derived bacteria is largely in this zone (Gola et al, Nature 2021).

Regarding the lack of immune cells, we would like to highlight again that implementing immune cells in assembloid cultures is still challenging for the whole field, and this has been achieved only recently and by a few labs and never, to our knowledge, with liver organoids. Additionally, we would like to reiterate that growing and maintaining immune cells in media compatible with assembloids is not trivial, and would require a completely separate project with a timeframe and resources that we believe are not compatible with a fair peer review process.

Several of the chemokines studied in the manuscript are produced by the Msc or epithelia (hepatocytes, cholangiocytes) themselves, as proven by the cytokine array (Extended Data Figure 12c and Supplementary Information File-Gels, n=2 of the revised version of the manuscript). We therefore find it an interesting feature of this system, which allows delineation of which cytokines are produced by which cells to recruit the immune response. We explain that actually not having macrophages in the system allows finding these interesting parameters. The cytokines expressed by epithelia are not artifacts as in Andrews et al. Journal of Hepatology 2024 (PMID: 38199298), the authors also shows that in human PSC (biliary fibrosis) the epithelia can also express some cytokines in the fibrotic response.

*We want to emphasise here that **we analysed these cytokines as per this same Reviewer's request in point 6 of the first revision.** Specifically asking for an experiment in the first round and then highlighting the inherent limitations of the requested experiments in the second round seems an unfair criticism.*

Here, we understand that the Reviewer is not satisfied with our analysis of the cytokines, and considered that these “may be due to the absence of leukocytes in the assembloid model as

many of receptors are largely on immune cells”. We have now once more acknowledged this limitation in the manuscript in:

Line 264: “in the absence an immune compartment”

Line 331: “as expected given that our system lacks the immune compartment^{2,67}.”

Line 341: “and in the absence of the immune compartment.”

Lines 417-429: “Future studies are needed to incorporate more cell types and recapitulate the whole spectrum of the liver organisation. “

4. The mechanisms how fibroblast drive hepatocyte death and whether this is an artifact of the system remain unclear. The authors added data using caspase inhibitors but the mode of hepatocyte death and the impact of cell death on fibrosis and other cell types remain vague (cell death is a main driver of liver disease progression).

In our manuscript we demonstrated that an increase in the numbers of mesenchymal cells results in a fibrotic-like phenotype in vitro. We further investigated that preventing Msc-Msc interactions prior to assembly prevents this phenotype, despite the numbers of Msc cells remaining constant. Amongst the different fibrotic phenotypes, we had observed collagen deposition, bile flow obstruction and also hepatocyte death. The previous comments from this Reviewer regarding hepatocyte death (point 7) were: “Evidence for cleaved caspase 3 and hepatocyte is weak (there is no quantification comparing homeostatic- and fibrotic-like assembloids) and there is no functional investigation (e.g. using caspase inhibitors) for confirmation and for determination of the role of apoptotic hepatocyte death in fibrogenesis in this model system. And, How do Msc induce hepatocyte death?”

We had provided the quantification in the presence of caspase inhibitors, as requested, and we had also acknowledged that other forms of death were detected (lines 370-372 in the previous version). Now, here the Reviewer asks 4 further points:

- 1) the mechanisms of how fibroblasts drive hepatocyte death*
- 2) the in vivo relevance (“artifact”)*
- 3) the mode of hepatocyte death in assembloids*
- 4) the impact of hepatocyte death on other cells.*

For point 1) “mechanism of how fibroblasts drive death”:
*unfortunately, **this is a full project per se** that would expand over years and beyond the scope of this manuscript. We had discussed that in the manuscript (previous version lines 536-539, now lines 443-446) where we can already envision that these mechanisms would go from biophysical/mechanical/ biochemical or all combined.*

[Redacted text]

Figure R2 for Reviewer only below.

[Redacted figure]

Figure R2 for Reviewer only: Cell death in mouse tissue in the model of biliary fibrosis.

a. Immunofluorescence staining of *Mdr2*^{+/+} (top) or *Mdr2*^{-/-} (bottom) periportal region of mouse liver tissue, stained for apoptotic marker Cleaved caspase 3 (magenta). Nuclei were stained with DAPI (cyan), cell borders with Phalloidin (grey) and ductal cells with Osteopontin, OPN (yellow). Scale bar, 50 μ m, insert 10 μ m.

b. Immunofluorescence staining of *Mdr2*^{+/+} (top) or *Mdr2*^{-/-} (bottom) periportal region of mouse liver tissue, stained for necrotic marker HMGB1 (magenta). Note that the presence of HMGB1 signal inside of cytoplasm is indicator of necrotic cell. Nuclei were stained with DAPI (cyan), cell borders with Phalloidin (grey) and ductal cells with Osteopontin, OPN (yellow). Scale bar, 50 μ m, insert 10 μ m.

For point 3) “mode of death of hepatocytes in assembloids”

Now, we investigated the mode of cell death using live imaging and staining for apoptosis (Annexin V), and immunofluorescence staining for necrosis (HMGB1), and necroptosis (pMLKL) markers, in addition to previously presented immunofluorescence data for cleaved caspase 3 (revised manuscript Fig.5g). As shown in **Figure R3 for Reviewer only**, some fibrotic-like

assembloids—but not homeostasis-like assembloids—exhibited Annexin V positivity, consistent with our previous caspase-3 staining, but also indicative that fibrotic-like assembloid hepatocytes can be dying by other cell death modes. Furthermore, staining for pMLKL (indicative of necroptosis) and HMGB1 (a marker of general damage and potential necrosis) suggested that hepatocytes in fibrotic-like assembloids can also undergo those modes of cell death. We have acknowledged this limitation in:

Lines 306-308 (previous manuscript version lines 370-372): “Cleaved caspase 3 staining indicated that hepatocyte death was, at least partially, through apoptosis, although gene expression analysis also indicates that other forms of cell death might be involved (Fig.5g and Suppl. Fig. 14b).”

Lines 327-331: “Taken together, all the described features, namely (i) fibrotic gene expression similar to *in vivo* models of biliary fibrosis, (ii) hepatocyte death *at least partially driven by apoptosis*, (iii) bile flow obstruction, (iv) duct cell expansion and (v) collagen deposition indicate that fibrotic assembloids recapitulate *in vitro* many aspects of the *in vivo* biliary/cholestasis fibrosis, except for the inflammatory reaction, as expected given that our system lacks the immune compartment^{2,67}.”

Lines 435-437: *Interestingly, in fibrotic-like conditions we observe hepatocyte cell death, at least partially mediated by apoptosis, although our gene expression data suggest that other types of cell death might also be involved.*

Figure R3
Figure R3 for Reviewer only: Cell death mode analysis in fibrotic-like periportal assembloids.

a. Live imaging of fibrotic-like and homeostatic assembloids stained with Annexin-V-AlexaFluor647-conjugate in $n=2$ biological replicates; two examples are shown; white arrowhead – fibrotic-like assembloid hepatocytes dying of apoptosis, orange arrowhead – fibrotic-like assembloid hepatocytes negative for annexin V signal, indicative of another cell death mode; scale bar, 100 μm . **b.c.** Immunofluorescence staining for HMGB1 (**b**) and pMLKL (**c**) in fibrotic-like and homeostatic assembloids shown in orange as well as in Fire LUT for clarity; nuclei (DAPI, blue), Msc (Pdgfra-H2BGFP, green), hepatocytes (mem-tdTomato, magenta), cholangiocytes (mem-mCFP, cyan) and actin (Phalloidin, white) are also shown; scale bar, 50 μm .

For point 4) “the impact of hepatocyte death on other cells”.

To address this point, we would need to selectively induce cell death in the hepatocyte population within healthy assembloids and assess its impact on other cell types, namely cholangiocytes and mesenchyme. This experiment requires a method to specifically target hepatocytes—whether through chemical injury, genetic ablation, or laser ablation—without affecting the surrounding cells.

Given the time constraints of this revision, we initially attempted chemical inhibition and genetic ablation but found that optimizing drug concentration and treatment conditions was more complex than anticipated. Ensuring selective hepatocyte damage without unintended toxicity to other cells requires extensive optimization, which is not feasible within the current review timeline.

While this experiment is technically achievable with our model, the necessary refinements would delay the revision process. We recognize this as an excellent direction for future studies and have addressed this point in the discussion.

Lines 438-440: In the future it would be of great interest to also investigate the effect of hepatocyte cell death on the other cells, which could be performed by genetic, chemical or laser ablation of hepatocytes in our model.

Referee #3 (Remarks to the Author):

I would like to congratulate the authors on an outstanding and comprehensive body of work which will advance the field. In particular, the large amount of new data added in revision has significantly enhanced the manuscript. However, I would like the authors to consider my further specific comments which are listed below:

*We thank the Reviewer for the positive comments on the revised version of the manuscript and **acknowledging that our work will advance the field.***

1) Original reviewer comment:

‘Regarding point 11: Line 129: It is not clear whether the cholangiocyte-like structures are due to differentiation of hepatocytes or from existing cholangiocytes/BDs in the media aggregating with hepatocytes. This should be explored in more detail.’

Author rebuttal states: ‘However, the reviewer has a valid point indicating that this could also be also due to cholangiocytes transdifferentiating from hepatocytes. While we believe studying transdifferentiation would be a subject of a full new project in itself, we have attempted to address the question of the reviewer by generating hepatocyte organoid cultures from an induced Prom1- CreRT2xfl-fl-ZsGreen mouse line’.

- I do think the question of hepatocyte / cholangiocyte transdifferentiation in this system is an important one, particularly given the recent publication in Nature demonstrating epithelial plasticity in human chronic liver disease (Acquisition of epithelial plasticity in human chronic liver disease, Nature 2024 Jun;630(8015):166-173), and that the authors are promoting this as a new model system that will help increase our understanding of human liver disease pathogenesis, and also represent a very useful system for drug testing in models of human liver disease.

To this end, in the revised manuscript, the authors have used a Prom1-CreRT2xfl-fl-ZsGreen reporter mouse line to investigate whether cholangiocytes give rise to cholangiocyte organoids in their system – they conclude that ‘Cholangiocyte organoids in HM-Wnt media arise from cholangiocytes in the hepatocyte isolation prep’. However, I don’t think the authors can conclude this for the following reasons:

a) Prom1 is not specific for cholangiocytes, as it also labels mouse and human hepatocytes (eg Hepatocyte-specific Prominin-1 protects against liver injury-induced fibrosis by stabilizing SMAD7. PMID: 36038590). Therefore, Prom1 CreERT2 will trace both cholangiocytes and hepatocytes, not just cholangiocytes.

*Unfortunately, we respectfully disagree with the Reviewer here. The paper cited by the Reviewer relates to a damaged liver scenario. Instead, we isolate the hepatocytes from adult healthy liver. In Zhu et al., 2016, using lineage tracing experiments with the same mouse line that we have, the group of Richard Gilbertson showed that **only in the embryonic liver Prom1 can trace both hepatocytes and duct cells, while in the adult healthy liver (our case), Prom1 tracing is restricted to the ductal compartment** (see Figure 1F from Zhu et al., Cell 2016 paper, reproduced here below in the **Figure R4 for Reviewer only**). This result was confirmed by*

ourselves in Aloia *et al.*, *Nature Cell Biology* 2019 and in Extended Data Figure 9b. Additionally, we also showed that to the reviewer in Rebuttal Figure 24b from our own previous work (Aloia *et al.*, 2019, *Nature Cell Biology*). Therefore, our experiment shows that, if green cysts grow, they are derived from pre-existing cholangiocytes, not hepatocytes.

Figure R4

a adapted from Aloia *et al.*, 2019

b adapted from Zhu *et al.*, 2016

Figure R4 for Reviewer only: *Prom1* specifically marks cholangiocytes in adult mouse livers.

a. Reproduced from Extended Data 9b, Aloia *et al.*, (2019); immunofluorescence staining of a adult mouse liver section showing a clear co-localisation of *Prom1*-Cre-recombined ZsGreen (green) with a cholangiocyte-specific osteopontin staining (OPN, red); DAPI (blue); scale bar, 50 μ m. **b.** Reproduced from Fig.1F, Zhu *et al.*, (2016); *Prom1*+ Cell Properties in Major Organs of *Prom1*C-L; *Rosa*ZsG Mice, showing the results for neonatal (left panel) and adult (right panel) recombination; scale bars, 50 μ m.

b) Furthermore, the cholangiocyte recombination efficiency is only 45%, which still does not help answer this question as some of the Zs Green negative cholangiocyte-like structures may have either derived from unlabelled (non-recombined) cholangiocytes or hepatocytes. to answer this question, the authors could employ a fairly straightforward approach using IV injection of AAV8-TBG-Cre in Zs green reporter mice. This would label ~99-100% of hepatocytes green at the start of the experiment, and then these green hepatocytes could be traced throughout the organoid timecourse. This would give a clear readout of whether the cholangiocyte-like structures are due to differentiation of hepatocytes or from existing cholangiocytes/BDs in the media aggregating with hepatocytes.

The Reviewer suggested experiment is very elegant and, following from the suggestion, we

aimed to perform this experiment. However, our current mouse licence does not allow us to perform the stated experiment, and it would require a lengthy process to obtain the necessary permits, according to the German law regulations.

Therefore, to give some partial answer to this question, we have additionally performed analysis of organoid structure numbers, where we observed 70% of cholangiocyte organoids arising in hepatocyte organoid culture to be ZsGreen-positive. We show these results below in **Figure R5 for Reviewer only**. Given these results, while still possible that some cholangiocytes derive from trans-differentiation, this possibility is small given that the number of unlabelled cholangiocyte organoids that grew is 30%, below the 50% recombination efficiency, suggesting that the non-labelled cholangiocyte organoids derive from non-recombined cholangiocytes.

Figure R5

Quantification of organoid numbers
from live microscopy analysis

Figure R5 for Reviewer only: Cholangiocyte organoids in HM-Wnt media arise from cholangiocytes in the hepatocyte isolation prep. Percentage of Zs-Green-positive cholangiocyte organoids in hepatocyte culture from Prom1-CreRT2xfl-fl-ZsGreen animals recombined with tamoxifen reveals 70% of all organoids to be marked.

Therefore, given this difficulty and that transdifferentiation of hepatocytes to cholangiocytes is not a claim in our manuscript, we have clarified this limitation in:

Lines 146-149: “we observed the emergence of hepatocyte organoids containing cholangiocyte-like organoids in close proximity, which most likely arose from contaminating cholangiocytes in the hepatocyte isolation, although we cannot rule out an emergence of cholangiocytes through hepatocytes trans-differentiation.”

2) Lines 435-438: ‘Additionally, CDH11 knockdown also resulted in a marked reduction in collagen deposition and a reduction in hepatocyte cell death in assembloids cultured with 10-fold excess mesenchyme (Fig.6e and Suppl. Fig.15 j-k), which now resembled homeostatic control assembloids’.

- I could not see any quantitation / graphs presented here regarding SHG assessment of collagen and hepatocyte death – quantitation should be presented.

We provide quantification of SHG signal assessment in Cdh11-siRNA in Suppl. Fig.12j, as well as below in **Figure R6** for the Reviewer’s assessment. In the light of the new cell death analysis data, we have not provided cleaved caspase 3 staining and removed this data for Cdh11-siRNA

from the manuscript, as we cannot rule out other modes of cell death in our system. As expected from the figures presented in the previous version of the manuscript, the graphs and statistical analysis confirms the SHG results presented in the previous round of revision.

Figure R6

Figure R6: Quantification of fibrillar collagen.

a. Quantification of fibrillar collagen from the images presented in Figure 6e; Graph represents the area of SHG-positive signal over the area of whole assembloid, for the non-targeting pool control group and the Cdh11-KD group at day3 (assembloid seeding). Dot, an individual portal assembloid. Non-targeting pool (n=15) and Cdh11-KD (n=14), combined from two biological replicates, **, P=0,0052 using a two-tailed Mann-Whitney test.

Referee #4 (Remarks to the Author):

Referee #4 to the manuscript Dowbaj et al., Functional periportal assembloids...

Comments to authors and editors

General summary

The authors have not convincingly addressed my comments. The manuscript remains a description of a three-dimensional cell culture system (named HepaOrg). This cell culture system recapitulates some features of liver tissue *in vivo*, such as formation of bile canaliculi into which a bile acid analogue is secreted and also polarized cells can be maintained for longer periods; however, these features have already been reported in previous publications using similar cell culture systems. The claim that the organoids “are reminiscent of the defects observed in cholestatic injury and in mouse models of biliary/cholestatic fibrosis” is not convincing, because critical histological features of these diseases are not seen in the presented images of the cell culture system. Moreover, the HepaOrg system does not recapitulate critical structural and functional features of a liver lobule (the smallest functional unit of the liver) and does not show typical features of liver tissue such as zoned expression of specific genes (meaning that expression levels differ in a specific pattern between the centre and the periphery of the lobules). Experiments to understand the molecular mechanisms responsible for tissue organization were not performed in this study.

*We understand that the Reviewer does not appreciate the novelty of our system and dismisses the impact of a 3D system vs a 2D system. He/she keeps insisting, as in the previous round, on comparing our 3D-HepOrg with the published 2D-hepatocyte sandwich culture method, and considers that the advance of our method is minimal. **We understand the Reviewer does not appreciate the Organoid field in general**, including the organoids we present here, neither understands that in Organoids the cells self-organize to generate a part of the liver tissue *in vitro*.*

Once more, in the revised version, we had compared our 3D organoid system to the Reviewer’s suggested 2D-hepatocyte sandwich culture system, where cells grow as 2D on a collagen layer, and provided additional comparisons for the Reviewer only. We demonstrated that our 3D-HepOrg system is superior to the 2D-sandwich cultured hepatocytes in terms of cellular function, bile canaliculi architecture and also long-term expansion in Fig.1c, Supplementary Fig. 1i and Supplementary Fig. 4h on the revised version of the manuscript, as well as in figure R31 from the previous rebuttal, a figure for the Reviewers only).

***To our knowledge, such a system as ours with physiological bile canaliculi architecture, combining 3 different liver cells, showing bile canaliculi-bile duct connection and modelling many aspects of biliary fibrosis has not been published yet, neither in 2D in nor 3D.** We would like to invite the Reviewer to indicate the relevant literature in that regard. Of note, the relevance of studying 3D organoids instead of 2D cells has been reviewed extensively by the leaders in the field including Clevers, Knoblich, Sasai, Wells, Takebe, Arlotta, Lutolf, Little, Jensen, Lancaster and many others, including ourselves. Below, we provide a selection from the extensive list of relevant publications in that regard:*

<https://www.nature.com/articles/s41580-020-0259-3>,
<https://www.nature.com/articles/s41578-021-00279-y>,
<https://pubmed.ncbi.nlm.nih.gov/27315476/>
<https://www.science.org/doi/10.1126/science.aaw7567>
<https://www.sciencedirect.com/science/article/pii/S2213671123001881>
<https://pubmed.ncbi.nlm.nih.gov/32640930/>
<https://www.science.org/doi/10.1126/science.1247125#:~:text=Lancaster%20and%20Knoblich%20review%20organoids,the%20formation%20of%20these%20structures.>)

Next, I address in more detail the four comments to which the authors have responded.

Comment 1: (“It is not correct that this is the first study showing that primary liver cells form functional bile canaliculi in vitro and that bile canaliculi link to bile ducts.”) The authors agreed to my comment that they have not convincingly emphasized the novelty of their study. However, also the revised version does not convincingly show novelty. Before discussing the authors’ arguments in detail, I would like to give some examples what would be versus is not sufficiently novel to justify publication in a top journal. Having an organoid (or any culture system) that recapitulates a part of liver tissue, such as a group of liver lobules or at least one liver lobule would be of relevant novelty. However, this is clearly not achieved by the culture system presented in the present study. Only bile canaliculi or even bile canaliculi that connect to bile ducts (as demonstrated in the present work) are not novel and have already been reported previously. Also, liver cells that maintain polar features over a longer period of time are not novel. The authors now explain their novelty as a combination of 7 features (long term passaging while retaining bile canaliculi; the bile canalicular diameter is between 1.5 and 2 μm ; no cholestatic features; elongated interconnected bile canaliculi; functional connection of bile canaliculi to bile ducts; integration of fibroblasts; no immune system involvement). However, these features are not novel and several previous publications reached already a similar phenotype. It is also not novel that bile canaliculi in culture can have a similar diameter as in vivo (1-2 μm), since there are several publications where this has been shown (together with some functional features, such as transport of bile acids). It is also not sufficient that the authors used a 3D (instead of a 2D) system, since there are already dozens of publications with 3D “organoids” published. The authors included an additional experiment showing that deoxycholic acid led to bile canaliculi dilution and other morphological changes. However, several previously published studies have shown the influence of bile acids leading to altered (e.g. widened) bile canaliculi. For example doi.org/10.1016/j.bcp.2012.07.008 write: “Cells cultured for 3 days in the presence of UDCA, the bile canaliculi lumen was enlarged and the bile canaliculi ...”. In conclusion, the novelty remains absent or moderate.

*Once more, the Reviewer shows no appreciation for the organoid field in general, and specifically ignores the advance of our model compared to previously published 2D and 3D models as outlined above and in the previous rebuttal. Generation of 3D bile canaliculi network is one of the hallmarks of hepatocytes in vivo, in liver tissue. **Until now, none of the hepatocyte organoids generated using either the Hu et al., or the Peng et al., method have investigated in depth, or focused on generating, 3D bile canaliculi network in an in vitro organoid system. As we show in our manuscript, the present published models do not present bile canaliculi of sufficient size or network to guarantee bile duct connection (Extended Data Fig. 1b-f).***

Thus, we have described in detail the advance of our system in regards to bile canaliculi, compared to these previous methods.

This Reviewer also questions novelty on several other fronts “long term passaging while retaining bile canaliculi; the bile canalicular diameter is between 1.5 and 2 μm ; no cholestatic features; elongated interconnected bile canaliculi; functional connection of bile canaliculi to bile ducts; integration of fibroblasts; no immune system involvement). However, these features are not novel and several previous publications reached already a similar phenotype”, but does not provide literature to support their claim, just mentions the “dozen 3D organoid publications” and including immune cells in the system.

We had already extensively discussed in the previous response to this Reviewer, the differences between the previously described 2D-culture systems and our HepOrg. We had also described the differences between the previous bile canaliculi-bile duct connection by Tanimizu et al. 2021 (reference from the Reviewer in the previous version), where hepatocytes are cultured in 2D. We had shown comparisons of the differences in bile canaliculi architecture and also time scales, and the clear advance of our system. As we demonstrated in the previous version, 2D-hepatocytes, as the ones in Tanimizu’s 2021 do not have bile canaliculi network, are cholestatic, and when the bile canaliculi connect to the ductal structure, the connection is not physiological, i.e., it takes hours to transfer bile acid to the bile duct structure, a process that in vivo occurs in seconds/min as published by Vartak et al. 2021 (doi:10.1002/hep.31422 (2021)). As shown in Video 12 of our manuscript, in our assembloids system bile acid drains in sec/min to the bile duct, resembling the in vivo timescales.

We have now edited the manuscript to further clarify the novelty on those exact points:

For bile canaliculi and absence of cholestatic features:

Lines 196-201: “Taken together, our optimized hepatocyte organoid system enables the growth of adult hepatocytes that preserve hepatocyte polarity, show partial hepatocyte zonation and generate a physiological and functional bile canaliculi 3D network akin to the *in vivo* adult liver tissue, without emergence of cholestatic features found in published 2D-hepatocyte cultures. Critically, our results highlight the importance of how mimicking physiological properties at the cellular and tissue-scale level impacts the overall fitness of our hepatocyte culture model.”

Lines 406-414: “Remarkably, our system recapitulates for the first time the complexity of the physiological 3D network architecture of the *in vivo* tissue. Notably, when HepOrg are generated from the cholestatic *Mdr2*^{-/-} mouse model, they readily recapitulate, with unprecedented precision, the specific histological features of cholestatic liver disease^{80,87}. This is in stark contrast to current 2D-cultured hepatocyte models, all of which are cholestatic at baseline (present bulkheads)⁸⁰, lack the 3D- network organization typical of transport networks⁸⁸⁻⁹⁰, and deteriorate within days in culture^{27,39}. Also, published hepatocyte organoids models lack bile canaliculi network. To our knowledge, ours is the first study reporting a 3D-physiological bile canaliculi network that recapitulates many features of *in vivo* physiology and pathophysiology of liver disease.”

For functional connection of bile canaliculi and to bile duct and integration of fibroblasts:

Lines 399-407: “Here, we generated a periportal assembloid model that combines, for the first time, portal mesenchyme, cholangiocytes and hepatocytes, and readily recapitulates the

cellular and mesoscale architecture of the liver's periportal region, albeit lacking the portal endothelium and resident immune cells. By improving hepatocyte organoid (HepOrg) models to retain hepatocyte polarity and form bile canaliculi 3D-network, and mixing them with cholangiocytes and portal mesenchyme, we reconstruct, for the first time, the physiological and functional connection between the bile canaliculi and bile duct. All of these features are absent in published methods".

In addition the reviewer mentions' "Having an organoid (or any culture system) that recapitulates a part of liver tissue, such as a group of liver lobules or at least one liver lobule would be of relevant novelty."

We have not claimed mechanisms of tissue organization at the level of liver lobule, only recapitulation of a very specific liver region. We have also not claimed full zonation of the structures, which are too small to model full liver axis, even though some extent of zonation exists in our system. As per the Reviewer's request, we have stained our HepOrg with the markers the Reviewer suggested, and we reported that there is discrepancy in hepatocytes within the organoid structure. While some are clearly positive, some hepatocytes do not show the presence of for example GS marker. The Reviewer request to reconstitute liver lobule and zonation will most likely involve another 5 different intermediate models, and 10-20 years of work, which is not reasonable to ask in the timescale of this manuscript.

Finally, the reviewer also mentions: "The authors included an additional experiment showing that deoxycholic acid led to bile canaliculi dilation and other morphological changes. However, several previously published studies have shown the influence of bile acids leading to altered (e.g. widened) bile canaliculi. For example doi.org/10.1016/j.bcp.2012.07.008 write: "Cells cultured for 3 days in the presence of UDCA, the bile canaliculi lumen was enlarged and the bile canaliculi ...". In conclusion, the novelty remains absent or moderate."

*The bile acid addition with DCA experiments were not included in the manuscript, and were shown only for the Reviewer in the previous version, to demonstrate that the system also reacts, as in vivo, to the excess bile acid. **It was not our intention to claim novelty regarding this specific point, but we performed the experiments requested by the Reviewer to show that this specific experiment leads to similar results also in our system.***

Comment 2: ("This reviewer does not agree to the statement that the "Periportal assembloids model cholestatic liver fibrosis ..."). Also in the revised version the authors do not show characteristic features of fibrosis and ductular reaction which are seen in vivo. In vivo, a very characteristic formation of new ducts (appearing as new tubes and a denser network of tubes) is seen. This is not seen in Fig R30. Please consider that "cholangiocyte cell expansion" is not the same as ductular reaction.

As mentioned in the manuscript (Fig. 5 and previous Suppl. Fig. 14, now Extended Data Fig 11), our model recapitulates the following characteristics of a fibrotic liver: (i) fibrotic gene expression similar to in vivo models of biliary fibrosis, (ii) hepatocyte death, (iii) bile flow obstruction, (iv) duct cell expansion and (v) collagen deposition. All together indicate that fibrotic assembloids recapitulate in vitro many aspects of the in vivo biliary /cholestasis

fibrosis, except for the inflammatory reaction, as expected given that our system lacks the immune compartment.

Regarding the formation of “tubes”, we would invite the Reviewer to revisit Fig.5b, where the assembloids exhibit cholangiocyte tube formation in the fibrotic-like conditions. To facilitate the Reviewer’s assessment, we provide below the z-stacks of manuscript’s Figure 5b as well as 3D rendering and higher resolution pictures, to better illustrate this point (Figure R7 for Reviewer only).

Figure R7

Figure R7 for Reviewer only: Ductal cells in fibrotic-like assembloid form tubular structures.

Left, schematic illustrations of fibrotic-like assembloid (top) and homeostatic-like assembloid (bottom). Hepatocytes (white), ductal cells (magenta). Middle, 3D visualisation of ductal structures in the fibrotic-like and homeostatic-like assembloid (top and bottom, respectively), from Figure 5b. Left, three individual z-stacks are shown for each assembloid. White dotted lines and letters (A, B or C) indicate which Z-stack corresponds to which part of the 3D visualised assembloid. Cell borders were stained with Phalloidin (grey) and ductal cells are nucTom (magenta). Scale bar, 50 µm.

Comment 3: (It is not convincing that the organoids “recapitulate the mesoscale hepatic architecture...”).

The authors agree that their culture system does not recapitulate a liver lobule. The problem I see is that features at a smaller scale (bile canaliculi, polar cells, interactions between cell types, response of hepatocytes to Wnt) have already been reported in previous publications. The data based on which the authors claim “zonation” (Fig. R2e, f) are not convincing. In vivo, e.g., CYP2E1 is expressed in the central part of the liver lobule. However, this is not the case for the organoids (HepOrg), where CYP2E1 occurs predominantly the peripheral cells and less in the center. However, also (suspiciously) ECAD is also less expressed in the centre. In contrast, glutamine synthetase (GS) is expressed only in an inner ring of hepatocytes around the central vein in vivo. However, in the organoid (HepOrg) GS appears unspecifically

in the periphery and in the centre. It remains unclear, why the authors think that they obtained zonation similar as in vivo.

As mentioned already above, we have not claimed mechanisms of tissue organization at the level of liver lobule, only recapitulation of a very specific liver region. We have also not claimed full zonation of the structures, which are too small to model the full liver axis.

Traditionally, the hexagon-shaped liver lobule is considered basic structural and functional unit, where in the centre of the hexagon lies the central vein (hence, the pericentral region name), and on the tips of hexagon lie portal triads containing portal vein (hence, the periportal region name). Wnt signalling is the strongest next to central vein and gradually decreases towards the portal regions.

*In our system, since we add Wnt signalling molecules through the media, the hepatocytes that are on the periphery (surface) of the organoid are the ones that show more pericentral signature. So, in that sense, we have a reverse orientation of pericentral and periportal signature, where pericentral is present on the surface. Regarding the orientation of the expression of CYP2E1 and GS in the periphery, the Reviewer did not appreciate that the peripheral cells are the ones exposed to the medium that contains Wnt and, consequently, these represent the pericentral region, while the centre of the organoid has less access to Wnt signals, and expressed periportal region markers such as ECAD. Hence, the staining is consistent with the Wnt-gradient orientation and, in more general terms, this result is in the agreement **with other organoid models**, including the seminal work on intestinal organoids by the Clevers and the Wells labs, where the apical domain is inside the structure, and not outside, because of the ECM being in the outside.*

In the Figure 2a of the manuscript, and previous revised figure Fig. R2e, f, we had shown the maximum intensity projection of the whole organoid. This could have made it less clear to the Reviewer where the “pericentral” cells are. Therefore, we have now explained it better in the text and also in the figure legend that the pericentral marker (GS) is visible in the peripheral cells of the organoid as follows:

Lines 175-180: “RNAscope and immunofluorescence analysis for some periportal (Gls2 – RNA, ALB, ECAD - protein) and pericentral (Cyp1a1 – RNA, CYP2E1, GS -protein) gene markers confirmed that some cells expressed more pericentral while other express more periportal markers (Fig.2a and Extended Data Fig.5e-f), with the pericentral markers generally more expressed on the hepatocytes at the surface/periphery of the organoid, which are more exposed to the Wnt ligand from the media.”

Legend Figure 2a (Lines 1208-1210): Images are maximum intensity projections. Note that the scheme represents a z-stack of the middle of the organoid where GS positive cells are detected in the periphery.

*Additionally, we provide here, **for the Reviewer only**, two z-slices of the same organoid from Figure 2a one of the surface of the organoid and another one from the center (**Figure R8 for Reviewer only**). Would the Editor or Reviewer consider a requirement that these panels are also presented in the manuscript, we would be delighted to follow the Editor or Reviewer’s advice and include these in Figure 2a.*

Figure R8

a

b

Figure R8 for Reviewer only: Optimized HepOrg cultures show a degree of zonation.

a. Left, schematic representing the position and distribution of zonation markers along CV-PV axis in liver. Immunofluorescence staining of periportal (ALB, Albumin, green) and pericentral (GS, Glutamine synthetase, magenta) in liver tissue (top) or in HepOrg cultures grown in HM-Wnt and cultured in MM media (bottom) for 7 days. Both ALB and GS are also shown in Fire LUT for the ease of viewing. Maximum intensity projections of representative images are shown. Scale bar, 50 μm (tissue) and 20 μm organoid.

b. Left, schematic representing the 3D representation of HepOrg and cross section showing the position and distribution of pericentral zonation marker (Glutamine synthetase, magenta, to match the pseudo-colouring on the images on the right) within HepOrg. Right, immunofluorescence staining of periportal (ALB, Albumin, green) and pericentral (GS, Glutamine synthetase, magenta) on the surface (top) or in the centre of HepOrg grown in

HM-Wnt and cultured in MM media (bottom) for 7 days. Both ALB and GS are also shown in Fire LUT for the ease of viewing. Single Z-slice of the image from panel a is shown. Scale bar, 20 μ m.

Comment 4: (This reviewer finds it unclear if “assembloids ... recapitulate the ... gene expression of the liver tissue”).

I am aware of the supplemental figure but would criticize that the procedure to compare cells from organoids (HepOrg) to primary (isolated) hepatocytes is not adequate. For the analysis of RNAseq data, I would expect (among others) a comparison of cells from HepOrg and primary hepatocytes with a list of the differentially expressed genes, including *fd*-adjusted p-values and fold-changes. A statement such as “our hepatocytes cluster with publicly available hepatocytes” is not sufficient since advanced biostatistical methods for quantification of similarity are available; but one would begin with the basics, especially a list of differentially expressed genes.

We now provide the requested additional datasets, which contains the differential gene expression analysis of our bulk RNAseq, comparing HepOrg to freshly isolated hepatocytes (Supplementary dataset 6).